# An improved global Remote Sensing-based Surface Soil Moisture (RSSSM) dataset covering 2003~2018

Yongzhe Chen[1,2], Xiaoming Feng[1,*], Bojie Fu[1,2]

[1] State Key Laboratory of Urban and Regional Ecology, Research Center for Eco-Environmental Sciences, Chinese Academy
of Sciences, Beijing 100085, PR China.

[2] University of Chinese Academy of Sciences, Beijing 100049, PR China.

*Correspondence to:* Xiaoming Feng (fengxm@rcees.ac.cn)

**Abstract.** Soil moisture is an important variable linking the atmosphere and terrestrial ecosystems. However, long-term satellite monitoring of surface soil moisture at the global scale needs improvement. In this study, we conducted data calibration
and data fusion of 11 well-acknowledged microwave remote sensing soil moisture products since 2003 through a neural network approach, with Soil Moisture Active Passive (SMAP) soil moisture data applied as the primary training target. The training efficiency was high ($R^2$ =0.95) due to the selection of 9 quality impact factors of microwave soil moisture products and the complicated organizational structure of multiple neural networks (5 rounds of iterative simulations; 8 substeps; 67 independent neural networks; and more than one million localized subnetworks). Then, we developed the global Remote
Sensing-based Surface Soil Moisture dataset (RSSSM) covering 2003~2018 at 0.1° resolution. The temporal resolution is approximately 10 days, meaning that 3 data records are obtained within a month, for days 1~10, 11~20 and from 21 to the last day of that month. RSSSM is proved comparable to the in situ surface soil moisture measurements of the International Soil Moisture Network sites (overall $R^2$ and RMSE values of 0.42 and 0.087 m³/m³), while the overall $R^2$ and RMSE values for the existing popular similar products are usually within the ranges of 0.31~0.41 and 0.095~0.142 m³/m³, respectively. RSSSM
generally presents advantages over other products in arid and relatively cold areas, which is probably because of the difficulty in simulating the impacts of thawing and transient precipitation on soil moisture, and during the growing seasons. Moreover, the persistent high quality during 2003~2018 as well as the complete spatial coverage ensure the applicability of RSSSM to studies on both the spatial and temporal patterns (e.g., long-term trend). RSSSM data suggests an increase in the global mean surface soil moisture. Moreover, without considering the deserts and rainforests, the surface soil moisture loss on consecutive
rainless days is highest in summer over the low latitudes (30°S~30°N) but mostly in winter over the mid-latitudes (30°N~60° N; 30°S~60°S). Notably, the error propagation is well controlled with the extension of the simulation period to the past, indicating that the data fusion algorithm proposed here will be more meaningful in the future when more advanced microwave sensors become operational. RSSSM data can be accessed at https://doi.pangaea.de/10.1594/PANGAEA.912597 (Chen, 2020).

## 1 Introduction

Soil moisture plays an important role in modulating the exchange of water, carbon and energy between the land surface and atmosphere, and it also links the global water, carbon and energy cycles (Dorigo et al., 2012; Karthikeyan et al., 2017a). Soil moisture has been endorsed by the Global Climate Observing System (GCOS) as an essential climate variable (Bojinski et al., 2014), because it can indicate the climatic impact on the ecosystems, such as during ecological droughts (Martínez-Fernández et al., 2016; Samaniego et al., 2018). Current research requires high-quality soil moisture information in terms of data accuracy

and spatial-temporal coverage (Hashimoto et al., 2015; Stocker et al., 2019).

Reanalysis-based land surface model products are frequently used, including the Global Land Data Assimilation System (GLDAS, with 0.25° resolution) (Rodell et al., 2004), European Reanalysis (ERA)-interim (0.75°) (Balsamo et al., 2015) and its successors ERA5 (0.25°) and ERA5-Land (0.1°) (Hoffmann et al., 2019)). These products can often predict temporal variations well due to the incorporation of the time variance of environmental factors, e.g., precipitation. In addition, the

modeling approach can also provide information on the soil moisture in soil layers deeper than the surface layer (< 5 cm). The uncertainties arise from meteorological forcing data, model parameters, as well as inadequacies in model physics (Cheng et al., 2017). Moreover, the anthropogenic impacts from irrigation and land cover changes are rarely considered (Kumar et al., 2015; Qiu et al., 2016).

With advances of remote sensing technology, microwave remote sensing became an alternative to soil moisture monitoring.

Currently, global-scale soil moisture can be acquired from either passive sensors (e.g., SMMR, SSM/I, TMI, WindSAT, AMSR-E, AMSR2, SMOS, SMAP, see Table 1 for the full names) or active sensors (e.g., ERS and ASCAT), with that within the top 5 cm of soil being detectable (Feng et al., 2017; Jiao et al., 2016; Piles et al., 2018). The data quality and spatial coverage are improved step by step (Karthikeyan et al., 2017b). However, valid temporal spans of all these sensors are limited, and the data quality and spatial coverage were considered to be unsatisfactory until the launch of AMSR-E in June 2002

(Karthikeyan et al., 2017b; Kawanishi et al., 2003). Currently, ASCAT sensors have produced the longest continuous record of global surface soil moisture of microwave remote sensing (Bartalis et al., 2007), with the temporal span from 2007 until present. Satellite-based soil moisture retrievals may also suffer from various disturbances, such as lower quality over dense

vegetation cover, high open water fractions and complex topography (Draper et al., 2012; Fan et al., 2020; Ye et al., 2015). Difference in the algorithms dealing with the disturbances make different microwave soil moisture products hardly comparable with each other (Kim et al., 2015a; Mladenova et al., 2014). New sensors, such as SMOS (Kerr et al., 2001) and SMAP (Entekhabi et al., 2010), can produce significantly improved estimates because L-band microwaves (1.4 GHz (Kerr et al., 2001)) penetrate the vegetation canopy better than shorter wavelengths (Burgin et al., 2017; Chen et al., 2018; Karthikeyan et al., 2017b; Kerr et al., 2016; Kim et al., 2018; Leroux et al., 2014a; Stillman and Zeng, 2018). However, SMOS data are noisy and lacks data in Eurasia due to high radio frequency interference (RFI) (Oliva et al., 2012). While the SMAP passive product has achieved an unbiased RMSE that is close to its target of 0.04 $m^3/m^3$, and has incorporated hardware RFI mitigation (Chen et al., 2018; Colliander et al., 2017), the data are only available since March 2015.

Interest in fusing satellite-observed and modeled soil moisture has increased recently. The European Space Agency (ESA) published a long-term surface soil moisture dataset called the Climate Change Initiative (CCI), and the latest version (v4.5) covers the time period of 1978~2018. Two steps contribute to the combined CCI product. The first step involves rescaling the soil moisture of all microwave sensors against the reference data (GLDAS Noah product) by cumulative distribution function (CDF) matching, while the second step merges the rescaled products together by selecting the best product in each subperiod or averaging the products weighted by the estimated errors (Dorigo et al., 2017; Gruber et al., 2017; Gruber et al., 2019; Liu et al., 2012). CCI utilized almost all the available microwave soil moisture datasets to form a long time series, and generally agrees well with measured values at some sites, e.g.,, the Irish grassland sites and the grassland and agricultural fields in the United States, France, Spain, China and Australia (Albergel et al., 2013; An et al., 2016; Dorigo et al., 2017; Pratola et al., 2015). Valid microwave observations were quite limited before June 2002 due to satellite sensor constraints (Dorigo et al., 2017). Through CDF matching, the CCI soil moisture references the spatial patterns of all the satellite products relative to that of GLDAS (Gruber et al., 2019; Liu et al., 2012; Liu et al., 2011b). The temporal variation in each satellite product is retained, although the data averaging (Liu et al., 2012) cannot efficiently distinguish between the divergent interannual variations in various products (Feng et al., 2017). Soil Moisture Operational Product System (SMOPS) v3.0 is another global blended surface soil moisture dataset that was developed in the similar way (Yin et al., 2019). SMOPS v3.0 is a daily/6-hourly temporal

interval dataset with a complete global land coverage since March 2017. The overall performance, which is indicated by a RMSE of 0.035~0.066 $m^3/m^3$, is slightly lower than that of CCI (with RMSE of 0.031~0.06 $m^3/m^3$) (Wang et al., 2021). The Global Land Evaporation Amsterdam Model (GLEAM) surface soil moisture was produced by assimilating CCI data into a

land surface model- GLEAM (Burgin et al., 2017; Martens et al., 2017; Miralles et al., 2011) through an optimized Newtonian Nudging approach (Martens et al., 2016). The general performance of the GLEAM soil moisture product is satisfactory (Beck et al., 2020). In the current version, the CCI soil moisture anomalies (the deviations to the seasonal climatology, which indicate whether the soil moisture at a time point is more humid or drier than the multiyear average) are assimilated instead of the original CCI time series (Martens et al., 2017). Therefore, satellite observations play a much smaller role than modelling in

forming the GLEAM product. For further improvements in the efficiency of soil moisture assimilation, a high-quality long-term surface soil moisture dataset basically derived from microwave remote sensing is highly needed.

In addition to the CDF matching algorithm, at least four methods have been proposed that target the use of the information acquired by one sensor to produce soil moisture data that are compatible with the data retrieved from another. Based on physical-based equations (Wigneron et al., 2004), the regression between SMOS soil moisture and dual-polarized brightness

temperature (Tb) data from AMSR-E is applied to match the AMSR-E soil moisture time series to SMOS (R-square =0.36) (Al-Yaari et al., 2016). An example of the second method uses the Land Parameter Retrieval Model (LPRM) (Owe et al., 2008) to retrieve soil moisture from SMOS and then match the 'SMOS-LPRM' data with the AMSR-E-LPRM product by calibrating the LPRM parameters and then applying a linear regression (Van der Schalie et al., 2017). Thirdly, Copulas functions allow to model the structure of the dependence between two different Tb or soil moisture datasets and thus could perform better for

the extreme values, thereby reducing the RMSE (Gao et al., 2007; Leroux et al., 2014b; Lorenz et al., 2018; Verhoest et al., 2015). To better characterize the nonlinear relationship between two datasets (Rodriguez-Fernandez et al., 2015), researchers built a neural network that links SMOS soil moisture to the Tb at different polarizations and frequencies of AMSR-E to produce a calibrated soil moisture data product that covers 9 years (2003~2011) (Rodríguez-Fernández et al., 2016). This approach proves to be efficient according to the connection between precipitation and the soil moisture changes, as evaluated

based on a data assimilation technique and triple collocation analysis result (Van der Schalie et al., 2018).

A global long-term observational-based soil moisture product was recently developed by building a neural network between the SMOS product and the Tb data from AMSR-E (2003~September 2011) and AMSR2 (July 2012~2015) (Yao et al., 2017). Environmental factors, including the land surface temperature (LST) derived from the Tb at 36.5 GHz (Holmes et al., 2009) and the microwave vegetation index (MVI, an indicator of vegetation cover), were also incorporated as ancillary inputs. The

training R-square value ($R^2$) of this product was only 0.45 (or correlation coefficient, $r$, equals 0.67), and the validation against in situ measurements showed a temporal $r$ of 0.52 and temporal RMSE of 0.084. Soil moisture data are partially missing due to the gap between the temporal spans of AMSR-E and AMSR2 and the lack of SMOS data in Asia. As SMAP observations have become increasingly available, SMAP soil moisture data have been chosen as the training target, thereby improving the training $R^2$ to 0.55, while the overall $r$ and RMSE against measurements are 0.44 and 0.113 (Yao et al., 2019). Another study

rebuilt a soil moisture time series over the Tibetan Plateau by using SMAP data as the reference for a random forest (Qu et al., 2019). For the environmental factors, while vegetation cover is not considered, elevation, IGBP land use cover type, grid location and the day of a year (DOY) were chosen as ancillary inputs. The training $R^2$ in this region reached 0.9 with a high temporal accuracy (temporal $r$=0.7; RMSE=0.07 in the unfrozen season). However, these data are regional (for the Tibetan Plateau only), and have a temporal gap between AMSR-E and AMSR2 data (October 2011~June 2012).

Therefore, although previous studies have focused on developing long-term satellite-based surface soil moisture products using machine learning, major concerns remain to be addressed. 1) Training designed for soil moisture estimation at the global scale should be more complex than that for only a specific region to ensure a satisfactory training efficiency; 2) microwave observations are often limited to three sensors, leading to temporal and spatial gaps at the global scale and the limited training efficiency; 3) the environmental factors that should be incorporated as ancillary inputs have not been clarified. In this study,

11 high-quality microwave soil moisture products starting from 2003 are incorporated into iterative 5-round neural networks to produce a spatially and temporally continuous dataset for 2003~2018, and as many sources of microwave observational data as possible are used as predictors in each neural network. The quality impact factors of microwave soil moisture retrievals are also determined and then incorporated as ancillary inputs to improve the training efficiency. Moreover, we designed localized subnetworks instead of one global-scale neural network to account for the regional differences in training rules.

## 2 Data and Methods

### 2.1 Data for the production of global long-term surface soil moisture data

### 2.1.1 Satellite-based surface soil moisture data products

SMAP currently has the highest quality of all remote sensing-based soil moisture products (Al-Yaari et al., 2019) and is thus chosen as the primary training target. The SMAP Enhanced L3 Radiometer Global Daily 9 km EASE-Grid Soil Moisture V002 (SPL3SMP_E_002, hereinafter SMAP_E for short), which was developed by improving the spatial interpolation of the original 36 km resolution SMAP soil moisture data (Chan et al., 2018), was adopted in this study. SMAP_E was reprojected from the EASE-Grid 2.0 projection with 9 km resolution to the WGS1984 geographic coordinate system with 0.1° resolution. The nominal penetration depth of SMAP_E is ~5 cm.

Previous studies often used Tb observations at various bands as network inputs (Rodríguez-Fernández et al., 2016). However, in this study, the well-acknowledged surface soil moisture products retrieved through mature algorithms (see Figure 1) are directly applied instead of Tb because 1) the primary goal of this study is to calibrate and then fuse the existing popular microwave soil moisture products and 2) the Tb signals at multiple bands contain too much information that is not related to soil moisture, which may weaken the training efficiency and lead to overfitting. Although the drawback is that the final soil moisture products may inherit the uncertainties associated with each retrieval method, this problem can be generally solved by including quality impact factors (see section 2.1.2). The first satellite soil moisture product that is used as a predictor is the ASCAT soil water index (ASCAT-SWI) product, which was developed by the European Meteorological Satellite Organization (EUMETSAT) and provided by the ESA-Copernicus Land Monitoring Service (Albergel et al., 2008; Wagner et al., 1999). The saturation degree in the top soil layer (SWI_001) was converted to volumetric soil moisture by multiplication with soil porosity data included in the SMAP L4 Global Surface and Root Zone Soil Moisture Land Model Constants V004 dataset (hereinafter, 'SMAP Constant'; note that porosity data were not provided in the ASCAT-SWI). Second, AMSR2-JAXA is the AMSR2 soil moisture retrieved by the Japan Aerospace Exploration Agency (JAXA) using Tb at the X-band (10.65 GHz) (Fujii et al., 2009), and version 3 data on the Global Portal System (G-Portal) were used. Third, AMSR2-LPRM-X stands for the AMSR2 soil moisture produced by applying the LPRM algorithm at the X-band (Parinussa et al., 2014) (X-band retrievals

may not perform well in high-vegetated areas, but C-band data such as AMSR2-LPRM-C or AMSR-E-LPRM-C were not applied due to the high RFI, especially in the United States, Japan, and the Middle East (Njoku et al., 2005)), and is obtained from NASA's Earthdata Search web. The fourth predictor, SMOS-IC (SMOS INRA-CESBIO), is a new SMOS soil moisture product created by INRA (Institut National de la Recherche Agronomique) and CESBIO (Centre d'Etudes Spatiales de la BIOsphère) with the main goal of being as independent as possible from the auxiliary data, including the simulated soil moisture (Fernandez-Moran et al., 2017a; Fernandez-Moran et al., 2017b; Wigneron et al., 2007). The accuracy of SMOS-IC has been proven to be higher than that of other SMOS products (Al-Yaari et al., 2019; Ma et al., 2019), and the data version 105 offered by Centre Aval de Traitement des Données SMOS (CATDS) is adopted. TMI-LPRM-X is the X-band LPRM product of TMI and was created by the NASA Goddard Space Flight Center (GSFC), which is used as the 5th predictor. Fengyun 3B is a Chinese meteorological satellite with a Microwave Radiation Imager (MWRI) onboard (Yang et al., 2011; Yang et al., 2012). The National Satellite Meteorological Center product is retrieved using the Tb at 10.7 GHz, and it is denoted by 'FY-3B-NSMC' (the 6th predictor product). WindSat is onboard the Coriolis satellite (Gaiser et al., 2004), and the soil moisture retrieved by LPRM at the X-band (Parinussa et al., 2012) is provided by NASA (the 7th predictor). Three AMSR-E products are used, including the NASA product (AE_Land3) created by the National Snow and Ice Data Center (AMSR-E-NSIDC) (Njoku et al., 2003), the JAXA product (AMSR-E-JAXA) (Fujii et al., 2009; Koike et al., 2004) obtained from G-Portal and the LPRM product (AMSR-E-LPRM) available at the NASA Earthdata Search. All these data are reprojected to the WGS-1984 reference coordinate system and resampled to 0.1°.

To reduce noise and fill the gaps between sensor observation tracks (at least 3 days are required for a microwave sensor to cover the whole globe), for every soil moisture product, both the daytime and nighttime observations within each 10-day period are combined by data averaging (the relative superiority of daytime and nighttime retrievals is not considered). For example, for SMAP, 11% of the global land surface has data for only 5 days or less within a 10-day period. Therefore, the temporal resolution of the dataset developed in this study is approximately 10 days, meaning that 3 data records are obtained within a month for days 1~10, 11~20 and from 21 to the last day of that month. This format is exactly the same as that of the ASCAT-SWI and many other products developed by the Copernicus Land Monitoring Service (https://land.copernicus.eu).

**2.1.2 Quality impact factors of soil moisture retrievals**

Environmental factors, including elevation, LST and vegetation cover (indicated by the Normalized Difference Vegetation

Index or MVI, etc.), were used as ancillary neural network inputs to improve the soil moisture simulation (Lu et al., 2015; Qu

et al., 2019; Yao et al., 2017). According to these studies, these factors alone may not predict surface soil moisture well without

the incorporation of any microwave remote sensing data because although they are somewhat related to soil moisture (e.g.

soil moisture is generally limited in areas with low vegetation cover but high in forests (McColl et al., 2017)), the relationships

are rather uncertain (e.g., at smaller scales, the leaf area index (LAI) may have a negative influence on soil moisture due to

the variation in evapotranspiration (Naithani et al., 2013), or may not have a clear impact (Zhao et al., 2010); also, soil moisture

can be either high or low in summers when vegetation peaks (Baldocchi et al., 2006; Méndez-Barroso et al., 2009)). However,

these factors are quite essential due to their direct impacts on microwave-based soil moisture retrieval through the radiative

transfer model and other models (Fan et al., 2020; Karthikeyan et al., 2017a); thus, they are retrieval quality impact factors.

Detailed explanations are as follows. 1) The bias of soil moisture estimates derived from a certain sensor or a specific algorithm

can be correlated with the degree of disturbances from various environmental factors. For example, in vegetated areas, LST

is overestimated by LPRM (Ma et al., 2019), whereas soil moisture is underestimated by JAXA (Kim et al., 2015a), and the

magnitudes of the biases are often determined by vegetation amount or vegetation optical depth (VOD). Therefore, the

environmental factors are essential for a better calibration of various products, especially when soil moisture, which contains

errors associated with the retrieval method, is directly applied instead of the Tb. 2) The relative performances of different

products is also controlled by environmental factors; for example, the ASCAT product is preferable to AMSR-E-LPRM in

vegetated areas (Dorigo et al., 2010), while LST influences the relative superiority of the LPRM and JAXA algorithms (Kim

et al., 2015a). Therefore, for improved data fusion, the weights assigned to different soil moisture (or Tb) predictor data

available at the same time should be determined by referring to these quality impact factors (Kim et al., 2015b).

In this study, 9 quality impact factors are incorporated: LAI, water fraction, LST, land use cover, tree cover fraction, non-tree

vegetation fraction, topographic complexity, soil sand fraction and clay fraction (see Figure 1). The reasons are as follows.

Based on the two criteria above, the first environmental factor to be included is the 'vegetation factor' (i.e., vegetation water

content, VWC). Plants can absorb or scatter radiation from soil and emit radiation, thereby reducing the sensitivities of both radiometers and radars to soil moisture (Du et al., 2000; Owe et al., 2001). However, L-band microwaves can penetrate the vegetation layer better due to their longer wavelengths (Konings et al., 2017; Piles et al., 2018). On the other hand, although vegetation effects can be somewhat corrected (Jackson and Schmugge, 1991), different methods have different efficiencies. Radiative transfer models such as LPRM may have difficulty describing the radiation attenuation by dense canopy due to the neglect of multiple scattering (Mo et al., 1982; Owe et al., 2008), whereas the TU-Wien change detection algorithm applied to ASCAT utilizes the quadratic polynomial dependence of backscatter on the incidence angle to better characterize the vegetation effect on backscatter and then remove it by identifying the reference angles (Hahn et al., 2017; Vreugdenhil et al., 2016). Microwave vegetation indexes may contain large uncertainty and have coarse resolutions (Liu et al., 2011a; Shi et al., 2008). The NDVI becomes saturated at high vegetation cover (Huete et al., 2002). Because the LAI stands for the total leaf area per unit land, which is closely related to the VWC assuming a relatively stable leaf equivalent water thickness (Yilmaz et al., 2008), LAI is a suitable surrogate. Copernicus global 1 km resolution LAI (called GEOV2-LAI, which consists of SPOT-VGT and PROBA-V LAI) data are adopted here due to the high accuracy and full coverage (Baret et al., 2013; Camacho et al., 2013; Verger et al., 2014). Because the sensor conversion from SPOT-VGT to PROBA-V in 2014 led to LAI data discontinuity in specific areas (Cammalleri et al., 2019), which may reduce neural network training and simulation efficiency, the Global LAnd Surface Satellite (GLASS) LAI product (Xiao et al., 2014; Xiao et al., 2016) from 2007~2017 is also used (Figure 1). The LAIs are averaged on a monthly scale and aggregated to 0.1° resolution. The second is the 'water fraction factor' (i.e., the fraction of water area in each pixel). Waters in land pixels dramatically decrease the Tb, thereby leading to overestimated soil moisture. Because different methods are used to detect and correct small areas of water, either open water, wetlands or partly inundated wetlands and croplands (Entekhabi et al., 2010; Kerr et al., 2001; Mladenova et al., 2014; Njoku et al., 2003), microwave soil moisture data calibration and weight assignment based on the water fraction within land pixels make sense (Ye et al., 2015). In addition, the water fraction is a direct indicator of surface soil moisture. In this study, the daily water area fraction derived from the Surface WAter Microwave Product Series (SWAMPS) v3.2 dataset (Schroeder et al., 2015) is applied. The third factor is the 'heat factor' (i.e., LST). Soil moisture retrievals from passive microwave sensors are

based on the correlation between the soil dielectric constant, which is influenced by soil moisture, and the emissivity estimated as the ratio of Tb to soil physical temperature (Ts) (Karthikeyan et al., 2017a). Ts is approximate to the LST and can be derived from the Tb at 36.5 GHz (Holmes et al., 2009; Parinussa et al., 2011) or from reanalysis datasets including ECMWF, MERRA and NCEP, or set as a constant of 293 K (Koike, 2013). Active microwave products are independent of LST (Ulaby et al.,

1978). Because different LST estimates are used in the retrievals of different soil moisture products, while the bias of each LST estimate compared to the actual LST is influenced by the actual LST, we assume that the actual LST can determine the accuracy of every LST estimate and finally the relative performances of various soil moisture products (Kim et al., 2015a). In this study, we averaged the MODIS monthly LST acquired from the ascending and descending passes of both TERRA and AQUA. The 4~6[th] factors are the 'land cover factors', which are added because the parameters essential for soil moisture

retrieval (vegetation effect correction) are set based on land use types (Griend and Wigneron, 2004; Jackson and Schmugge, 1991; Jackson et al., 1982; Panciera et al., 2009). Additionally, landscape heterogeneity influences the retrieval accuracy (Lakhankar et al., 2009; Lei et al., 2018; Ma et al., 2019). Here, both the annual MODIS land use cover maps and the MEaSUREs vegetation continuous fields (i.e., the cover fractions of trees and non-tree vegetation (Hansen and Song, 2018)) are adopted. Apart from the above dynamic factors, three (7~9[th]) static factors are included: the 'topographic factor' (i.e.,

topographic complexity) and the 'soil texture factors' (two factors, sand fraction and clay fraction) (Neill et al., 2011). Both factors can influence the relationship between soil moisture and emissivity or the dielectric constant (Dobson et al., 1985; Karthikeyan et al., 2017a; Njoku and Chan, 2006), but they are characterized and corrected differently, leading to different relative performances of various soil moisture products (Das and O'neill, 2010; Gao et al., 2006; Kim et al., 2015a). For topographic complexity, the static layer of the Copernicus ASCAT-SWI product (hereinafter the ASCAT Constant) is adopted

while for soil texture, the SMAP Constant is used (topographic complexity data are not available from SMAP Constant while soil texture is not provided by ASCAT Constant). The contribution analysis results show that because various microwave soil moisture retrievals have already been included, precipitation data are not an essential indicator of soil moisture and are not utilized as a physically based 'quality impact factor' either (see Text S1 for detailed explanations).

**2.2 Methods for the production of global long-term surface soil moisture data**

Global long-term surface soil moisture data production includes three basic parts: 1) preprocessing: the production of high-quality neural network inputs, including the training target soil moisture, predictor soil moisture products and the quality impact factors (i.e., 9 environmental factors); 2) neural network operation: the training of localized neural networks (i.e., the rules for soil moisture prediction are separately trained in different 1°×1° zones) followed by surface soil moisture simulation based on the localized neural networks; and 3) postprocessing: the correction of potential errors or deficiencies in the soil

moisture simulation outputs.

The temporal span of the primary training target SMAP does not overlap with that of TMI, FY-3B, WindSat or AMSR-E (see Figure 1), while most microwave soil moisture products are not available from the beginning year 2003 (e.g., AMSR2 data are only available since July 2012). Therefore, to fully utilize the 10 predictor surface soil moisture products retrieved from 7 different microwave sensors and form a temporally continuous soil moisture dataset covering 2003~2018, several iterative

rounds of simulations are performed. Here, 'iterative' means that the simulated soil moisture data in a round were also converted to part of the training targets of the next round's neural network (hereinafter the 'secondary training targets'), thus extending the potential temporal span of the target soil moisture data. Accordingly, the postprocessing steps which are intended to transform the simulation outputs to reliable secondary training targets can be seen as preprocessing steps as well. The basic flow of this process is shown in Figure 2.

**2.2.1 Neural network design (1): localized neural networks**

In this study, instead of a universal network, we devised localized neural networks. The data within each individual zone are used to train a zonal neural network (hereinafter a subnetwork), which is used for soil moisture simulation at that zone. By comparison, localized neural networks help improve the training efficiency; however, a smaller zonal size does not indicate a better simulation accuracy. We noticed that over arid regions, the surface soil moisture values retrieved by the LPRM algorithm

(AMSR2/TMI/WindSat/AMSR-E-LPRM-X) can be obviously different on the two sides of each edge of 1°×1° sized squares, which was probably attributed to the spatial distribution of key parameters (i.e., some parameters are at 1° resolution). This

finding suggests that subnetworks should be built at the 1°×1° scale. Therefore, we divided the global extent except the polar areas (80°N~60°S) into 140×360 zones. Here, for a 0.1° pixel during a specific 10-day period, if all the input data (input soil moisture products and quality impact factors) have valid values, one valid data point is provided. Therefore, the maximal

number of valid data points applied to train a subnetwork = 100 × the number of 10-day periods within the training period. The subnetworks with less than 100 valid data points (e.g., those in oceans) were dropped, leaving usually >15,000 zonal subnetworks included in an independent neural network. The training was performed in MATLAB 2016a-using the Neural network fitting toolbox, and the number of nodes in the hidden layer (between the input and output layers (Stinchcombe and White, 1989)) of each subnetwork was 7. We chose the gradient descent backpropagation algorithm as the training function.

**2.2.2 Preprocessing and postprocessing steps**

After standardization of the original soil moisture data, to improve the neural network training efficiency, the potential salt and pepper noises are removed. For each map (a specific 10-day period), within each 1°×1° zone, the soil moisture values are filtered to the level of three standard deviations relative to the mean in that zone (the principle is that 99.87% of the data appear within this range for a normal distribution (Howell et al., 1998). Also note that the filter applied spatially rather than

temporally to detect and delete the extreme values, which are usually noise in mountain areas. Therefore, the extreme climatic events will not be mistakenly removed). This preprocessing step is thus called '3σ denoising'.

After neural network operation, boundary fuzzification is first applied, and it is a step in both preprocessing and postprocessing. Because the localized 1°×1° network is applied instead of the global network, the boundary between nearby zones may be too obvious over some areas. To blur the boundary, a simple algorithm is applied as shown in Figure S1. The soil moisture data

with fuzzified boundaries are transformed into both the final product and the next round's training target. To produce the final product, two postprocessing steps are essential: filling of missing values and data masking. Because '3σ denoising' deleted suspicious soil moisture retrievals, the simulation outputs also contain few missing values, which can be simply filled by sequentially searching and averaging nearby valid values (Chen et al., 2019). While the snow/ice mask of the ASCAT-SWI product can be transferred to the simulation output, the potential snow or ice cover before 2007 should be identified. For a

pixel in a specific ten-day period, if ice cover is reported by ASCAT-SWI in most years, it should also be covered by snow/ice unless the thaw state is observed in the MEaSUREs Global Record of Daily Landscape Freeze/Thaw Status V4 dataset. The simulated soil moisture in the rainforests identified in the 'ASCAT Constant' is retained but not recommended due to the high uncertainty. On the other hand, to avoid error propagation with the training times by ensuring a high-quality training target for the next round's simulation, we remove all suspicious values for every simulated result. This preprocessing step is performed by first obtaining the maximal and minimum values of SMAP_E soil moisture in each pixel. If the simulated value is out of the range of the SMAP data during 2015~2018, then the value is considered suspicious and not used as a training target. Subsequently, '3σ denoising' is performed again before the simulated soil moisture becomes secondary training target, which are referred to as SIM-1T, SIM-2T, and so on ('SIM' stands for the simulated soil moisture, the number after the hyphen indicates the round of simulation, and 'T' means it is applied as training target; the temporal spans of SIM-XT and SIM-X are the same, as shown in Figure 1).

### 2.2.3 Neural network design (2)- five rounds of simulations

The 11 microwave soil moisture data products with different temporal spans are incorporated, and utilized as fully as possible through up to 5 rounds of neural network-based simulations, with at least four different soil moisture products retrieved from three different sensors applied as predictors in each round (see Figure 1). Although increasing the sources of soil moisture data inputs can improve the training efficiency, the spatial coverage of the simulation output is sacrificed because the overlapping area decreases as the number of soil moisture products increases. After all, most products have missing data in specific regions (e.g., mountains, wetlands and urban settlements), and some sensors are even unable to produce data at the global scale (e.g., TMI is limited to [N40°, S40°]; SMOS have many missing values in Eurasia). To resolve this dilemma, we classified all 0.1° pixels according to the predictor soil moisture products that have a valid value over a 10-day period (for example, if there are four predictor soil moisture datasets in one round, there should be 4+6+4+1=15 combinations. Here, '1' indicates the condition that all four products have a valid value in the 0.1° pixel, and there are '6' conditions when only two of the four predictors have valid value in the pixel). However, to avoid soil moisture simulation under snow or ice cover (see

section 2.2.2), not all combinations are considered. Then, an independent neural network corresponding to each selected combination is trained. For data simulations in a 0.1° pixel, the most preferable independent neural network is expected to be

trained using all the available soil moisture data sources in that pixel (i.e., if valid values are provided by three soil moisture products, then the preferable neural network is the one trained using those three predictors). However, in the 1° zone in which the 0.1° pixel is located, the subnetwork belonging to that preferable independent neural network may not exist due to limited valid data points (see section 2.2.1). Then, an alternative subnetwork driven by the combination of fewer soil moisture data inputs should be applied instead. Hence, we should determine the neural network collocation that is the best choice for every

pixel. Apart from applicability, the relative priority order of different neural networks was obtained by comprehensively considering the number and quality of input soil moisture products, the variety of sensors, the quantity of training samples indicated by the number of 10-day periods, and the relative quality of the training targets (the training target quality declines monotonically: SMAP>SIM-1T>SIM-2T>SIM-3T>SIM-4T). Occasionally, the two most likely priority orders are given and the simulation results of the corresponding two substeps are integrated later. Specifically, when the LAI data source changes,

the division of a single round into two substeps is also essential. Based on these principles, five rounds of neural networks are designed as follows, with 8 substeps containing a total of 67 independent neural networks. The training period for each neural network and the simulation period for each substep are shown in Figure 1 (below the timeline), and the details are as follows. For the first round's neural network (labeled as NN1), the potential training period is 2015D10~2018 ('D' is the ordinal of the 10-day period; therefore, '2015D10' represents the period from April 1st to April 10th in 2015) because SMAP soil moisture

data that cover only that period are applied as the training target, while ASCAT-SWI10 (abbreviated as ASCAT), SMOS-IC (SMOS), AMSR2-JAXA and AMSR2-LPRM-X (AMSR2-LPRM) are the four soil moisture products used as predictors (details are in Tables S1~S2). Because all four predictors have data since 2012D19, the potential soil moisture simulation period is 2012D19~2018, which is further divided into two parts: 2014~2018 (substep1), for which the PROBA-V LAI data that begins in 2014 are applied; and 2012D19~2013 (substep2), for which GLASS LAI data are used (note: because GLASS

LAI covers from the beginning of our study period until 2017, the training period for substep 2 is 2015D10~2017). The simulation results of the two substeps (SIM-1-1 and SIM-1-2) are combined as SIM-1, which is then transformed into a

secondary training target, denoted as SIM-1T. In the second round of simulation, the training target can be either SMAP or SIM-1T, while the soil moisture input data are ASCAT, SMOS, TMI-LPRM-X (TMI) and FY-3B-NSMC (FY). The simulation output SIM-2, covers the period of 2011D20~2012D18, which is constrained by the common period of the four predictors

(Tables S3~S4). SIM-2 was also converted into the training target SIM-2T. In the third round of neural network operation, the simulation period is 2010D16~2011D19. SMAP, SIM-1T and SIM-2T are combined and used as the training targets (the training periods are within the range of 2011D20~2017D36), while the predictor soil moisture data are ASCAT, SMOS, TMI and WindSat-LPRM-X (WINDSAT). There are two substeps in round 3 that are distinguished by whether the priority order of the neural networks is determined mainly based on the training sample quantity and the training target quality (SIM-3-1),

or by first considering the number of predictor soil moisture products (SIM-3-2, Tables S5~S8). Because these two methods emphasize different aspects of neural network quality, in some pixels, SIM-3-1 will be advantageous, whereas in others, SIM-3-2 could be better. Hence, an algorithm is devised to combine the advantages of both simulations (SIM-3), which is described in Table S9. Next, the 4th round is for the simulations from 2007D01 to 2010D15. SIM-2T and SIM-3T are combined to be the training target, and ASCAT, WINDSAT, TMI, AMSR-E-JAXA, AMSR-E-LPRM-X (AMSR-E-LPRM) and AMSR-E-

NSIDC are all applied as predictors (LAI data now come from SPOT-VGT). Two substeps are also considered. In the first substep, neural networks are sorted by focusing on the number of soil moisture inputs and the sensors they are derived from, while the training sample size and training target quality are prioritized to create an alternative estimate (Tables S10~S13). Afterwards, SIM-4 is obtained by reasonably integrating these two results. In the final round, the soil moisture simulation is extended to as early as 2003. SIM-2T, SIM-3T and SIM-4T together are the training targets, while the predictor soil moisture

data entering the neural networks consist of WINDSAT, TMI, AMSR-E-JAXA, AMSR-E-LPRM and AMSR-E-NSIDC (Tables S14~S15).

## 2.3 Methods for the validation of surface soil moisture products

For the evaluation of global-scale soil moisture data, we adopted the International Soil Moisture Network (ISMN) dataset (Dorigo et al., 2011; Dorigo et al., 2013). Because the training target SMAP represents the soil moisture within 0~5 cm, the

simulated soil moisture is intended for that surface soil layer as well. Accordingly, the measurements used for validation are limited to ≤ 5 cm in depth. Records outside of the RSSSM data period (2003~2018), such as those from Russian networks, are ignored as well. The quality flags of ISMN (Dorigo et al., 2013) are also checked to retain only the 'good quality' data. After data screening and processing (e.g., the pixels with average annual maximal water area fractions greater than 5% are excluded, please see Text S2), more than 100,000 ~10-day-averaged soil moisture records obtained from 728 stations of 29 networks are applied for validation of the soil moisture products. The detailed information of these stations and the periods of the data used are listed in Table S16, while the spatial distribution of these stations is shown in Figure 3. The major climate types of the sites are determined from the Köppen-Geiger climate classification map (see Table 2 for the description (Kottek et al., 2006)). Next, we further aggregated the site-scale 10-day averaged soil moisture data to a 0.1° pixel-scale by averaging all the measurements made by different stations or different sensors within the pixel (Gruber et al., 2020). Specifically, if soil moisture is not simulated due to snow or ice cover, then the corresponding measurement is useless. This process resulted in a final collection of ~40,000 pixel-scale 10-day period soil moisture records within the validation dataset.

The soil moisture datasets to be evaluated include the RSSSM product in this study (Remote Sensing Surface Soil Moisture, covering 2003~2018); SMAP_E (the primary training target, covering April 2015~2018); the longest record of satellite-based soil moisture: ASCAT-SWI (converted to volumetric fraction; data period is 2007~2018); the reanalysis-based soil moisture: GLDAS Noah V2.1 and ERA5-Land (data were resampled, 10-day averaged and then evaluated during 2003~2018); as well as the soil moisture datasets developed by combining both satellite observations and model simulations: CCI v4.5 and GLEAM v3.3 (for v3.3a, the radiation and air temperature forcing data come from ERA5, whereas for v3.3b, all meteorological data are satellite-based, yet the data after September 2018 are not available). The overall performance of any soil moisture product is first evaluated using all of the validation datasets, with Pearson R-square ($R^2$) and RMSE values (unit: $m^3\ m^{-3}$) adopted as the main indicators. The next step is temporal pattern validation. For pixels with enough (>20) 10-day averaged in situ records, we compare the estimated soil moisture during all periods against the corresponding measurements, with the calculated Pearson correlation coefficient ($r$) and RMSE. Several supplementary indexes are also added, including bias, unbiased RMSE (ubRMSE) and the correlation coefficient between the anomalies (anomalies $r$, abbreviated here as 'A.R'; A.R can better

indicate the simulation accuracy of interannual variations; soil moisture anomalies are calculated by Eq. 1). Next, we compare
the means and medians of the above evaluation indexes for different soil moisture products and test whether the differences
are significant. Moreover, the relative performances of various products in different climatic zones are analyzed. Finally, we
perform spatial pattern validation. In detail, for every 10-day period, we compare all the soil moisture measurements that are
upscaled to 0.1° during that period with the corresponding estimated values. The spatial pattern evaluation indexes include
the correlation coefficient ($r$), RMSE, bias and ubRMSE values (Eq. 2). The relative superiority of all products during different
10-day periods in a year and the changes in data coverage as well as data quality with time are also investigated.

$$\overline{SSM(k)} = \frac{\sum_{y=1}^{ny} SSM(y,k)}{ny} \quad (ny \geq 3) \ ; \ SSM \ is \ either \ estimated \ or \ measured$$

$SSM$: surface soil moisture; $k$: the ordinal of 10 day period in a year; $y$: a year with measured SSM in $k^{th}$ 10 day period; $ny$: number of those years

$$SSM_{anom}(y,k) = SSM(y,k) - \overline{SSM(t)}$$

$SSM_{anom}(y,t)$: the anomalies of surface soil moisture during the $t^{th}$ 10 day period in year $y$    (Eq. 1)


$$\overline{SSM_{est}} = \frac{\sum_{i=1}^{ng} SSM_{est,i}}{ng} \ ; \quad \overline{SSM_{act}} = \frac{\sum_{i=1}^{ng} SSM_{act,i}}{ng} \ (ng \geq 20)$$

$i$: a grid with upscaled surface soil moisture measurements during a specific 10 day period; $ng$: the number of those grids in the globe

$$ubRMSE_{spatial} = \sqrt{\sum_{i=1}^{ng} [(SSM_{est,i} - \overline{SSM_{est}}) - (SSM_{est,i} - \overline{SSM_{act}})]^2/ng} \quad (Eq. \ 2)$$

## 2.4 Methods for the intra-annual variation analysis of surface soil moisture

Because the original resolution of SMAP soil moisture is ~0.4° while that of most predictor soil moisture products is 0.25°,
the intra-annual variation analysis of RSSSM is performed at 0.5° resolution. We also exclude high-latitude areas (60°N~90°N)
where the available data are limited due to frequent ice cover. Fourier functions can characterize intra-annual variation well
(Brooks et al., 2012; Hermance et al., 2007). Therefore, for the remaining areas (60°S~60°N), based on a total of 36×16 (years)
=576 data points, we fit the intra-annual cycle of soil moisture using the Fourier function, with the period fixed to 1 year (36
10-day periods). The number of terms is set to 1 unless the intra-annual cycle is obviously asymmetrical and can be much
better characterized by a two-term Fourier function. Subsequently, the highest peak and lowest trough values of surface soil

moisture as well as the corresponding locations in time (the ordinal of 10 days) are exported.

The direct driving factor of the variation in surface soil moisture is precipitation, for which we adopted the GPM (Global Precipitation Measurement) IMERG (Integrated Multi-satellitE Retrievals for GPM) Precipitation V06 Final Run data (Huffman et al., 2019). Apart from a direct correlation analysis, we also explored the relationship between the intra-annual cycles of precipitation and surface soil moisture using Fourier fitting (the derived fitting function is dropped if the adjusted $R^2$ is lower than 0.1), with the peak time difference in each 0.5° grid cell calculated (if both cycles have two peaks, the average locations of the two peaks are calculated). Because RSSSM indicates the average soil moisture condition during every 10-day period, we evaluate the surface soil moisture decline after 20 consecutive days (i.e., two adjacent 10-day periods) without effective precipitation to explore the impact of dry periods on surface soil moisture. Effective precipitation is calculated by precipitation minus canopy interception, which is estimated by the modified Merriam canopy interception model (Kozak et al., 2010; Merriam, 1960). If the total effective precipitation within two consecutive 10-day periods (20 days) is less than a given threshold (initially set to 10 mm), we consider that the soil moisture change in the latter period compared to the previous period is mostly due to surface evaporation and percolation (capillary rise is negligible (McColl et al., 2017)); thus, it should be negative. Hence, for a 0.5° grid cell, if the number of negative values does not meet two times the number of positive values, the precipitation threshold is reduced by 1 mm until that condition is satisfied. This loop is terminated when there are less than 36 available data points in dry periods (the maximal number of data points is 576), and then the grid cell is excluded from the analysis. In desert areas, the random noise of the surface soil moisture product can hide the signal of moisture changes, while in wet areas (e.g., rainforests), 20 days without effective precipitation seldom occurs, thus leading to no results over most areas. In the remaining areas, the intra-annual variation in the surface soil moisture loss during dry days can be fitted by the Fourier function as well, which is then analyzed using the above methods.

## 3 Results

### 3.1 Neural network training efficiency: a comparison between RSSSM and SMAP

To examine the training and simulation efficiency of the neural network, we compare the neural network simulated surface

soil moisture (RSSSM) with the training target SMAP (note: these two datasets are not completely independent because SMAP data are used as the training target while RSSSM data are the simulation results) during April 2015~2018. The $R^2$ reaches up to 0.95, while the RMSE is 0.031 $m^3/m^3$ (Figure 4a). If only the pixels with measured data are considered, the consistency between RSSSM and SMAP becomes even stronger, with an $R^2$ of 0.97 and an RMSE of 0.016 (Figure 4b). When validated against site measurements, the $R^2$ and RMSE values are 0.46 and 0.083, respectively, for both RSSSM and SMAP (Figure 4c and 4d). All these findings justify the high training and prediction efficiency of the neural network set designed in this study.

According to Table 3, RSSSM is just slightly lower than SMAP in terms of temporal accuracy (the differences in the five indicators, $r$, RMSE, bias, ubRMSE and A.R, are all nonsignificant). Figure 5 indicates generally the same level of temporal accuracy for RSSSM and SMAP under all climates. RSSSM cannot adequately characterize the temporal variation in soil moisture in the 'Dfc' (snow climate, fully humid, see Table 2) region because the training target, SMAP, does not have a high temporal accuracy in this area, probably due to frequent freezing and melting processes.

Next, we compare the spatial accuracy of RSSSM and SMAP. The spatial correlation of RSSSM is somewhat reduced compared to the training target, while the RMSE is slightly increased (Table 4), indicating a subtle loss of detailed spatial information through neural network operation. Because ISMN stations are mostly located in the middle to high latitudes of the Northern Hemisphere, Figure 6 shows that: 1) the accuracy of RSSSM is highest in summers (growing seasons) and lowest in winters, which is inherited from its origin (SMAP) probably due to the impact of freezing on soil moisture retrieval; and 2) RSSSM has a similar spatial accuracy as SMAP in most periods except for May to June and November to December.

### 3.2 Accuracy comparison between RSSSM and popular global long-term soil moisture products

### 3.2.1 Data quality comparison between RSSSM and the satellite-derived product

The satellite-derived global surface soil moisture product ASCAT-SWI now covers 12 years, 2007~2018. During that period, the overall $R^2$ and RMSE for RSSSM are 0.44 and 0.086, respectively (Figure 7), which appear to be much better than those for ASCAT-SWI ($R^2$=0.33, RMSE=0.100). If the data period of SMAP (2015D10~2018) is excluded, the overall $R^2$ and RMSE for RSSSM are 0.43 and 0.087, respectively, which are still better than those for ASCAT-SWI ($R^2$=0.33, RMSE=0.1). However,

RSSSM overestimates soil moisture when low moisture occurs, which is a problem inherited from the SMAP product (Figure 4), and is a bit nonlinearly correlated with the measured values (Figure 7a).

According to the temporal validation results (Table 3), the evaluation indexes including $r$, RMSE, bias and ubRMSE, are all significantly ($p<0.05$) better for RSSSM than ASCAT-SWI (anomalies $r$ for RSSSM is also higher, but not significant). The temporal accuracy of RSSSM appears to be obviously higher in all climatic zones except for polar areas (Dsb, Dwc and ET). Specifically, in arid areas (BWh and BWk), the temporal correlation coefficients for ASCAT-SWI are much lower and even negative (Figure 8). This problem is known and might be related to the different scattering mechanisms in dry soils invalidating the assumptions of change detection method (Al-Yaari et al., 2014).

The spatial accuracy of RSSSM is found significantly higher than that of ASCAT-SWI when any evaluation index is considered (Table 4). Moreover, the results show that RSSSM is generally superior to ASCAT-SWI throughout the year, especially during the growing seasons (Figure 9).

### 3.2.2 Data quality comparison between RSSSM and land surface model products

First, the overall accuracies of RSSSM and GLDAS Noah V2.1 surface soil moisture data from 2003 to 2018 are compared. While RSSSM is nonlinearly correlated with measured soil moisture, the relationship between GLDAS soil moisture and the measurements appears to be slightly more nonlinear, resulting in a smaller $R^2$ of 0.39 and higher RMSE of 0.097 for GLDAS product compared to RSSSM ($R^2$: 0.42; RMSE: 0.087, see Figure 10). When excluding the SMAP (training target) data period, the $R^2$ and RMSE for RSSSM are 0.41 and 0.089, respectively, which are also superior to those for GLDAS ($R^2$: 0.37; RMSE: 0.099).

The higher temporal accuracy of RSSSM than GLDAS can be justified by comparing the indicators, including $r$, RMSE and ubRMSE (Table 3). The advantage of RSSSM over GLDAS could be identified in almost all climatic regions, especially the cold areas such as BWk, Dfa, Dfc, Dwc and ET (Figure 11), perhaps because the soil thawing and freezing processes are not simulated well. The spatial accuracy of RSSSM, indicated by $r$, RMSE, bias and ubRMSE, is found to be significantly higher than GLDAS as well (Table 4). The spatial correlation of RSSSM is somewhat higher than that of GLDAS during March to

May and September to November, and the spatial RMSE is lower all year round except in January and February (Figure 12). ERA5-Land is a newly published reanalysis-based model product with 0.1° resolution. The overall quality validation (Figure S2) reveals a frequent overestimation of soil moisture by ERA5-Land as well as a nonlinear relationship between the predicted and measured values. Accordingly, although the $R^2$ for ERA5-Land is 0.41, which is only slightly lower than that of RSSSM (0.42), the RMSE for ERA5-Land is 0.123, much higher than that for RSSSM (0.087) during their common period. Without

considering the SMAP period, the conditions are the same (the $R^2$ for RSSSM and ERA5-Land are 0.41 and 0.38; the RMSE values for these two products are 0.089 and 0.125, respectively). The temporal correlation indicated by $r$ and A.R is somewhat higher for ERA5-Land in general (Table 3), but in most cold areas (Dfa, Dwc and ET), the opposite condition occurs (Figure S3a, S3d). The temporal ubRMSE values for RSSSM and ERA5-Land do not differ significantly, but RSSSM usually performs better in relatively arid places (Figure S3c). While the relative temporal accuracies of RSSSM and ERA5-Land are unclear,

the spatial pattern of RSSSM is more accurate than that of ERA5-Land considering the significantly better spatial correlation, RMSE, bias and ubRMSE (Table 4). The considerable advantage of RSSSM over ERA5-Land exists throughout the year, especially during the growing seasons from March to November (Figure S4).

### 3.2.3 Data quality comparison between RSSSM and the soil moisture products derived from both satellite data and model simulations

CCI is a typical surface soil moisture dataset developed by combining satellite observations and model simulations. However, validation against measurements indicates that the CCI product is not of very good quality because the overall $R^2$ is only 0.31 with an RMSE value of up to 0.095 (Figure S5, when the SMAP data period is excluded, the $R^2$ and RMSE for CCI are 0.28 and 0.098, compared to 0.41 and 0.089 for RSSSM). The temporal pattern of RSSSM, indicated by $r$ and RMSE, is found to be significantly better than that of CCI (Table 3), and under all climate conditions (Figure S6). Our results indicate that RSSSM

also shows a consistently higher spatial accuracy than the CCI, especially during the growing seasons (Table 4 and Figure S7). Next, we focus on the interannual change in data quality. According to Figure 13a~c, while the correlation coefficient for RSSSM does not vary significantly among different years, the RMSE and ubRMSE values in earlier periods are somewhat higher compared to those after 2012. Although the data quality of RSSSM can hardly be maintained as well, the degradation

degree is much slighter than that of CCI. The comparison of the spatial coverages of the 10-day scale RSSSM and CCI data

(rainforests are excluded) shows that RSSSM covers all land surfaces except for permafrost while the interannual variation in

coverage is also negligible throughout the entire period (the intra-annual cycles of data coverages result from the changes in

frozen areas), which are preferable to the CCI, whose data coverage before 2007 is limited (Figure 13d).

GLEAM products also contain satellite information due to the assimilation of CCI data, although model simulations play a

much more important role. By validation, the overall $R^2$ and RMSE values for the GLEAM v3.3a product (2003~2018) are

0.38 and 0.142, respectively, whereas those for the v3.3b product are 0.36 and 0.13, respectively. Both estimates are

nonlinearly correlated with and generally higher than the measured values (Figure S8). Therefore, with an $R^2$ of 0.42 and

RMSE of 0.087, RSSSM is found to be superior to GLEAM v3.3a/b in general (if the SMAP data period is excluded, RSSSM's

$R^2$ and RMSE values are 0.41 and 0.089, respectively, which are still better than both GLEAM v3.3a ($R^2$: 0.35; RMSE: 0.141)

and GLEAM v3.3a ($R^2$: 0.34; RMSE: 0.128)). The temporal and spatial accuracies of GLEAM products and RSSSM are

compared in Tables 3~4. The advantage of GLEAM is its ability to characterize the temporal variations in soil moisture, with

higher temporal correlation achieved in most climatic regions (Figure S9a and S9d). However, the main potential disadvantage

is the obvious overestimation, which leads to significantly higher RMSE values compared with RSSSM in all regions and all

periods (Figure S9b and Figure S10b). Moreover, the spatial pattern of GLEAM products is less convincing than that of

RSSSM, considering the lower spatial correlation coefficients, especially in spring (March to May) and autumn (September

to November) (Figure S10a). Therefore, the potential advantages of RSSSM can exceed those of GLEAM.

In conclusion, surface soil moisture developed mainly based on land surface models (GLEAM and ERA5-Land) has high

temporal accuracy, but relatively unreliable absolute values and spatial patterns; however, RSSSM shows good performances

in all aspects. Generally, this study indicates that the expected order of data applicability among various global long-term

surface soil moisture products is RSSSM (applicable to all studies)> GLEAM (suitable for temporal variation studies)> ERA5-

Land (applicable to temporal pattern studies)> GLDAS Noah V2.1 (somewhat applicable to all studies)> ASCAT-SWI> CCI.

The training $R^2$ of the previous neural networks designed for global surface soil moisture mapping is 0.45~0.55, while the

temporal $r$ and RMSE values against measurements are 0.52 and 0.084 (Yao et al., 2017), and the overall $R^2$ and RMSE are

0.2 and 0.113 (Yao et al., 2019). In this study, by elaborating the neural network, the training $R^2$ is elevated to 0.95, with improvements in the temporal *r* and RMSE (0.69 and 0.08) as well as the overall $R^2$ and RMSE (0.42 and 0.087) values. In addition, our 10-day period average product is both spatially and temporally continuous over 16 years, has a high spatial resolution, and covers all land except for frozen ground. Hence, our product could be more useful than previous machine learning products.

### 3.3 Spatial and temporal patterns of the calculated surface soil moisture

For the calculated global surface soil moisture, the spatial pattern averaged during 2003~2018 is shown in Figure 14a (the maps for separate months are shown in Figure S11a). The above validation results show that except for RSSSM, GLDAS has the highest spatial accuracy, so the spatial pattern of GLDAS surface soil moisture is also shown below (Figure S11b). By comparison, the spatial patterns of RSSSM and GLDAS are similar, but some differences also exist (see the regions circled in red). Obviously, RSSSM has a higher spatial heterogeneity and probably more reflections on wetlands and irrigated fields (e.g., the Hetao Irrigation Area in China), whereas GLDAS appears patchy in arid areas. The latitudinal pattern comparison in Figure S12a also implies that RSSSM contains more detailed spatial information.

For the interannual variation, because the GLEAM v3.3a product is proven to have the best accuracy in characterizing the temporal anomalies of soil moisture and covers the whole world, this product is selected as the reference to justify our calculation. According to Figure S12b, both GLEAM and RSSSM support a significant rising trend in global mean surface soil moisture during 2003~2018, while the average rates are both approximately 0.03 m$^3$ m$^{-3}$ yr$^{-1}$ (Figure S12b). The spatial patterns of the interannual trends in RSSSM and GLEAM are shown in Figure 14c~d, and they are generally consistent. Soil moisture gains are found over the border between the USA and Canada, as well as over Paraguay, Kazakhstan, Northeastern and Southern China (the regions circled in blue), while soil moisture declines are observed in North Asia and eastern Brazil (the regions with red circles). The main discrepancy between the soil moisture trends predicted by the two products lies in Central Africa, the Arabian Peninsula and northwestern Canada.

Because the validation against measurements proves that the intra-annual soil moisture variation in the 'Dfc' climate region

cannot be captured by SMAP or RSSSM, the acquired intra-annual analysis results in this region are not considered. Over low-latitude areas (30°S~30°N), surface soil moisture peaks in summers (seasons are opposite in the Northern and Southern hemispheres); however, in midlatitude areas (30°S~60°S; 30°N~60°N) except for eastern Asia (i.e., east of the Yenisei River), the soil moisture is high in winters (nongrowing seasons) and low in summers (Figure 15a and Figure S13a). The intra-annual

range of surface soil moisture is largest in the tropical monsoon climate regions, including the African savannas, the Orinoco Plain, the Ganges plain and the plain areas in the Indochina Peninsula, as well as some seasonal frozen areas, whereas it is lowest in arid places (Figure 15b; Figure S13b~c). Precipitation is a direct driver of surface soil moisture changes (Figure S14a~b), and the intra-annual cycle of soil moisture often strictly follows that of precipitation as long as it exists (Figure 15c and Figure S14c). Considering that at low latitudes, precipitation is often highest in summer, whereas in the westerlies, rainfall

is even among different seasons (eastern Asia is an exception probably due to the monsoon climate and topographic conditions) yet much higher evapotranspiration occurs in summer, the global intra-annual patterns of soil moisture can be explained. The peak time difference between surface soil moisture and precipitation is approximately one 10-day period, or six days on average at global scale (Figure 15d), which is expected to be related to the 'time lag' effect. On dry days, the fastest surface soil moisture decline is expected in summers when evapotranspiration is high. However, this study reveals that at midlatitudes,

the opposite condition occurs: the surface soil moisture loss without rain is lowest in summer (Figure 15e and Figure S15a). Further analysis identified a positive correlation between surface soil moisture and its rate of decline, with $r>0.8$ over 85% of the area (Figure S15b~c), indicating that because soil moisture in the westerlies is often high in winters, the available surface soil water for evaporation and percolation loss is limited in summer, and plants tend to utilize water in deeper soil layers. When droughts occur during a random period, the mean surface soil moisture decline is highest in the tropical monsoon

climate regions (Figure 15f). Therefore, if sufficient water during rainy seasons is lacking there, then significant water loss (Figure S15d) may destroy the local ecosystem.

## 4 Discussion and conclusions

### 4.1 Contributions of microwave observations and environmental characteristics to the neural network prediction

In this study, we developed an improved global long-term remote sensing-based surface soil moisture dataset, named RSSSM. The key algorithm calibrates and fuses various sources of microwave surface soil moisture products through multiple neural networks. Several environmental factors are also chosen as ancillary neural network inputs because they are quality impact factors of microwave soil moisture retrievals, or also director indicators of surface soil moisture. To explore the relative roles of soil moisture data retrieved from microwave observations and the environmental characteristics, we performed contribution tests on all the input features at the global scale (for each predictor, we added a random error that is controlled within the standard deviation of the predictor. Then the increased mean squared error (MSE) in neural network training can be used to determine the relative contribution of that variable). Taking the first independent neural network (NN1-1-1, a primary NN) as an example, the results (Figure 16) indicate that SMOS soil moisture plays the dominant role in the neural network training (55.5%), while the four predictor soil moisture products explained 62.7% in total. The remaining 37.3% of the training efficiency could be attributed to the environmental characteristics, among which the water fraction accounts for the most (13.4%) since it is both a quality impact factor and a direct indicator of soil moisture. The tree cover fraction is an important neural network input as well and reduces the MSE by 7.8%, which is probably due to the strong impact of forest cover on microwave soil moisture retrievals.

### 4.2 Requirement of further validations

Our product is generally more comparable to the in-situ measurements at ISMN stations than the existing global long-term surface soil moisture datasets in general, when all indicators on both spatial and temporal accuracy are considered. However, we can neither conclude that our product is superior to the existing products, nor determine the performance of our product at the global scale. This is mainly because the ISMN measurements are unevenly distributed globally (Figure 3) and incompatible at a spatial scale with the scales of passive microwave observations and land surface modeling (0.1°~0.25°). We validated the soil moisture products against the ISMN's point-scale data just because only such in situ measurements are currently available,

and the ISMN dataset (Dorigo et al., 2011; Dorigo et al., 2013) is the most frequently used in the assessments of large-scale soil moisture data (Al-Yaari et al., 2019; Albergel et al., 2012; Dorigo et al., 2015; Fernandez-Moran et al., 2017b; Gao et al., 2020; Karthikeyan et al., 2017b; Kerr et al., 2016; Kim et al., 2015b; Kolassa et al., 2018; Lievens et al., 2017; Zhang et al., 2019). In this study, to alleviate the impact of spatial scale differences on the evaluation, dense networks are more utilized (19 out of 29 networks, see Text S2 for details) that contain multiple stations within the same 0.1° pixel. The pixels with

nonnegligible water area are also excluded in case of high spatial variability in surface soil moisture. In addition, more than 90% of the selected stations are located in relatively flat areas with a topographic complexity less than 10%. The Cosmic-Ray Neutron Sensing method (CRNS) can provide soil moisture estimates at a scale of hundreds of meters in diameter (Andreasen et al., 2017). Hence, the in situ networks generated using this method, e.g., COSMOS, are more suitable for the validation of satellite-based or modeled coarse resolution soil moisture products. We hope that additional records obtained from cosmic-

ray neutron stations become available in the future so that our product may be better evaluated.

### 4.3 Approaches towards more accurate soil moisture predictions

By referring to the ISMN measurements, the accuracy ($R^2$=0.42; RMSE=0.087) of RSSSM requires further improvement. The target RMSE for surface soil moisture set by GCOS is 0.04 m$^3$ m$^{-3}$, indicating the need to further improve the global soil moisture data quality.

Fortunately, this study provides a novel approach that has the potential to lead to increasingly better soil moisture products in future. The RMSE and ubRMSE values in earlier periods are somewhat higher than those after 2012, which is because: 1) five rounds of simulations were performed, with the output converted into the training target of the next round's neural networks, thus leading to a little error propagation as the simulation period extended to the past; and 2) the quality of microwave soil moisture data is generally lower in earlier periods due to the relatively unadvanced microwave sensors with low signal-to-

noise ratio (SNR). However, due to the design of localized networks and the full use of 11 microwave soil moisture products and quality impact factors, etc., high training efficiency is achieved, resulting in limited amplification of noise and high maintenance of valid information during 16 years of simulation. The overall data accuracy of RSSSM is only slightly lower

than that of SMAP, which is the primary training target. Therefore, if microwave sensors with higher SNR or better penetration of the vegetation canopy than SMAP are launched in the future (e.g., the upcoming P-band microwave sensors (Etminan et al., 2020; Ye et al., 2020)), we can develop a temporally continuous soil moisture dataset beginning in 2003 by using the soil moisture or Tb retrieved from the new sensors as the reference. This upcoming product is expected to have even higher accuracy than the SMAP product (we will update the complete RSSSM product then). In that sense, the data fusion algorithm proposed here will be even more meaningful in the future.

Remote sensing may provide more detailed spatial information on surface soil moisture, whereas reanalysis-based models have advantages in characterizing temporal variations, even on a daily scale. Furthermore, root-zone soil moisture, which often plays a more important role in ecosystems, cannot be directly retrieved through microwave remote sensing. Therefore, combining the advantages of satellite observation and model simulation helps to improve the data accuracies of both surface and root-zone soil moisture. To realize a better combination, one possible approach is to use the pixel-specific confidence range and the spatial pattern of RSSSM to constrain the model parameters or add supplementary modules if necessary. In detail, RSSSM can be used as the initial base map of surface soil moisture. Then, after each time of soil moisture simulation in multiple layers (both root-zone and surface), the model efficiency is examined through a spatial correlation test between the simulated surface soil moisture and RSSSM. In addition, whether the simulated values fall within the confidence range (e.g., ±20%) reported by RSSSM should also be tested. Using recurrent adjustments, the model parameters in each pixel can be optimized. For irrigated croplands, if irrigation is not considered in the models, the simulated surface soil moisture will soon fall below the confidence range, and the correlation will also decline regardless of the parameters that are provided. Therefore, a well-designed irrigation module (Chen et al., 2019) should be introduced. Finally, for regions with human-induced land cover changes (e.g., afforestation), optical remote sensing should be applied to better estimate evapotranspiration.

**5 Data availability**

The global surface soil moisture dataset RSSSM, is available at: https://doi.pangaea.de/10.1594/PANGAEA.912597 (Chen, 2020). In the ZIP file, data maps are all provided in Geotiff format, and we also attached a csv table relating the filename and the nominal time period of the file.

**Author contributions**

Yongzhe Chen conducted the research, completed the original draft and revised it. Xiaoming Feng, the corresponding author, and Bojie Fu supervised the research and revised the draft. Bojie Fu administered the project and funded the research. All coauthors reviewed the manuscript and contributed to the writing process.

**Competing interests**

The authors declare that they have no known competing financial interests or personal relationships that could have appeared to influence the work reported in this paper.

**Acknowledgments**

This work was supported by the National Science Foundation of China (41991233, 41722104) and the Chinese Academy of Sciences (QYZDY-SSW-DQC025). We are grateful to all the data contributors who made it possible to complete this work.

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

**Tables**

**Table 1: Abbreviations for the name of satellites, remote sensors and missions.**

| Abbreviation | Full name |
|---|---|
| SMMR | Scanning Multichannel Microwave Radiometer |
| SSM/I | Special Sensor Microwave/Imager |
| TMI | Tropical Rainfall Measuring Mission (TRMM)'s Microwave Imager |
| AMSR-E | Advanced Microwave Scanning Radiometer for the Earth Observing System |
| AMSR2 | Advanced Microwave Scanning Radiometer 2 |
| SMOS | Soil Moisture Ocean Salinity |
| SMAP | Soil Moisture Active Passive |
| ERS | European Remote Sensing- Active Microwave Instrument Wind Scatterometer |
| ASCAT | Advanced Scatterometer |
| MODIS | Moderate-resolution Imaging Spectroradiometer |
| MEaSUREs | Making Earth System Data Records for Use in Research Environments |

**Table 2: Description of the Köppen-Geiger climate classification types at all the selected ISMN stations.**

| Climate_Köppen | General description |
|---|---|
| Aw | Equatorial savannah with dry winter |
| BSk | Steppe climate, cold and arid |
| BWh | Desert climate, hot and arid |
| BWk | Desert climate, cold and arid |
| Cfa | Warm temperate climate, fully humid, hot summer |
| Cfb | Warm temperate climate, fully humid, warm summer |
| Csa | Warm temperate climate with dry, hot summer |
| Csb | Warm temperate climate with dry, warm summer |
| Dfa | Snow climate, fully humid, hot summer |
| Dfb | Snow climate, fully humid, warm summer |
| Dfc | Snow climate, fully humid, cool summer and cold winter |
| Dsb | Snow climate with dry, warm summer |
| Dwc | Snow climate with cool summer and cold, dry winter |
| ET | Tundra climate |

**Table 3: Mean and median values of the five evaluation indexes (correlation coefficient: *r*, RMSE, bias, unbiased RMSE (ubRMSE), and the anomalies *r* (A.R)) on the temporal accuracy of the surface soil moisture simulated in this study (RSSSM) and the other surface soil moisture products, when validated using the ISMN in situ measurements. Note: 1) for the comparison between RSSSM and SMAP_E (SMAP) product, the validation period is from April 2015 to 2018; 2) for the comparison between RSSSM and ASCAT-SWI (ASCAT), the period is 2007~2018; 3) for the comparison between RSSSM and GLDAS Noah v2.1 (GLDAS), ERA5-Land (ERA5-L), CCI or GLEAM v3.3a (GLE-a) surface soil moisture product, the validation period is 2003~2018; 4) for the comparison between RSSSM and GLEAM v3.3b (GLE-b), the validation period is 2003 to September 2018. For each pair of comparisons based on each evaluation index, the product with the better performance and its values are highlighted in bold.**

| Index | *r* | | RMSE | | bias | | ubRMSE | | A.R | |
|---|---|---|---|---|---|---|---|---|---|---|
| Product | RSSSM | **SMAP** | RSSSM | **SMAP** | **RSSSM** | SMAP | RSSSM | SMAP | RSSSM | **SMAP** |
| Mean | 0.756 | **0.762** | 0.075 | **0.074** | **0.015** | 0.016 | 0.043 | 0.043 | 0.700 | **0.707** |
| Median | 0.795 | **0.798** | 0.067 | **0.066** | **0.009** | 0.013 | 0.043 | 0.043 | 0.720 | **0.744** |
| Product | **RSSSM** | ASCAT | **RSSSM** | ASCAT | **RSSSM** | ASCAT | **RSSSM** | ASCAT | **RSSSM** | ASCAT |
| Mean | **0.687** | 0.561 | **0.079** | 0.095 | **0.002** | -0.007 | **0.047** | 0.062 | **0.627** | 0.554 |
| Median | **0.735** | 0.627 | **0.074** | 0.088 | **-0.001** | -0.010 | **0.048** | 0.062 | **0.654** | 0.595 |
| Product | **RSSSM** | GLDAS | **RSSSM** | GLDAS | **RSSSM** | GLDAS | **RSSSM** | GLDAS | **RSSSM** | GLDAS |
| Mean | **0.689** | 0.613 | **0.080** | 0.091 | **0.001** | 0.028 | **0.047** | 0.051 | **0.620** | 0.519 |
| Median | **0.737** | 0.661 | **0.075** | 0.082 | **-0.002** | 0.029 | **0.048** | 0.049 | **0.661** | 0.567 |
| Product | RSSSM | **ERA5-L** | **RSSSM** | ERA5-L | **RSSSM** | ERA5-L | **RSSSM** | ERA5-L | RSSSM | **ERA5-L** |
| Mean | 0.689 | **0.734** | **0.080** | 0.112 | **0.001** | 0.082 | **0.047** | 0.050 | 0.620 | **0.648** |
| Median | 0.737 | **0.758** | **0.075** | 0.094 | **-0.002** | 0.073 | **0.048** | 0.049 | 0.661 | **0.672** |
| Product | **RSSSM** | CCI | **RSSSM** | CCI | **RSSSM** | CCI | RSSSM | CCI | **RSSSM** | CCI |
| Mean | **0.690** | 0.642 | **0.080** | 0.091 | **0.002** | -0.002 | 0.047 | 0.049 | **0.620** | 0.530 |
| Median | **0.735** | 0.666 | **0.074** | 0.080 | **-0.002** | 0.006 | 0.049 | 0.047 | **0.658** | 0.552 |
| Product | RSSSM | **GLE-a** | **RSSSM** | GLE-a | **RSSSM** | GLE-a | RSSSM | GLE-a | RSSSM | **GLE-a** |
| Mean | 0.689 | **0.735** | **0.080** | 0.126 | **0.001** | 0.093 | 0.047 | 0.047 | 0.620 | **0.681** |
| Median | 0.737 | **0.771** | **0.075** | 0.119 | **-0.002** | 0.104 | 0.048 | 0.046 | 0.661 | **0.715** |
| Product | RSSSM | **GLE-b** | **RSSSM** | GLE-b | **RSSSM** | GLE-b | RSSSM | **GLE-b** | RSSSM | **GLE-b** |
| Mean | 0.688 | **0.729** | **0.080** | 0.117 | **0.001** | 0.077 | 0.047 | **0.046** | 0.618 | **0.670** |
| Median | 0.730 | **0.762** | **0.075** | 0.112 | **-0.002** | 0.091 | 0.048 | **0.045** | 0.659 | **0.705** |

**Table 4: Mean and median values of the four evaluation indexes (*r*, RMSE, bias and ubRMSE) on the spatial pattern accuracy of RSSSM and the other global long-term surface soil moisture products (SMAP_E, ASCAT-SWI, GLDAS Noah v2.1, ERA5-Land, CCI, GLEAM v3.3a and GLEAM v3.3b) in every 10-day period. For each pair of comparisons, the evaluation indexes are for the common period of the two products, and the product with better performance is highlighted in bold (the same as Table 3). The abbreviations for the products are also the same as those in Table 3.**

| Index | *r* | | RMSE | | bias | | ubRMSE | |
|---|---|---|---|---|---|---|---|---|
| Product | RSSSM | **SMAP** | RSSSM | SMAP | RSSSM | SMAP | RSSSM | **SMAP** |
| Mean | 0.652 | **0.659** | 0.084 | 0.084 | 0.016 | 0.016 | 0.082 | **0.081** |
| Median | 0.655 | **0.664** | 0.082 | 0.081 | 0.019 | 0.019 | 0.080 | **0.078** |
| Product | **RSSSM** | ASCAT | **RSSSM** | ASCAT | **RSSSM** | ASCAT | **RSSSM** | ASCAT |
| Mean | **0.636** | 0.561 | **0.087** | 0.102 | **0.005** | -0.010 | **0.085** | 0.097 |
| Median | **0.650** | 0.572 | **0.086** | 0.100 | **0.007** | -0.009 | **0.085** | 0.095 |
| Product | **RSSSM** | GLDAS | **RSSSM** | GLDAS | **RSSSM** | GLDAS | RSSSM | GLDAS |
| Mean | **0.617** | 0.593 | **0.090** | 0.097 | **-0.005** | 0.035 | 0.086 | 0.087 |
| Median | **0.643** | 0.630 | **0.089** | 0.096 | **0.001** | 0.041 | 0.086 | 0.086 |
| Product | **RSSSM** | ERA5-L | **RSSSM** | ERA5-L | **RSSSM** | ERA5-L | **RSSSM** | ERA5-L |
| Mean | **0.616** | 0.575 | **0.090** | 0.125 | **-0.005** | 0.077 | **0.086** | 0.095 |
| Median | **0.641** | 0.633 | **0.089** | 0.125 | **0.001** | 0.082 | **0.086** | 0.092 |
| Product | **RSSSM** | CCI | **RSSSM** | CCI | RSSSM | CCI | **RSSSM** | CCI |
| Mean | **0.618** | 0.497 | **0.090** | 0.099 | -0.004 | 0.003 | **0.086** | 0.093 |
| Median | **0.647** | 0.554 | **0.089** | 0.098 | 0.002 | 0.006 | **0.086** | 0.093 |
| Product | **RSSSM** | GLE-a | **RSSSM** | GLE-a | **RSSSM** | GLE-a | **RSSSM** | GLE-a |
| Mean | **0.617** | 0.576 | **0.090** | 0.139 | **-0.005** | 0.105 | **0.086** | 0.089 |
| Median | **0.643** | 0.616 | **0.089** | 0.142 | **0.001** | 0.112 | **0.086** | 0.088 |
| Product | **RSSSM** | GLE-b | **RSSSM** | GLE-b | **RSSSM** | GLE-b | **RSSSM** | GLE-b |
| Mean | **0.616** | 0.560 | **0.090** | 0.128 | **-0.005** | 0.088 | **0.086** | 0.090 |
| Median | **0.643** | 0.613 | **0.089** | 0.130 | **0.001** | 0.094 | **0.086** | 0.089 |

**Figures**

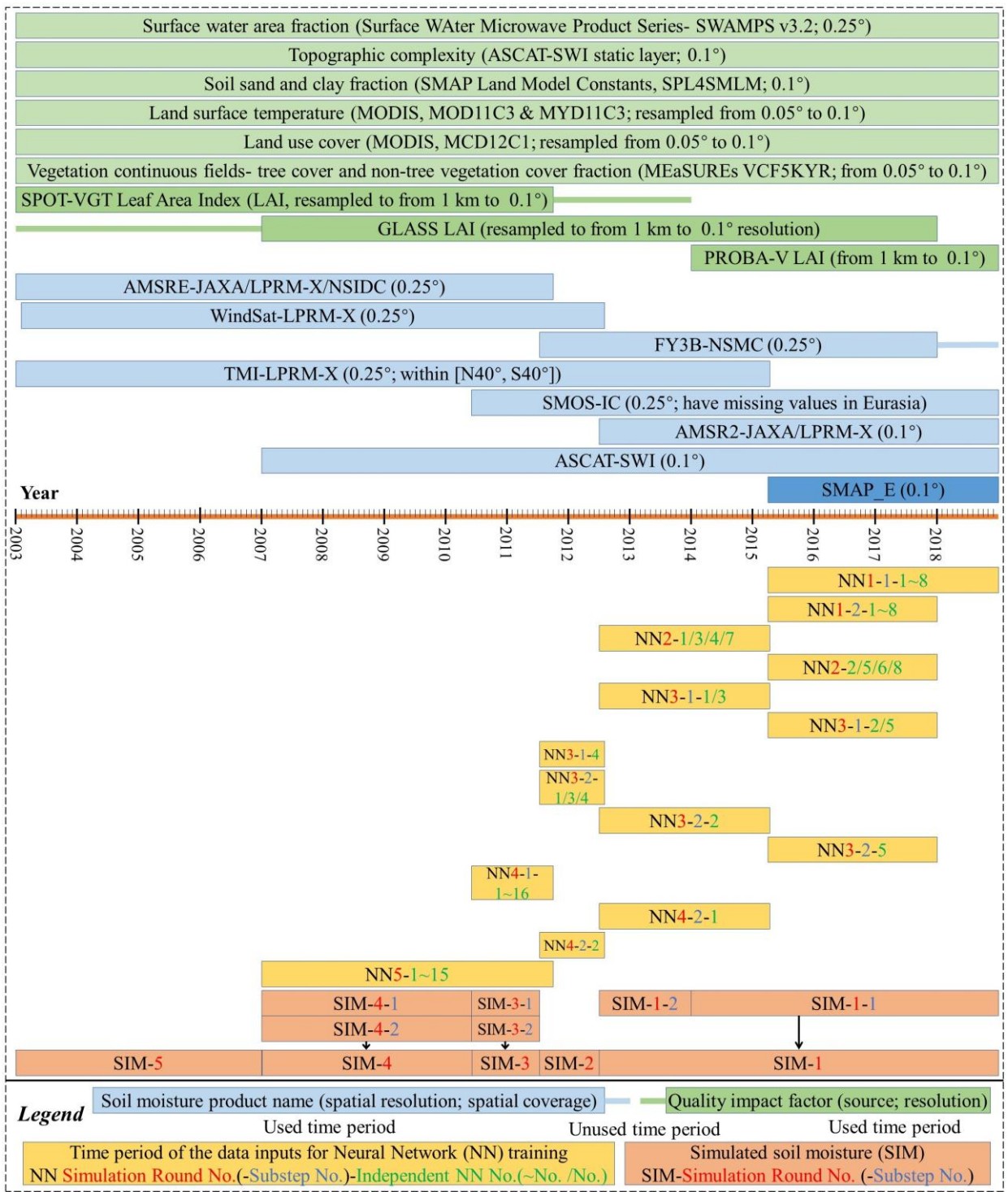

**Figure 1: Overview of the time periods of different soil moisture datasets and the 'quality impact factor' products (e.g., LAI dataset) used in this study (listed above the timeline), as well as the periods of data applied for the training of the 67 independent neural networks and the neural network simulation outputs (i.e., simulated soil moisture) in eight substeps (listed below the timeline).**

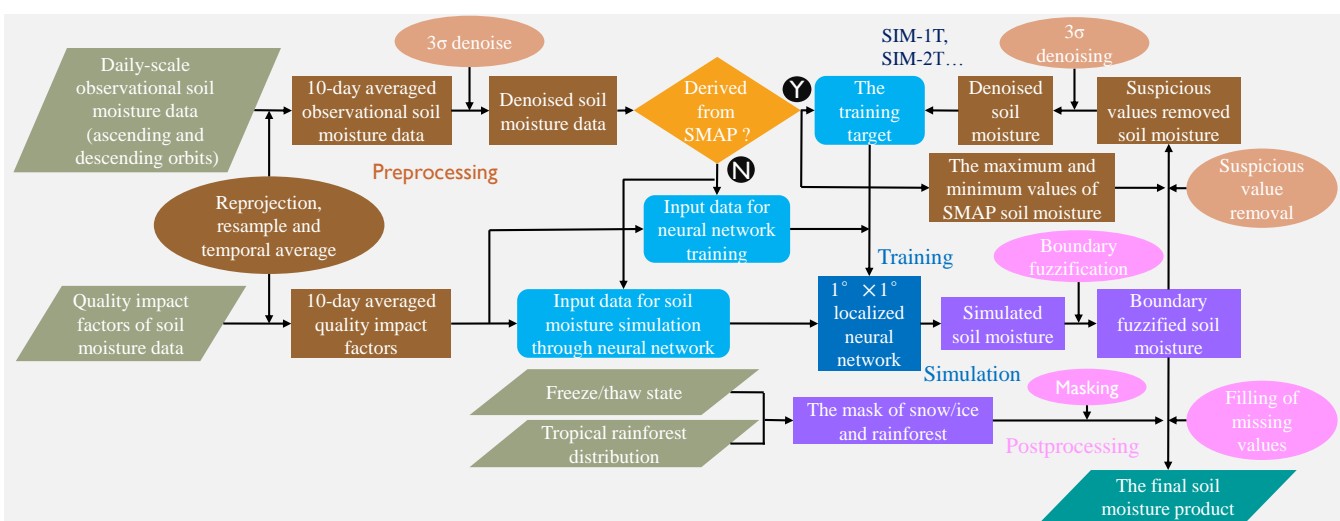

**Figure 2: Flow chart for the production of global surface soil moisture data (RSSSM).**

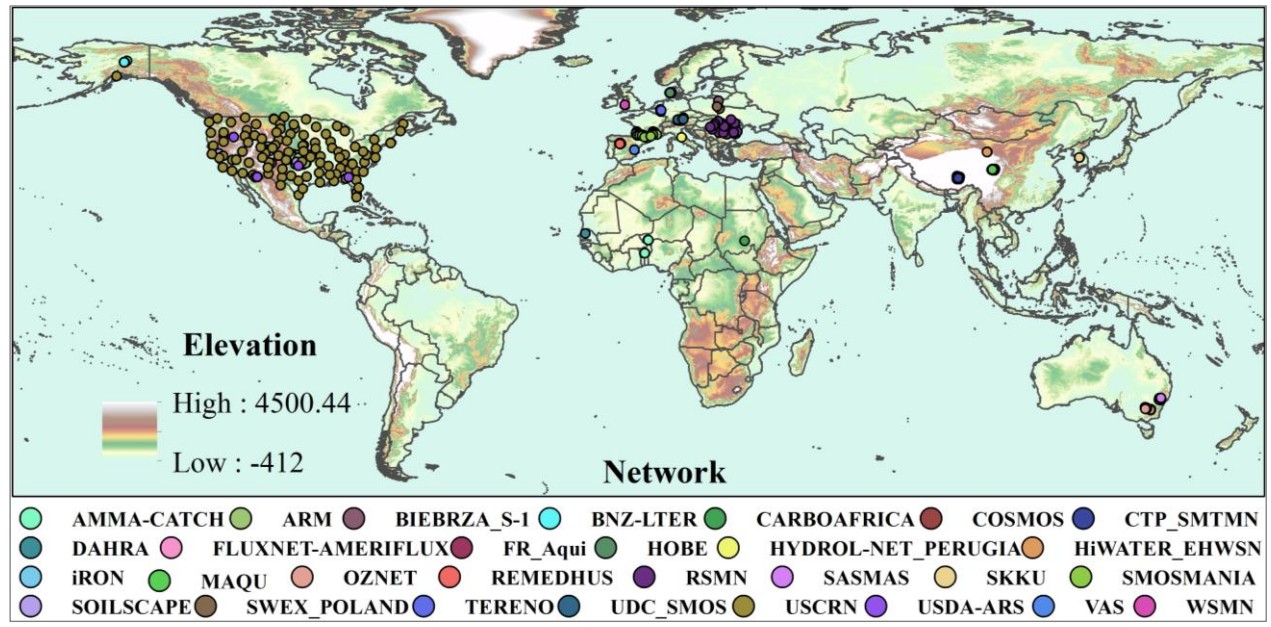

**Figure 3. Global distribution of ISMN networks and stations.**

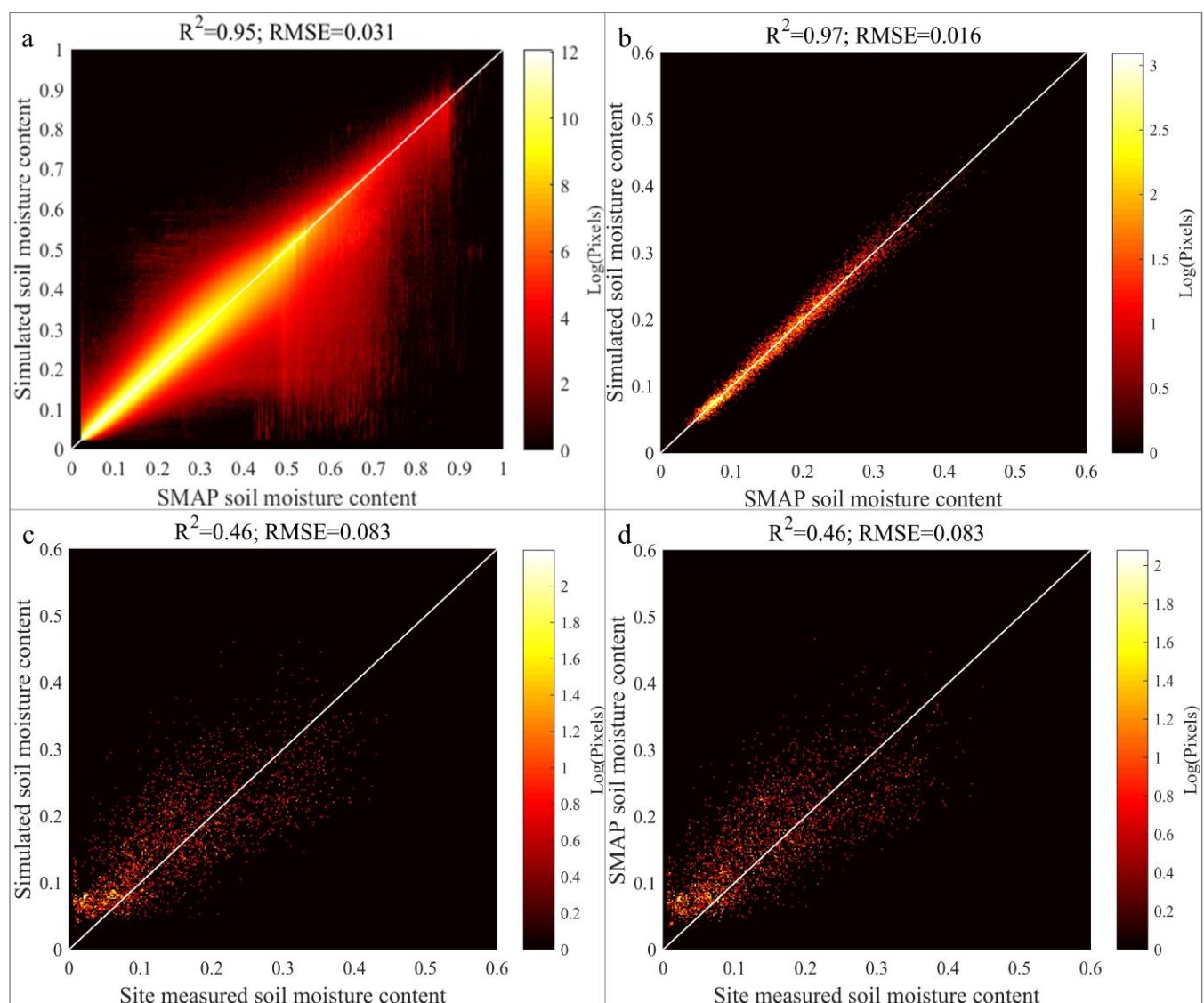

**Figure 4: Comparison between the neural network simulated surface soil moisture (RSSSM) and SMAP data. The scatter plots are between (a) RSSSM and SMAP values at all pixels; b) RSSSM and SMAP values at only the pixels with measurements; (c) RSSSM and the site-measured soil moisture from April 2015 to 2018; and (d) SMAP and the site measurements during April 2015~2018. All plots are represented as the point density on a logarithmic scale, while the units for soil moisture content and RMSE values are m³ m⁻³.**


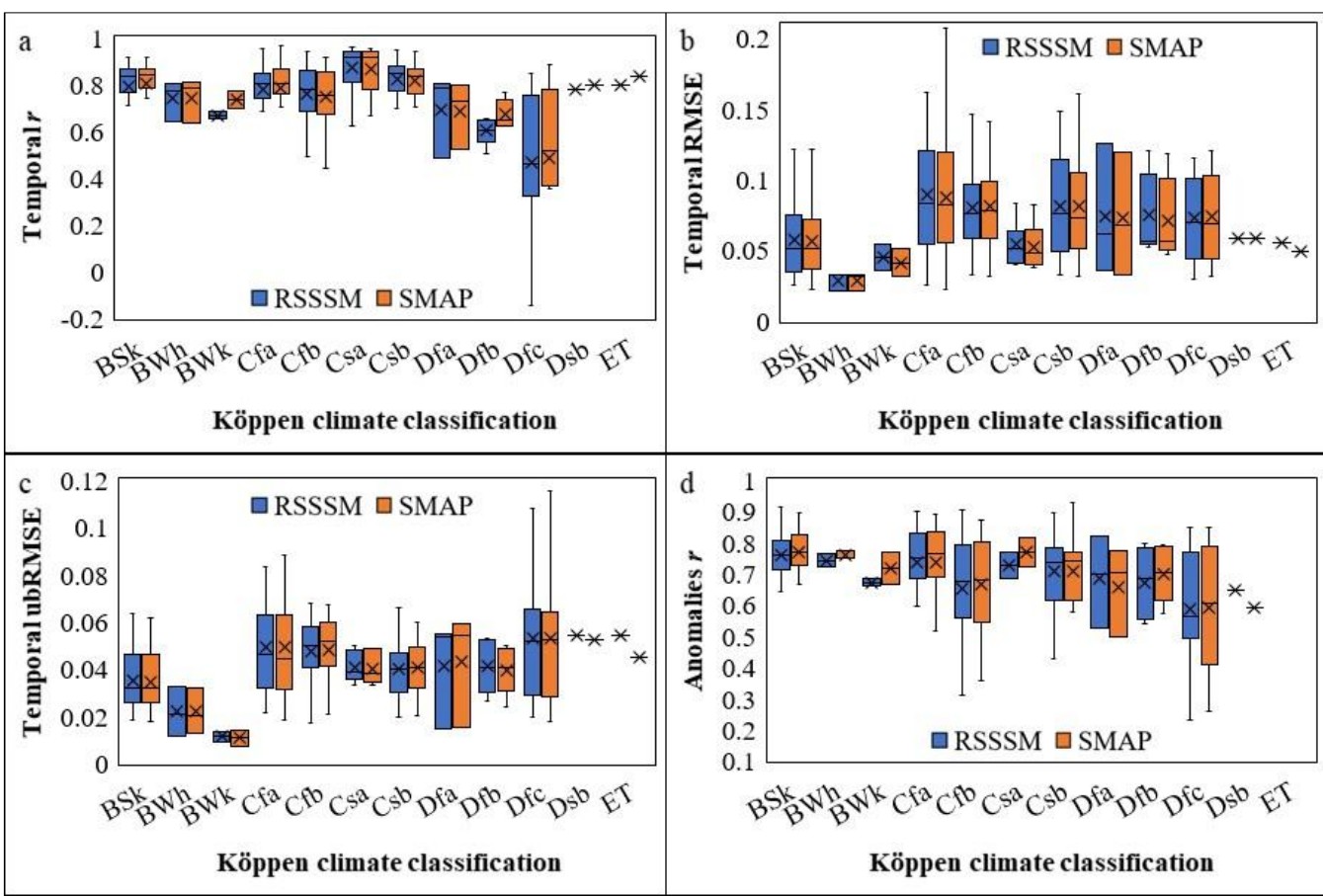

**Figure 5: Comparison of the temporal accuracy between RSSSM and SMAP in regions with different Köppen-Geiger climate types. The four indexes are (a) *r*, (b) RMSE, (c) ubRMSE and (d) Anomalies *r* (A.R). The lengths of the error bars are 1.5 times that of the interquartile range, while the upper and lower boundaries and the central lines of the boxes indicate the 75th, 50th and 25th percentile values, with mean values marked by '×' (the forms of all the following boxplots are the same).**

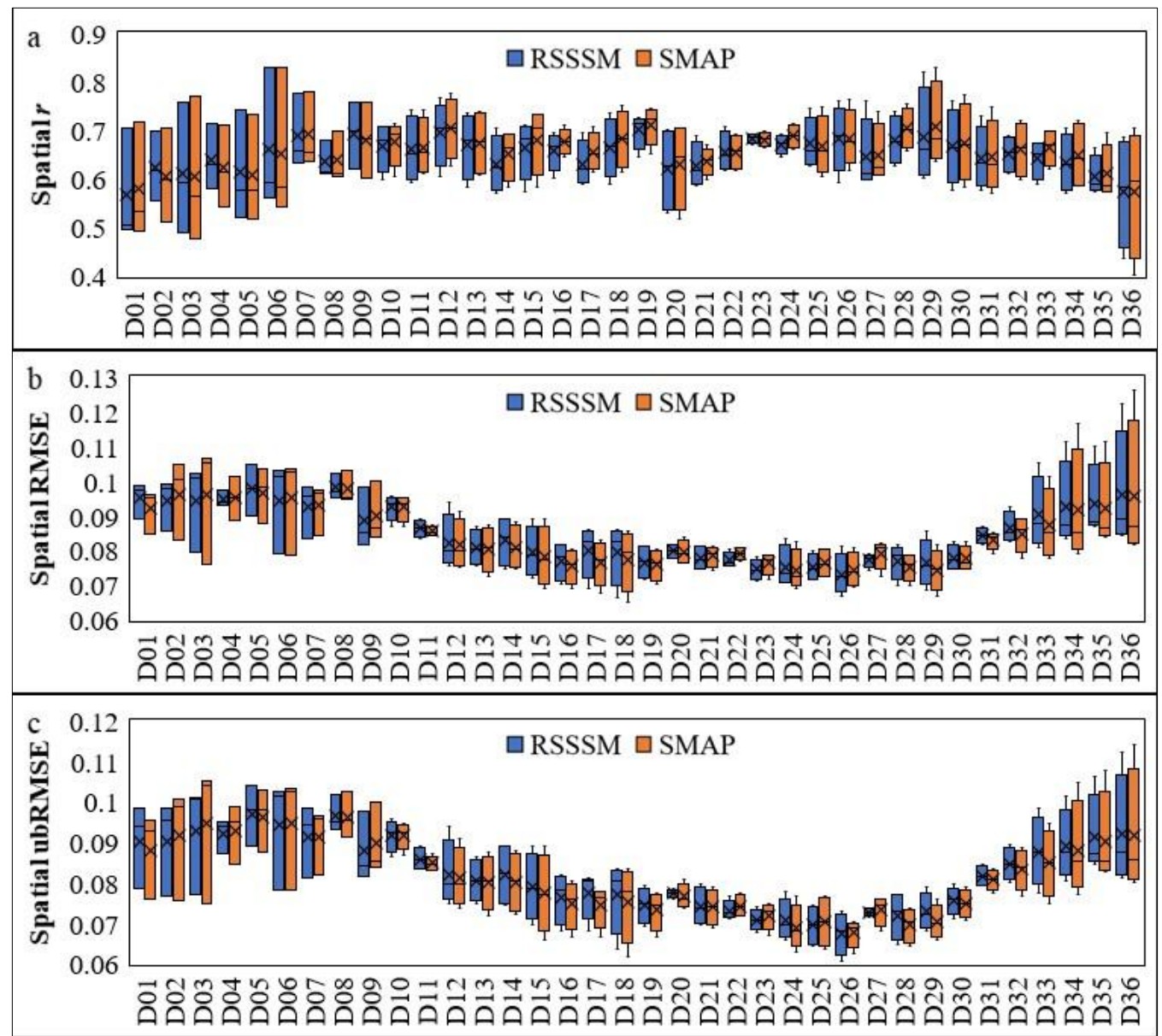

Figure 6: Comparison of the spatial pattern accuracy between RSSSM and SMAP in different 10-day periods from April 2015 to 2018. The three evaluation indexes are (a) *r*, (b) RMSE and (c) ubRMSE. The length of each box/error bar is determined from the evaluation index values in three (January to March) or four (April to December) years.

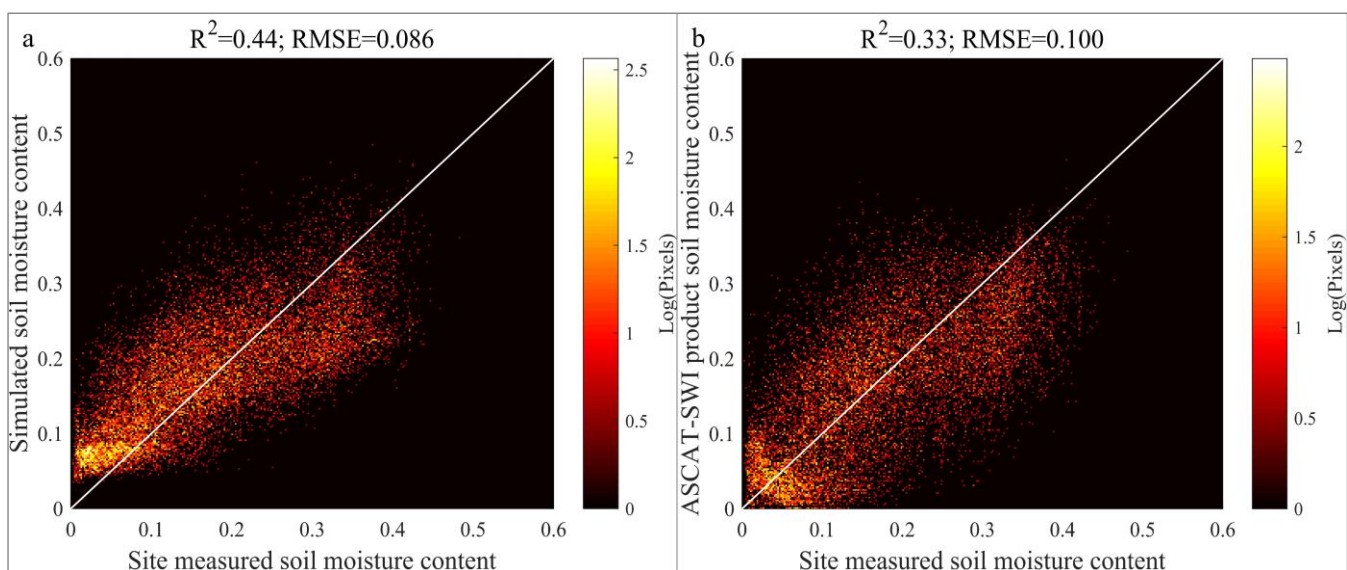

**Figure 7: Overall data accuracy comparison between RSSSM and the ASCAT-SWI data product. The scatter plot is**
**between (a) RSSSM or (b) ASCAT-SWI soil moisture and the site measured values during 2007~2018. The unit of all**
**plots is the density of points on a logarithmic scale.**

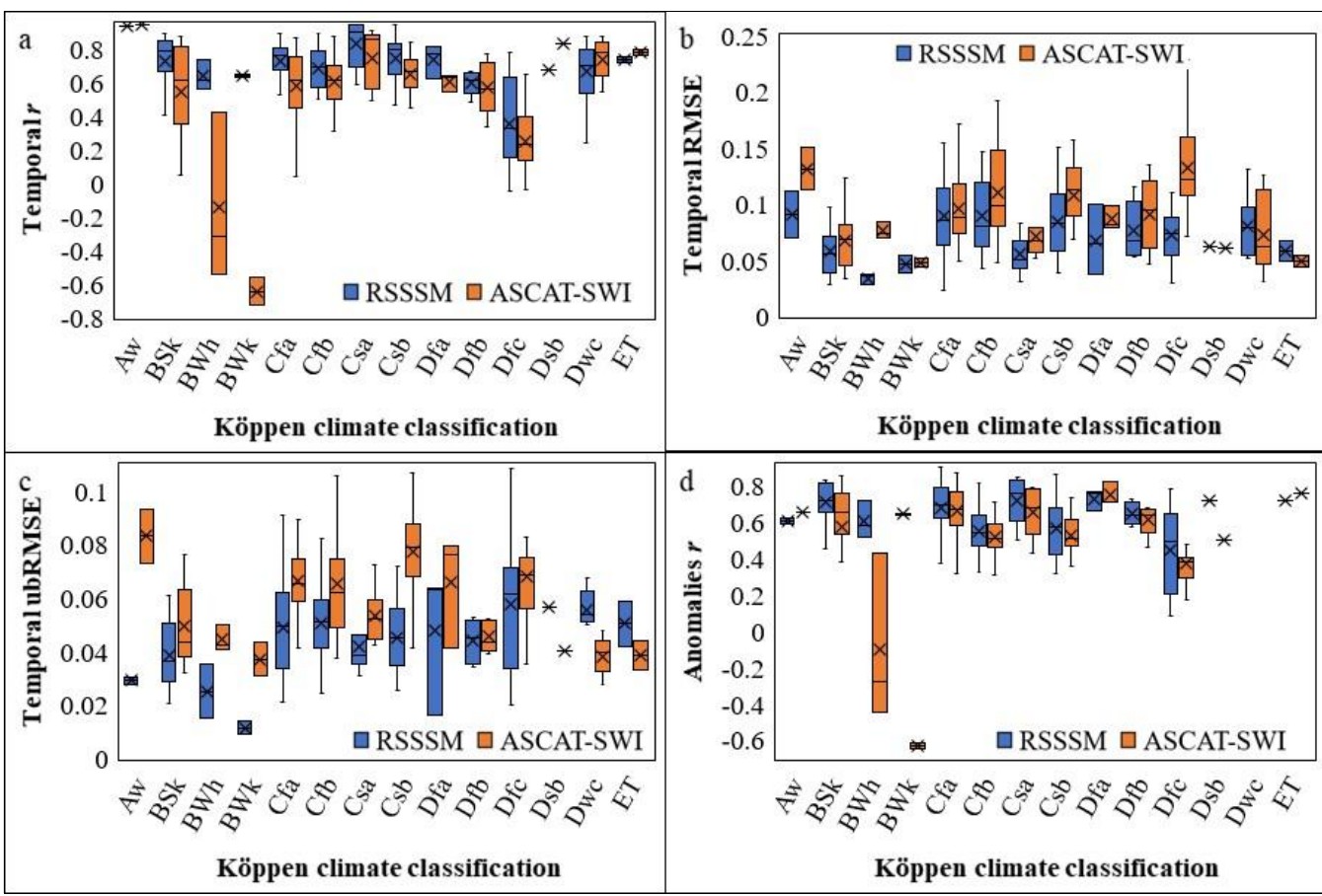

**Figure 8: Comparison of the temporal accuracy between RSSSM and ASCAT-SWI in different Köppen-Geiger climatic regions. The four indexes are (a) *r*, (b) RMSE, (c) ubRMSE, and (d) Anomalies R (A.R).**

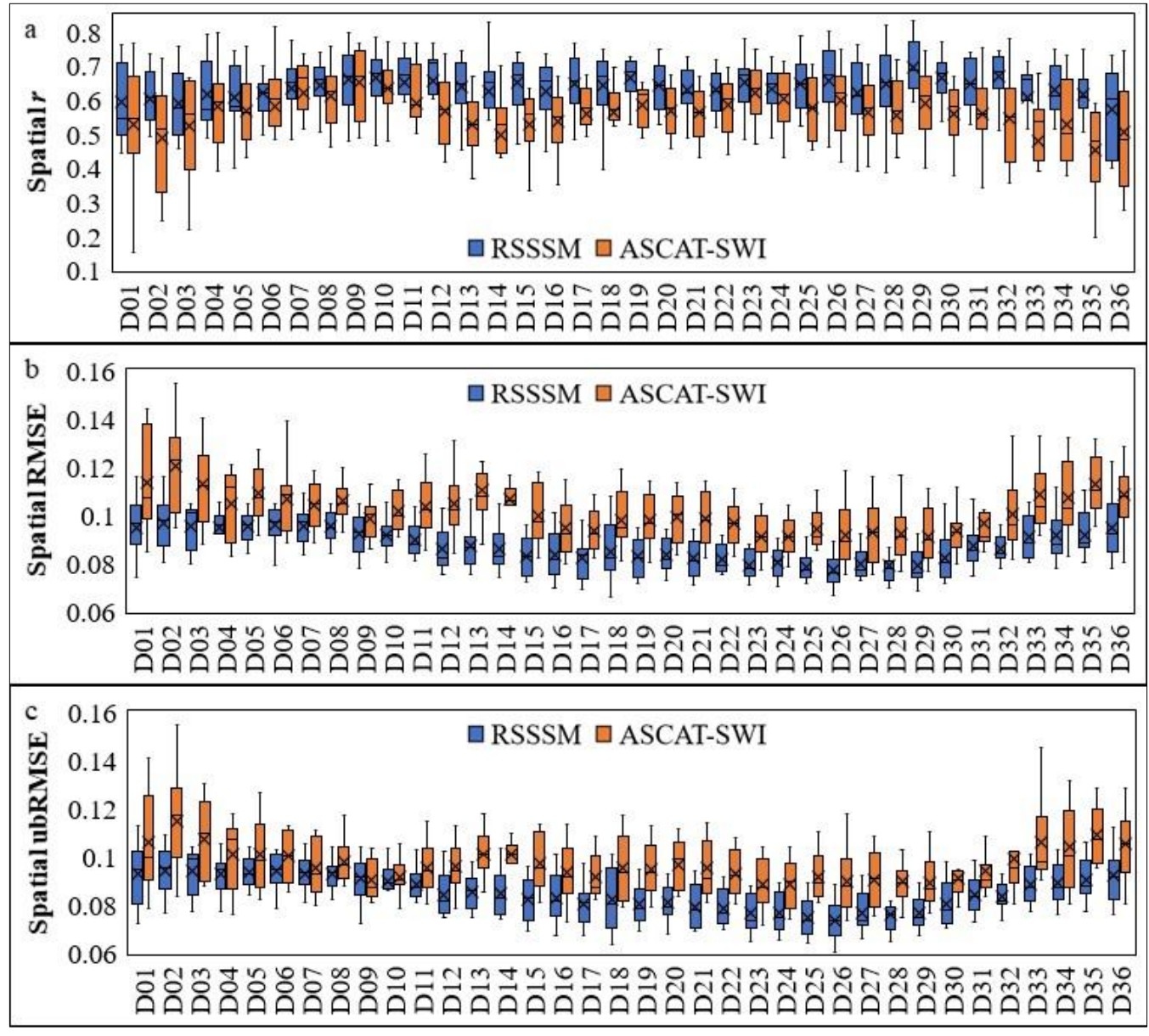

**Figure 9: Comparison of the spatial accuracy between RSSSM and ASCAT-SWI during different 10-day periods. The evaluation indexes are (a) *r*, (b) RMSE, and (c) ubRMSE.**

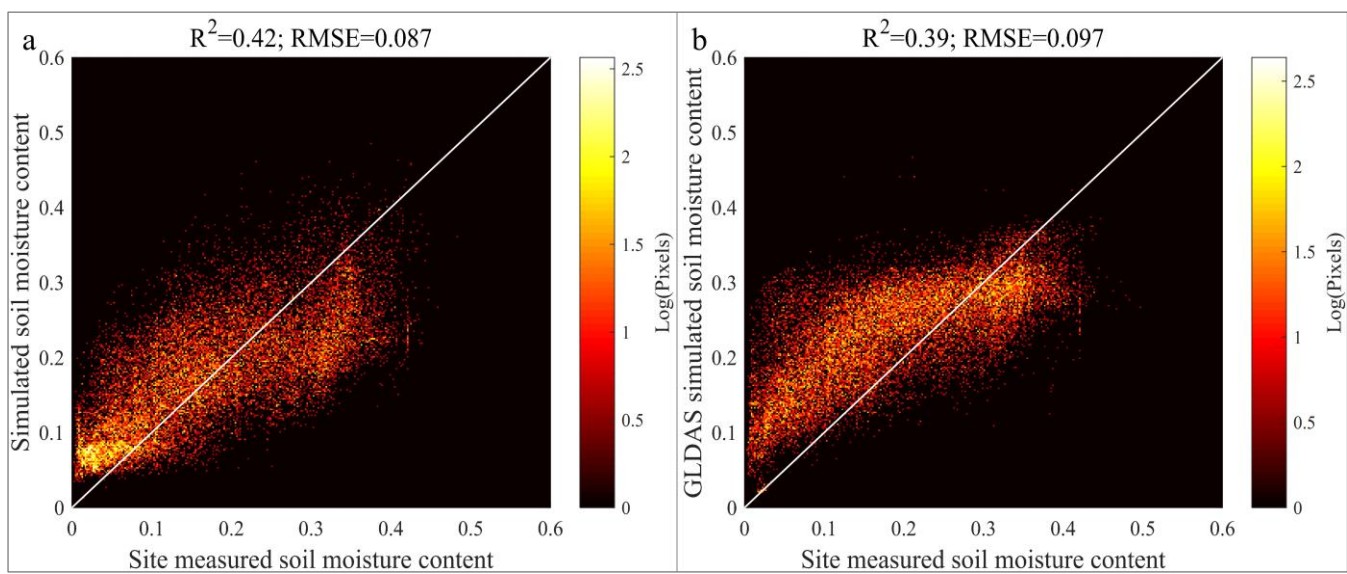

1120

**Figure 10: Overall data accuracy comparison between RSSSM and the surface soil moisture simulated by GLDAS Noah V2.1. The scatter plot is between the (a) RSSSM or (b) GLDAS soil moisture and the measured soil moisture from 2003 to 2018.**

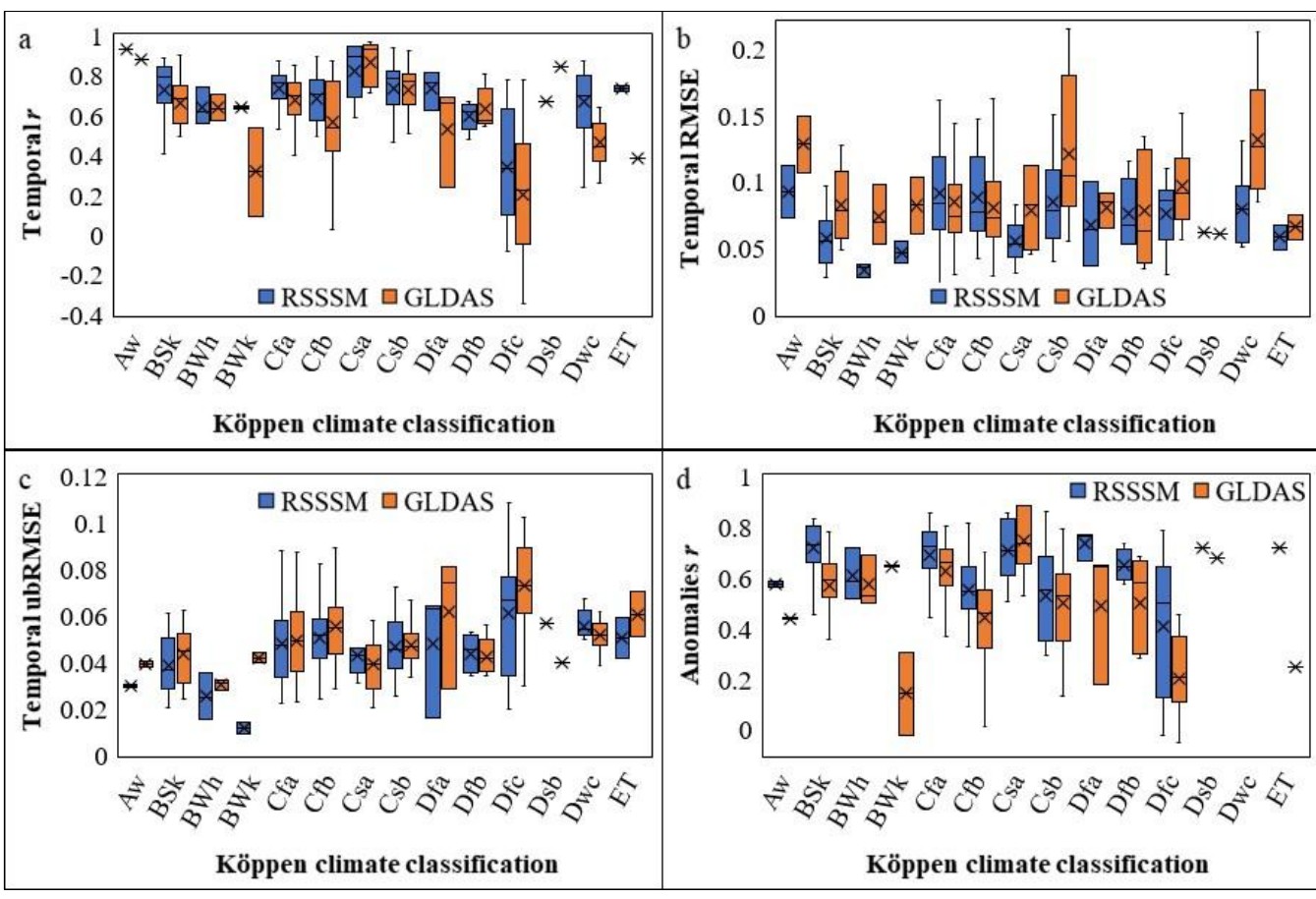

1125

**Figure 11: Comparison of the temporal accuracy between RSSSM and GLDAS surface soil moisture in regions with different Köppen-Geiger climate types. The four indexes are (a) *r*, (b) RMSE, (c) ubRMSE, and (d) Anomalies *r*.**

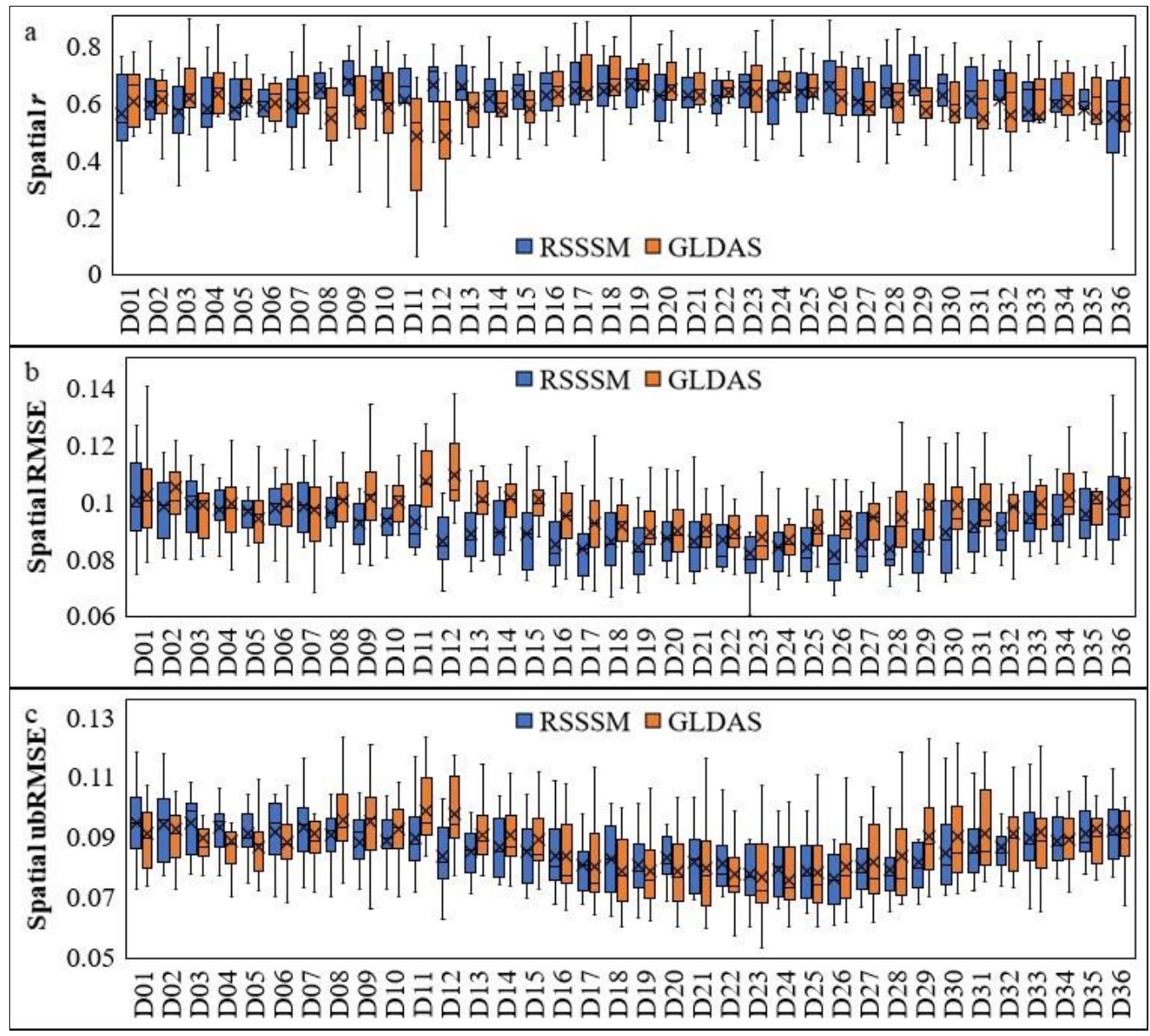

**Figure 12: Comparison of the spatial accuracy between RSSSM and GLDAS during different 10-day periods. The evaluation indexes are the same as those in Figure 7.**

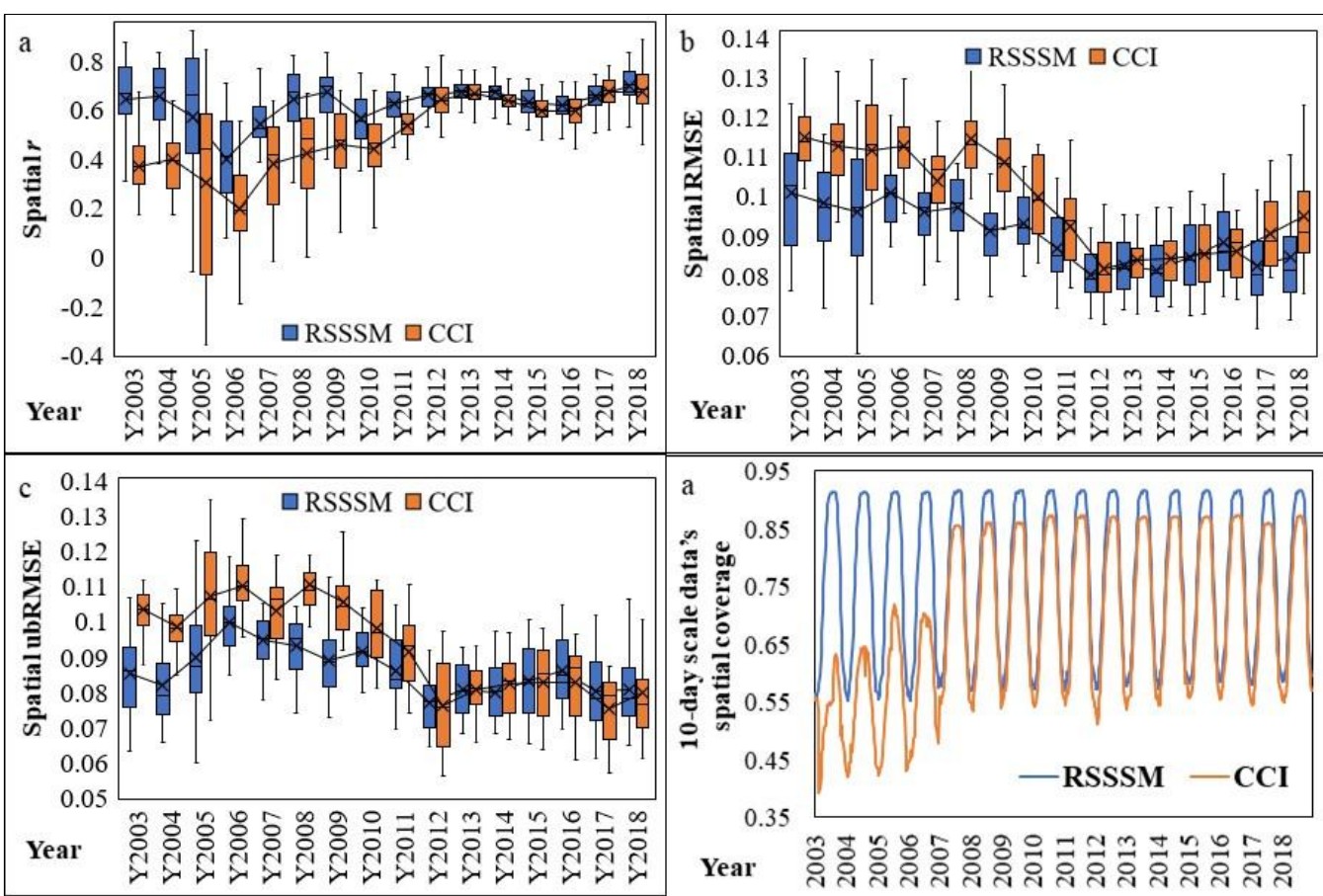

**Figure 13: Changes in the data quality and data spatial coverages of RSSSM and CCI soil moisture with year. The interannual changes in (a) spatial correlation coefficients (*r*), (b) spatial RMSE, (c) spatial ubRMSE values, and (d) the spatial coverages of 10-day period data for RSSSM and CCI.**

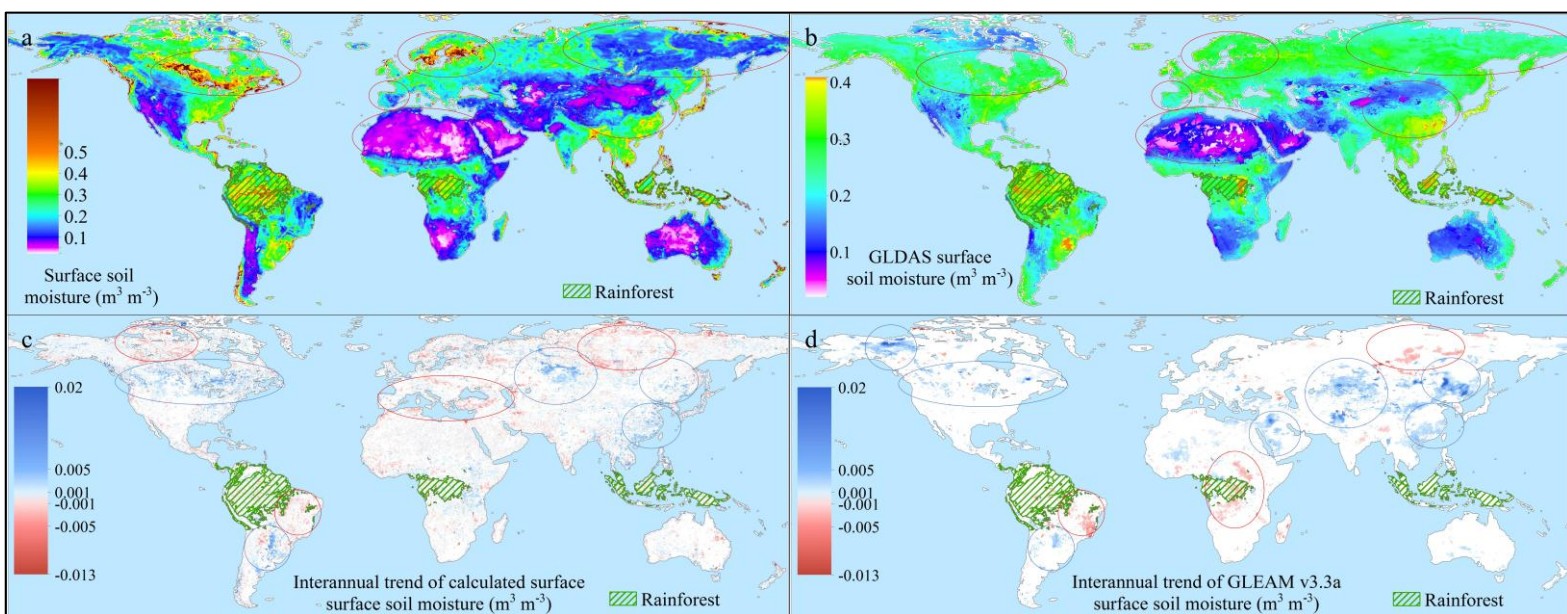

**Figure 14: Spatial and temporal patterns of the neural network simulated surface soil moisture (RSSSM) and comparison against other products: (a~b) the global map of (a) calculated RSSSM and (b) GLDAS Noah V2.1 soil moisture (averaged during 2003~2018); and (c~d) interannual trend map of (c) calculated RSSSM and (d) GLEAM v3.3a soil moisture from 2003 to 2018. The circled regions in (a~b) are the places with obvious differences between RSSSM and the other products, while the circled regions in (c~d) are those with significant trends.**

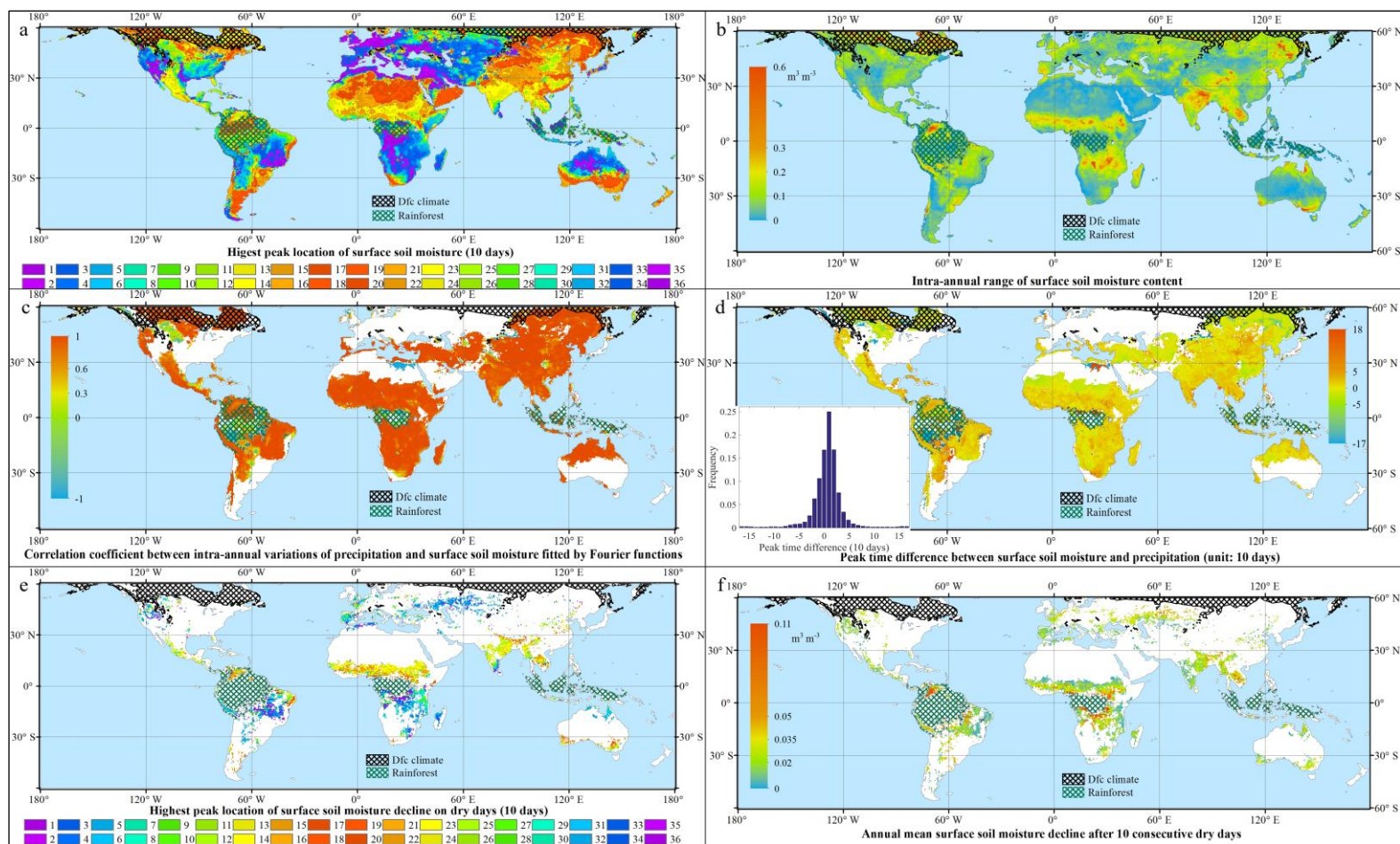

**Figure 15: Intra-annual variation in global surface soil moisture and its relationship with precipitation. (a) Spatial pattern of the time when surface soil moisture reaches its maximum in a year (unit: 10 days, note that the seasons are opposite in the Northern and Southern hemispheres); (b) intra-annual variation range of surface soil moisture; (c) map of the correlation coefficient between the intra-annual variations in precipitation and surface soil moisture (both are fitted by Fourier periodic functions); (d) peak time difference between the surface soil moisture and precipitation (unit: 10 days), with the frequency histogram shown as the inset; (e) 10-day period with the fastest surface soil moisture loss on rainless days in every 0.5° grid cell over the world; and (f) map of the annual mean surface soil moisture decline after 10 consecutive dry days (assuming that the dry period occurs randomly throughout a year).**

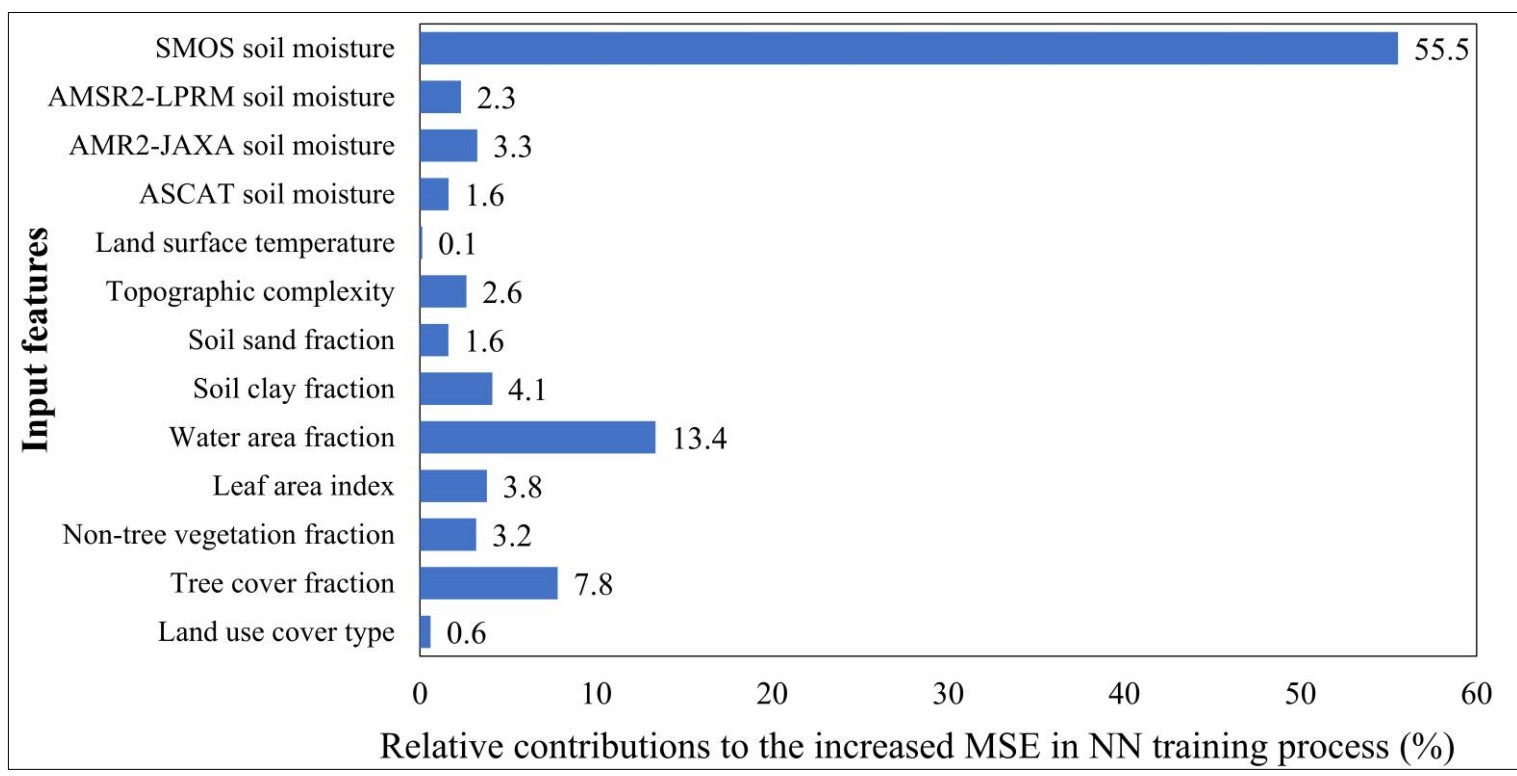

**Figure 16: Relative contributions of the 13 input features (i.e., four predictor soil moisture products retrieved from microwave remote sensing and 9 environmental factors that are quality impact factors of microwave soil moisture retrieval or also indicators of soil moisture) to the training efficiency of the first round's primary neural network (NN1-1-1).**

1155