# Peer review of "An improved global Remote Sensing-based Surface Soil Moisture (RSSSM) dataset covering 2003~2018"

_Earth System Science Data, 2020_

## Referee Comment (RC1) · Anonymous Referee #1 · 9 Jun 2020

Review comments for essd-2020-59 "A new dataset of satellite observation-based global surface soil moisture covering 2003–2018 " by Chen, Y., Feng, X., and Fu, B.

The authors propose a global dataset of top (0-5cm) soil moisture with 10 day temporal and 0.1° spatial resolution, covering the period 2003-2018. The dataset was produced gradually, backward in time, through machine learning methods (neural networks) for 5 periods that correspond with the availability of 11 different passive and active satellite remote sensing soil moisture products. Besides satellite observations (starting with SMAP in step one), 9 environmental properties were fed to the neural network, and from step two on, previously modeled soil moisture was included to enable the expansion backward in time. The final product is evaluated with observations of the international soil moisture network and, in comparison to other merged products,

rated superior, however, the potential for further improvement is also emphasized. Altogether, the work seems sound and the developed method and dataset appear valuable for further scientific studies and applications. Nevertheless, some of the steps in the processing chain need further clarification and the data structure needs to be improved before the manuscript can be considered to suffice for publication.

Specific comments

Title

I suggest to remove "new" from the title, since all dataset proposed in this journal are somewhat new. You may consider to name it "combined" or "improved" or "complete" or "optimal". Have you thought about giving the product an acronym? That improves recognizability and makes it easier to reuse it in other studies and publications.

Abstract

L14: more than 10**6 not correctly displayed in the online abstract (here it reads 106)
L15: Please state also the temporal resolution (10 days)

Introduction

L32: resolution of ERA INTERIM is rather 0.75°

L34-37: I agree that these products have many shortcomings, but other than the dataset provided by the authors, the models provide also information about the deeper soil layers. This important point should not be omitted here.

L75-76: "data averaging" - what type of averaging is meant, spatial or temporal? "can hardly unify the temporal variations." Please specify what the "temporal variations" refer to. Is it the temporal variations of the different soil moisture data products?

Data an methods

Instead of Table 1 or in addition, it would be good to have a timeline figure from 2003-

2018 that shows a bar for every dataset used in the process of creating the final product, including the 11 soil moisture products, the time-varying quality impact factors and the intermediate modeling products (SIM-1T, SIM-2T,...).

L110: Specify why SMAP is mentioned as the "best product" here. Is it because of the spatial resolution, the algorithms or with respect to the in situ observations? Can you add a citation to corroborate this statement?

L143: change to "reference coordinate system"

L182: "based on the correlation between soil dielectric conductivity" - do you mean soil dielectric permittivity or or soil electric conductivity?

L186-188: "Because ..." this sentence is unclear.

L205: Figure 1 is never referenced in the manuscript. This should be done here or later at L225.

L219: Do the 140x360 zones include water (ocean) areas?

L220: A subnetwork has 100 pixels, but ("for a 0.1° pixel in a given 10-day period, if all the subnetwork inputs have valid..."), how can one pixel have more subnetworks? Please improve the formulation.

L222: What is an "individual neural network"? Is it the collective of all zonal neural networks for one simulation (SIM-T1, SIM-T2, ...)? Is the maximum possible number of subnetworks 50.400 or less because of ocean cells?

L223: For reproducibility, it is required to state exactly the MATLAB version and the toolbox version and method/function name that was used for training the neural network.

L256: "we classified all pixels" -> "we classified all 0.1° pixels", I suggest to add the resolution information that it is clear which of the different grids is addressed.

L259: Again, I thought that a pixel is the smallest unit in the process (i.e. subnetwork). So how can a pixel have a subnetwork? Not clear to me.

L261-262: "Hence, it is a ..." sentence seems incorrect. I think you should better write "neural network collocation" or "neural network constitution" to make it more clear that these are neural network realizations with identical configuration but different ingredients.

L272,815 and other occurrences: it is not clear how the 10 day periods are defined and how they relate to the ordinal numbering. A month has between 29 an 31 days, so how are the periods split and how does that affect the last 3rd where the number of days is variable? How does this variable length averaging affect the results and what are the implications for validation?

L270-292: also this section would greatly benefit from a timeline bar plot that shows all the soil moisture products and simulated models, so that the overlaps can be grasped immediately

L318: define how R**2 is computed (based on Spearman or Pearson).

L321: lower case r should be used for the correlation coefficient (based on Pearson?). Why are you mixing r and R**2 and do not use R**2 for all analyses?

L322: please provide formula for A.R computation

L326: "in all grids", grids or pixels (1 x 1 or 0.1 x 0.1°)?

L326: please provide formulas for spatial pattern validation (at least in the supplement)

Results

Figure 3: Use identical labels for the x-axis, add missing lower frame.

Figure 4: If the color key is put below the figure, the figure can be increased in the horizontal direction which leads to wider bars. You could even remove the x-axis labels

and names and leave only the lowermost. By this you can increase the size of the bars and hence the readability (reduce redundancy).

L381: How is the performance of SIM if the SMAP training period is omitted, i.e. from 2003 until 2015D01, as compared to ASCAT-SWI?

Figure 7,10: As for fig. 4 place color key below the plots and increase the bars horizontally

Discussion

Do you see any chance to improve the temporal resolution of the product in the future? If not, what are the constraints?

L499-500: Is SIM also superior to the other products if only the prior to SMAP period is considered (2003 until 2015D01)?

Are there plans to update the data-set on a regular basis?

Dataset

The dataset is organized as an archive of geotiff files. The problem with this structure is that the time identifier is only contained in the file name, but without practical formatting. If one wants to import a time series for a region or a single pixel, the data structure is quite unhandy. Also from the readme file and the metadata it is not quite clear what the 10 days ordinal numbering means exactly. Is it always the [1-9],[10-19],[20-29] or [1-10],[11-20],[21-30] periods? How are the months with variable length considered (28,29,30,31 days)? That's not clear also not from the manuscript.

Further, I would suggest to add a table (csv) that links the different file names to their specific period using ISO 8601 https://en.wikipedia.org/wiki/ISO_8601 notation:

e.g., a file named inventory.dat with a list like the following one:

Period, Filename 2003-01-01/2003-01-10, SMY2003DECA01.tif 2003-01-11/2003-01-

20, SMY2003DECA02.tif ...

Also the numbering should be formatted as %02d so that, e.g.,

SMY2003DECA1.tif becomes SMY2003DECA01.tif. This is important if one wants create a chronological file list for looping over time. With the current scheme, the order would become SMY2003DECA1.tif SMY2003DECA10.tif, SMY2003DECA11.tif, ...

This should be also applied to all tables in the manuscript (e.g., 2005D01 instead of 2005D1).

Supplement

Figure S1: The figure and description is not completely clear. I assume that every number (yellow and blue frames) is one pixel ($0.1°$ x $0.1°$)? I think it would become more clear if you superimposed a light gray mesh for the pixels over the 1x1° zones. But then, why are there 4 steps required to smooth the borders? It means that every boarder gets smoothed twice, and every corner point even four times.

Figures S5, S8, S11: put the color-key to the bottom of the figure (a single key would be sufficient for all sub-figures), you could even remove the x-axis labels and names and leave only the lowermost. By this you can increase the size of the bars and hence the readability (reduce redundancy).

Language and bibliography

There are often blanks missing between words. DOIs are completely missing in the reference list.

---

## Referee Comment (RC2) · Anonymous Referee #2 · 29 Jun 2020

The authors tried to generate long-term surface soil moisture at a global scale, via data fusion of 11 microwave remote sensing-based soil moisture products since 2003 through neural network approach, and SMAP soil moisture products were used as the training target. The idea is very interesting and should be encouraged to explore further how much extent the machine learning can help in Earth Observation for delivering physically-consistent (or physic-aware) products. However, the way the current manuscript is written, organized is still far from clarity, structured for this reviewer to comprehend their contributions. I would suggest rejection and encourage the author to continue along this line of effort. In the following, I listed some major concerns: 1. The author claimed that "This new dataset, once validated against the International Soil Moisture Network (ISMN) records, is supposed to be superior to the existing products

(ASCAT-SWI, GLDAS Noah, ERA5-Land, CCI/ECV and GLEAM), and is applicable to studying both the spatial and temporal patterns. " This assumption is too strong. On the other hand, it seems the author referred to the validation of the NN-based 10-d soil moisture products versus the 10-d averaged ISMN in-situ observations (as seen Figure 5, Figure 8, Figure S3, S6, and S9). Is it true? In any case, it should be specified under what conditions the generated product is performing better than other products. "supposed to be superior" is really not a scientific statement.

2. There were some strange 'terminologies' the author used for discussion, for example: a. 'penetrability of microwave' (which is seldom found in the literature. A more widely used term is 'microwave penetration depth'); b. "Soil moisture retrieval from passive microwave sensors is based on the correlation between soil dielectric conductivity, that is influenced by soil moisture . . ..". Following the theoretical development of soil moisture retrievals from remote sensing, the relationship between soil moisture and dielectric constant is the fundamental (not soil dielectric conductivity).

3."However, this data is regional, with a large temporal gap, and cannot be seen as observational-based only since precipitation data is incorporated." This is a very strange argument. We all know there is a strong link between precipitation and soil moisture variation. Physically speaking, one used the antecedent precipitation index to understand how precipitation events drive the variation of soil moisture. This is like one of 'quality impact factors'. If the above argument is true, we can argue that the author's approach in this manuscript is also not 'observation-based', as they used LAI, land cover, LST, and many other factors.

4. " are these factors used as direct spatial predictors of soil moisture or just because they are related to the errors of satellite soil moisture retrievals (i.e., the quality impact factors of soil moisture)? We insist on the latter, proposing two main reasons for the incorporation of environmental factors." This is very confusing and not necessarily correct, and not well grounded. We know the soil moisture retrieval from remote sensing is using a radiative transfer model to account for scattering and emissions from both

soil and vegetation, which is conflicting with the author's statements.

5. 'Water Body' was used as one of the predictors (it should be predictor, rather than quality impact factors). This is very strange. As we know, water body map in either SMOS or SMAP soil moisture products were used to mark out those locations to avoid soil moisture retrievals over these water bodies (otherwise, it would be physically no sense, in terms of soil moisture). This is wrong and not physically sound to include water bodies as one of predictor for predicting surface soil moisture.

6. For 'topographic complexity' 'soil texture', the author used from different sources, one from ASCAT ancillary data and the other use SMAP ancillary data. This reviewer is wondering why such a choice? Why not making it consistent (i.e., get ancillary data from one single product, instead of two?)

7. '$3\sigma$ denoise'. what is the effect of such a filter on identifying extreme years? for example, during 2003, 2010, 2018, 2019 there are extreme heat events in Europe and the soil moisture is so dry which can be beyond the 3 standard deviations.

8.NN design. SMAP is only available after 2015, so I am not sure what is the meaning of simulation period 2012D19~2013D36, but also 2014-2018. I guess this is constrained by the available data (PROBA-V and GLASS LAIs)? But in any case, it does not represent any physical meaning to predict 2015data with 2012-2013 data. At least, the NN design is not clear on why it is designed as such.

There are some other specific comments can be found in the attached PDF.

Please also note the supplement to this comment:
https://essd.copernicus.org/preprints/essd-2020-59/essd-2020-59-RC2-supplement.pdf

**Supplement:**

[revised manuscript text omitted]

https://doi.org/10.5194/essd-2020-59
This is just a preview and not the published preprint.

[Figure]

**Figures**

[Figure]

835

**Figure 1: The flow chart of global surface soil moisture data production.**

[Figure]

**Figure 2: Comparison between the neural network simulated surface soil moisture (SIM) and SMAP data. The scatter plots are between: (a) SIM and SMAP values at all grids; b) SIM and SMAP values at only the grids with measurements; (c) SIM and the site measured soil moisture; and (d) SMAP and the site measurements during April 2015~2018. All plots are represented as the point density on a logarithmic scale, while the unit of soil moisture content and RMSE values is m³ m⁻³.**

[Figure]

845

**Figure 3: Comparison between the temporal accuracy of SIM and SMAP in regions with different Köppen-Geiger climate types. The four indexes are (a) R, (b) RMSE, (c) ubRMSE and (d) Anomalies R (A.R). The lengths of the error bars are 1.5 times that of the interquartile range, while the upper and lower boundaries and the central lines of the boxes indicate the 75th, 50th and 25th percentile values, with mean values marked by '×' (the forms of all the**

850 **following boxplots are the same).**

https://doi.org/10.5194/essd-2020-59
This is just a preview and not the published preprint.

**Figure 4: Comparison between the spatial pattern accuracy of SIM and SMAP in different 10-day periodsduring April 2015~2018. The three evaluation indexes are: (a) R, (b) RMSE and (c) ubRMSE. The length of each box/error 855 bar is determined from the evaluation index values in three (January to March) or four (April to December) years.**

https://doi.org/10.5194/essd-2020-59
This is just a preview and not the published preprint.

[Figure]

[Figure]

**Figure 5: The overall data accuracy comparison between SIM and the ASCAT-SWI data product. The scatter plot is between: (a) SIM or (b) ASCAT-SWI soil moisture and the site measured values during 2007~2018. The unit of all plots is the density of points on a logarithmic scale.**

[Figure]

860

**Figure 6: Comparison between the temporal variation accuracy of SIM and ASCAT-SWI in different Köppen-Geiger climatic regions. The four indexes are: (a) R, (b) RMSE, (c) ubRMSE, and (d) Anomalies R (A.R).**

https://doi.org/10.5194/essd-2020-59

[Figure]

[Figure]

[Figure]

**Figure 7: Comparison between the spatial accuracy of SIM and ASCAT-SWI during different 10-day periods. The evaluation indexes are: (a) R, (b) RMSE, and (c) ubRMSE.**

https://doi.org/10.5194/essd-2020-59
This is just a preview and not the published preprint.

[Figure]

[Figure]

[Figure]

**Figure 8: The overall data accuracy comparison between SIM and the surface soil moisture simulated by GLDAS Noah V2.1. The scatter plot is between the (a) SIM or (b) GLDAS soil moisture and the measured soil moisture during 2003~2018.**

[Figure]

**Figure 9: Comparison between the temporal accuracy of SIM and GLDAS surface soil moisture in regions with different Köppen-Geiger climate types. The four indexes are: (a) R, (b) RMSE, (c) ubRMSE, and (d) Anomalies R.**

[Figure]

**Figure 10: Comparison between the spatial accuracy of SIM and GLDAS during different 10-day periods. The evaluation indexes are the same as those in Figure 7.**

https://doi.org/10.5194/essd-2020-59

[Figure]

**Figure 11: Changes in the data quality and data spatial coverages of SIM and CCI soil moisture with year. The interannual changes in (a) spatial correlation coefficients, (b) spatial RMSE, (c) spatial ubRMSE values, and (d) the spatial coverages of 10-day period data of SIM and CCI.**

880

[Figure]

**Figure 12: The spatial and temporal patterns of the neural network simulated surface soil moisture and comparison against other products: (a~b) the global map of (a) calculated soil moisture and (b) GLDAS Noah V2.1 soil moisture (averaged during 2003~2018); (c~d) the interannual trend map of (c) calculated soil moisture and (d) GLEAM v3.3a soil moisture from 2003 to 2018. The circled regions in (a~b) are the places with obvious differences, while the circled regions in (c~d) are those with significant trends.**

885

[Figure]

[Figure]

**Figure 13: The intra-annual variation in surface soil moisture and its relationship with precipitation at global scale. (a) The global**
890 **distribution of the highest peak location in time of the calculated surface soil moisture (unit: 10 days, note that the seasons are opposite**
**in the Northern and Southern Hemispheres); (b) the global spatial pattern of the intra-annual range of surface soil moisture; (c) the**
**map of the correlation coefficient between the intra-annual variations in precipitation and surface soil moisture fitted by Fourier**
**periodic functions; (d) the peak time difference between the intra-annual variations in surface soil moisture and precipitation (unit: 10**
**days), and the inset is the frequency histogram; (e) the spatial distribution of the highest peak location of surface soil moisture decline**
895 **on dry days (unit: 10 days); and (f) map of the estimated annual mean decline in surface soil moisture after 10 consecutive dry days**
**(assuming the dry period occurs randomly over a year).**

---

## Referee Comment (RC3) · Anonymous Referee #3 · 24 Jul 2020

The authors used an iterative neural network approach to produce a new satellite based soil moisture dataset using 11 microwave soil moisture products, using SMAP data for training and ISMN database for validation. The approach is quite original and efficient resulting is a improvement in the accuracy of the spatio-temporal patterns at the global scale, and at a 0.1 degree resolution. However, the manuscript will need to be improved before acceptance, in its structure, clarity and tone.

1- The introduction would need to be improved. Several statements need to be supported by existing literature, others would need to be clarified. Finally the introduction would need to end with a brief description of the approach used in the study and how this approach will address the three major concerns raised from existing soil moisture products. See detailed comments below for details. 2- The tone of the manuscript

when referring to the new product and to past studies is not always appropriate. For instance, stating that the present product is "superior to the existing products" is useless, not informative and condescending. I would encourage the authors to rather explain how their product is an improvement to the global estimation of soil moisture, without necessarily condemn other products. In the result section, while nonlinearities between estimate and in-situ soil moisture measurements are identified for other products, it is not reported for the author's product which I find quite biased. 3- The validation approach is based on site specific comparison. However, soil moisture being so spatially variable within a 0.1 degree pixel, validation based on single site observations within 0.1 degree pixels can be quite meaningless. This might be particularly true when one considers that site selection for in-situ measurement is rarely motivated by representativity of the surrounding landscape, but by specific ecological reasons.

—- Abstract. —- "This new dataset, once validated against the International Soil Moisture Network (ISMN) records, is supposed to be superior to the existing products" → do you mean this validation hasn't been done yet? Superior in what way?

"reveals that the surface moisture decline on rainless days is highest in summers over the low-latitudes but highest in winters over most mid-latitude areas." Soil moisture being so spatially variable , I find the impact of this statement quite limited – e.g. low latitude regions range from tropical / equatorial rain forests to deserts and one would expect as much differences in the sensitivity of soil moisture to precipitation between a desert and a tropical forest than between a tropical forest and a temperate prairie.

—- Introduction —-

L47: "due to various disturbances": what type of disturbances? L49: " Although new sensors, SMOS . . .." -> "Although new sensors such as SMOS. . ." L 50: "better penetrability" -> please be more specific: what depth? L66: "Because the temporal variation in soil moisture is often better captured by model simulations than remote sensing inversions" : please include a reference that support this statement. L67: "CCI may

undesirably combine the disadvantages of both." Be more specific here (low accuracy of temporal variations from remote sensing products and low spatial accuracy from model simulations – am I right?). And please include another reference here for this second statement. L70: "are assimilated instead": instead of what? this sentence is not clear. L85: "Among these three approaches, machine learning proves to be probably the best choice" based on what criteria – again, please be more specific. L102: "substantial success has not been achieved yet." This is a rather strong and yet vague statement that denies the merits of a large body of research. Please remove this statement. L102/103:" the high-quality microwave observations are not fully utilized": this is not clear from the literature review – please develop this point in earlier sections of the introduction (i.e. in what way high-quality microwave observations haven't been fully utilized, and how the authors are proposing to utilize them more efficiently). L106-107: This statement should be removed from the introduction section.- this is rather a concluding statement. Instead please describe your approach in a couple sentences and how this approach addresses the three major concerns identified.

—- Data and Methods —- – Section 2.1 L110: please add citation to literature supporting this statement. L115-118: This sentence is too long and too complex. Please split into shorter and clearer sentences. – Section 2.2 L147/160: the purpose of this argumentation is quite unclear as the environmental predictors that are selected are also important drivers of soil moisture dynamic. L201: since precipitation is such an important driver of soil moisture, the reasons why this variable hasn't been included as a quality impact factor should be included in the main document. L202/203: This sentence should start this section, not end it. A table similar to table 1 but for the quality impact factors would be useful. The table would indicate the source of the data, the resolution and the temporal span for the dynamic factors. – Section 2.3 Sections 2.4, 2.5 and 2.6 should be included in section 2.3 as it details the different steps of the calculation flow. A clear justification on why a neural network approach was adopted should be included in this section. L223: what is a hidden layer? L242: reference required for the "suspicious value removal". L259: Here you are referring to a 1 degree

pixel a presume ? Please specify. L268: why 74 networks? Please explain. – Section 2.7 Figure S2 should be included in the main manuscript as the spatial distribution of validation data is critically important to evaluate the overall strength of this validation. It is surprising that none of the Canadian sites of the Russian sites made it to the final set of sites for validation. L304/306: This is an important point. Soil moisture being so spatially variable, validation based on single site observations within 0.1 degree pixels can be quite meaningless. Especially when one considers that site selection for in-situ measurement is rarely motivated by representativity of the surrounding landscape, but by specific ecological reasons. – Section 2.8 L335: "probably the best choice for periodic function fitting" : please support this statement by adequate reference to literature. L344/345: I don't understand this argument. Why restricting this analysis to 10 consecutive rainless days and not the whole range of 10-days sum of precipitation?

—- Results —- Tables 3 to 8 could be synthesized into only two tables: one for temporal accuracy assessment and one for spatial accuracy assessment for the three products comparisons with SIM. Similarly, it would be nice to have figures 3, 4, 6, 7, 9 and 10 summarized in 2 figures where all four products appear (SIM, SMAP, GLDAS and ASCAT). This would facilitate comparison between products.

L380-383: However, it looks like the relationship between SIM estimates and in situ observations is nonlinear (Figure 5a). Furthermore, SIM seems to overestimate soil moisture in the lowest range (winter?) when a density of pixels is quite high. Please include these remarks in the results.

L393: the relationship between SIM and in situ measurements is also obviously non-linear. Please include this remark in the text for fairness.

Why table S19 and Figure S7 do not appear in the main document like the other product comparison? Please move them to the main manuscript.

L422-434: This belongs to the discussion section.

---

## Author Comment (AC1) · 14 Aug 2020

**To Reviewer #1:**

We thank referee #1 for the valuable comments that will help us improve the quality and readability of the manuscript. We have carefully revised the MS following your comments and suggestions. We provide detailed responses to the Referee's comments in the Supplement.

**General comment.** The authors propose a global dataset of top (0-5cm) soil moisture with 10-day temporal and 0.1 spatial resolution, covering the period 2003-2018. The dataset was produced gradually, backward in time, through machine learning methods (neural networks) for 5 periods that correspond with the availability of 11 different passive and active satellite remote sensing soil moisture products. Besides satellite observations (starting with SMAP in step one), 9 environmental properties were fed to the neural network, and from step two on, previously modeled soil moisture was included to enable the expansion backward in time. The final product is evaluated with observations of the international soil moisture network and, in comparison to other merged products, rated superior, however, the potential for further improvement is also emphasized. Altogether, the work seems sound and the developed method and dataset appear valuable for further scientific studies and applications. Nevertheless, some of the steps in the processing chain need further clarification and the data structure needs to be improved before the manuscript can be considered to suffice for publication.

**Response:** Thank you for your careful reading and the positive comments on our work. We agree that some steps in the methods were unclearly written, while the data structure is not easy for other researchers to use. We have added the missing important details in the revision to further clarify the processing chain and reuploaded the dataset with filename changes and table additions, according to your valuable suggestions. Please see the details in the responses below.

**Specific comment 1.** I suggest to remove "new" from the title, since all dataset proposed in this journal are somewhat new. You may consider to name it "combined" or "improved" or "complete" or "optimal". Have you thought about giving the product an acronym? That improves recognizability and makes it easier to reuse it in other studies and publications.

**Response:** Thank you for the suggestion. We have changed it to 'improved'. We also named the product 'RSSSM' since it is a remote sensing-based surface soil moisture product. In other parts of the article, all the instances of 'SIM' have been changed to 'RSSSM' as well, including those in the figures and tables.

**Specific comment 2.** L14: more than 10**6 not correctly displayed in the online abstract (here it reads 106)

**Response:** We have revised it to '*more than one million*'.

**Specific comment 3.** L15: Please state also the temporal resolution (10 days)

**Response:** We have added this important information.

**Specific comment 4.** L32: resolution of ERA INTERIM is rather 0.75°

**Response:** Thank you for reminding us. We have corrected this accordingly.

**Specific comment 5.** L34-37: I agree that these products have many shortcomings, but other than the dataset provided by the authors, the models provide also information about the deeper soil layers. This important point should not be omitted here.

**Response:** We agree that models can simulate soil moisture in deep layers, which is an important advantage. We have added this point to the revised manuscript: '*Apart from surface soil moisture that can be observed by satellites, the modeling method also provides information on the moisture in deep soil layers.*'

**Specific comment 6.** L75-76: "data averaging" - what type of averaging is meant, spatial or temporal? "can hardly unify the temporal variations." Please specify what the

"temporal variations" refer to. Is it the temporal variations of the different soil moisture data products?

**Response:** We apologize for the unclear expressions. It is neither spatial nor temporal averaging. Instead, the CCI product is achieved by rescaling the soil moisture data retrieved from each microwave sensor first and then averaging the rescaled soil moisture data products during the same period (i.e., the common period for two or more products) based on some criteria (e.g., the estimated error) (Dorigo et al., 2017; Gruber et al., 2017; Gruber et al., 2019; Liu et al., 2012). The 'temporal variations' in this sentence refer to the temporal variations in the different soil moisture data products. Following your question, we have revised the sentence to: '*Rescaling the soil moisture data retrieved from each sensor by using CDF matching followed by averaging the rescaled data during a common period, which is adopted in CCI, will result in problems when unifying the temporal variations in different soil moisture products.*'

**Specific comment 7.** Instead of Table 1 or in addition, it would be good to have a timeline figure from 2003-2018 that shows a bar for every dataset used in the process of creating the final product, including the 11 soil moisture products, the time-varying quality impact factors and the intermediate modeling products (SIM-1T, SIM-2T,...).

**Response:** Thank you for this suggestion. Following your comment, we have added a timeline figure showing the temporal coverages (including the used data periods and unused data periods) of all 11 soil moisture products, the time-varying quality impact factor (i.e., three LAI products) and that of the intermediate products. The figure below is attached as Figure 1 in the revised manuscript. Table 1 has thus been removed.

[Figure]

*Figure R1: The timeline figure showing the time periods of the soil moisture datasets and the 'quality impact factor' products (e.g., LAI dataset) used in this study (listed above the timeline), as well as the periods of data applied for the training of the 67 independent neural networks and the neural network simulation outputs (i.e., simulated soil moisture) of eight substeps (listed below the timeline).*

**Specific comment 8.** L110: Specify why SMAP is mentioned as the "best product" here. Is it because of the spatial resolution, the algorithms or with respect to the in-situ

observations? Can you add a citation to corroborate this statement?

**Response:** SMAP is the 'highest quality product' with respect to in situ observations. In the Introduction section, this point has been stated: '*Although new sensors such as SMOS (Stillman and Zeng, 2018) and SMAP (Entekhabi et al., 2010) can produce significantly improved estimates because L-band microwaves (1~2 GHz) can penetrate the vegetation canopy better than other bands (Burgin et al., 2017; Chen et al., 2018; Karthikeyan et al., 2017; Kerr et al., 2016; Kim et al., 2018; Leroux et al., 2014; Stillman and Zeng, 2018), the applicability of both products is still limited. SMOS data have too much noise and too many missing values in Eurasia due to high radio frequency interference (RFI) (Oliva et al., 2012). While SMAP has the highest quality (the unbiased RMSE of the passive product can be close to its target of 0.04 m³/m³) and has filtered RFI (Chen et al., 2018; Colliander et al., 2017), ...*' Following your advice, we revised this sentence and added two new references supporting the best performance of the SMAP product. It reads: '*SMAP currently has the highest quality of all remote sensing-based soil moisture products (Al-Yaari et al., 2019; Liu et al., 2019)...*'.

**Specific comment 9.** L143: change to "reference coordinate system".
**Response:** We have changed it accordingly.

**Specific comment 10.** L182: "based on the correlation between soil dielectric conductivity" - do you mean soil dielectric permittivity or soil electric conductivity?
**Response:** We apologize for this mistake. It should be 'soil dielectric permittivity' or 'soil dielectric constant'. We have corrected it to '*soil dielectric constant*' in the revision.

**Specific comment 11.** L186-188: "Because ..." this sentence is unclear.
Response: We apologize for the unclear expression. This sentence explains that the actual LST can determine the bias of every LST estimate, which is used in the corresponding soil moisture retrieval. Hence, the actual LST will influence the biases of different soil moisture products. We have revised the sentence as follows: '*Because different LST estimates are used in the retrievals of different soil moisture products,*

*while the bias of each LST estimate compared to the actual LST is influenced by the actual LST, we assume that the actual LST can determine the accuracy of every LST estimate and finally the relative performances of various soil moisture products (Kim et al., 2015).'* We hope this sentence will be easier for readers to understand.

**Specific comment 12.** L205: Figure 1 is never referenced in the manuscript. This should be done here or later at L225.

**Response:** Thank you for the reminder. We have added: *'The basic flow of this process is shown in Figure 2'* (note: Figure 2 is Figure 1 in the original manuscript) at the beginning of section 2.2 in the revised manuscript.

**Specific comment 13.** L219: Do the 140x360 zones include water (ocean) areas?

**Response:** Yes, the 140×360 zones include water (ocean) areas. However, for zones with no land or very limited land, the number of valid pixels is less than 100, so these ocean zones are not applicable for subnetworks and are excluded.

**Specific comment 14.** L220: A subnetwork has 100 pixels, but ("for a 0.1pixel in a given 10-day period, if all the subnetwork inputs have valid..."), how can one pixel have more subnetworks? Please improve the formulation.

**Response:** We apologize for the confusing expression. We have rewritten the paragraph as follows: '*Therefore, we divided the global extent into 140×360 zones in all regions except the polar areas (80°N~60°S). Here, for a 0.1° pixel during a specific 10-day period, if all the input data (soil moisture products and quality impact factors) have valid values, one valid data point is provided. Therefore, the maximal number of valid data points applied to train a subnetwork = 100 × the number of 10-day periods within the training period. The subnetworks with less than 100 valid data points (e.g., those in oceans) were dropped, leaving usually >15,000 zonal subnetworks included in an independent neural network.'*

**Specific comment 15.** L222: What is an "individual neural network"? Is it the collective

of all zonal neural networks for one simulation (SIM-T1, SIM-T2, ...)? Is the maximum possible number of subnetworks 50.400 or less because of ocean cells?

**Response:** We have changed it to '*independent neural network*' to make it consistent with the expression in the abstract. An independent neural network is the collective of all zonal subnetworks. Several independent neural networks constitute a simulation substep (for example, NN-1-1, NN-1-2, …, NN-1-8 are applied in Round 1- Substep 1), while each substep is responsible for one simulation (there are eight simulations: SIM-1-1, SIM-1-2, SIM-2, SIM-3-1, SIM-3-2, SIM-4-1, SIM-4-2 and SIM-5; for example, SIM-1-1 is the output of Round 1- Substep 1).

The number of subnetworks in each independent neural network is far less than 50400, not only because of ocean zones but also because some soil moisture data are only available in a region (e.g., TMI is available within [-40°S~40°N]). The paragraph has been revised in response to Specific comment 14.

(**Please also note that SIM-1T, SIM-2T, …, SIM-4T are only the postprocessing results that are intended to be used as secondary training targets, while SIM-1, SIM-2, …, SIM-5 are combined to constitute our soil moisture products.)

**Specific comment 16.** L223: For reproducibility, it is required to state exactly the MATLAB version and the toolbox version and method/function name that was used for training the neural network.

**Response:** We have added MATLAB version 2016a accordingly.

**Specific comment 17.** L256: "we classified all pixels" -> "we classified all 0.1 pixels", I suggest to add the resolution information that it is clear which of the different grids is addressed.

**Response:** We have revised it accordingly.

**Specific comment 18.** L259: Again, I thought that a pixel is the smallest unit in the process (i.e. subnetwork). So how can a pixel have a subnetwork? Not clear to me.

**Response:** We apologize for the unclear description. Actually, the subnetwork belongs

to a 1° ×1° zone, not a pixel. We have revised it as follows: '*For data simulation in a 0.1° pixel, the most preferable independent neural network is expected to be trained using all the available soil moisture data sources in that pixel. However, in the 1° zone where it is located, the subnetwork belonging to that preferable independent neural network may not exist due to limited valid data points (see section 2.2.1). Then, an alternative subnetwork driven by the combination of fewer soil moisture data inputs should be applied instead.*'

**Specific comment 19.** L261-262: "Hence, it is a ..." sentence seems incorrect. I think you should better write "neural network collocation" or "neural network constitution" to make it more clear that these are neural network realizations with identical configuration but different ingredients.

**Response:** We have changed it to 'neural network collocation' accordingly.

**Specific comment 20.** L272,815 and other occurrences: it is not clear how the 10 day periods are defined and how they relate to the ordinal numbering. A month has between 29 and 31 days, so how are the periods split and how does that affect the last 3rd where the number of days is variable? How does this variable length averaging affect the results and what are the implications for validation?

**Response:** The first and second 10-day periods in a month both contain exactly 10 days, but the last 10-day period has a variable number of days (9(8)~11). This difference, however, may not have a substantial effect on our results and data quality. This is because it takes at least three days for a microwave sensor to cover the globe. Additionally, for each grid, the days with observations are not the same among different sensors. Therefore, this study only took the average of the available soil moisture retrievals during a 10-day period (we have added a paragraph in section 2.1.1 as: '*To reduce noise and fill the gaps between sensor observation tracks (it takes at least 3 days for a microwave sensor to cover the whole globe), both the daytime and nighttime observations within each 10-day period are combined by data averaging (the relative superiority of daytime and nighttime retrievals is not considered) for every soil moisture*

*product. For example, for SMAP, 11% of the global land surface has data for only 5 days or less within a 10-day period.*' Moreover, surface soil moisture may vary significantly even in a day due to rainfall events, but the observations are transient. Therefore, either the 10-day averaged microwave soil moisture products or the simulated soil moisture data in this study can only roughly indicate the overall soil moisture condition and is not exactly equal to the mean soil moisture during 10 complete days. Hence, it does not matter whether the 'last 10-day period' in a month has exactly 10 days or not. In fact, this data format is exactly the same as that of the ASCAT-SWI soil moisture and many other products (e.g., LAI) developed by the ESA-Copernicus Land Monitoring Service (https://land.copernicus.eu). For the validation process based on ISMN measurements, the mean in situ soil moisture in the 'last 10-day period' of a month was also calculated by averaging the records over either 10 days or 11 days (or 8~9 days in February), which was consistent with the 'nominal' simulation period. Following this comment, we added this information: '*The temporal resolution is approximately 10 days, or to be specific, there are 3 data records within a month, for days 1~10, 11~20 and from 21 to the last day of that month*' to the abstract in the revision.

**Specific comment 21.** L270-292: also this section would greatly benefit from a timeline bar plot that shows all the soil moisture products and simulated models, so that the overlaps can be grasped Immediately.

**Response:** We have added Figure 1, which shows the timelines of the simulated soil moisture corresponding to 8 substeps and the periods of data inputs for the training of 67 independent neural networks. Please find the details in response to Specific comment 7. We also added a sentence: '*The training period for each neural network and the simulation period for each substep are shown in Figure 1.*' in this paragraph.

**Specific comment 22.** L318: define how R**2 is computed (based on Spearman or Pearson).

**Response:** The $R^2$ is computed based on Pearson's correlation, and we have added this

information in the revision.

**Specific comment 23.** L321: lower case r should be used for the correlation coefficient (based on Pearson?). Why are you mixing r and $R^2$ and do not use $R^2$ for all analyses?

**Response:** Following this comment, we have changed R to '$r$' to represent the Pearson correlation coefficient, including those in the figures and tables. In this way, we can better distinguish the correlation coefficient from $R^2$.

To evaluate the overall performance, we showed the scatter plot between the simulated soil moisture and the measured values. Here, instead of $r$, we used $R^2$ to better reveal the differences among the performances of the different soil moisture products. However, in the temporal and spatial validation, at some sites or during specific 10-day periods within a climatic region, the simulations and measurements were negatively correlated, as shown in the figures within the manuscript, which are actually of very low quality. However, if we only use $R^2$, these low-quality data will be overshadowed (for example, if $r$ is -0.6, $R^2$ can be as high as 0.36). Therefore, it is wiser to use temporal correlation and spatial correlation. Previous studies also used '$r$' to evaluate the spatial and temporal accuracy of surface soil moisture products against ISMN measurements, for example (Karthikeyan et al., 2017).

**Specific comment 24.** L322: please provide formula for A.R computation

**Response:** 'A.R' in this study represents the correlation coefficient ($r$) between the anomalies of simulated soil moisture and the anomalies of measured soil moisture at a specific ISMN station. Following this comment, we have added the equation below to show how the anomalies of simulated or measured surface soil moisture were calculated in the revised manuscript.

$$\overline{SSM(k)} = \frac{\sum_{y=1}^{ny} SSM(y, k)}{ny} \ (ny \geq 3); \ SSM \ is \ either \ estimated \ or \ measured$$

$$SSM: surface \ soil \ moisture; k: the \ ordinal \ of \ a \ 10 \ day \ period \ in \ a \ year;$$

$$y: a \ year \ with \ measured \ SSM \ in \ the \ k^{th} \ 10 \ day \ period; \ ny: number \ of \ those \ years$$

$$SSM_{anom}(y, k) = SSM(y, k) - \overline{SSM(t)}$$

$SSM_{anom}(y, t)$: the anomalies of surface soil moisture during the $t^{th}$ 10 day period in year y.

**Specific comment 25.** L326: "in all grids", grids or pixels (1 x 1 or 0.1 x 0.1)?

**Response:** Thank you for your careful reading. We revised it as '*in all 0.1° grids*'.

**Specific comment 26.** L326: please provide formulas for spatial pattern validation (at least in the supplement)

**Response:** We have provided more details for spatial pattern validation. Now it reads: '*Finally, we performed spatial pattern validation. In detail, for every 10-day period, we compared all the soil moisture measurements that were upscaled to 0.1° during that period with the corresponding estimated values. The spatial pattern evaluation indexes include the correlation coefficient (r), RMSE, bias and ubRMSE values (Eq. 2).*'

$$\overline{SSM_{est}} = \frac{\sum_{i=1}^{ng} SSM_{est,i}}{ng} \; ; \; \overline{SSM_{act}} = \frac{\sum_{i=1}^{ng} SSM_{act,i}}{ng} \; (ng \geq 20)$$

$i$: a grid with upscaled surface soil moisture measurements during a specific 10 day period;

$ng$: the number of those grids on the globe

$$ubRMSE_{spatial} = \sqrt{\sum_{i=1}^{ng}[(SSM_{est,i} - \overline{SSM_{est}}) - (SSM_{est,i} - \overline{SSM_{act}})]^2 / ng} \quad \text{(Eq. 2)}$$

**Specific comment 27.** Figure 3: Use identical labels for the x-axis, add missing lower frame.

**Response:** We unified the labels for the x-axis. The figures have been adjusted accordingly. The revised Figure 3 (Figure 5 in the revised manuscript) is shown below:

[Figure]

*Figure R2: Comparison between the temporal accuracy of RSSSM and SMAP in regions with different Köppen-Geiger climate types. The four indexes are (a) r, (b) RMSE, (c) ubRMSE and (d) Anomalies r (A.R). The lengths of the error bars are 1.5 times that of the interquartile range, while the upper and lower boundaries and the central lines of the boxes indicate the 75th, 50th and 25th percentile values, with mean values marked by '×' (the forms of all the following boxplots are the same).*

**Specific comment 28.** Figure 4: If the color key is put below the figure, the figure can be increased in the horizontal direction which leads to wider bars. You could even remove the x-axis labels and names and leave only the lowermost. By this you can increase the size of the bars and hence the readability (reduce redundancy).

**Response:** We moved the color key below the figure and removed the x-axis names to increase the size of the bars (shown in Figure R3, Figure 6 in the revised manuscript). Other figures were also revised and made larger and clearer.

[Figure]

*Figure R3: Comparison between the spatial pattern accuracy of RSSSM and SMAP in different 10-day periods during April 2015~2018. The three evaluation indexes are (a) r, (b) RMSE and (c) ubRMSE. The length of each box/error bar is determined from the evaluation index values in three (January to March) or four (April to December) years.*

**Specific comment 29.** L381: How is the performance of SIM if the SMAP training period is omitted, i.e. from 2003 until 2015D01, as compared to ASCAT-SWI?

**Response:** We have added this information to the manuscript accordingly. It reads as: '*If the data period of SMAP (2015D10~2018) is excluded, the overall $R^2$ and RMSE for RSSSM are 0.43 and 0.087, respectively, which are still better than those for ASCAT-SWI ($R^2$=0.33, RMSE=0.1).*'

**Specific comment 30.** Figure 7,10: As for fig. 4 place color key below the plots and

increase the bars horizontally.

**Response:** We have revised all the figures accordingly.

**Specific comment 31.** Do you see any chance to improve the temporal resolution of the product in the future? If not, what are the constraints?

**Response:** Currently, it is probably not a good choice to further increase the temporal resolution of the long-term microwave surface soil moisture product. As illustrated in the response to Specific comment 20, for each grid, both the days and the hours of the observations by different microwave sensors differ from each other. Because surface soil moisture has high variability in a short time period (even in a day) due to rainfall events, the actual soil moisture at the passing time of various sensors is not the same. However, this study used multiple sources of microwave surface soil moisture products as predictors in the neural networks and SMAP soil moisture data as the training target, meaning that the neural network training should be based on the assumption that the products retrieved by different sensors contain exactly the same actual soil moisture information. As we can see, a conflict exists. To solve this conflict, for each sensor, we took the average of its available retrievals during a certain time period, and the larger the amount of data applied for averaging is, the better the result represents the mean soil moisture during that period. Because 11% of global land has only 5 or fewer days with observations during a 10-day period, if the temporal resolution is improved, for example, to 5 days, there may be only 2~3 observations available. Considering the high temporal variability of surface moisture, the average of those limited data can hardly indicate the average soil moisture condition. This phenomenon will lead to large uncertainties in the neural network training and, finally, our soil moisture simulation results.

This problem is an inherent constraint of microwave remote sensing data integration. Therefore, to improve the temporal resolution, other data sources need to be incorporated. Soil moisture retrieved from other remote sensing techniques (e.g., optical) exhibits low quality over vegetated areas and is heavily affected by clouds. Therefore, model simulation may be the only solution to this problem. For example,

assimilating the observational-based surface soil moisture into models such as GLEAM can achieve surface/root-zone soil moisture mapping at a daily scale. Therefore, we have revised the last paragraph as follows: '*Another way to improve global surface soil moisture data accuracy as well as the temporal resolution is to combine satellite-based products with land surface models such as GLEAM. Remote sensing inversion can delineate more detailed spatial information on soil moisture, whereas reanalysis-based models have advantages in characterizing temporal variations and even on a daily scale, except for…*'.

**Specific comment 32.** L499-500: Is SIM also superior to the other products if only the prior to SMAP period is considered (2003 until 2015D01)?

**Response:** Thank you for this advice. We have added the following comparisons to the manuscript.

*1) If the data period of SMAP (2015D10~2018) is excluded, the overall $R^2$ and RMSE for RSSSM are 0.43 and 0.087, respectively, which are still better than those for ASCAT-SWI ($R^2$=0.33, RMSE=0.1).*

*2) When excluding the SMAP (training target) data period, the $R^2$ and RMSE for RSSSM are 0.41 and 0.089, respectively, which are also superior to those for GLDAS ($R^2$: 0.37; RMSE: 0.099).*

*3) Without considering the SMAP period, the conditions are the same (the $R^2$ values for RSSSM and ERA5-Land are 0.41 and 0.38; the RMSE values for these two products are 0.089 and 0.125, respectively).*

*4) when the SMAP data period is excluded, the $R^2$ and RMSE for CCI are 0.28 and 0.098, compared to 0.41 and 0.089 for RSSSM.*

*5) if the SMAP data period is excluded, RSSSM's $R^2$ and RMSE values are 0.41 and 0.089, respectively, which are still better than both GLEAM v3.3a ($R^2$: 0.35; RMSE: 0.141) and GLEAM v3.3a ($R^2$: 0.34; RMSE: 0.128).*

Therefore, the comparisons above can prove that our product (RSSSM) is superior to the other products even if the SMAP period is excluded.

**Specific comment 33.** Are there plans to update the data-set on a regular basis?

**Response:** Yes. We plan to update the whole dataset when more advanced microwave sensors (e.g., P-band sensors) are launched and global-scale higher-quality surface soil moisture data are available in the future. We have added this information to the discussion. Now it reads: '*Therefore, if microwave sensors with higher SNR or better penetration of vegetation canopy than SMAP are launched in the future (for example, the upcoming P-band microwave sensors (Etminan et al., 2020; Ye et al., 2020)), we can develop a temporally continuous soil moisture dataset beginning in 2003 by using the soil moisture or Tb retrieved from the new sensors as the reference. This newly developed product is expected to have even higher accuracy than the SMAP product (we will update the complete RSSSM product then). In that sense, the data fusion algorithm proposed here will be very meaningful in the future.*'

**Specific comment 34.** The dataset is organized as an archive of geotiff files. The problem with this structure is that the time identifier is only contained in the file name, but without practical formatting. If one wants to import a time series for a region or a single pixel, the data structure is quite unhandy. Also from the readme file and the metadata it is not quite clear what the 10 days ordinal numbering means exactly. Is it always the [1-9],[10-19],[20-29] or [1-10],[11-20],[21-30] periods? How are the months with variable length considered (28,29,30,31 days)? That's not clear also not from the manuscript.

Further, I would suggest to add a table (csv) that links the different file names to their specific period using ISO 8601 https://en.wikipedia.org/wiki/ISO_8601 notation:

e.g., a file named inventory.dat with a list like the following one:

Period, Filename 2003-01-01/2003-01-10, SMY2003DECA01.tif 2003-01-11/2003-01-20, SMY2003DECA02.tif ...

**Response:** Thank you for the suggestion on the naming and structure of our data. Actually, it is [1-10], [11-20], [21, the end of each month]. We have added a csv table named 'filename' linking the different file names to their specific period, following your instructions.

**Specific comment 35.** Also the numbering should be formatted as %02d so that, e.g., SMY2003DECA1.tif becomes SMY2003DECA01.tif. This is important if one wants create a chronological file list for looping over time. With the current scheme, the order would become SMY2003DECA1.tif SMY2003DECA10.tif, SMY2003DECA11.tif, ... This should be also applied to all tables in the manuscript (e.g., 2005D01 instead of 2005D1).

**Response:** We have changed the naming of the product as well as the abbreviations for each 10-day period, both in the manuscript and in the Supplement.

**Specific comment 36.** Figure S1: The figure and description is not completely clear. I assume that every number (yellow and blue frames) is one pixel (0.1x 0.1)? I think it would become more clear if you superimposed a light gray mesh for the pixels over the 1x1zones. But then, why are there 4 steps required to smooth the borders? It means that every boarder gets smoothed twice, and every corner point even four times.

**Response:** We apologize for the unclear description. The figure has been revised (see Figure R4), with light gray mesh superimposed for all the pixels over one 1°×1° zone. The four steps were used to process the four borders of each 1°×1° zone (please note that in each step, only the border colored in blue is smoothed). We have added more details to clarify this information, which now reads as: '*A sketch of the four substeps in boundary fuzzification. The 1°×1° zones are separated by solid black lines (the 0.1°×0.1° pixels in one zone are superimposed by light gray mesh). For each substep (subfigures a~d), the soil moisture value within each pixel that is colored in blue is recalculated as the average of its original surface soil moisture and the original soil moisture value in its most adjacent yellow color pixel, weighted by the corresponding numbers labeled (i.e., 2 and 1). In this way, every border of a 1°×1° zone gets smoothed once (substeps 'a~d' are for four borders, respectively, where a~b are for the horizontal borders while c~d are for the vertical borders), but the four corners get smoothed twice (both horizontally and vertically).*'

[Figure]

*Figure R4. A sketch of the four substeps in boundary fuzzification. The 1°×1° zones are separated by solid black lines (the 0.1°×0.1° pixels in one zone are superimposed by light gray mesh). For each substep (subfigures a~d), the soil moisture value within each pixel that is colored in blue is recalculated as the average of its original surface soil moisture and the original soil moisture value in its most adjacent yellow color pixel, weighted by the corresponding numbers labeled (i.e., 2 and 1). In this way, every border of a 1°×1° zone gets smoothed once (substeps 'a~d' are for four borders, respectively, where a~b are for the horizontal borders while c~d are for the vertical borders), but the four corners get smoothed twice (both horizontally and vertically).*

**Specific comment 37.** Figures S5, S8, S11: put the color-key to the bottom of the figure (a single key would be sufficient for all sub-figures), you could even remove the x-axis

labels and names and leave only the lowermost. By this you can increase the size of the bars and hence the readability (reduce redundancy).

**Response:** We have revised these figures, making the size of the bars much larger now by adjusting the locations of the color keys and removing the x-axis names.

**Specific comment 38.** There are often blanks missing between words. DOIs are completely missing in the reference list.

**Response:** We apologize for these mistakes. We have added the missing blanks and rewritten the reference list (added DOIs, abbreviated the journal names, corrected the incorrect references) to ensure that it meets the format requirement of ESSD. Thank you again for your careful reading and valuable suggestions.

**References**

[revised manuscript text omitted]

---

## Author Comment (AC2) · 14 Aug 2020

**To Reviewer #2:**

We thank referee #2 for the valuable comments that will help us improve the quality and readability of the manuscript. We have carefully revised the MS following your comments and suggestions. We provide a detailed response to the Referee's comments in the Supplement.

**\*\*\*\*\*\*\*\*\*\***

**General comment.** The authors tried to generate long-term surface soil moisture at a global scale, via data fusion of 11 microwave remote sensing-based soil moisture products since 2003 through neural network approach, and SMAP soil moisture products were used as the training target. The idea is very interesting and should be encouraged to explore further how much extent the machine learning can help in Earth Observation for delivering physically-consistent (or physic-aware) products. However, the way the current manuscript is written, organized is still far from clarity, structured for this reviewer to comprehend their contributions. I would suggest rejection and encourage the author to continue along this line of effort.

**Response:** We thank the reviewer for the positive comment on the idea of generating long-term surface soil moisture at a global scale in this study. We agree that the organization of this manuscript, especially the explanation of the methods, is not clear and we apologize for some confusing or incorrect terminologies. These problems will make the readers and reviewers misunderstand several rather complex algorithms, which are also the major innovation points of this study, including the selection of quality impact factors as neural network inputs and the design of five rounds of simulations (the organization structure of 67 independent neural networks). We have carefully followed your advice, revising the explanations of the key methods. In addition, for each of your doubts or queries, we explained our design of the method in greater detail and clarified the related sentences in the manuscript, hoping that our real

contributions could be comprehended by you and other readers. Please find the details in the response to each of the following comments.

**Major comment 1.** The author claimed that "This new dataset, once validated against the International Soil Moisture Network (ISMN) records, is supposed to be superior to the existing products (ASCAT-SWI, GLDAS Noah, ERA5-Land, CCI/ECV and GLEAM), and is applicable to studying both the spatial and temporal patterns. " This assumption is too strong. On the other hand, it seems the author referred to the validation of the NN-based 10-d soil moisture products versus the 10-d averaged ISMN in-situ observations (as seen in Figure 5, Figure 8, Figure S3, S6, and S9). Is it true? In any case, it should be specified under what conditions the generated product is performing better than other products. "supposed to be superior" is really not a scientific statement.

**Response:** Thank you for this comment. In this study, for our product (named RSSSM hereinafter) and each existing product, by referring to all valid ISMN sites' surface soil moisture measurements, we carefully conducted overall validation (evaluation indexes are overall $R^2$ and RMSE values), temporal variation validation (evaluated by temporal correlation coefficient, temporal RMSE and unbiased RMSE, etc.) and spatial pattern validation (evaluated by the spatial correlation coefficient, spatial RMSE and unbiased RMSE, etc.). Please see section 2.3 in the Methods in the revised manuscript for details, while the accuracy comparison among all products is in section 3.2 of the results. The validation results indicate that our RSSSM product is comparable to the site measurements, in terms of both $R^2$ and the RMSE (see Figure 7, Figure 10, Figure S3, Figure S6, Figure S9 in the revised manuscript and Supplementary). For temporal variation accuracy, RSSSM is proven to be better than ASCAT-SWI, GLDAS and CCI, both in temporal correlation and RMSE, especially in arid regions and relatively cold areas (Figure 8, Figure 11, Figure S7 and Table 2). The temporal accuracy of RSSSM is similar to that of ERA5-Land and GLEAM v3.3 products (the temporal correlation is somewhat lower, but the temporal RMSE value of our product is lower, see Figure S4, Figure S10 and Table 2). For spatial pattern accuracy, our RSSSM product is

superior to all other products (please refer to Figure 9, Figure 12, Figure S5, Figure S8, Figure S11 and Table 3) throughout almost the entire year, especially during the growing seasons. Based on these findings, we propose that our product (RSSSM) has better agreement with the site-measured surface soil moisture than the five existing soil moisture products (ASCAT-SWI, GLDAS Noah, ERA5-Land, CCI/ECV and GLEAM). Moreover, the observational-based soil moisture, CCI, has limited spatial coverage and significantly reduced data accuracy before 2012, while ASCAT-SWI is only available since 2003. These problems have been well solved by our estimation (the data quality is maintained during 2003~2018, see Figure 13). We agree with you that the phrase 'supposed to be superior' is not a scientific statement, and the claim is probably too strong and condescending. Therefore, we corrected the sentences as follows: '*This new dataset, named RSSSM, is comparable to the in situ surface soil moisture measurements at the International Soil Moisture Network sites (overall $R^2$ and RMSE values of 0.42 and 0.087 $m^3/m^3$), while the overall $R^2$ and RMSE values for the existing products (ASCAT-SWI, GLDAS Noah, ERA5-Land, CCI/ECV and GLEAM) are within the range of 0.31~0.41 and 0.095~0.142 $m^3/m^3$, respectively. The advantage of RSSSM is especially obvious in arid or relatively cold areas and during growing seasons. Moreover, the persistent high data quality as well as complete spatial coverage ensure the applicability of RSSSM to studies on both spatial and temporal patterns.*' We have also corrected all the relevant unclear statements (e.g., supposed superior, expected to be better, …) throughout the manuscript.

**Major comment 2.** There were some strange 'terminologies' the author used for discussion, for example: a. 'penetrability of microwave' (which is seldom found in the literature. A more widely used term is 'microwave penetration depth'); b. "Soil moisture retrieval from passive microwave sensors is based on the correlation between soil dielectric conductivity, that is influenced by soil moisture …'. Following the theoretical development of soil moisture retrievals from remote sensing, the relationship between soil moisture and dielectric constant is the fundamental (not soil dielectric conductivity). **Response:** We apologize for the unsuitable terminologies. We corrected the sentence

'the penetrability of microwaves is usually <5 cm of soil' to '*current satellite microwave sensors can detect only soil moisture within the top 5 cm of soil*' following this comment and your Specific comment 5, and corrected '*dielectric conductivity*' to '*dielectric constant*'.

**Major comment 3.** "However, this data is regional, with a large temporal gap, and cannot be seen as observational-based only since precipitation data is incorporated." This is a very strange argument. We all know there is a strong link between precipitation and soil moisture variation. Physically speaking, one used the antecedent precipitation index to understand how precipitation events drive the variation of soil moisture. This is like one of 'quality impact factors'. If the above argument is true, we can argue that the author's approach in this manuscript is also not 'observation-based', as they used LAI, land cover, LST, and many other factors.

**Response:** We checked the work by *Qu* et al. (Qu et al., 2019) carefully and found that precipitation data were actually not applied as an input for the random forest in that study.

Furthermore, other recent studies focusing on long-term soil moisture mapping based on microwave remote sensing data also did not incorporate precipitation as ancillary inputs for the neural networks (Santi et al., 2016; Yao et al., 2019; Yao et al., 2017).

Hence, we conducted research on the role of precipitation in neural network training to further explore the reasons.

Because it takes at least 3 days for a microwave sensor to cover the globe, for 11% of global land, there will be only 5 or fewer observations for random days within a 10-day period. By taking the average of these available data, this study focuses on only the mean soil moisture condition during that 10-day period. Then, to see how much the incorporation of precipitation data can improve the neural network training efficiency, we calculated the 10-day averaged GPM Final-Run precipitation, which can indicate the overall precipitation water availability (the antecedent precipitation index is not used because it must be calculated on a daily scale, and the attenuation coefficient is difficult to determine at a global scale (Kohler and Linsley, 1951)). Taking the first

primary independent neural network, NN1-1-1, as an example, we performed contribution tests on all the input features at the global scale (not for each separate zone), including 9 'quality impact factors', 4 soil moisture predictor products and precipitation - a potential ancillary soil moisture indicator. For each predictor, we added a random error that is controlled within the standard deviation of the predictor, and then an increase in MSE during neural network training can indicate the relative contribution of that variable. The results (see Figure R1a, that is Figure S1a in the revised Supplement) show that precipitation will contribute to only 1.7% of the training efficiency, which is much lower than the contribution of any soil moisture product (the total contribution fraction of the four soil moisture products is 61.2%) and is also lower than that of most 'quality impact factors'. This result suggests that microwave soil moisture datasets together with several 'quality impact factors' of microwave soil moisture retrieval are enough to predict the training target, SMAP soil moisture, and there is no need to add precipitation as another ancillary index of soil moisture.

'Quality impact factors' are defined in this study as the variables that will have a significant impact on the retrieval errors of soil moisture by microwave remote sensing (section 2.1.2). Although the relative performances of different soil moisture products are related to surface moisture conditions (Kim et al., 2015), it is found mainly due to the less vegetation in arid areas. After all, no explicit mechanism can support the idea that the retrieval errors of soil moisture are significantly influenced by water availability. Even if this is true, the soil water availability can already be indicated by the microwave soil moisture products. Therefore, it is unreasonable to incorporate the precipitation variable as a 'quality impact factor'. On the other hand, LAI, water area fraction, LST, land use cover, tree cover fraction, non-tree vegetation fraction, topographic complexity, and soil sand/clay fractions all have direct impacts on the microwave soil moisture retrieval errors, with solid physical mechanisms (see section 2.1.2). Therefore, theoretically, these variables should be added to the neural network, even though the land use cover type and soil sand fraction data have been proven to have limited contributions to NN training efficiency.

One may argue that if NARX (nonlinear autoregressive with external input) is applied

instead, in which the soil moisture in the previous 10-day period is also incorporated as a predictor, precipitation data can be very beneficial to neural network training. This result is true because precipitation directly contributes to increases in soil moisture. However, NARX is not suitable for global-scale long-term continuous soil moisture mapping because the base map (i.e., the soil moisture at the beginning of the simulation period) is difficult to determine. Moreover, in mid to high latitudes, the lack of soil moisture retrievals over frozen ground in winters will lead to missing data there in summers when soil moisture data are otherwise available. Therefore, if NARX is adopted, we can only estimate long-term soil moisture in the tropics and subtropics with air temperatures consistently higher than 0 ℃. Finally, if the soil moisture in the previous phase and the current precipitation amount are both incorporated, they will largely conceal the role of satellite-observed signals. As shown in Figure R1b (Figure S1b in the Supplement), the total contribution fraction of all four microwave soil moisture products is reduced to only 10.6%, while the roles of ASCAT, AMSR2-JAXA and AMSR2-LPRM are all negligible. Without taking full advantage of remote sensing, simulations based on previous soil moisture and current precipitation products will lead to errors in regions where soil moisture gains are mostly driven by glacier melting or in places with high levels of radiation-driven surface soil evaporation. The reliability of the derived soil moisture will be reduced in irrigated croplands and afforestation/deforestation areas as well.

[Figure]

[Figure]

*Figure R1: (Figure S1 in the revised Supplement): The roles of different input features in the soil moisture simulations based on BP neural networks and nonlinear autoregressive with external input (NARX) with microwave soil moisture products incorporated: (a) the contributions of different input features of a primary neural network: NN1-1-1, including 4 predictor soil moisture products, 9 quality impact factors of microwave soil moisture retrieval, plus 1 probable ancillary soil water indicator: 10-day averaged precipitation, to the neural network training efficiency indicated by the increased MSE; (b) the contributions of all the input features to the training efficiency if NN1-1-1 is changed into a NARX, in which the SMAP soil moisture for the previous period is also applied as a predictor.*

On account of all of the above, precipitation data are neither included as an ancillary soil moisture indicator nor added as a 'quality impact factor' in this study.

Following this comment, we have added the above explanations to the Supplementary Data (Text S1), replacing the previous paragraph. We also added a sentence in the revised manuscript: '*The contribution analysis results (Figure S1) show that because various microwave soil moisture data have already been included, precipitation data are not an essential indicator of soil moisture and are not utilized as a physically based 'quality impact factor' either (see Text S1 for detailed explanations).*'

In Qu et al.'s (2019) study, the random forest input features include only microwave Tb products, DEM, IGBP global vegetation classification, latitude, longitude, and DOY.

Therefore, we apologize that the latter half of this sentence was not correct, which we have deleted. The sentences now read: '*Another study rebuilt a soil moisture time series over the Tibetan Plateau by using SMAP data as the reference data for a random forest model (Qu et al., 2019). For the environmental factors, while vegetation cover was not considered, elevation (DEM), IGBP land use cover type, grid location and the day of a year (DOY) were chosen as ancillary inputs. The training $R^2$ in this region reached 0.9, with a temporal accuracy higher than that of other products (temporal r=0.7; RMSE=0.07 in the unfrozen season). However, these data are regional (for only the Tibetan Plateau) and suffer from the temporal gap between AMSR-E and AMSR2 data (October 2011~June 2012).*'.

**Major comment 4.** " are these factors used as direct spatial predictors of soil moisture or just because they are related to the errors of satellite soil moisture retrievals (i.e., the quality impact factors of soil moisture)? We insist on the latter, proposing two main reasons for the incorporation of environmental factors." This is very confusing and not necessarily correct, and not well grounded. We know the soil moisture retrieval from remote sensing is using a radiative transfer model to account for scattering and emissions from both soil and vegetation, which is conflicting with the author's statements.

**Response:** We apologize for the unsuitable sentences. We have rewritten them as follows: '*Environmental factors, including DEM, LST and vegetation cover (indicated by NDVI, MVI, etc.), were used as ancillary neural network inputs to improve the soil moisture simulation (Lu et al., 2015; Qu et al., 2019; Yao et al., 2017). According to these studies, these factors alone may not predict surface soil moisture well without the incorporation of any microwave remote sensing data, which can also be justified by the contribution analysis results (Figure S1a). This phenomenon occurs because although these data are somewhat related to soil moisture (e.g., soil moisture is generally limited in areas with low vegetation cover but high in forests (McColl et al., 2017)), the relationships are rather uncertain (e.g., at small scales, leaf area index (LAI) may have a negative influence on soil moisture due to the variation in evapotranspiration*

*(Naithani et al., 2013) or may not have clear impacts (Zhao et al., 2010); also, soil moisture can be either high or low in summers when vegetation peaks (Baldocchi et al., 2006; Méndez-Barroso et al., 2009)). However, these factors are essential due to their direct impacts on soil moisture retrieval through the radiative transfer model using microwave remote sensing data (Fan et al., 2020) and are factors that impact the retrieval quality. The detailed explanations are as follows: 1) …'*. On the other hand, we agree that the precipitation and open water fraction can directly indicate the surface soil moisture (but because the microwave soil moisture products are already applied, precipitation data were not included as a predictor; see the detailed explanations in response to Major comment 3).

We understand that soil moisture retrieval from microwave remote sensing accounts for the scattering and emissions from both soil and vegetation, so these factors can have direct impacts on the microwave soil moisture retrieval results and are closely related to the retrieval errors, as we explained in the revision.

**Major comment 5.** 'Water Body' was used as one of the predictors (it should be predictor, rather than quality impact factors). This is very strange. As we know, water body map in either SMOS or SMAP soil moisture products were used to mark out those locations to avoid soil moisture retrievals over these water bodies (otherwise, it would be physically no sense, in terms of soil moisture). This is wrong and not physically sound to include water bodies as one of predictor for predicting surface soil moisture.

**Response:** We apologize for the confusion. We agree that water is a direct indicator of surface soil moisture. However, 'water fraction', rather than 'water body', is used as both a quality impact factor and a potential indicator of surface soil moisture content. Water bodies (large lakes, oceans) are masked out in both existing (SMOS or SMAP) soil moisture products and our simulation product (RSSSM). However, in a grid with a size of $0.1° \times 0.1°$ (approximately 120 km$^2$), there could be a small fraction of water, which may be due to the presence of rivers, streams, ponds, partly inundated wetlands or paddy croplands. If we mask out all 0.1° resolution grids with even 1% water (or less), there will be no data in many parts of the world, especially over humid areas.

Previous soil moisture products also produce valid values in those grids. However, water can dramatically decrease the brightness temperature (Tb), while different retrieval algorithms correct the impact of water within the grid differently, leading to different biases and relative accuracy of soil moisture estimates in grids with water (Ye et al., 2015). For example, we noted a strong underestimation of soil moisture by the NSIDC (National Snow and Ice Data Center) method (e.g., AMSRE-NSIDC) over rivers and small lakes when compared to nearby lands. Moreover, the sensitivity of different microwave sensors to the water fraction within the grid may differ as well. Hence, the fraction of water (not only open water but also inundated wetlands or croplands) in the grid can significantly influence the retrieval errors and relative reliability of various soil moisture products (Ye et al., 2015), which exactly meets the definition of 'quality impact factor'.

Therefore, this study uses 'water fraction' as both a quality impact factor and an ancillary soil moisture indicator (please also note that the word 'predictor' in this manuscript refers to only the existing soil moisture products that are applied as the neural network inputs). This information has been added to the manuscript. For the 'water fraction', the Surface WAter Microwave Product Series (SWAMPS) dataset (Schroeder et al., 2015) was applied because it is microwave based, including not only open water but also partly inundated lands. The contribution analysis on all the input features (see Figure R1) proves that the calculated water fraction plays a very important role in neural network training.

Following this comment, we have clarified the descriptions. It now reads: '*The second factor is the 'water fraction factor' (i.e., the fraction of water area in each pixel). Waters in land pixels dramatically decrease the Tb, leading to overestimation of soil moisture. Because different methods are used to detect and correct small areas of water, either open water, wetlands or partly inundated wetlands and croplands (Entekhabi et al., 2010; Kerr et al., 2001; Mladenova et al., 2014; Njoku et al., 2003), microwave soil moisture data calibration and weight assignment based on the water fraction within land pixels make sense (Ye et al., 2015). In addition, the water fraction is a direct indicator of surface soil moisture. In this study, the daily water area fraction derived*

*from the Surface WAter Microwave Product Series (SWAMPS) v3.2 dataset (Schroeder et al., 2015) is applied.*' The confusing words 'water body' have been replaced by '*water area fraction*' in other parts of the manuscript as well.

**Major comment 6.** For 'topographic complexity' 'soil texture', the author used from different sources, one from ASCAT ancillary data and the other use SMAP ancillary data. This reviewer is wondering why such a choice? Why not making it consistent (i.e., get ancillary data from one single product, instead of two?

**Response:** The reason we used different sources of data for 'topographic complexity' and 'soil texture' was due to data availability. SMAP uses only GMTED 2010 DEM data to derive quality flags for data retrieved in mountainous areas, while topographic complexity is included as only the ancillary data of ESA's ASCAT-SWI product, which was calculated by normalizing the standard deviation of GTOPO30 elevation in each grid point to values from 0 to 100 (Scipal et al., 2005) and is closely related to errors of surface soil moisture retrieved from microwave remote sensing. On the other hand, soil texture data are not included in the static layers of the ASCAT-SWI product, so we had to obtain these important data from SMAP ancillary input collection. Moreover, because the data sources of topographic complexity and soil texture both have relatively high quality, we suggest that they could be used even if they come from different soil moisture products. In the revised manuscript, we added this information: '*(topographic complexity data are not available from SMAP Constant; soil texture is not provided by ASCAT Constant)*'.

**Major comment 7.** '3σ denoise'. what is the effect of such a filter on identifying extreme years? For example, during 2003, 2010, 2018, 2019 there are extreme heat events in Europe and the soil moisture is so dry which can be beyond the 3 standard deviations.

**Response:** The '3σ denoising' was conducted spatially, rather than temporally, to detect and delete the extreme values (usually salt and pepper noises in mountain areas) in each 1°×1° zone during a certain 10-day period. Therefore, if there are extreme heat or

precipitation events, as you noted, the whole 1°×1° zone will exhibit a sharp increase or decrease in soil moisture content in almost all the 0.1° grids within that zone (there are <100 grids in each zone). Therefore, due to the increase/decrease in the zonal mean soil moisture value, extreme weather events will not be removed by this 'spatial 3σ denoising' step.

We apologize for the confusion. We have added more detailed explanations. It reads as: '*After standardization of the original soil moisture data, to improve the neural network training efficiency, the potential salt and pepper noises are removed. For each map (a specific 10-day period), within each 1°×1° zone, the soil moisture values are filtered to the level of three standard deviations relative to the mean in that zone. This preprocessing step is thus called '3σ denoising' (note that denoising is conducted spatially, rather than temporally, so that the extreme events will not be treated).*'

**Major comment 8.** NN design. SMAP is only available after 2015, so I am not sure what is the meaning of simulation period 2012D19-2013D36, but also 2014-2018. I guess this is constrained by the available data (PROBA-V and GLASS LAIs)? But in any case, it does not represent any physical meaning to predict 2015data with 2012-2013 data. At least, the NN design is not clear on why it is designed as such.

**Response:** We apologize for the confusion. In the first round of simulation, the division of the simulation period into two subperiods, 2012D19~2013 and 2014~2018, is due to the available data periods of PROBA-V and GLASS LAI. However, we did not predict data in 2015 by using the data from 2012~2013. In this study, the common period for the predictor soil moisture products applied in each 'substep' of NN training always includes the corresponding soil moisture simulation period.

We agree that the design of 67 independent neural networks, which are embedded in 8 substeps applied for five rounds of simulations, is quite complex. However, it ensures long-term continuous satellite-based soil moisture mapping, almost full spatial coverage at the global scale, and high data accuracy. Following this comment, we have revised this section in the manuscript to clarify the NN design. It reads as:

[revised manuscript text omitted]

The following is a plain language description, which you may choose to read if would

like to understand the NN design deeper.

First, considering that the temporal spans of different microwave sensors are all limited (see Figure R2 for details), we designed five rounds of neural networks to achieve long-term continuous soil moisture mapping while ensuring that as many microwave soil moisture products as possible are applied as predictors in each round of the NN. In detail, SMAP soil moisture data are used as the training target of the first round NN (labeled NN1), with ASCAT-SWI, SMOS, AMSR2-JAXA and AMSR2-LPRM-X applied as soil moisture predictor products. The potential training period of NN1 is the time period of SMAP (2015D10~2018, Table S1). Because the four soil moisture predictors all have data since 2012D19, the potential soil moisture simulation period is 2012D19~2018. However, because PROBA-V LAI (quality impact factor) starts in 2014, the neural networks trained using PROBA-V LAI can be used for the simulation only during 2014~2018. For the remaining period (2012D19~2013), the applicable neural networks should be trained based on another LAI dataset, GLASS LAI, which covers the time from the beginning of our study period until 2017. Therefore, NN1 should be divided into two substeps. For substep 1 (marked by NN1-1), PROBA-V LAI is used, and the training period is 2015D10~2018D36 (Table S1), while the simulation period is 2014~2018 (Table S2). For substep 2 (denoted by NN1-2), GLASS LAI is applied instead, and the training period is 2015D10~2018D36, while the simulation period is 2012D19~2013. Because each predictor soil moisture product has missing values in some specific areas (e.g., SMOS-IC does not have values in Eurasia), there are 1~4 predictor soil moisture products available in every 0.1° grid. While the maximum number of combinations is 4+3+2+1=10, 8 of them are valid since the soil moisture retrievals over snow or ice are not recommended (Table S2). Corresponding to these 8 combinations, 8 independent neural networks are trained, each with a combination of predictor soil moisture products applied as neural network inputs (labeled NN1-1(2)-1 ~ NN1-1(2)-8; for example, NN1-1(2)-1 is trained using all four soil moisture predictor products and is the most preferable NN). However, even for a 0.1° grid with all four predictor soil moisture data available, we may not be able to simulate soil moisture there using NN1-1(2)-1. This problem occurs because the corresponding neural network, NN1-1(2)-1, may not exist in the 1°×1° zone where the grid is located due to limited valid data points available for zonal subnetwork training (please refer to revised Method section 2.2.1 for details on the localized neural networks). Under this condition, the other less preferable independent neural networks should be applied instead (the relative priority

order of all independent neural networks within a substep is determined by comprehensively considering the number and quality of input soil moisture products, the variety of sensors, the quantity of training samples indicated by the number of 10-day periods, and the relative accuracy of training targets). After simulation, we combined the results for substep 1 (NN1-1-1~8), which is denoted by SIM-1-1, and the results for substep 2 (NN1-2-1~8), which is denoted by SIM-1-2, to obtain SIM-1. After further processing steps (section 2.2.2), we convert SIM-1 into the secondary training target, SIM-1T. For the second round of NN, the training target can be either SMAP (primary training target), while the training period is 2015D10~2017 (GLASS LAI is used, ASCAT-SWI, SMOS and FY data products are applied as predictors), or SIM-1T (secondary training target), while the training period is 2012D19~2015D10 (ASCAT-SWI, SMOS, FY and TMI products can all be applied). There are 8 independent neural networks included in the round 2 NN (see Table S3), while the corresponding simulation output is SIM-2, covering the period from 2011D20~2012D18 since the FY data product has been available since 2011D20 (see Table S4).

The 3$^{rd}$ to 5$^{th}$ round of neural network training and simulations are even more complex (for example, in the 3$^{rd}$ round, the priority order of independent neural networks is not definite. Two probable orders are provided, leading to two substeps, the simulation results of which are combined by taking the relative accuracy in each grid into account), but the basic principles are similar to those explained above (see Table S5~S15).

**Specific comment 1.** In the abstract, change 'elaborate' to 'elaborated', delete 'various', change 'simulation' to 'simulations'

**Response:** We have corrected them accordingly.

**Specific comment 2.** In the abstract, 'This new dataset, once validated against the International Soil Moisture Network (ISMN) records, is supposed to be superior to the existing products (ASCAT-SWI, GLDAS Noah, ERA5-Land, CCI/ECV and GLEAM), and is applicable to studying both the spatial and temporal patterns.' This is a very strong assumption, and should be avoided. Otherwise, the corresponding results should be shown.

**Response:** We have corrected the unsuitable statement as follows: '*This new dataset,*

*named RSSSM, is comparable to the in situ surface soil moisture measurements at the International Soil Moisture Network sites (overall $R^2$ and RMSE values of 0.42 and 0.087 $m^3/m^3$), while the overall $R^2$ and RMSE values for the existing products (ASCAT-SWI, GLDAS Noah, ERA5-Land, CCI/ECV and GLEAM) are within the range of 0.31~0.41 and 0.095~0.142 $m^3/m^3$, respectively. The advantage of RSSSM is especially obvious in arid or relatively cold areas and during growing seasons. Moreover, the persistent high data quality as well as complete spatial coverage ensure the applicability of RSSSM to studies on both spatial and temporal patterns.*' Please also find the details in the response to Major Comment 1.

**Specific comment 3.** Lines 27~30: 'It has been endorsed by the Global Climate Observing System (GCOS) as an essential climate variable (Bojinski et al., 2014), probably the best indicator of ecological droughts (Martínez-Fernández et al., 2016; Samaniego et al., 2018). However, due to the large uncertainty in global-scale soil moisture data, its applicability in global ecosystem models are currently limited (Hashimoto et al., 2015; Stocker et al., 2019).' What do you want to say here? What are points? It is suggested to shorten the sentence.

**Response:** We apologize for the too complicated sentences. We have shortened the sentence as: *'Soil moisture has been endorsed by the Global Climate Observing System (GCOS) as an essential climate variable (Bojinski et al., 2014), as it is probably the best indicator of ecological droughts(Martínez-Fernández et al., 2016; Samaniego et al., 2018). However, due to the large uncertainty in global-scale soil moisture data, the applicability of these data in global ecosystem models is currently limited (Hashimoto et al., 2015; Stocker et al., 2019).'*

**Specific comment 4.** Lines 31~33: 'The reanalysis land surface model products (e.g., the Global Land Data Assimilation System (GLDAS, with spatial resolution of 0.25°) (Rodell et al., 2004), ECMWF ERA-interim (0.25°) (Balsamo et al., 2015) and its newly-published successors: ERA5 (0.25°) and the land product, ERA5-Land (0.1°)(Hoffmann et al., 2019)) are the most frequently used.' The sentence seems not

completed.

**Response:** We have revised this complicated sentence as follows: '*Reanalysis-based land surface model products are the most frequently used, mainly including the Global Land Data Assimilation System (GLDAS, with 0.25° resolution) (Rodell et al., 2004), European Reanalysis (ERA)-interim (0.75°) (Balsamo et al., 2015) and its successors - ERA5 (0.25°) and ERA5-Land (0.1°) (Hoffmann et al., 2019).*'

**Specific comment 5.** Lines 40: 'the penetrability of microwaves is usually <5 cm of soil'. This is particularly not true for L-band passive microwave like SMOS, SMAP, which are dedicated to soil moisture monitoring. 'Penetrability' is usually called penetration depth.

**Response:** Following this comment, we have carefully checked this information and determined that the L-band microwave is sensitive to the soil moisture within only <5 cm of surface soil. For example, 'L-band, the brightness temperature emission originates from the top ~5 cm of soil' for SMAP (Entekhabi et al., 2010). 'At L-band soil moisture in the first centimeters (typically 5 cm) impacts significantly on the emitted brightness temperature' for SMOS (Kerr et al., 2001). This depth is, however, larger than the observation depth of higher frequency microwaves, which is 1 cm (C band) or less (Piles et al., 2018). We agree that the word 'penetrability' is not scientific, while 'penetration depth' is not very accurate as well. Therefore, we have changed the sentence to '*current satellite microwave sensors can detect only soil moisture within the top 5 cm of soil)*' for clarity.

**Specific comment 6.** Line 45: change 'frommicrowave' to 'from microwave'.
**Response:** We have made the revision accordingly.

**Specific comment 7.** Line 45: 'Currently, the longest continuous record of global soil moisture retrieved frommicrowave remote sensing only is the ASCAT product'. Change 'only is'.
**Response:** Following this comment, we have corrected the sentence as: '*Currently, the*

*ASCAT product represents the longest continuous record of global surface soil moisture that is derived from only microwave remote sensing.*'

**Specific comment 8.** Line 63~64: 'Upon rescaling, the spatial patterns of the satellite products are almost replaced by those of GLDAS.' , Any citations?

**Response:** We have added the citations of (Gruber et al., 2019; Liu et al., 2012; Liu et al., 2011). Moreover, the statement has been changed to: '*Upon rescaling through CDF matching, the spatial patterns of the satellite products are generally replaced by those of GLDAS (Gruber et al., 2019; Liu et al., 2012; Liu et al., 2011).*'

**Specific comment 9.** Line 66~67: 'Because the temporal variation in soil moisture is often better captured by model simulations than remote sensing inversions, CCI may undesirably combine the disadvantages of both.' How? Any proof?

**Response:** We have deleted the unsuitable description accordingly. There is no strong evidence for the claim that the temporal variation in soil moisture is better captured by model simulations than remote sensing inversions.

**Specific comment 10.** Change 'deviationsto' to 'deviations to'.

**Response:** We have checked the manuscript carefully and added all missing blanks.

**Specific comment 11.** Lines 90~91: 'The training $R^2$ is only 0.45 (R=0.67)'. $R^2$ and R needs to specify.

**Response:** Thank you for this suggestion. We have revised it as follows: '*The training R-square value ($R^2$) of this product was only 0.45 (or correlation coefficient, r, equals 0.67).*' Following this comment, we have also corrected all the 'R' to '*r*' to distinguish it from $R^2$ in the revised manuscript.

**Specific comment 12.** Lines 99~100: 'this data is regional, with a large temporal gap, and cannot be seen as observational-based only since precipitation data is incorporated.' Well, this is arguable.

**Response:** Accordingly, the sentence has been revised to '*However, these data are regional (for only the Tibetan Plateau) and suffer from the temporal gap between AMSR-E and AMSR2 data (October 2011~June 2012).*' We have also rewritten Text S1 to more clearly explain why precipitation was not included as an input feature in the neural networks in this study. Please also find the details in the responses to Major Comment 3.

**Specific comment 13.** Lines 148~150: 'are these factors used as direct spatial predictors of soil moisture or just because they are related to the errors of satellite soil moisture retrievals (i.e., the quality impact factors of soil moisture)?' What do you mean?

**Response:** We have revised the sentences to '*Environmental factors, including DEM, LST and vegetation cover (indicated by NDVI, MVI, etc.), were used as ancillary neural network inputs to improve the soil moisture simulation (Lu et al., 2015; Qu et al., 2019; Yao et al., 2017). According to these studies, these factors alone may not predict surface soil moisture well without the incorporation of any microwave remote sensing data, which can also be justified by the contribution analysis results (Figure S1a). This phenomenon occurs because although these data are somewhat related to soil moisture (e.g., soil moisture is generally limited in areas with low vegetation cover but high in forests (McColl et al., 2017)), the relationships are rather uncertain (e.g., at small scales, leaf area index (LAI) may have a negative influence on soil moisture due to the variation in evapotranspiration (Naithani et al., 2013) or may not have clear impacts (Zhao et al., 2010); also, soil moisture can be either high or low in summers when vegetation peaks (Baldocchi et al., 2006; Méndez-Barroso et al., 2009)). However, these factors are essential due to their direct impacts on soil moisture retrieval through the radiative transfer model using microwave remote sensing data (Fan et al., 2020) and are factors that impact the retrieval quality. The detailed explanations are as follows: 1) ...*'.

**Specific comment 14.** Lines 178: 'Water bodies dramatically lower the Tb, leading to overestimation of soil moisture'. water bodies are marked/flagged out for soil moisture

retrieval. The consideration of water bodies in your approach seems very strange.

**Response:** We have replaced 'water bodies' with 'water area fraction' for clarity. Please also refer to our response to Major Comment 5.

**Specific comment 15.** Line 182, are you sure it is soil dielectric conductivity, not soil dielectric constants?

**Response:** We have corrected 'soil dielectric conductivity' to '*soil dielectric constant*' accordingly.

**Specific comment 16.** Lines 199~200, 'For topographic complexity, the static layer of the Copernicus ASCAT-SWI product (hereinafter the ASCAT Constant) is adopted while for soil texture, the SMAP Constant is used.' Why not using static layers from the same satellite product?

**Response:** Thank you for the question. This process is used because topographic complexity and soil texture data cannot be obtained from one product. Additionally, the quality for the two data are satisfying (Reichle et al., 2018; Scipal et al., 2005). Please see the response to Major Comment 6.

**Specific comment 17.** Lines 227~228, 'For each map, soil moisture values are filteredto the level of three standard deviations relative to themean in each zone. This preprocessing step is thus called '3σ denoise'.' What is the effect of such filter on identifying extreme years? for example, during 2003, 2010, 2018, 2019 there are extreme heat events in Europe and the soil moisture is so dry which can be beyond the 3 standard deviations.

**Response:** This is spatial (zonal) '3σ denoising', which helps to mask out the incorrect retrievals in mountain areas (zonal extreme values); however, it will not mask out extreme climatic events. Please find the detailed explanation in the response to Major Comment 7. Following this comment, we have made the clarification as follows: '*After standardization of the original soil moisture data, to improve the neural network training efficiency, the potential salt and pepper noises are removed. For each map (a*

*specific 10-day period), within each 1°×1° zone, the soil moisture values are filtered to the level of three standard deviations relative to the mean in that zone. This preprocessing step is thus called '3σ denoising' (note that denoising is conducted spatially, rather than temporally, so that the extreme events will not be treated).'*

**Specific comment 18.** Line 273: 'SMAP soil moisture is the training target'. SMAP is only available after 2015, so I am not sure what is the meaning of simulation period 2012D19~2013D36, but also 2014-2018. I guess this is constrained by the available data (PROBA-V and GLASS LAIs)? But in any case, it does not represent any physical meaning to predict 2015data with 2012-2013 data.

**Response:** Here, we did not predict data in 2015 by using the data from 2012~2013. In this study, the common data period for the predictor soil moisture products in each substep always contains the period of the simulated soil moisture. There are two substeps in Round 1, which were separated due to the differences in the data periods of different LAI products, while each substep was responsible for a simulation period (2012D19~2013D36 and 2014-2018, respectively). Following this comment, we have revised that section, for example, '*Because all four predictors have data since 2012D19, the potential soil moisture simulation period is 2012D19~2018, which is further divided into two parts: one is 2014~2018 (substep1), for which the PROBA-V LAI data that begins in 2014 are applied, whereas the other is 2012D19~2013 (substep2), for which GLASS LAI data are used (note: because GLASS LAI covers the period from the beginning of our study period until 2017, the training period for substep 2 is 2015D10~2017).*' For more details, please see the response to Major Comment 8.

**Specific comment 19.** Line 278: '2011D20 to 2012D18 (Table S3~S4). In the third round (2010D16~2011D19)'. Why and how are these time spans defined?

**Response:** The time spans of the soil moisture simulation period corresponding to the different rounds of neural networks were determined based on the temporal coverages of the different microwave sensors that were utilized as predictors. For example, the simulation period (not the training period) of the second round NN is constrained by

the common period of ASCAT, SMOS, FY and TMI data. Following this comment, we have made some clarifications, such as: '*In the second round of simulation, the training target can be either SMAP or SIM-1T, while the soil moisture input data are ASCAT, SMOS, TMI-LPRM-X (TMI) and FY-3B-NSMC (FY). The simulation output, SIM-2, covers the period from 2011D20~2012D18, which is constrained by the common period of the four predictors (Table S3~S4).*' We have also attached the details for the design of five rounds of neural network operations in Tables S1~S15. Please see the details in the response to Major Comment 8.

**Specific comment 20.** Line 278, 'In the third round (2010D16~2011D19), SMAP…' Again, SMAP is only available after 2015.

**Response:** For the 3$^{rd}$ round, the simulation period is 2010D16~2011D19, but the neural network training period could be 2015D10~2017D36, 2012D19~2015D10, or 2011D20~2012D21, depending on whether the training target is SMAP, SIM-1T (SIM-1T is the postprocessed simulation output of the first round NN), or both SIM-1T and SIM-2T. Detailed information on NN training and soil moisture simulation in Round 3 is provided in Tables S5~S8. We have also revised the sentence in the manuscript as follows: '*In the third round of neural network operation, the simulation period is 2010D16~2011D19. SMAP, SIM-1T and SIM-2T are combined and used as the training targets (the training periods are within the range of 2011D20~2017D36), while the soil moisture predictor data are ASCAT, SMOS, TMI and WindSat-LPRM-X (WINDSAT)…*'

---

## Author Comment (AC3) · 14 Aug 2020

**To Reviewer #3:**

We thank referee #3 for the valuable comments that will help us improve the quality and readability of the manuscript. We have carefully revised the MS following your comments and suggestions. We provide a detailed response to the Referee's comments in the Supplement.

**\*\*\*\*\*\*\*\*\*\***

**General comment:** The authors used an iterative neural network approach to produce a new satellite-based soil moisture dataset using 11 microwave soil moisture products, using SMAP data for training and ISMN database for validation. The approach is quite original and efficient resulting is a improvement in the accuracy of the spatio-temporal patterns at the global scale, and at a 0.1 degree resolution. However, the manuscript will need to be improved before acceptance, in its structure, clarity and tone.

**Response:** Thank you for your positive comments on our work. We have adjusted the article structure, revised the Methods section to make it clearer, and modified the tone of the expressions of the comparison of our product against other products. In addition, we have revised the figures and tables following each of the comments. Please find the details in the responses to the following comments.

**Major comment 1:** The introduction would need to be improved. Several statements need to be supported by existing literature, others would need to be clarified. Finally the introduction would need to end with a brief description of the approach used in the study and how this approach will address the three major concerns raised from existing soil moisture products. See detailed comments below for details.

**Response:** We have carefully addressed all the identified problems, including adding additional references, clarifying the confusing phrases, and briefly introducing how the approach addressed the three major concerns. Please see our responses to Specific

comments 3~11 for details. Thank you for these suggestions.

**Major comment 2:** The tone of the manuscript when referring to the new product and to past studies is not always appropriate. For instance, stating that the present product is "superior to the existing products" is useless, not informative and condescending. I would encourage the authors to rather explain how their product is an improvement to the global estimation of soil moisture, without necessarily condemn other products. In the result section, while nonlinearities between estimate and in-situ soil moisture measurements are identified for other products, it is not reported for the author's product which I find quite biased.

**Response:** We apologize for the inappropriate descriptions. We have corrected the descriptions in the abstract as follows: '*This new dataset, named RSSSM, is comparable to the in situ surface soil moisture measurements at the International Soil Moisture Network sites (overall $R^2$ and RMSE values of 0.42 and 0.087 $m^3/m^3$), while the overall $R^2$ and RMSE values for the existing products (ASCAT-SWI, GLDAS Noah, ERA5-Land, CCI/ECV and GLEAM) are within the range of 0.31~0.41 and 0.095~0.142 $m^3/m^3$, respectively. The advantage of RSSSM is especially obvious in arid or relatively cold areas and during growing seasons. Moreover, the persistent high data quality as well as complete spatial coverage ensure the applicability of RSSSM to studies on both spatial and temporal patterns.*' We have also reported the nonlinearity of the relationship between our data and in situ measurements, following: '*However, RSSSM overestimates soil moisture when it is low, which is a problem inherited from the SMAP product (Figure 4) and is slightly nonlinearly correlated with the measured values (Figure 7a).*'

**Major comment 3.** The validation approach is based on site specific comparison. However, soil moisture being so spatially variable within a 0.1 degree pixel, validation based on single site observations within 0.1 degree pixels can be quite meaningless. This might be particularly true when one considers that site selection for in-situ measurement is rarely motivated by representativity of the surrounding landscape, but

by specific ecological reasons.

**Response:** We agree that although only the 'good' quality data records were used, due to high spatial variability in surface soil moisture, it is not very reasonable to compare the 0.1° resolution soil moisture product against the ISMN site-scale measurements. However, the currently available global-scale soil moisture products are all coarse resolution, usually approximately 0.25°. To evaluate these coarse resolution products, previous studies also had to rely on the site-measured soil moisture, especially the ISMN dataset, while the validation process and the evaluation indicators are almost the same as those used in this study (Al-Yaari et al., 2019; Albergel et al., 2012; Dorigo et al., 2015; Fernandez-Moran et al., 2017; Gao et al., 2020; Karthikeyan et al., 2017; Kerr et al., 2016; Kim et al., 2015b; Kolassa et al., 2018; Lievens et al., 2017; Zhang et al., 2019). For 29 ISMN networks used for validation in this study, 19 are dense networks (usually with multiple stations within one 0.1° pixel (Dorigo et al., 2015)), including AMMA-CATCH (Cappelaere et al., 2009; De Rosnay et al., 2009; Lebel et al., 2009; Mougin et al., 2009; Pellarin et al., 2009), BIEBRZA_S-1 (http://www.igik.edu.pl/en), BNZ-LTER (Van Cleve et al., 2015) (http://www.lter.uaf.edu/), CTP_SMTMN (Yang et al., 2013), FLUXNET-AMERIFLUX (http://ameriflux.lbl.gov/), FR_Aqui (Al-Yaari et al., 2018), HiWATER_EHWSN (Jin et al., 2014; Kang et al., 2014), HOBE (Bircher et al., 2012), HYDROL-NET_PERUGIA (Morbidelli et al., 2014), iRON (Osenga et al., 2019), MAQU (Su et al., 2011), OZNET (Smith et al., 2012; Young et al., 2008), REMEDHUS (http://campus.usal.es/~hidrus/), SASMAS (Rüdiger et al., 2007), SKKU (Hyunglok et al., 2016), SOILSCAPE (Moghaddam et al., 2010; Moghaddam et al., 2016), SWEX_POLAND (Marczewski et al., 2010), VAS (http://nimbus.uv.es/) and WSMN (http://www.aber.ac.uk/wsmn). This information has been added to Text S2. Therefore, the average of the data obtained from two or more stations within a 0.1° pixel, which was calculated in this study, can best represent the grid-scale soil moisture conditions (Gruber et al., 2020).

In addition, to avoid the errors induced by the high spatial variability of soil moisture as much as possible, we excluded pixels with nonnegligible open water, wetland or inundated fields. In Supplementary Text- Text S2, the related details now read: '*It has been acknowledged that the scale difference between the records at ISMN sites and the 0.1° pixel-scale soil moisture data may lead to incomparability, especially for pixels with open water and inundated land (Loew, 2008). If the measurement site is located*

*on land, away from water, yet the corresponding pixel contains much water, the pixel-scale soil moisture can be significantly higher than the site-measured values. Conversely, if the site is in or close to the open water or inundated areas but land also exists in the pixel, the soil moisture measured at the station will be much higher than the average pixel value. The absolute values are unmatchable, and the temporal variations cannot be directly compared as well, because the moisture conditions of riverside (or wetland) soil and the land soil may change with precipitation differently. Therefore, the sites located in the pixels with an average annual maximal water area fraction greater than 5% according to SWAMPS data are excluded (for example, some sites in wetlands in Canada).*' We also added more explanations in the manuscript as follows: 1) '*After data screening and processing (for example, in the case of high spatial variability in soil moisture, we excluded the pixels with average annual maximal water area fractions greater than 5%, see Text S2), ...*'; 2) *More than 90% of the stations are located in relatively flat areas with topographic complexity less than 10%;* and 3) '*Hence, to make full use of all the high-quality records and to reduce the problem caused by the scale difference between simulation and measurement, the site-scale 10-day averaged soil moisture data are further aggregated to a 0.1° pixel-scale by averaging all the data (different stations or different sensors) within the pixel (Gruber et al., 2020).*'

**Specific comment 1:** "This new dataset, once validated against the International Soil Moisture Network (ISMN) records, is supposed to be superior to the existing products". Do you mean this validation hasn't been done yet? Superior in what way?

**Response:** We apologize for the confusion. Following this comment, we show the situation that our product is preferred, according to the validation against site measurements, instead of the general description that our product is superior to other products. We have made the clarification as follows: '*This new dataset, named RSSSM, is comparable to the in situ surface soil moisture measurements at the International Soil Moisture Network sites (overall $R^2$ and RMSE values of 0.42 and 0.087 $m^3/m^3$), while the overall $R^2$ and RMSE values for the existing products (ASCAT-SWI, GLDAS Noah, ERA5-Land, CCI/ECV and GLEAM) are within the range of 0.31~0.41 and 0.095~0.142 $m^3/m^3$, respectively. The advantage of RSSSM is especially obvious in arid*

*or relatively cold areas and during growing seasons. Moreover, the persistent high data quality as well as complete spatial coverage ensure the applicability of RSSSM to studies on both spatial and temporal patterns.*'

**Specific comment 2:** "reveals that the surface moisture decline on rainless days is highest in summers over the low-latitudes but highest in winters over most mid-latitude areas." Soil moisture being so spatially variable, I find the impact of this statement quite limited – e.g. low latitude regions range from tropical/equatorial rain forests to deserts and one would expect as much differences in the sensitivity of soil moisture to precipitation between a desert and a tropical forest than between a tropical forest and a temperate prairie.

**Response:** We apologize for the unclear sentences. In the calculation of soil moisture decline on consecutive rainless days, because '*In desert areas, the random noise of the surface soil moisture product can hide the signal of moisture changes, while in wet areas (e.g., rainforests), 20 days without effective precipitation seldom occurs, leading to no results over most areas.*', the acquired results can represent the conditions in only regions excluding deserts and rainforests. Following this comment, we have revised the sentence as follows: '*These data also reveal that without considering the deserts and rainforests, the surface moisture decline on consecutive rainless days is highest in summers over the low latitudes (30°S~30°N) but highest in winters over most midlatitude areas (30°N~60°N; 30°S~60°S).*'

**Specific comment 3:** "L47: "due to various disturbances": what type of disturbances?
**Response:** Following your comment, we have revised this phrase as '*due to various disturbances, such as high vegetation cover, high open water fractions and complex topography (Draper et al., 2012; Fan et al., 2020; Ye et al., 2015)*' to improve the clarity.

**Specific comment 4:** "L49: " Although new sensors, SMOS : : :." -> "Although new sensors such as SMOS: : :"
**Response:** We have revised this phrase accordingly.

**Specific comment 5:** L 50: "better penetrability" -> please be more specific: what depth?

**Response:** We apologize for the unclear expression. It does not indicate the nominal soil depth of the microwave soil moisture. We have revised it to '*… because L-band microwaves (1~2 GHz) can penetrate the vegetation canopy better than other bands,'* by referring to (Piles et al., 2018).

**Specific comment 6:** L66: "Because the temporal variation in soil moisture is often better captured by model simulations than remote sensing inversions": please include a reference that support this statement. L67: "CCI may undesirably combine the disadvantages of both." Be more specific here (low accuracy of temporal variations from remote sensing products and low spatial accuracy from model simulations – am I right?). And please include another reference here for this second statement.

**Response:** We apologize for the arbitrary statements. We agree that there is no strong evidence supporting the claim that the temporal variation in soil moisture is better captured by model simulations than remote sensing inversions. Following this comment, we have deleted these two sentences in the Introduction of the manuscript. Thank you for the reminder!

**Specific comment 7:** L70: "are assimilated instead": instead of what? this sentence is not clear.

**Response:** Following this comment, we have revised the sentence as follows: '*Currently, anomalies of CCI soil moisture (the deviations to the seasonal climatology that indicate whether the soil moisture at a time point is more humid or drier than the multi-year average) are assimilated instead of the original CCI time series (Martens et al., 2017).'*

**Specific comment 8:** L85: "Among these three approaches, machine learning proves to be probably the best choice" based on what criteria – again, please be more specific

**Response:** Following this comment, we revised the sentence as follows: '*Among these*

*three approaches, machine learning has been proven to be the best choice according to the connection between precipitation and the changes in soil moisture, as evaluated through a data assimilation technique and triple collocation analysis results (Van der Schalie et al., 2018).*'

**Specific comment 9:** "L102: "substantial success has not been achieved yet." This is a rather strong and yet vague statement that denies the merits of a large body of research. Please remove this statement.

**Response:** We apologize for the confusion. We have revised the sentence as follows: '*In conclusion, while previous studies have focused on developing long-term satellite-based surface moisture products using machine learning, some major concerns remain that need to be solved. 1)…*'

**Specific comment 10:** "L102/103:" the high-quality microwave observations are not fully utilized": this is not clear from the literature review – please develop this point in earlier sections of the introduction (i.e. in what way high-quality microwave observations haven't been fully utilized, and how the authors are proposing to utilize them more efficiently).

**Response:** To avoid potential misunderstanding, we have revised the sentence as follows: '*1) The microwave observations from only three sensors at most are utilized, leading to large temporal and spatial gaps and limited the training efficiency.*' This point has been illustrated in the earlier sections of the introduction: '*A global long-term observational-based soil moisture product was recently developed by building a neural network between the SMOS product and the Tb data from AMSRE (2003~September 2011) and AMSR2 (July 2012~2015) (Yao et al., 2017).… The gap between the temporal spans of AMSRE and AMSR2 and the lack of SMOS data in Asia resulted in large quantities of missing data.*'

**Specific comment 11:** "L106-107: This statement should be removed from the introduction section. - this is rather a concluding statement. Instead please describe your

approach in a couple sentences and how this approach addresses the three major concerns identified.

**Response:** Thank you for your advice. We have rewritten the paragraph as: '... *some major concerns remain that need to be solved. 1) The microwave observations from only three sensors at most are utilized, leading to large temporal and spatial gaps and limited the training efficiency; 2) it remains unclear which environmental factors should be incorporated as ancillary inputs and why; and 3) the training designed for soil moisture estimation at the global scale should be more complex than that for only a specific region to ensure satisfactory training efficiency. In this study, 11 high-quality microwave soil moisture products since 2003 are incorporated into 5 rounds of neural networks to achieve a spatially and temporally continuous simulation for 2003~2018, using as many sources of microwave observational data as possible as predictors in each neural network. The factors impacting the quality of microwave soil moisture retrievals are also determined and then utilized as ancillary inputs to improve the training efficiency. Moreover, we designed localized subnetworks instead of only one global-scale neural network to account for the regional differences in training rules.*'

**Specific comment 12:** "Section 2.1 L110: please add citation to literature supporting this statement.

**Response:** In the Introduction section, the best overall quality of SMAP soil moisture has been stated as follows: '*Although new sensors such as SMOS (Stillman and Zeng, 2018) and SMAP (Entekhabi et al., 2010) can produce significantly improved estimates because L-band microwaves (1~2 GHz) can penetrate the vegetation canopy better than other bands (Burgin et al., 2017; Chen et al., 2018; Karthikeyan et al., 2017; Kerr et al., 2016; Kim et al., 2018; Leroux et al., 2014; Stillman and Zeng, 2018), the applicability of both products is still limited. SMOS data have too much noise and too many missing values in Eurasia due to high radio frequency interference (RFI) (Oliva et al., 2012). While SMAP has the highest quality (the unbiased RMSE of the passive product can be close to its target of 0.04 $m^3/m^3$) and has filtered RFI (Chen et al., 2018; Colliander et al., 2017), ...*' Following your advice, we have revised this sentence and

added two new references supporting the best performance of SMAP product. It reads: '*SMAP currently has the best quality of all remote sensing-based soil moisture products (Al-Yaari et al., 2019; Liu et al., 2019)...*'

**Specific comment 13:** L115-118: This sentence is too long and too complex. Please split into shorter and clearer sentences.

**Response:** We apologize for the complex sentence. We have revised it to: '*However, in this study, the well-acknowledged surface soil moisture products retrieved through mature algorithms (see Figure 1) are directly applied instead of Tb. These products are chosen because 1) the primary goal of this study is to calibrate and then fuse the existing popular microwave soil moisture products, and 2) the Tb signals at multiple bands contain too much information that is not related to soil moisture, which may weaken the training efficiency and lead to overfitting.*'

**Specific comment 14:** Section 2.2 L147/160: the purpose of this argumentation is quite unclear as the environmental predictors that are selected are also important drivers of soil moisture dynamic.

**Response:** We apologize for the arbitrary sentence. Here, we would like to express the idea that without the incorporation of any microwave remote sensing product, these factors (LAI, topographic complexity, LST, etc.) alone may not predict surface soil moisture very well. This phenomenon occurs because although we agree that they are somewhat related to soil moisture (e.g., soil moisture is usually limited in areas with low vegetation cover), they can hardly be considered direct indexes of surface soil moisture content (since the relationships are rather uncertain; for example, as found in this study, in Europe, the soil moisture is low in summers when vegetation peaks). On the other hand, however, we admit that the water area fraction is a direct indicator of surface soil moisture and have corrected it by adding the sentence '*Environmental factors, including DEM, LST and vegetation cover (indicated by NDVI, MVI, etc.), were used as ancillary neural network inputs to improve the soil moisture simulation (Lu et al., 2015; Qu et al., 2019; Yao et al., 2017). According to these studies, these factors*'

*alone may not predict surface soil moisture well without the incorporation of any microwave remote sensing data, which can also be justified by the contribution analysis results (Figure S1a). This phenomenon occurs because although these data are somewhat related to soil moisture (e.g., soil moisture is generally limited in areas with low vegetation cover but high in forests (McColl et al., 2017)), the relationships are rather uncertain (e.g., at small scales, leaf area index (LAI) may have a negative influence on soil moisture due to the variation in evapotranspiration (Naithani et al., 2013) or may not have clear impacts (Zhao et al., 2010); also, soil moisture can be either high or low in summers when vegetation peaks (Baldocchi et al., 2006; Méndez-Barroso et al., 2009)). However, these factors are essential due to their direct impacts on soil moisture retrieval through the radiative transfer model using microwave remote sensing data (Fan et al., 2020) and are factors that impact the retrieval quality. The detailed explanations are as follows: 1) ...'*

**Specific comment 15:** L201: since precipitation is such an important driver of soil moisture, the reasons why this variable hasn't been included as a quality impact factor should be included in the main document.

**Response:** We have revised the sentence as follows: '*The contribution analysis results (Figure S1) show that because various microwave soil moisture data have already been included, precipitation data are not an essential indicator of soil moisture and are not utilized as a physically based 'quality impact factor' either (see Text S1 for detailed explanations).*'

The Text S1 is a bit long, so we did not move all the information to the manuscript, but we summarized the key reasons in the revised manuscript, following your advice. Text S1 has been revised as follows:

'*Because it takes at least 3 days for a microwave sensor to cover the globe, for 11% of global land, there will be only 5 or fewer observations for random days within a 10-day period. By taking the average of these available data, this study focuses on only the mean soil moisture condition during that 10-day period. Then, to see how much the incorporation of precipitation data can improve the neural network training efficiency,*

*we calculated the 10-day averaged GPM Final-Run precipitation, which can indicate the overall precipitation water availability (the antecedent precipitation index is not used because it must be calculated on a daily scale, and the attenuation coefficient is difficult to determine at a global scale (Kohler and Linsley, 1951)). Taking the first primary independent neural network, NN1-1-1, as an example, we performed contribution tests on all the input features at the global scale (not for each separate zone), including 9 'quality impact factors', 4 soil moisture predictor products and precipitation - a potential ancillary soil moisture indicator. For each predictor, we added a random error that is controlled within the standard deviation of the predictor, and then an increase in MSE during neural network training can indicate the relative contribution of that variable. The results (Figure S1a) show that precipitation will contribute to only 1.7% of the training efficiency, which is much lower than the contribution of any soil moisture product (the total contribution fraction of the four soil moisture products is 61.2%) and is also lower than that of most 'quality impact factor'. This result suggests that various microwave soil moisture datasets together with several 'quality impact factors' of microwave soil moisture retrieval are enough to predict the training target, SMAP soil moisture, and there is no need to add precipitation as another ancillary indicator of soil moisture.*

*'Quality impact factors' are defined in this study as the variables that will have a significant impact on the retrieval errors of soil moisture by microwave remote sensing (section 2.1.2). Although the relative performances of different soil moisture products are related to surface moisture conditions (Kim et al., 2015a), it is found mainly due to the less vegetation in arid areas. After all, no explicit mechanism can support the idea that the retrieval errors of soil moisture are significantly influenced by water availability. Even if this is true, the soil water availability can already be indicated by the microwave soil moisture products. Therefore, it is unreasonable to incorporate the precipitation variable as a 'quality impact factor'. On the other hand, LAI, water area fraction, LST, land use cover, tree cover fraction, non-tree vegetation area fraction, topographic complexity, and soil sand/clay fractions all have direct impacts on the microwave soil moisture retrieval errors, with solid physical mechanisms (see section 2.1.2). Therefore, theoretically, these variables should be added to the neural network, even though the land use cover type and soil sand fraction data have been proven to have limited contributions to NN training efficiency.*

*One may argue that if NARX (nonlinear autoregressive with external input) is applied instead, in which the soil moisture in the previous 10-day period is also incorporated as a predictor, precipitation data can be very beneficial to neural network training. This is true because precipitation directly contributes to increases in soil moisture. However, NARX is not suitable for global-scale long-term continuous soil moisture mapping because the base map (i.e., soil moisture at the beginning of the simulation period) is difficult to determine. Moreover, in mid to high latitudes, the lack of soil moisture retrievals over frozen ground in winters will lead to missing data there in summers when soil moisture data are otherwise available. Therefore, if NARX is adopted, we can only estimate long-term surface soil moisture in the tropics and subtropics with air temperatures consistently higher than 0 ℃. Finally, if the soil moisture in the previous phase and the current precipitation amount are both incorporated, they will largely conceal the role of satellite-observed signals. As shown in Figure S1b, the total contribution fraction of all four microwave soil moisture products is reduced to only 10.6%, while the roles of ASCAT, AMSR2-JAXA and AMSR2-LPRM are all negligible. Without taking full advantage of remote sensing, simulations based on previous soil moisture and current precipitation products will lead to errors in regions where soil moisture gains are mostly driven by glacier melting or in places with high levels of radiation-driven soil evaporation. The reliability of the derived soil moisture will be reduced in irrigated croplands and afforestation/deforestation areas as well.*

*On account of all of the above, precipitation data are neither included as an ancillary soil moisture indicator nor added as a 'quality impact factor' in this study.'*

[Figure]

*Figure R1 (Figure S1 in the revised Supplement): The roles of different input features*

*in the soil moisture simulations based on BP neural networks and nonlinear autoregressive with external input (NARX) with microwave soil moisture products incorporated: (a) the contributions of different input features of a primary neural network: NN1-1-1, including 4 predictor soil moisture products, 9 quality impact factors of microwave soil moisture retrieval, plus 1 probable ancillary soil water indicator: 10-day averaged precipitation, to the neural network training efficiency indicated by the increased MSE; (b) the contributions of all the input features to the training efficiency if NN1-1-1 is changed into a NARX, in which the SMAP soil moisture for the previous period is also applied as a predictor.*

**Specific comment 16:** L202/203: This sentence should start this section, not end it. A table similar to table 1 but for the quality impact factors would be useful. The table would indicate the source of the data, the resolution and the temporal span for the dynamic factors.

**Response:** Following your suggestion, we have revised this section, moving this concluding sentence to the beginning. It now reads as:

'*In this study, 9 quality impact factors, LAI, water fraction, LST, land use cover, tree cover fraction, non-tree vegetation fraction, topographic complexity, and sand and clay fractions, are selected and incorporated (see Figure 1). The reasons for the selection of these factors are as follows.*

*Based on the two criteria above, the first environmental factor to be included is the 'vegetation factor' (i.e., vegetation water content, VWC) … Because the LAI stands for the total leaf area per unit land, which is closely related to VWC assuming a relatively stable leaf equivalent water thickness (Yilmaz et al., 2008), LAI is a suitable surrogate….*'

We also added a timeline figure (Figure 1 in the revised manuscript) to show the temporal spans, sources and spatial resolution of all microwave soil moisture products and the data for 9 quality impact factors. This figure presents the information more clearly than a table.

[Figure]

*Figure R2 (Figure 1 in the revised manuscript): The timeline figure showing the time periods of the soil moisture datasets and the 'quality impact factor' products (e.g., LAI dataset) used in this study (listed above the timeline), as well as the periods of data applied for the trainings of the 67 independent neural networks and the neural network simulation outputs (i.e., simulated soil moisture) of eight substeps (listed below the timeline).*

**Specific comment 17:** Section 2.3 Sections 2.4, 2.5 and 2.6 should be included in

section 2.3 as it details the different steps of the calculation flow. A clear justification on why a neural network approach was adopted should be included in this section.

**Response:** We have made the revision accordingly. The structure for the Data and Method Section is now as follows:

*'2 Data and methods*

*2.1 Data for the production of global long-term surface soil moisture data*

*2.1.1 Satellite-based surface soil moisture data products*

*2.1.2 The quality impact factors of soil moisture retrievals*

*2.2 Methods for the production of global long-term surface soil moisture data*

*The steps taken to produce global long-term surface soil moisture data include three basic parts, which are as follows. 1) Preprocessing: the production of high-quality neural network inputs; 2) neural network operation: the network training and soil moisture simulation; and 3) postprocessing: the correction of potential errors or deficiencies in the soil moisture simulation outputs…*

*2.2.1 Neural network design (1): localized neural networks*

*2.2.2 Preprocessing and postprocessing steps*

*2.2.3 Neural network design (2)- five rounds of simulations*

*2.3 Methods for the validation of surface soil moisture products*

*2.4 Methods for the intra-annual variation analysis of surface soil moisture'*.

The justification of the neural network approach is now in the Introduction: '*Among these three approaches, machine learning has been proven to be the best choice according to the connection between precipitation and the changes in soil moisture, as evaluated through a data assimilation technique and triple collocation analysis results (Van der Schalie et al., 2018).*' In section 2.1.2, we have already mentioned the use of neural network approach in this study, '…*were used as ancillary neural network inputs to improve the soil moisture simulation (Lu et al., 2015; Qu et al., 2019; Yao et al., 2017)*'.

**Specific comment 18:** L223: what is a hidden layer?

**Response:** In neural networks, the hidden layer is located between the input and output

layers. This layer is the result of nonlinear transformations of the input data through the activation function and can also be transformed into the output data. We have revised the sentence as '*…, and the number of nodes in the hidden layer (between the input and output layers (Stinchcombe and White, 1989)) of each subnetwork was set to 7.*'

**Specific comment 19:** L242: reference required for the "suspicious value removal".

**Response:** We apologize for the confusing phrase. Actually, it is a detailed method invented by this study to ensure that only the most reliable estimates are applied as the training target of the next round of neural networks to avoid significant error propagation along the subsequence of neural network rounds. Because multiple rounds of the neural network is a characteristic of this study, this processing step was not found in previous studies (i.e., no reference). We have revised the sentence, removing the phrase '*suspicious value removal*'. It now reads: '*On the other hand, to avoid error propagation with training times by ensuring a high-quality training target for the next round's simulation, we remove all suspicious values for every simulated result. This preprocessing step is performed by first obtaining the maximal and minimum values of SMAP_E soil moisture in each pixel. If the simulated value is out of the range of the SMAP data during 2015~2018, the value is considered suspicious and is not used as a training target.*'

**Specific comment 20:** L259: Here you are referring to a 1 degree pixel a presume? Please specify.

**Response:** We apologize for the confusing sentences. We have revised the sentences to: '*For data simulation in a 0.1° pixel, the most preferable independent neural network is expected to be trained using all the available soil moisture data sources in that pixel. However, in the 1° zone where it is located, the subnetwork belonging to that preferable independent neural network may not exist due to limited valid data points (see section 2.2.1). Then, an alternative subnetwork driven by the combination of fewer soil moisture data inputs should be applied instead..*' In the revised manuscript, the word 'pixel' only stands for 0.1-degree resolution, while the word 'zone' indicates 1° scale.

**Specific comment 21:** Please specify. L268: why 74 networks? Please explain.

**Response:** We have corrected it to '67 independent neural networks' because the remaining 7 networks are optional and not independent. There are multiple independent networks included in each round. That is because '... *While increasing the sources of soil moisture data inputs can be beneficial to the training efficiency, the spatial coverage of the simulation output is sacrificed because the overlapping area decreases with the increase in the number of soil moisture products. After all, most products have missing data in specific regions ... To solve that dilemma, we classified all 0.1° pixels according to the available predictor soil moisture products over a 10-day period (for example, if there are at most four soil moisture data inputs in one round, there should be 4+3+2+1=10 combinations). However, to avoid soil moisture simulation under snow or ice cover (Section 2.2.2), not all combinations are considered. Then, corresponding to each selected combination, an independent neural network is trained*' (Lines 269~277 in the revised manuscript). The revised explanations are clearer than the original version.

In addition, the training periods of the 67 independent neural networks are now shown in Figure R2 (Figure 1 in the revised manuscript). Therefore, this sentence is now revised to: '*Based on these principles, five rounds of neural networks are designed as follows, with 8 substeps containing a total of 67 independent neural networks. The training period for each neural network and the simulation period for each substep are shown in Figure 1 (below the timeline), and the details are as follows: ...*'

**Specific comment 22:** Section 2.7 Figure S2 should be included in the main manuscript as the spatial distribution of validation data is critically important to evaluate the overall strength of this validation. It is surprising that none of the Canadian sites of the Russian sites made it to the final set of sites for validation.

**Response:** Thank you for this advice. We have moved the figure showing the spatial distribution of the validation sites to the main manuscript (shown in Figure 3).

The soil moisture measurement depths for the Russian sites are 0~10 cm (RUSWET-

GRASS) or 0~20 cm (RUSWET-AGRO and RUSWET-VALDAI) (Robock et al., 2000), which do not match the nominal simulation depth (0~5 cm) of our soil moisture product (constrained by SMAP).

On the other hand, the Canadian sites are actually very limited (Dorigo et al., 2011; Dorigo et al., 2013) (https://www.geo.tuwien.ac.at/insitu/data_viewer/) and are often wetland sites with nonnegligible water fractions, which should be removed. Previous studies also utilized very few sites in Canada and Russia (Kolassa et al., 2018; Lievens et al., 2017; Zhang et al., 2019).

In the revised manuscript, we added the information '*Accordingly, the measurements used for validation are limited to ≤ 5 cm in depth (e.g., the Russian networks were not applicable for this reason).*'

**Specific comment 23:** L304/306: This is an important point. Soil moisture being so spatially variable, validation based on single site observations within 0.1 degree pixels can be quite meaningless. Especially when one considers that site selection for in situ measurement is rarely motivated by representativity of the surrounding landscape, but by specific ecological reasons.

**Response:** Thank you for your careful consideration. Because grid-scale soil moisture measurements are unavailable, almost all recent studies rely on in situ soil moisture records for the validation of remote sensing soil moisture products. Moreover, we have taken the average of the data at multiple stations in a 0.1° grid and excluded grids with potentially high spatial heterogeneity of surface soil moisture. Please see the response to Major comment 3 for details.

**Specific comment 24:** Section 2.8 L335: "probably the best choice for periodic function fitting" : please support this statement by adequate reference to literature.

**Response:** We have revised it to '*Fourier functions can characterize intra-annual variation well (Brooks et al., 2012; Hermance et al., 2007). Therefore, for the remaining areas …, we fit the intra-annual cycle of soil moisture using the Fourier function…*'

**Specific comment 25:** L344/345: I don't understand this argument. Why restricting this analysis to 10 consecutive rainless days and not the whole range of 10-days sum of precipitation?

**Response:** This study explored the impact of dry periods (20 consecutive days without effective rainfall) on the surface soil moisture in different areas. Moreover, without effective precipitation, the surface soil moisture changes are mainly driven by evaporation and deep percolation and thus should be negative. This process is simpler and can help us exclude unreliable soil moisture values from the analysis. However, if we consider the sum of precipitation over 10 days, the surface soil moisture changes will be rather complex, and there would be much more erroneous data included, leading to unreliable analysis results.

We have revised the sentence to: '*Because RSSSM indicates the average soil moisture condition during every 10-day period, we evaluate the surface soil moisture decline after 20 consecutive days (i.e., two adjacent 10-day periods) without effective precipitation to explore the impact of dry periods on surface soil moisture.*'

**Specific comment 26:** Tables 3 to 8 could be synthesized into only two tables: one for temporal accuracy assessment and one for spatial accuracy assessment for the three products comparisons with SIM. Similarly, it would be nice to have figures 3, 4, 6, 7, 9 and 10 summarized in 2 figures where all four products appear (SIM, SMAP, GLDAS and ASCAT). This would facilitate comparison between products.

**Response:** Thank you for this advice. The periods corresponding to these comparisons differ from each other. SMAP has data since March 2015, ASCAT is available from 2007, while SIM (named 'RSSSM' in the revised manuscript), GLDAS, ERA5 Land and GLEAM v3.3a all cover the whole study period (2003~2018), but with missing values in different areas. Although CCI (ECV) also covers the entire period, it lacks data in many places, especially before 2007 (Figure 13d). Therefore, the in situ surface soil moisture measurements entering the comparison between RSSSM and SMAP are limited to March 2015~2018, whereas only the ISMN site data during 2007~2018 were applied for the accuracy comparison between RSSSM and ASCAT. Additionally, when

comparing RSSSM against CCI (ECV), we included the soil moisture records in only the grids with both RSSSM and CCI data during the specific period. As we can see, the overall accuracy of RSSSM in Figure 4c (for the comparison with SMAP, during April 2015~2018) is $R^2=0.46$, RMSE=0.083, but in Figure 10 (for comparison against GLDAS, during 2003~2018), the overall $R^2$ of RSSSM is 0.42, and the RMSE is 0.087. In addition, the temporal accuracy of RSSSM in all climatic regions (Figure 5, Figure 8, Figure 11 in the revised paper) and its spatial accuracy during all seasons (Figure 6, Figure 9, Figure 12) are different among the comparisons against different soil moisture products. For that reason, if the 6 figures are combined into 2 figures, they will be too crowded, and the comparison will be not clear enough.

Following your comment, we have combined the tables as follows:

[revised manuscript text omitted]

**Specific comment 27:** L380-383: However, it looks like the relationship between SIM estimates and in situ observations is nonlinear (Figure 5a). Furthermore, SIM seems to overestimate soil moisture in the lowest range (winter?) when a density of pixels is quite high. Please include these remarks in the results.

**Response:** Thank you for the reminder. We have added a sentence accordingly. It reads: '*However, RSSSM overestimates soil moisture when it is low, which is a problem inherited from SMAP product (Figure 4), and is a bit nonlinearly correlated with the measured values (Figure 7a).*'

**Specific comment 28:** L393: the relationship between SIM and in situ measurements is also obviously nonlinear. Please include this remark in the text for fairness.

**Response:** We have revised the description accordingly. It now reads: '*While RSSSM is nonlinearly correlated with measured soil moisture, the relationship between GLDAS soil moisture and the measurements appears to be slightly more nonlinear, resulting in a smaller $R^2$ of 0.39 and higher RMSE of 0.097 for the GLDAS product than those for RSSSM ($R^2$: 0.42; RMSE: 0.087, see Figure 10).*'

**Specific comment 29:** Why table S19 and Figure S7 do not appear in the main document like the other product comparison? Please move them to the main manuscript

**Response:** We have combined Table S17, Table S19, Table S21, and Table S22 in the original Supplement to Table 2 in the revised manuscript (see Table R1). Table S18, Table S20, Table S23 and Table S24 are integrated into Table 3 as well (see Table R2). For the reasons described in the response to Specific comment 26 (the time periods and spatial extents for the comparisons of RSSSM in this study against the different existing surface soil moisture products), the figures were not combined to avoid over-crowding the figures and unnecessary confusion. Thus, we retained Figure S7, Figure S8, etc. in

the Supplement to prevent the main manuscript from being too long and too complex to read.

**Specific comment 30:** L422-434: This belongs to the discussion section.

**Response:** We have moved the sentences belonging to the discussion section to the Discussion section. The revised sentences in the Results section now read: '*Next, we focus on the interannual change in data quality. According to Figure 13a~c, while the correlation coefficient for RSSSM does not vary significantly among different years, the RMSE and ubRMSE values in earlier periods are somewhat higher than those after 2012. Although the data quality of RSSSM is difficult to maintain, the degree of degradation is much lower than that of CCI. The comparison of the spatial coverages of the 10-day scale RSSSM and CCI data (rainforests are excluded) shows that RSSSM covers all land surfaces except for permafrost, while the interannual variation in coverage is also negligible throughout the period (the intra-annual cycles of data coverages result from the changes in frozen areas), which are preferable to CCI, whose data coverage before 2007 is limited (Figure 13d).*'

After the removal, the Discussion section has been revised as follows: '*In this study, an improved global long-term satellite-based surface soil moisture dataset, .... Our product is temporally continuous during 2003~2018 and covers the whole globe except for frozen grounds (CCI has limited spatial coverage before 2007, when ASCAT data are unavailable), ensuring its applicability to global long-term studies or ecosystem modeling.... The RMSE and ubRMSE values in earlier periods are somewhat higher than those after 2012 because 1) five rounds of simulations were performed, with the output converted into the training target of the next round's neural networks, leading to little error propagation as the simulation period extended to the past; and 2) the quality of microwave soil moisture data is generally low in early periods due to the relatively unadvanced microwave sensors with low signal-to-noise ratio (SNR). However, due to the elaborate design of the neural network set (localized networks, full use of 11 microwave soil moisture products, the determination of quality impact factors and the organization of 67 independent neural networks), high training efficiency is achieved,*

*resulting in limited amplification of noise and high maintenance of valid information during 16 years of simulation. This method turns out to be better than the simple CDF matching algorithm, which may not efficiently calibrate the low-quality soil moisture data retrieved from early sensor measurements.'*

---

## Author Response (AR1)

Dear editor,

Thank you for editing our manuscript. The first part of this document includes the point-by-point responses to the reviews (Reviewer 1, Reviewer 2, Reviewer 3). Comments of the referees are marked as e.g. << Reviewer 1 Major comment 1>> followed by the answer from the authors, which includes the changes made in the manuscript to fulfill the referees' suggestions.

5   The section of responses to the referees is followed by a marked-up version of the manuscript.

Best regards

Yongzhe Chen, Xiaoming Feng and Bojie Fu

  **To Reviewer #1:**

We thank referee#1 for the valuable comments that will help us in improving the quality and readably of the manuscript. We have carefully revised the MS following your comments and suggestions. We provide a detailed response to the Referee's comments in the Supplement.

15  **Reviewer 1 General comment.** The authors propose a global dataset of top (0-5cm) soil moisture with 10-day temporal and 0.1 spatial resolution, covering the period 2003-2018. The dataset was produced gradually, backward in time, through machine learning methods (neural networks) for 5 periods that correspond with the availability of 11 different passive and active satellite remote sensing soil moisture products. Besides satellite observations (starting with SMAP in step one), 9 environmental properties were fed to the neural network, and from step two on, previously modeled soil moisture was included

20  to enable the expansion backward in time. The final product is evaluated with observations of the international soil moisture network and, in comparison to other merged products, rated superior, however, the potential for further improvement is also emphasized. Altogether, the work seems sound and the developed method and dataset appear valuable for further scientific studies and applications. Nevertheless, some of the steps in the processing chain need further clarification and the data structure needs to be improved before the manuscript can be considered to suffice for publication.

25  **Response:** Thank you for your careful reading and the positive comments on our work. We agree that some steps in the method was unclearly written, while the data structure is not easy for other researchers to use. We have added the missing important details in the revision for further clarification of the processing chain, and reuploaded the dataset with filename changed and table added, according to your valuable suggestions. Please see the details in the responses below.

30  **Reviewer 1 Specific comment 1.** I suggest to remove "new" from the title, since all dataset proposed in this journal are somewhat new. You may consider to name it "combined" or "improved" or "complete" or "optimal". Have you thought about giving the product an acronym? That improves recognizability and makes it easier to reuse it in other studies and publications.

**Response:** Thank you for the suggestion, we have changed it to 'improved'. We also named the product as 'RSSSM' since it is a remote sensing-based surface soil moisture. In other parts of the article, all the 'SIM' have been changed to 'RSSSM' as

35 well, including those in figures and tables.

**Reviewer 1 Specific comment 2.** L14: more than 10**6 not correctly displayed in the online abstract (here it reads 106)

**Response:** We have revised it to '*more than one million*'.

40 **Reviewer 1 Specific comment 3.** L15: Please state also the temporal resolution (10 days)

**Response:** We have added this important information.

**Reviewer 1 Specific comment 4.** L32: resolution of ERA INTERIM is rather 0.75°

**Response:** Thank you for reminding us. We have corrected it accordingly.

**Reviewer 1 Specific comment 5.** L34-37: I agree that these products have many shortcomings, but other than the dataset

provided by the authors, the models provide also information about the deeper soil layers. This important point should not be

omitted here.

**Response:** We agree that models can simulate soil moisture at deeper layers, which is an important advantage. We have added

50 this point in the revised manuscript: '*Apart from surface soil moisture that can be observed by satellites, the modeling way*

*provides also the information on the moisture in deeper soil layers.*'

**Reviewer 1 Specific comment 6.** L75-76: "data averaging" - what type of averaging is meant, spatial or temporal? "can hardly

unify the temporal variations." Please specify what the "temporal variations" refer to. Is it the temporal variations of the

55 different soil moisture data products?

**Response:** Sorry for the unclear expressions. It's neither spatial nor temporal averaging. Instead, CCI product is achieved by

rescaling the soil moisture data retrieved from each microwave sensor first and then averaging the rescaled soil moisture data

products during the same period (i.e., the common period for two or more products) based on some criteria (e.g. the estimated

error) (Dorigo et al., 2017; Gruber et al., 2017; Gruber et al., 2019; Liu et al., 2012). The 'temporal variations' in this sentence refers to the temporal variations of the different soil moisture data products. Following your question, we have revised the sentence to: '*Rescaling the soil moisture data retrieved from each sensor by using CDF matching followed by averaging the rescaled data during one common period, which is adopted in CCI, can hardly unify the temporal variations of different soil moisture products.*'.

**Reviewer 1 Specific comment 7.** Instead of Table 1 or in addition, it would be good to have a timeline figure from 2003-2018 that shows a bar for every dataset used in the process of creating the final product, including the 11 soil moisture products, the time-varying quality impact factors and the intermediate modeling products (SIM-1T, SIM-2T,...).

**Response:** Thank you for this nice suggestion. Following your comment, we have added a timeline figure showing the temporal coverages (including used data periods and unused data periods) of all 11 soil moisture products, the time-varying quality impact factor (i.e. three LAI products), and that of the intermediate products. The figure below is attached as Figure 1 in the revised manuscript. Table 1 is thus removed.

[Figure]

*Figure R1: The timeline figure showing the time periods of the soil moisture datasets and the 'quality impact factor' products*

*(e.g. LAI dataset) used in this study (listed above the timeline), as well as the periods of data applied for the trainings of 67*

75   *independent neural networks and the neural network simulation outputs (i.e. simulated soil moisture) of eight substeps (listed*

*below the timeline).*

**Reviewer 1 Specific comment 8.** L110: Specify why SMAP is mentioned as the "best product" here. Is it because of the spatial resolution, the algorithms or with respect to the in-situ observations? Can you add a citation to corroborate this statement?

**Response:** SMAP is the 'highest quality product', with respect to in-situ observations. In the Introduction part, this point has been stated: '*Although new sensors such as SMOS (Stillman and Zeng, 2018) and SMAP (Entekhabi et al., 2010), can produce significantly improved estimates because L-band microwaves (1~2 GHz) can better penetrate the vegetation canopy (Burgin et al., 2017; Chen et al., 2018; Karthikeyan et al., 2017; Kerr et al., 2016; Kim et al., 2018; Leroux et al., 2014; Stillman and Zeng, 2018), the applicability of both products is still limited. SMOS data have too much noise and too many missing values in Eurasia due to high radio frequency interference (RFI) (Oliva et al., 2012). While SMAP has the highest quality (the unbiased RMSE of the passive product can be close to its target of 0.04 m³/m³) and has filtered RFI (Chen et al., 2018; Colliander et al., 2017), ...*' Following your advice, we revised this sentence and added two new references supporting the best performance of SMAP product. It reads: '*SMAP has currently the highest quality of all remote sensing-based soil moisture products (Al-Yaari et al., 2019; Liu et al., 2019)...*'.

**Reviewer 1 Specific comment 9.** L143: change to "reference coordinate system".
**Response:** We have changed it accordingly.

**Reviewer 1 Specific comment 10.** L182: "based on the correlation between soil dielectric conductivity" - do you mean soil dielectric permittivity or soil electric conductivity?
**Response:** Sorry for that mistake. It should be 'soil dielectric permittivity', or 'soil dielectric constant'. We have corrected it to '*soil dielectric constant*' in the revision.

100 **Reviewer 1 Specific comment 11.** L186-188: "Because ..." this sentence is unclear.

Response: Sorry for the unclear expression. This sentence is to explain that actual LST can determine the bias of every LST estimate, which is used in the corresponding soil moisture retrieval. Hence, the actual LST will influence the biases of different soil moisture products. We have revised the sentence as: '*Because in the retrievals of different soil moisture products, different LST estimates are used, while the bias of each LST estimate compared to the actual LST is influenced by actual LST, we*

105 *suppose that the actual LST can determine the accuracy of every LST estimate, and finally the relative performances of various soil moisture products (Kim et al., 2015)*'. We hope it will be easier for readers to understand.

**Reviewer 1 Specific comment 12.** L205: Figure 1 is never referenced in the manuscript. This should be done here or later at L225.

110 **Response:** Thanks for reminding. We have added: '*The basic flow is shown in Figure 2.*' (note: Figure 2 is Figure 1 in the original manuscript) at the beginning of the section 2.2 in the revised manuscript.

**Reviewer 1 Specific comment 13.** L219: Do the 140x360 zones include water (ocean) areas?

**Response:** Yes, the 140×360 zones include water (ocean) areas. However, for zones with no land or very limited land, the

115 number of valid pixels is lower than 100, so these ocean zones are not applicable for subnetworks, and are excluded.

**Reviewer 1 Specific comment 14.** L220: A subnetwork has 100 pixels, but ("for a 0.1pixel in a given 10-day period, if all the subnetwork inputs have valid..."), how can one pixel have more subnetworks? Please improve the formulation.

**Response:** Sorry for the confusing expression. We have rewritten the paragraph as: '*Therefore, we divided the global extent*

120 *except the polar areas (80°N~60°S) into 140×360 zones. Here, for a 0.1° pixel during a specific 10-day period, if all the input data (soil moisture products and quality impact factors) have valid values, it can provide one valid data point. So, the maximal number of valid data points applied to train a subnetwork = 100 × the number of 10-day periods within the training period. The subnetworks with valid data points less than 100 (e.g. those in oceans) were dropped, leaving usually >15,000*

*subnetworks included in an independent neural network.*'

**Reviewer 1 Specific comment 15.** L222: What is an "individual neural network"? Is it the collective of all zonal neural networks for one simulation (SIM-T1, SIM-T2, ...)? Is the maximum possible number of subnetworks 50.400 or less because of ocean cells?

**Response:** We have changed it to '*independent neural network*' to make it consistent with the expression in the abstract. An independent neural network is the collective of all zonal subnetworks. Several independent neural networks constitute a simulation substep (for example, NN-1-1, NN-1-2, …, NN-1-8 are applied in Round 1- Substep 1), while each substep is responsible for one simulation (there are eight simulations: SIM-1-1, SIM-1-2, SIM-2, SIM-3-1, SIM-3-2, SIM-4-1, SIM-4-2 and SIM-5, for example, SIM-1-1 is the output of Round 1- Substep 1).

The number of subnetworks in each independent neural network is far below 50400, not only because of ocean zones, but also because some soil moisture data is only available in a region (e.g. TMI is available within [-40°S~40°N]). The paragraph is revised as in the response to Reviewer 1 Specific comment 14.

(**Please also note that SIM-1T, SIM-2T, …, SIM-4T are only the postprocessing results that are intended to be used as secondary training targets, while SIM-1, SIM-2, …, SIM-5 are combined to constitute our soil moisture products.)

140 **Reviewer 1 Specific comment 16.** L223: For reproducibility, it is required to state exactly the MATLAB version and the toolbox version and method/function name that was used for training the neural network.

**Response:** We have added the MATLAB version 2016a accordingly.

**Reviewer 1 Specific comment 17.** L256: "we classified all pixels" -> "we classified all 0.1 pixels", I suggest to add the resolution information that it is clear which of the different grids is addressed.

145

**Response:** We have revised it accordingly.

**Reviewer 1 Specific comment 18.** L259: Again, I thought that a pixel is the smallest unit in the process (i.e. subnetwork). So how can a pixel have a subnetwork? Not clear to me.

150 **Response:** We are sorry for the unclear description. Actually, the subnetwork belongs to a 1° ×1° zone, not a pixel. We have revised it as: '*For data simulation in a 0.1° pixel, the most preferable independent neural network is expected to be trained using all the available soil moisture data sources in that pixel. However, in the 1° zone where it is located, the subnetwork belonging to that preferable independent neural network may not exist due to limited valid data points (see section 2.2.1). Then, an alternative subnetwork driven by the combination of fewer soil moisture data inputs should be applied instead.*'

155

**Reviewer 1 Specific comment 19.** L261-262: "Hence, it is a ..." sentence seems incorrect. I think you should better write "neural network collocation" or "neural network constitution" to make it more clear that these are neural network realizations with identical configuration but different ingredients.

**Response:** We have changed it to 'neural network collocation' accordingly.

160

**Reviewer 1 Specific comment 20.** L272,815 and other occurrences: it is not clear how the 10 day periods are defined and how they relate to the ordinal numbering. A month has between 29 and 31 days, so how are the periods split and how does that affect the last 3rd where the number of days is variable? How does this variable length averaging affect the results and what are the implications for validation?

165 **Response:** The first and second 10-day periods in a month both contain exactly 10 days, but the last 10-day period has variable number of days (9(8)~11). This, however, may not have substantial effect on our results and data quality. This is because it takes at least three days for a microwave sensor to cover the globe. Also, for each grid, the days with observations are not the same among different sensors. Therefore, this study only took the average of the available soil moisture retrievals during a 10-day period (we have added a paragraph in section 2.1.1 as: '*To reduce noises and fill the gaps between sensor observing*

170 *tracks (it takes at least 3 days for a microwave sensor to cover the whole globe), for every soil moisture product, both the daytime and nighttime observations within each 10-day period are combined by data averaging (the relative superiority of*

*daytime and nighttime retrievals is not considered). For example, for SMAP, 11% of global land surface has data for only 5 days or less within a 10-day period.*'). Moreover, surface soil moisture may vary significantly even in a day, due to rainfall events, but the observations are transient. Therefore, either the 10-day averaged microwave soil moisture products or the

175 simulated soil moisture data in this study can only roughly indicate the overall soil moisture condition, not exactly equals to the mean soil moisture during 10 complete days. Hence, it doesn't matter whether the 'last 10-day period' in a month has exactly 10 days or not. In fact, this data format is exactly the same as the ASCAT-SWI soil moisture and many other products (e.g. LAI) developed by the ESA- Copernicus Land Monitoring Service (https://land.copernicus.eu). For the validation process based on ISMN measurements, the mean in-situ soil moisture in the 'last 10-day period' of a month was also calculated by

180 averaging the records in either 10 days or 11 days (or 8~9 days in February), which was consistent with the 'nominal' simulation period. Following this comment, we added this information: '*The temporal resolution is approximately 10 days, or to be specific, there are 3 data records within a month, for days 1~10, 11~20 and from 21 to the last day of that month, respectively.*' to the abstract in the revision.

185 **Reviewer 1 Specific comment 21.** L270-292: also this section would greatly benefit from a timeline bar plot that shows all the soil moisture products and simulated models, so that the overlaps can be grasped Immediately.

**Response:** We have added Figure 1 which shows the timelines of the simulated soil moisture corresponding to 8 substeps and the periods of data inputs for the trainings of 67 independent neural networks. Please find the detail in response to Reviewer 1 Specific comment 7. We also added a sentence: '*The training period of each neural network and the simulation period of*

190 *each substep are shown in Figure 1.*' in this paragraph.

**Reviewer 1 Specific comment 22.** L318: define how R**2 is computed (based on Spearman or Pearson).

**Response:** The $R^2$ is computed based on Pearson, we have added this information in the revision.

195 **Reviewer 1 Specific comment 23.** L321: lower case r should be used for the correlation coefficient (based on Pearson?). Why

are you mixing r and $R^2$ and do not use $R^2$ for all analyses?

**Response:** Following this comment, we have changed R to '$r$' to represent the Pearson correlation coefficient, including those in figures and tables. In this way, we can better distinguish the correlation coefficient from $R^2$.

To evaluate the overall performance, we showed the scatter plot between the simulated soil moisture and the measured values. Here, instead of $r$, we used $R^2$ better reveal the differences among the performances of different soil moisture products. However, in the temporal and spatial validation, at some sites or during some specific 10-day periods within a climatic region, the simulations and measurements can be negatively correlated, as shown in the figures within manuscript, which are actually of very low quality. But if we only use $R^2$, these low-quality data will be overshadowed (for example, if the $r$ is -0.6, the $R^2$ can be as high as 0.36). Therefore, it's wiser to use temporal correlation and spatial correlation. Previous studies also used '$r$' to evaluate the spatial and temporal accuracy of surface soil moisture products against ISMN measurements, for example (Karthikeyan et al., 2017).

**Reviewer 1 Specific comment 24.** L322: please provide formula for A.R computation

**Response:** 'A.R' in this study stands for the correlation coefficient ($r$) between the anomalies of simulated soil moisture and the anomalies of measured soil moisture at a specific ISMN station. Following this comment, we have added the equation blow to show how the anomalies of simulated or measured surface soil moisture were calculated in the revised manuscript.

$$\overline{SSM(k)} = \frac{\sum_{y=1}^{ny} SSM(y,k)}{ny} \quad (ny \geq 3) \; ; \; SSM \; is \; either \; estimated \; or \; measured$$

$$SSM: surface \; soil \; moisture; \; k: the \; ordinal \; of \; 10 \; day \; period \; in \; a \; year;$$

$$y: a \; year \; with \; measured \; SSM \; in \; the \; k^{th} \; 10 \; day \; period; \; ny: number \; of \; those \; years$$

$$SSM_{anom}(y,k) = SSM(y,k) - \overline{SSM(t)}$$

$$SSM_{anom}(y,t): the \; anomalies \; of \; surface \; soil \; moisture \; during \; the \; t^{th} \; 10 \; day \; period \; in \; year \; y.$$

**Reviewer 1 Specific comment 25.** L326: "in all grids", grids or pixels (1 x 1 or 0.1 x 0.1)?

**Response:** Thanks for careful reading. We revised it as '*in all 0.1° grids*'.

220

**Reviewer 1 Specific comment 26.** L326: please provide formulas for spatial pattern validation (at least in the supplement)

**Response:** We have provided more details for spatial pattern validation. Now it reads: '*Finally, we performed spatial pattern validation. In detail, for every 10-day period, we compared all the soil moisture measurements that were upscaled to 0.1° during that period with the corresponding estimated values. The spatial pattern evaluation indexes include correlation*

225 *coefficient (r), RMSE, bias and ubRMSE values (Eq. 2).*'

$$\overline{SSM_{est}} = \frac{\sum_{i=1}^{ng} SSM_{est,i}}{ng} \; ; \;\; \overline{SSM_{act}} = \frac{\sum_{i=1}^{ng} SSM_{act,i}}{ng} \; (ng \geq 20)$$

$$i: a\; grid\; with\; upscaled\; surface\; soil\; moisture\; measurements\; during\; a\; specific\; 10\; day\; period;$$

$$ng: the\; number\; of\; those\; grids\; in\; the\; globe$$

$$ubRMSE_{spatial} = \sqrt{\sum_{i=1}^{ng} [(SSM_{est,i} - \overline{SSM_{est}}) - (SSM_{est,i} - \overline{SSM_{act}})]^2 / ng} \quad (Eq.\; 2)$$

230

**Reviewer 1 Specific comment 27.** Figure 3: Use identical labels for the x-axis, add missing lower frame.

**Response:** We uniformed the labels for the x-axis. The figures have been adjusted accordingly. The revised Figure 3 (Figure 5 in the revised manuscript) is shown below:

[Figure]

235 *Figure R2: Comparison between the temporal accuracy of RSSSM and SMAP in regions with different Köppen-Geiger climate types. The four indexes are (a) r, (b) RMSE, (c) ubRMSE and (d) Anomalies r (A.R). The lengths of the error bars are 1.5 times that of the interquartile range, while the upper and lower boundaries and the central lines of the boxes indicate the 75th, 50th and 25th percentile values, with mean values marked by '×' (the forms of all the following boxplots are the same).*

240 **Reviewer 1 Specific comment 28.** Figure 4: If the color key is put below the figure, the figure can be increased in the horizontal direction which leads to wider bars. You could even remove the x-axis labels and names and leave only the lowermost. By this you can increase the size of the bars and hence the readability (reduce redundancy).

**Response:** We moved the color key below the figure, and removed the x-axis names to increase the size of the bars (shown in Figure R3, Figure 6 in the revised manuscript). Other figures were also revised to be larger and clearer.

[Figure]

245

*Figure R3: Comparison between the spatial pattern accuracy of RSSSM and SMAP in different 10-day periods during April 2015~2018. The three evaluation indexes are: (a) r, (b) RMSE and (c) ubRMSE. The length of each box/error bar is determined from the evaluation index values in three (January to March) or four (April to December) years.*

250 **Reviewer 1 Specific comment 29.** L381: How is the performance of SIM if the SMAP training period is omitted, i.e. from 2003 until 2015D01, as compared to ASCAT-SWI?

**Response:** We have added this information in the manuscript accordingly. It reads as: '*If the data period of SMAP (2015D10~2018) is excluded, the overall $R^2$ and RMSE for RSSSM are 0.43 and 0.087, still better than ASCAT-SWI ($R^2$=0.33,*

*RMSE=0.1).'*

255

**Reviewer 1 Specific comment 30.** Figure 7,10: As for fig. 4 place color key below the plots and increase the bars horizontally.

**Response:** We have revised all the figures accordingly.

**Reviewer 1 Specific comment 31.** Do you see any chance to improve the temporal resolution of the product in the future? If

260   not, what are the constraints?

**Response:** Currently, it's probably not a good choice to further increase the temporal resolution of the long-term microwave surface soil moisture product. As illustrated in the response to Reviewer 1 Specific comment 20, for each grid, both the days and the hours of observations by different microwave sensors differ from each other. Because surface soil moisture has high variability in a short time period (even in a day) due to rainfall events, the actual soil moisture at the passing time of various

265   sensors are not the same. However, this study used multiple sources of microwave surface soil moisture products as predictors of neural networks and SMAP soil moisture data as the training target, meaning that the neural network training should be based on the assumption that the products retrieved by different sensors contain exactly the same actual soil moisture information. As we can see, there exists a conflict. To solve it, for each sensor, we took the average of its available retrievals during a certain time period, and the larger number of data applied for averaging is, the better can the result represents the

270   mean soil moisture during that period. Because 11% of global land only have 5 or less days with observations during a 10-day period, if the temporal resolution is improved, for example, to 5 days, there may be only 2~3 observations available. Considering the high temporal variability of surface moisture, the average of those limited data can hardly indicate the average soil moisture condition. This will lead to large uncertainties in the neural network training, and finally, our soil moisture simulation results.

275   This problem is an inherent constraint of microwave remote sensing data integration. So, to improve the temporal resolution, other data sources need to be incorporated. As we know, soil moisture retrieved from other remote sensing techniques (e.g. optical) have low quality over vegetated areas, and are heavily interfered by clouds. Therefore, model simulation may be the

only solution to this problem. For example, assimilating the observational-based surface soil moisture into models such as
GLEAM can achieve surface/root-zone soil moisture mapping at daily scale. Therefore, we have revised the last paragraph as:

280 '*Another way to improve global surface soil moisture data accuracy as well as the temporal resolution is to combine satellite-based products with land surface models such as GLEAM. Remote sensing inversion can delineate more detailed spatial information on soil moisture, whereas the reanalysis-based models have advantages in characterizing temporal variations, and even on daily scale, except for…*'.

285 **Reviewer 1 Specific comment 32.** L499-500: Is SIM also superior to the other products if only the prior to SMAP period is considered (2003 until 2015D01)?

**Response:** Thank you for this advice. We have added the following comparisons to the manuscript.

*1) if the data period of SMAP (2015D10~2018) is excluded, the overall $R^2$ and RMSE for RSSSM are 0.43 and 0.087, still better than ASCAT-SWI ($R^2$=0.33, RMSE=0.1).*

290 *2) when excluding the SMAP (training target) data period, the $R^2$ and RMSE for RSSSM are 0.41 and 0.089, also superior to those for GLDAS ($R^2$: 0.37; RMSE: 0.099).*

*3) without considering the SMAP period, the condition is the same ($R^2$ for RSSSM and ERA5-Land are 0.41 and 0.38; RMSE for these two products are 0.089 and 0.125).*

*4) when the SMAP data period is excluded, the $R^2$ and RMSE for CCI are 0.028 and 0.098, compared to 0.41 and 0.089 for*
295 *RSSSM.*

*5) if the SMAP data period is excluded, RSSSM's $R^2$ and RMSE are 0.41 and 0.089, still better than both GLEAM v3.3a ($R^2$: 0.35; RMSE: 0.141) and GLEAM v3.3a ($R^2$: 0.34; RMSE: 0.128.*

So, the comparisons above can prove that our product (RSSSM) is superior to the other products even if the SMAP period is excluded.

300

**Reviewer 1 Specific comment 33.** Are there plans to update the data-set on a regular basis?

**Response:** Yes. We plan to update the whole dataset when more advanced microwave sensors (e.g. P-band sensors) are launched and the global-scale higher-quality surface soil moisture data is available in future. We have added this promise in the discussion. Now it reads: '*Therefore, if microwave sensors with higher SNR or better penetration of vegetation canopy*

305     *than SMAP are launched in future (for example, the upcoming P-band microwave sensors (Etminan et al., 2020; Ye et al., 2020)), by using the soil moisture or Tb retrieved from the new sensors as the reference, we can develop a temporally continuous soil moisture dataset since 2003, which is expected to have even higher accuracy than the SMAP product (we will update the complete RSSSM product then). In that sense, the data fusion algorithm proposed here will be very meaningful in future.*'

310

**Reviewer 1 Specific comment 34.** The dataset is organized as an archive of geotiff files. The problem with this structure is that the time identifier is only contained in the file name, but without practical formatting. If one wants to import a time series for a region or a single pixel, the data structure is quite unhandy. Also from the readme file and the metadata it is not quite clear what the 10 days ordinal numbering means exactly. Is it always the [1-9],[10-19],[20-29] or [1-10],[11-20],[21-30]

315     periods? How are the months with variable length considered (28,29,30,31 days)? That's not clear also not from the manuscript. Further, I would suggest to add a table (csv) that links the different file names to their
specific period using ISO 8601 https://en.wikipedia.org/wiki/ISO_8601 notation:
e.g., a file named inventory.dat with a list like the following one:
Period, Filename 2003-01-01/2003-01-10, SMY2003DECA01.tif 2003-01-11/2003-01-20, SMY2003DECA02.tif ...

320     **Response:** Thank you for the suggestion on the naming and structure of our data. Actually, it is [1-10], [11-20], [21, the end of each month]. We have added a csv table named 'filename' linking the different file names to their specific period, following your instructions.

**Reviewer 1 Specific comment 35.** Also the numbering should be formatted as %02d so that, e.g., SMY2003DECA1.tif

325     becomes SMY2003DECA01.tif. This is important if one wants create a chronological file list for looping over time. With the

current scheme, the order would become SMY2003DECA1.tif SMY2003DECA10.tif, SMY2003DECA11.tif, ... This should be also applied to all tables in the manuscript (e.g., 2005D01 instead of 2005D1).

**Response:** We have changed the naming of the product as well as the abbreviations for each 10-day period, both in the manuscript and in the Supplement.

330

**Reviewer 1 Specific comment 36.** Figure S1: The figure and description is not completely clear. I assume that every number (yellow and blue frames) is one pixel (0.1x 0.1)? I think it would become more clear if you superimposed a light gray mesh for the pixels over the 1x1zones. But then, why are there 4 steps required to smooth the borders? It means that every boarder gets smoothed twice, and every corner point even four times.

335 **Response:** Sorry for the unclear description. The figure has been revised (see Figure R4), with light gray mesh superimposed for all the pixels over one 1°×1° zone. The four steps were used to process respectively the four borders of each 1°×1° zone (please note that in each step, only the border colored in blue is smoothed). We have added more details to make it clearer, which now reads as: '*The sketch of the four substeps in boundary fuzzification. The 1°×1° zones are separated by black solid lines (the 0.1°×0.1° pixels in one zone are superimposed by light grey mesh). For each substep (subfigures a~d), the soil*

340 *moisture value within each pixel that is colored in blue are recalculated as the average of its original surface soil moisture and the original soil moisture value in its most adjacent yellow color pixel, weighted by the corresponding numbers labelled (i.e. 2 and 1). In this way, every border of a 1°×1° zone gets smoothed once (substeps 'a~d' are for four borders, respectively, where a~b are for the horizontal borders while c~d are for the vertical borders), but the four corners get smoothed twice (both horizontally and vertically).*'

[Figure]

345

*Figure R4. The sketch of the four substeps in boundary fuzzification. The 1°×1° zones are separated by black solid lines (the*

*0.1°×0.1° pixels in one zone are superimposed by light grey mesh). For each substep (subfigures a~d), the soil moisture value*

*within each pixel that is colored in blue are recalculated as the average of its original surface soil moisture and the original*

*soil moisture value in its most adjacent yellow color pixel, weighted by the corresponding numbers labelled (i.e. 2 and 1). In*

350 *this way, every border of a 1°×1° zone gets smoothed once (substeps 'a~d' are for four borders, respectively, where a~b are*

*for the horizontal borders while c~d are for the vertical borders), but the four corners get smoothed twice (both horizontally*

*and vertically).*

**Reviewer 1 Specific comment 37.** Figures S5, S8, S11: put the color-key to the bottom of the figure (a single key would be
355  sufficient for all sub-figures), you could even remove the x-axis labels and names and leave only the lowermost. By this you
can increase the size of the bars and hence the readability (reduce redundancy).

**Response:** We have revised these figures, making the size of the bars much larger now by adjusting the place of the color keys
and removing the x-axis names.

360  **Reviewer 1 Specific comment 38.** There are often blanks missing between words. DOIs are completely missing in the
reference list.

**Response:** Sorry for those mistakes. We have added the missing blanks and rewritten the reference list (add DOIs, abbreviate
the journal names, correct wrong references) to make sure it meets the format requirement of ESSD. Thank you again for your
careful reading and valuable suggestions.

365

**References**

[revised manuscript text omitted]

**To Reviewer #2:**

435 We thank referee#2 for the valuable comments that will help us in improving the quality and readably of the manuscript. We have carefully revised the MS following your comments and suggestions. We provide a detailed response to the Referee's comments in the Supplement.

\*\*\*\*\*\*\*\*\*\*

440

**Reviewer 2 General comment.** The authors tried to generate long-term surface soil moisture at a global scale, via data fusion of 11 microwave remote sensing-based soil moisture products since 2003 through neural network approach, and SMAP soil moisture products were used as the training target. The idea is very interesting and should be encouraged to explore further how much extent the machine learning can help in Earth Observation for delivering physically-consistent (or physic-aware)

445 products. However, the way the current manuscript is written, organized is still far from clarity, structured for this reviewer to comprehend their contributions. I would suggest rejection and encourage the author to continue along this line of effort.

**Response:** We thank the reviewer for the positive comment on the idea of generating long-term surface soil moisture at a global scale in this study. We agree that the organization of this manuscript, especially the explanation of the method is not clear, and sorry for some confusing or wrong terminologies. These problems will make the readers and reviewers

450 misunderstand several rather complex algorithms, which however, are also the major innovation points of this study, including the selection of quality impact factors as neural networks inputs and the design of five rounds of simulations (the organization structure of 67 independent neural networks). We have carefully followed your advice, revising the explanations of the key methods. In addition, for each of your doubt or query, we explained our design of the method in greater detail, and have clarified the related sentences in the manuscript, hoping that our real contributions could be comprehended by you and other

455 readers. Please find the details in the response to each of the following comments.

**Reviewer 2 Major comment 1.** The author claimed that "This new dataset, once validated against the International Soil Moisture Network (ISMN) records, is supposed to be superior to the existing products (ASCAT-SWI, GLDAS Noah, ERA5-

Land, CCI/ECV and GLEAM), and is applicable to studying both the spatial and temporal patterns. " This assumption is too strong. On the other hand, it seems the author referred to the validation of the NN-based 10-d soil moisture products versus the 10-d averaged ISMN in-situ observations (as seen in Figure 5, Figure 8, Figure S3, S6, and S9). Is it true? In any case, it should be specified under what conditions the generated product is performing better than other products. "supposed to be superior" is really not a scientific statement.

**Response:** Thank you for this comment. In this study, for our product (named RSSSM hereinafter) and each existing product, by referring to all valid ISMN sites' surface soil moisture measurements, we carefully conducted overall validation (evaluation indexes are overall $R^2$ and RMSE values), temporal variation validation (evaluated by temporal correlation coefficient, temporal RMSE and unbiased RMSE, etc.) and spatial pattern validation (evaluated by spatial correlation coefficient, spatial RMSE and unbiased RMSE, etc.). Please see Method section 2.3 in the revised manuscript for details, while the accuracy comparison among all products are in Result section 3.2. The validation results indicate that our RSSSM product is more comparable to the site measurements, both in terms of $R^2$ and RMSE (see Figure 7, Figure 10, Figure S3, Figure S6, Figure S9 in the revised manuscript and Supplementary). For temporal variation accuracy, RSSSM is proven to be better than ASCAT-SWI, GLDAS and CCI, both in temporal correlation and RMSE, especially in arid regions, relatively cold areas (Figure 8, Figure 11, Figure S7 and Table 2). The temporal accuracy of RSSSM is similar to ERA5-Land and GLEAM v3.3 products (temporal correlation is somewhat lower, but the temporal RMSE value of our product is lower, see Figure S4, Figure S10 and Table 2). For spatial pattern accuracy, our RSSSM product is found superior to all other products (please refer to Figure 9, Figure 12, Figure S5, Figure S8, Figure S11 and Table 3), almost all year round, especially during the growing seasons. Based on these findings, we propose that our product (RSSSM) have better agreement with the site-measured surface soil moisture than the five existing soil moisture products (ASCAT-SWI, GLDAS Noah, ERA5-Land, CCI/ECV and GLEAM). Moreover, the observational-based soil moisture, CCI, has limited spatial coverage and significantly reduced data accuracy before 2012, while ASCAT-SWI is only available since 2003. These problems have been well solved by our estimation (the data quality is maintained during 2003~2018, see Figure 13). We agree with you that the phrase 'supposed to be superior' is not a scientific statement, and the claim is probably too strong and condescending. So, we corrected the sentences as: '*This*

*new dataset, named RSSSM, is proved comparable to the in-situ surface soil moisture measurements at sites of the International Soil Moisture Network (overall $R^2$ and RMSE values of 0.42 and 0.087 $m^3$ $m^{-3}$), while the overall $R^2$ and RMSE values for the existing products (ASCAT-SWI, GLDAS Noah, ERA5-Land, CCI/ECV and GLEAM) are within the range of 0.31~0.41 and 0.095~0.142 $m^3$ $m^{-3}$, respectively. The advantage of RSSSM is especially obvious in arid or relatively cold areas, and during growing seasons. Moreover, the persistent high data quality as well as complete spatial coverage ensure the applicability of RSSSM to both the spatial and temporal pattern studies.'* We have also corrected all the relevant unclear statements (e.g. supposed superior, expected to be better, …) throughout the manuscript.

490

**Reviewer 2 Major comment 2.** There were some strange 'terminologies' the author used for discussion, for example: a. 'penetrability of microwave' (which is seldom found in the literature. A more widely used term is 'microwave penetration depth'); b. "Soil moisture retrieval from passive microwave sensors is based on the correlation between soil dielectric conductivity, that is influenced by soil moisture …'. Following the theoretical development of soil moisture retrievals from remote sensing, the relationship between soil moisture and dielectric constant is the fundamental (not soil dielectric conductivity).

**Response:** We are sorry for the unsuitable terminologies. we corrected the sentence 'the penetrability of microwaves is usually <5 cm of soil' to '*current satellite microwave sensors can only detect soil moisture within top 5 cm of soil*' following this comment and your Specific comment 5, and also corrected '*dielectric conductivity*' to '*dielectric constant*'.

500

**Reviewer 2 Major comment 3.** "However, this data is regional, with a large temporal gap, and cannot be seen as observational-based only since precipitation data is incorporated." This is a very strange argument. We all know there is a strong link between precipitation and soil moisture variation. Physically speaking, one used the antecedent precipitation index to understand how precipitation events drive the variation of soil moisture. This is like one of 'quality impact factors'. If the above argument is true, we can argue that the author's approach in this manuscript is also not 'observation-based', as they used LAI, land cover, LST, and many other factors.

**Response:** We checked the work by *Qu et al.* (Qu et al., 2019) carefully, and found that precipitation data was actually not applied as an input of the random forest in that study.

Furthermore, other recent studies focusing on long-term soil moisture mapping based on microwave remote sensing data did not incorporate precipitation as ancillary inputs of neural networks as well (Santi et al., 2016; Yao et al., 2019; Yao et al., 2017).

Hence, we conducted a research on the role of precipitation in neural network training to further explore the reasons.

Because it takes at least 3 days for a microwave sensor to cover the whole globe, for 11% of global land, there will be only 5 or less observations for random days within a 10-day period. By taking the average of these available data, this study only focuses on the mean soil moisture condition during that 10-day period. Then, to see how much can the incorporation of precipitation data improve the neural network training efficiency, we calculated 10-day averaged GPM Final-Run precipitation, which can well indicate the overall precipitation water availability (the antecedent precipitation index is not used, because it must be calculated on daily scale, and the attenuation coefficient is hard to determine at global scale (Kohler and Linsley, 1951)). Taking the first primary independent neural network, NN1-1-1, as an example, we performed contribution tests on all the input features at the global scale (not for each separate zone), including 9 'quality impact factors', 4 predictor soil moisture products and precipitation - a potential ancillary soil moisture indicator. For each predictor, we added a random error that is controlled within the standard deviation of the predictor, and then the increased MSE in neural network training can indicate the relative contribution of that variable. The results (see Figure R1a, that is Figure S1a in the revised Supplement) show that precipitation will only contribute to 1.7% of the training efficiency, which is much lower than the contribution of any soil moisture product (the total contribution fraction of the four soil moisture products is 61.2%), and is also lower than that of most 'quality impact factors'. This suggests that microwave soil moisture datasets together with several 'quality impact factors' of microwave soil moisture retrieval are enough to predict the training target- SMAP soil moisture, and there is no need to add precipitation as another ancillary index of soil moisture.

'Quality impact factors' are defined in this study as the variables that will have a significant impact on the retrieval errors of soil moisture by microwave remote sensing (section 2.1.2). Although the relative performances of different soil moisture

products is related to surface moisture condition (Kim et al., 2015), it is found mainly due to the less vegetation in arid areas. After all, no explicit mechanism can support the idea that the retrieval errors of soil moisture are significantly influenced by water availability. Even if this is true, the soil water availability can be already indicated by the microwave soil moisture products. So, it is unreasonable to incorporate the precipitation variable as a 'quality impact factor'. On the other hand, LAI,

535     water area fraction, LST, land use cover, tree cover fraction, non-tree vegetation fraction, topographic complexity, and soil sand/clay fractions all have direct impacts on the microwave soil moisture retrieval errors, with solid physical mechanism (see section 2.1.2). Therefore, theoretically they should be added to the neural network, even though the land use cover type and soil sand fraction data prove to have limited contributions to NN training efficiency.

One may argue that if NARX (nonlinear autoregressive with external input) is applied instead, in which the soil moisture in

540     the previous 10-day period is also incorporated as a predictor, precipitation data can be very beneficial to the neural network training. This is true, because precipitation directly contributes to soil moisture increases. However, NARX is not suitable for global-scale long-term continuous soil moisture mapping, because the base map (i.e., the soil moisture in the beginning of the simulation period) is hard to determine. Moreover, in mid to high latitudes, the lack of soil moisture retrievals over frozen ground in winters will lead to missing data there in summers when soil moisture data is otherwise available. So, if NARX is

545     adopted, we can only estimate long-term soil moisture in tropics and sub-tropics with air temperature consistently higher than 0 ℃. Last but not least, if the soil moisture in the previous phase and the current precipitation amount are both incorporated, they will largely conceal the role of satellite-observed signals. As shown in Figure R1b (Figure S1b in the Supplement), the total contribution fraction of all four microwave soil moisture products is reduced to only 10.6%, while the roles of ASCAT, AMSR2-JAXA and AMSR2-LPRM are all negligible. Without taking full advantage of remote sensing, the simulations based

550     on previous soil moisture and current precipitation will lead to errors over places where soil moisture gains are mostly driven by glacier melting, or in places with high levels of radiation-driven surface soil evaporation. The reliability of the derived soil moisture will be lower in irrigated croplands and afforestation/deforestation areas as well.

[Figure]

[Figure]

 *Figure R1: (Figure S1 in the revised Supplement): The roles of different input features in the soil moisture simulations based on BP neural networks and nonlinear autoregressive with external input (NARX) with microwave soil moisture products incorporated: (a) the contributions of different input features of a primary neural network: NN1-1-1, including 4 predictor soil moisture products, 9 quality impact factors of microwave soil moisture retrieval, plus 1 probable ancillary soil water indicator: 10-day averaged precipitation, to the neural network training efficiency indicated by the increased MSE; (b) the*
 *contributions of all the input features to the training efficiency, if NN1-1-1 is changed into a NARX, in which the SMAP soil*

*moisture for the previous period is also applied as a predictor.*

On account of all above, precipitation data is neither included as an ancillary soil moisture indicator, nor added as a 'quality impact factor' in this study.

Following this comment, we have added the above explanations to the Supplementary Data (Text S1), replacing the previous paragraph. We also added a sentence in the revised manuscript: '*The contribution analysis results (Figure S1) show that because various microwave soil moisture data have already been included, precipitation data is not an essential indicator of soil moisture, and is not utilized as a physically-based 'quality impact factor' either (see Text S1 for detailed explanations).*'

Actually, in Qu et al. (2019)'s study, the random forest's input features only include: microwave Tb products, DEM, IGBP global vegetation classification, latitude, longitude, and DOY. So, we feel sorry that the latter half of this sentence was not correct, which we have deleted. The sentences now reads: '*Another study rebuilt a soil moisture time series over the Tibetan Plateau by using SMAP data as the reference of a random forest (Qu et al., 2019). For environmental factors, while vegetation cover is not considered, elevation (DEM), IGBP land use cover type, grid location and the day of a year (DOY) are chosen as ancillary inputs. The training $R^2$ in this region reached 0.9, with a temporal accuracy higher than other products (temporal r=0.7; RMSE=0.07 in the unfrozen season). However, this data is regional (for Tibetan Plateau only), and with a temporal gap between AMSR-E and AMSR2 (October 2011~June 2012).*'.

**Reviewer 2 Major comment 4.** " are these factors used as direct spatial predictors of soil moisture or just because they are related to the errors of satellite soil moisture retrievals (i.e., the quality impact factors of soil moisture)? We insist on the latter, proposing two main reasons for the incorporation of environmental factors." This is very confusing and not necessarily correct, and not well grounded. We know the soil moisture retrieval from remote sensing is using a radiative transfer model to account for scattering and emissions from both soil and vegetation, which is conflicting with the author's statements.

**Response:** Sorry for the unsuitable sentences. We have re-written them as follows: '*Environmental factors, including DEM, LST and vegetation cover (indicated by NDVI, MVI, etc.), were used as ancillary neural network inputs for improved soil moisture simulation (Lu et al., 2015; Qu et al., 2019; Yao et al., 2017). According to these studies, these factors alone may not*

585 *predict surface soil moisture well without the incorporation of any microwave remote sensing data, which can also be justified*

*by the contribution analysis results (Figure S1a). This is because although they are somewhat related to soil moisture (e.g.*

*soil moisture is limited in areas with low vegetation cover in general but high in forests (McColl et al., 2017)), the relationships*

*are rather uncertain (e.g. at smaller scales, LAI may however have a negative influence on soil moisture due to the variation*

*in evapotranspiration (Naithani et al., 2013), or without clear impacts (Zhao et al., 2010); also, soil moisture can be either*

590 *high or low in summers when vegetation peaks (Baldocchi et al., 2006; Méndez-Barroso et al., 2009)). However, these factors*

*are quite essential due to their direct impacts on soil moisture retrieval through radiative transfer model using microwave*

*remote sensing data (Fan et al., 2020), and are retrieval quality impact factors. The detailed explanations are: 1) ...'.* On the

other hand, we agree that the precipitation and open water fraction can directly indicate the surface soil moisture (but because

the microwave soil moisture products are already applied, precipitation data was not included as a predictor, see the detailed

595 explanations in response to Reviewer 2 Major comment 3).

We understand that soil moisture retrieval from microwave remote sensing accounts for the scattering and emissions from

both soil and vegetation, so that these factors can have direct impacts on the microwave soil moisture retrieval results, and are

closely related to the retrieval errors, as we explained in the revision.

600 **Reviewer 2 Major comment 5.** 'Water Body' was used as one of the predictors (it should be predictor, rather than quality

impact factors). This is very strange. As we know, water body map in either SMOS or SMAP soil moisture products were used

to mark out those locations to avoid soil moisture retrievals over these water bodies (otherwise, it would be physically no

sense, in terms of soil moisture). This is wrong and not physically sound to include water bodies as one of predictor for

predicting surface soil moisture.

605 **Response:** We are sorry for the confusion. We agree that water is a direct indicator of surface soil moisture. However, 'water

fraction', rather than 'water body', is used as both a quality impact factor and a potential indicator of surface soil moisture

content. Water bodies (large lakes, oceans) are masked out in both existing (SMOS or SMAP) soil moisture products and our

simulation product (RSSSM). However, in a grid with size of $0.1°×0.1°$ (approximately 120 km$^2$), there could be a small

fraction of water, which may be rivers, streams, ponds, partly inundated wetlands or just paddy croplands. If we mask out all

610  0.1° resolution grids with even 1% of water (or less), there will be no data in many parts of the world, especially over humid

areas. Previous soil moisture products also produce valid values in those grids. However, water can dramatically lower the

brightness temperature (Tb), while different retrieval algorithms correct the impact of waters within grid differently, leading

to different biases and relative accuracy of soil moisture estimates in grids with water (Ye et al., 2015). For example, we noted

a strong underestimation of soil moisture by the NSIDC (National Snow and Ice Data Center) method (e.g., AMSRE-NSIDC)

615  over rivers and small lakes in compared to the nearby lands). Moreover, the sensitivity of different microwave sensors to water

fraction within grid may differ as well. Hence, the fraction of water (not only open water, but also inundated wetlands or

croplands) in the grid can significantly influence the retrieval errors and relative reliability of various soil moisture products

(Ye et al., 2015), which exactly meets the definition of 'quality impact factor'.

Therefore, this study uses 'water fraction' both as a quality impact factor and as an ancillary soil moisture indicator (please

620  also note that the word 'predictor' in this manuscript only refers to the existing soil moisture products that are applied as the

neural network inputs). This information has been added to the manuscript. For 'water fraction', the Surface WAter Microwave

Product Series (SWAMPS) dataset (Schroeder et al., 2015) was applied because it is microwave based, including not only

open water, but also partly-inundated lands. The contribution analysis on all the input features (see Figure R1) prove that the

calculated water fraction plays a very important role in neural network training.

625  Following this comment, we have clarified the descriptions. It now reads: '*The second is the 'water fraction factor' (i.e., the*

*fraction of water area in each pixel). Waters in land pixels dramatically lower the Tb, leading to overestimation of soil moisture*

*there. Because there are different methods used for detection and correction of small area of water, either open water, wetlands*

*or partly inundated wetlands and croplands (Entekhabi et al., 2010; Kerr et al., 2001; Mladenova et al., 2014; Njoku et al.,*

*2003), microwave soil moisture data calibration and weight assignment based on the water fraction within land pixels make*

630  *sense (Ye et al., 2015). In addition, water fraction is a direct indicator of surface soil moisture. In this study, daily water area*

*fraction derived from the Surface WAter Microwave Product Series (SWAMPS) v3.2 dataset (Schroeder et al., 2015) was*

*applied.*'. The confusing words 'water body' have been replaced by '*water area fraction*' in other parts of the manuscript as

well.

635 **Reviewer 2 Major comment 6.** For 'topographic complexity' 'soil texture', the author used from different sources, one from ASCAT ancillary data and the other use SMAP ancillary data. This reviewer is wondering why such a choice? Why not making it consistent (i.e., get ancillary data from one single product, instead of two?

**Response:** The reason we used different sources of data for 'topographic complexity' and 'soil texture' is the data availability. SMAP only uses GMTED 2010 DEM data to derive quality flags for data retrieved in mountainous areas, while topographic

640 complexity is only included as the ancillary data of ESA's ASCAT-SWI product, which was calculated by normalizing the standard deviation of GTOPO30 elevation in each grid point to values from 0 to 100 (Scipal et al., 2005), and is closely related to errors of surface soil moisture retrieved from microwave remote sensing. On the other hand, soil texture data is not contained in the static layers of ASCAT-SWI product, so we have to obtain this important data from SMAP ancillary input collection. Moreover, because the data sources of topographic complexity and soil texture both have relatively high quality, we suggest

645 that they could be used even they come from different soil moisture products. In the revised manuscript, we added this information: '*(topographic complexity data is not available from SMAP Constant; soil texture is not provided by ASCAT Constant)*'.

**Reviewer 2 Major comment 7.** '3σ denoise'. what is the effect of such a filter on identifying extreme years? For example,

650 during 2003, 2010, 2018, 2019 there are extreme heat events in Europe and the soil moisture is so dry which can be beyond the 3 standard deviations.

**Response:** The '3σ denoise' was conducted spatially, rather than temporally, to detect and delete the extreme values (usually salt and pepper noises in mountain areas) in each 1°×1° zone, during a certain 10-day period. So, if there are extreme heat or precipitation events, as you noted, the whole 1°×1° zone will see a sharp increase or decrease in soil moisture content in almost

655 all the 0.1° grids within that zone (there are <100 grids in each zone). Therefore, due to the increase/decrease in the zonal mean soil moisture value, the extreme weather events will not be removed by this 'spatial 3σ denoise' step.

We are sorry for the confusion. We have added more detailed explanations. It reads as: '*After standardization of the original soil moisture data, to improve the neural network training efficiency, the potential salt and pepper noises are removed. For each map (a specific 10-day period), within each 1°×1° zone, the soil moisture values are filtered to the level of three standard deviations relative to the mean in that zone. This preprocessing step is thus called '3σ denoise' (note that the denoise is conducted spatially, rather than temporally, so that the extreme events will not be treated).*'

**Reviewer 2 Major comment 8.** NN design. SMAP is only available after 2015, so I am not sure what is the meaning of simulation period 2012D19-2013D36, but also 2014-2018. I guess this is constrained by the available data (PROBA-V and GLASS LAIs)? But in any case, it does not represent any physical meaning to predict 2015data with 2012-2013 data. At least, the NN design is not clear on why it is designed as such.

**Response:** Sorry for the confusion. In the first round of simulation, the division of the simulation period into two subperiods: 2012D19~2013 and 2014~2018, is due to the available data periods of PROBA-V and GLASS LAI. However, we did not predict data in 2015 by using the data in 2012~2013. In this study, the common period of predictor soil moisture products applied in each 'substep' of NN training always includes the corresponding soil moisture simulation period.

We agree that the design of 67 independent neural networks, which are embedded in 8 substeps applied for five rounds of simulations, is quite complex. However, it ensures the long-term continuous satellite-based soil moisture mapping, almost full spatial coverage at the global scale, as well as high data accuracy. Following this comment, we have revised this section in the manuscript to make the NN design clearer. It reads as:

[revised manuscript text omitted]

The following is a plain language description, which you may choose to read if would like to understand the NN design deeper. First, considering that the temporal spans of different microwave sensors are all limited (see Figure R2 for details), we designed five rounds of neural networks to achieve long-term continuous soil moisture mapping, while ensuring that as

735   many microwave soil moisture products as possible are applied as predictors of each round of NN. In detail, SMAP soil moisture data is used as the training target of the first round NN (labeled as NN1), with ASCAT-SWI, SMOS, AMSR2-JAXA and AMSR2-LPRM-X applied as predictor soil moisture products. The potential training period of NN1 is the time period of SMAP (2015D10~2018, Table S1). Because the four soil moisture predictors all have data since 2012D19, the potential soil moisture simulation period is 2012D19~2018. However, because PROBA-V LAI (quality impact factor)

740   starts in 2014, the neural networks trained using PROBA-V LAI can only be used for the simulation during 2014~2018. For the remaining period (2012D19~2013), the applicable neural networks should be trained based on another LAI dataset- GLASS LAI, which covers from the beginning of our study period until 2017. Therefore, NN1 should be divided into two substeps. For substep 1 (marked by NN1-1), PROBA-V LAI is used, and the training period is 2015D10~2018D36 (Table S1), while the simulation period is 2014~2018 (Table S2). For substep 2 (denoted by NN1-

745   2), GLASS LAI is applied instead, and the training period is 2015D10~2018D36, while the simulation period is 2012D19~2013. Because each predictor soil moisture product has missing values in some specific areas (e.g. SMOS-IC do not have values in Eurasia), there are 1~4 predictor soil moisture products available in every 0.1° grid. While the maximum number of combinations are 4+3+2+1=10, 8 of them are valid since the soil moisture retrievals over snow or ice is not recommended (Table S2). Corresponding to these 8 combinations, 8 independent neural networks are trained,

750   each with a combination of predictor soil moisture products applied as neural network inputs (labeled as NN1-1(2)-1 ~ NN1-1(2)-8; for example, NN1-1(2)-1 is trained using all four predictor soil moisture products, and is the most preferable NN). However, even for a 0.1° grid with all four predictor soil moisture data available, we may not be able to simulate soil moisture there using NN1-1(2)-1. That is because the corresponding neural network, NN1-1(2)-1, may

not exist in the 1°×1° zone where the grid is located, due to limited valid data points available for zonal subnetwork training (please refer to revised Method section 2.2.1 for details on the localized neural networks). Under this condition, the other less preferable independent neural networks should be applied instead (the relative priority order of all independent neural networks within a substep is determined by comprehensively considering the number and quality of input soil moisture products, the variety of sensors, the quantity of training samples indicated by the number of 10-day periods, and the relative accuracy of training targets). After simulation, we combined the results for substep 1 (NN1-1-1~8), which is denoted by SIM-1-1 and the results for substep 2 (NN1-2-1~8): SIM-1-2 to obtain SIM-1. After further processing steps (section 2.2.2), we convert SIM-1 into the secondary training target, SIM-1T. For the second round of NN, the training target can be either SMAP, (rimary training target), while the training period is 2015D10~2017 (GLASS LAI is used, ASCAT-SWI, SMOS and FY data products are applied as predictors), or SIM-1T (secondary training target), while the training period is 2012D19~2015D10 (ASCAT-SWI, SMOS, FY and TMI products can all be applied). There are 8 independent neural networks included in round 2 NN (see Table S3), while the corresponding simulation output is SIM-2, covering the period of 2011D20~2012D18 since FY data product is available since 2011D20 (see Table S4).

The 3rd to 5th round of neural network training and simulations are even more complex (for example, in the 3rd round, the priority order of independent neural networks is not definite. Two probable orders are provided, leading to two substeps, and the simulation results of which are combined by taking the relative accuracy in each grid into account), but the basic principles are similar to those explained above (see Table S5~S15).

**Reviewer 2 Specific comment 1.** In the abstract, change 'elaborate' to 'elaborated', delete 'various', change 'simulation' to 'simulations'

**Response:** we have corrected them accordingly.

**Reviewer 2 Specific comment 2.** In the abstract, 'This new dataset, once validated against the International Soil Moisture Network (ISMN) records, is supposed to be superior to the existing products (ASCAT-SWI, GLDAS Noah, ERA5-Land,

CCI/ECV and GLEAM), and is applicable to studying both the spatial and temporal patterns.' This is a very strong assumption, and should be avoided. Otherwise, the corresponding results should be shown.

780 **Response:** We have corrected the unsuitable statement as: '*This new dataset, named RSSSM, is proved comparable to the in-situ surface soil moisture measurements at sites of the International Soil Moisture Network (overall $R^2$ and RMSE values of 0.42 and 0.087 $m^3$ $m^{-3}$), while the overall $R^2$ and RMSE values for the existing products (ASCAT-SWI, GLDAS Noah, ERA5-Land, CCI/ECV and GLEAM) are within the range of 0.31~0.41 and 0.095~0.142 $m^3$ $m^{-3}$, respectively. The advantage of RSSSM is especially obvious in arid or relatively cold areas, and during growing seasons. Moreover, the persistent high data*

785 *quality as well as complete spatial coverage ensure the applicability of RSSSM to both the spatial and temporal pattern studies.*'. Please also find the details in the response to Major Comment 1.

**Reviewer 2 Specific comment 3.** Lines 27~30: 'It has been endorsed by the Global Climate Observing System (GCOS) as an essential climate variable (Bojinski et al., 2014), probably the best indicator of ecological droughts (Martínez-Fernández

790 et al., 2016; Samaniego et al., 2018). However, due to the large uncertainty in global-scale soil moisture data, its applicability in global ecosystem models are currently limited (Hashimoto et al., 2015; Stocker et al., 2019).' What do you want to say here? What are points? It is suggested to shorten the sentence.

**Response:** Sorry for the too complicated sentences. We have shorten the sentence as: '*It has been endorsed by the Global Climate Observing System (GCOS) as an essential climate variable (Bojinski et al., 2014), probably the best indicator of*

795 *ecological droughts (Martínez-Fernández et al., 2016; Samaniego et al., 2018). However, due to the large uncertainty in global-scale soil moisture data, its applicability in global ecosystem models are currently limited (Hashimoto et al., 2015; Stocker et al., 2019).*'

'

**Reviewer 2 Specific comment 4.** Lines 31~33: 'The reanalysis land surface model products (e.g., the Global Land Data

800 Assimilation System (GLDAS, with spatial resolution of 0.25°) (Rodell et al., 2004), ECMWF ERA-interim (0.25°) (Balsamo et al., 2015) and its newly-published successors: ERA5 (0.25°) and the land product, ERA5-Land (0.1°)(Hoffmann et al.,

2019)) are the most frequently used.' The sentence seems not completed.

**Response:** We have revised this complicated sentence as: '*The reanalysis-based land surface model products are the most frequently used, mainly including the Global Land Data Assimilation System (GLDAS, with 0.25° resolution) (Rodell et al.,*

805 *2004), European Reanalysis (ERA)-interim (0.75°) (Balsamo et al., 2015) and its successors- ERA5 (0.25°) and ERA5-Land (0.1°) (Hoffmann et al., 2019)).*'

**Reviewer 2 Specific comment 5.** Lines 40: 'the penetrability of microwaves is usually <5 cm of soil'. This is particularly not true for L-band passive microwave like SMOS, SMAP, which are dedicated to soil moisture monitoring. 'Penetrability' is

810 usually called penetration depth.

**Response:** Following this comment, we have checked carefully to find that the L-band microwave can only be sensitive to the soil moisture within <5 cm of surface soil. For example, 'L-band, the brightness temperature emission originates from the top ~5 cm of soil' for SMAP (Entekhabi et al., 2010). 'At L-band soil moisture in the first centimeters (typically 5 cm) impacts significantly on the emitted brightness temperature' for SMOS (Kerr et al., 2001). This depth is however, larger than the

815 observation depth of higher frequency microwave, which is 1 cm (C band) or less (Piles et al., 2018). We agree that the word 'penetrability' is not scientific, while 'penetration depth' are not very accurate as well. Therefore, we have changed the sentence to '*current satellite microwave sensors can only detect soil moisture within top 5 cm of soil*' for make it clearer.

**Reviewer 2 Specific comment 6.** Line 45: change 'frommicrowave' to 'from microwave'.

820 **Response:** We have made the revision accordingly.

**Reviewer 2 Specific comment 7.** Line 45: 'Currently, the longest continuous record of global soil moisture retrieved frommicrowave remote sensing only is the ASCAT product'. Change 'only is'.

**Response:** Following this comment, we have corrected the sentence as: '*Currently, the longest continuous record of global*

825 *surface soil moisture that is derived only from microwave remote sensing is the ASCAT product*'.

**Reviewer 2 Specific comment 8.** Line 63~64: 'Upon rescaling, the spatial patterns of the satellite products are almost replaced by those of GLDAS.', Any citations?

**Response:** We have added the citations of (Gruber et al., 2019; Liu et al., 2012; Liu et al., 2011). Moreover, the statement has been changed into: '*Upon rescaling through CDF matching, the spatial patterns of the satellite products are generally replaced by those of GLDAS (Gruber et al., 2019; Liu et al., 2012; Liu et al., 2011)*'.

**Reviewer 2 Specific comment 9.** Line 66~67: 'Because the temporal variation in soil moisture is often better captured by model simulations than remote sensing inversions, CCI may undesirably combine the disadvantages of both.' How? Any proof?

**Response:** We have deleted the unsuitable description accordingly. There is no strong evidence for the claim that the temporal variation in soil moisture is better captured by model simulations than remote sensing inversions.

**Reviewer 2 Specific comment 10.** Change 'deviationsto' to 'deviations to'.

**Response:** We have checked the manuscript carefully and added all the missing blanks.

**Reviewer 2 Specific comment 11.** Lines 90~91: 'The training $R^2$ is only 0.45 (R=0.67)'. $R^2$ and R needs to specify.

**Response:** Thank you for this suggestion. We have revised it as: '*The training R-square value ($R^2$) is only 0.45 (or correlation coefficient, r, equals to 0.67)*'. Following this comment, we have also corrected all the 'R' into '*r*', to distinguish it from $R^2$ in the revised manuscript.

**Reviewer 2 Specific comment 12.** Lines 99~100: 'this data is regional, with a large temporal gap, and cannot be seen as observational-based only since precipitation data is incorporated.' Well, this is arguable.

**Response:** Accordingly, the sentence has been revised to '*However, this data is regional (for Tibetan Plateau only), and with a temporal gap between AMSR-E and AMSR2 (October 2011~June 2012).*' We have also re-written Text S1 to more clearly

850    explain why precipitation was not included as an input feature of the neural networks in this study. Please also find the details

in the responses to Major Comment 3.

**Reviewer 2 Specific comment 13.** Lines 148~150: 'are these factors used as direct spatial predictors of soil moisture or just

because they are related to the errors of satellite soil moisture retrievals (i.e., the quality impact factors of soil moisture)?'

855    What do you mean?

**Response:** We have revised the sentences to: '*Environmental factors, including DEM, LST and vegetation cover (indicated*

*by NDVI, MVI, etc.), were used as ancillary neural network inputs for improved soil moisture simulation (Lu et al., 2015; Qu*

*et al., 2019; Yao et al., 2017). According to these studies, these factors alone may not predict surface soil moisture well without*

*the incorporation of any microwave remote sensing data, which can also be justified by the contribution analysis results*

860    *(Figure S1a). This is because although they are somewhat related to soil moisture (e.g. soil moisture is limited in areas with*

*low vegetation cover in general but high in forests (McColl et al., 2017)), the relationships are rather uncertain (e.g. at smaller*

*scales, LAI may however have a negative influence on soil moisture due to the variation in evapotranspiration (Naithani et*

*al., 2013), or without clear impacts (Zhao et al., 2010); also, soil moisture can be either high or low in summers when*

*vegetation peaks (Baldocchi et al., 2006; Méndez-Barroso et al., 2009)). However, these factors are quite essential due to*

865    *their direct impacts on soil moisture retrieval through radiative transfer model using microwave remote sensing data (Fan et*

*al., 2020), and are retrieval quality impact factors. The detailed explanations are: 1) ...*'.

**Reviewer 2 Specific comment 14.** Lines 178: 'Water bodies dramatically lower the Tb, leading to overestimation of soil

moisture'. water bodies are marked/flagged out for soil moisture retrieval. The consideration of water bodies in your approach

870    seems very strange.

**Response:** We have replaced 'water bodies' with 'water area fraction' for clarity. Please also refer to our response to Major

Comment 5.

**Reviewer 2 Specific comment 15.** Line 182, are you sure it is soil dielectric conductivity, not soil dielectric constants?

875   **Response:** We have corrected 'soil dielectric conductivity' to '*soil dielectric constant*' accordingly.

**Reviewer 2 Specific comment 16.** Lines 199~200, 'For topographic complexity, the static layer of the Copernicus ASCAT-SWI product (hereinafter the ASCAT Constant) is adopted while for soil texture, the SMAP Constant is used.' Why not using static layers from the same satellite product?

880   **Response:** Thank you for the question. That is because topographic complexity and soil texture data cannot be obtained from one product. Also, the quality for the two data are satisfying (Reichle et al., 2018; Scipal et al., 2005). Please see the response to Major Comment 6.

**Reviewer 2 Specific comment 17.** Lines 227~228, 'For each map, soil moisture values are filteredto the level of three standard
885   deviations relative to themean in each zone. This preprocessing step is thus called '3σ denoise'.' What is the effect of such filter on identifying extreme years? for example, during 2003, 2010, 2018, 2019 there are extreme heat events in Europe and the soil moisture is so dry which can be beyond the 3 standard deviations.

**Response:** This is a spatial (zonal) '3σ denoise', which helps to masks out the wrong retrievals in mountain areas (zonal extreme values), however will not mask out the extreme climatic events. which will not mask out the extreme climatic events.
890   Please find the detailed explanation in the response to Major Comment 7. Following this comment, we have made the clarification as: "*After standardization of the original soil moisture data, to improve the neural network training efficiency, the potential salt and pepper noises are removed. For each map (a specific 10-day period), within each 1°×1° zone, the soil moisture values are filtered to the level of three standard deviations relative to the mean in that zone. This preprocessing step is thus called '3σ denoise' (note that the denoise is conducted spatially, rather than temporally, so that the extreme events will
895   not be treated).*'

**Reviewer 2 Specific comment 18.** Line 273: 'SMAP soil moisture is the training target'. SMAP is only available after 2015,

so I am not sure what is the meaning of simulation period 2012D19~2013D36, but also 2014-2018. I guess this is constrained by the available data (PROBA-V and GLASS LAIs)? But in any case, it does not represent any physical meaning to predict 900 2015data with 2012-2013 data.

**Response:** Here, we did not predict data in 2015 by using the data in 2012~2013. In this study, the common data period of predictor soil moisture products in each substep always contains the period of the simulated soil moisture. There are two substeps in Round 1, which were separated due to the data period of different LAI products, while each substep was responsible for a simulation period (2012D19~2013D36 and 2014-2018, respectively). Following this comment, we have revised that 905 section, for example, '*Because all the four predictors have data since 2012D19, the potential soil moisture simulation period is 2012D19~2018, which was further divided into two parts: one is 2014~2018 (substep1), for which the PROBA-V LAI data that starts from 2014, is applied, whereas the other is 2012D19~2013 (substep2), for which GLASS LAI data is used (note: because GLASS LAI covers from the beginning of our study period till 2017, the training period for substep 2 is 2015D10~2017).*'. For more details, please see response to Major Comment 8.

910

**Reviewer 2 Specific comment 19.** Line 278: '2011D20 to 2012D18 (Table S3~S4). In the third round (2010D16~2011D19)'. Why and how are these time spans defined?

**Response:** The time spans of the soil moisture simulation period corresponding to the different rounds of neural networks were determined based on the temporal coverages of the different microwave sensors which are utilized as predictors. For 915 example, the simulation period (Note: not the training period) of the second round NN is constrained by the common period of ASCAT, SMOS, FY and TMI data. Following this comment, we have made some clarifications, such as: '*In the second round of simulation, the training target can either be SMAP or SIM-1T, while the input soil moisture data are ASCAT, SMOS, TMI-LPRM-X (TMI) and FY-3B-NSMC (FY). The simulation output, SIM-2, covers the period of 2011D20~2012D18, that is constrained by the common period of the four predictors (Table S3~S4).*' We have also attached the details for the design of 920 five rounds of neural network operations in Tables S1~S15. Please see the details in the response to Major Comment 8.

**Reviewer 2 Specific comment 20.** Line 278, 'In the third round (2010D16~2011D19), SMAP...' Again, SMAP is only available after 2015.

**Response:** For the 3$^{rd}$ Round , the simulation period is 2010D16~2011D19, but the neural network training period could be 2015D10~2017D36, 2012D19~2015D10, or 2011D20~2012D21, depending on whether the training target is SMAP, or SIM-1T (SIM-1T is the postprocessed simulation output of the first round NN), or both SIM-1T and SIM-2T. The detailed information on NN training and soil moisture simulation in Round 3 were provided in Tables S5~S8. We have also revised the sentence in the manuscript as: '*In the third round of neural network operation, the simulation period is 2010D16~2011D19. SMAP, SIM-1T and SIM-2T are combined to be the training targets (the training periods are within the range of 2011D20~2017D36), while the predictor soil moisture data are ASCAT, SMOS, TMI and WindSat-LPRM-X (WINDSAT)....*'

1030

**To Reviewer #3:**

We thank referee#3 for the valuable comments that will help us in improving the quality and readably of the manuscript. We have carefully revised the MS following your comments and suggestions. We provide a detailed
1035    response to the Referee's comments in the Supplement.

\*\*\*\*\*\*\*\*\*\*

**Reviewer 3 General comment:** The authors used an iterative neural network approach to produce a new satellite-based soil
1040    moisture dataset using 11 microwave soil moisture products, using SMAP data for training and ISMN database for validation.

The approach is quite original and efficient resulting is a improvement in the accuracy of the spatio-temporal patterns at the

global scale, and at a 0.1 degree resolution. However, the manuscript will need to be improved before acceptance, in its

structure, clarity and tone.

**Response:** Thank you for your positive comments on our work. We have adjusted the article structure, revised the Method
1045    part to make it clearer, and modified the tone of the expressions in the comparison of our product against other products. In

addition, we have revised the figures and tables following each of the comment. Please find the details in the responses to the

following comments.

**Reviewer 3 Major comment 1:** The introduction would need to be improved. Several statements need to be supported by
1050    existing literature, others would need to be clarified. Finally the introduction would need to end with a brief description of the

approach used in the study and how this approach will address the three major concerns raised from existing soil moisture

products. See detailed comments below for details.

**Response:** We have carefully addressed all the raised problem, including adding some more references, clarifying the

confusing phrases, and briefly introducing the approach on how it addressed the three major concerns. Please see our responses
1055    to Specific comment 3~11 for details. Thank you for these nice suggestions!

**Reviewer 3 Major comment 2:** The tone of the manuscript when referring to the new product and to past studies is not always appropriate. For instance, stating that the present product is "superior to the existing products" is useless, not informative and condescending. I would encourage the authors to rather explain how their product is an improvement to the global estimation of soil moisture, without necessarily condemn other products. In the result section, while nonlinearities between estimate and in-situ soil moisture measurements are identified for other products, it is not reported for the author's product which I find quite biased.

**Response:** We are sorry for the inappropriate descriptions. We have corrected the descriptions in the abstract as: '*This new dataset, named RSSSM, is proved comparable to the in-situ surface soil moisture measurements at sites of the International Soil Moisture Network (overall $R^2$ and RMSE values of 0.42 and 0.087 $m^3/m^3$), while the overall $R^2$ and RMSE values for the existing products (ASCAT-SWI, GLDAS Noah, ERA5-Land, CCI/ECV and GLEAM) are within the range of 0.31~0.41 and 0.095~0.142 $m^3/m^3$, respectively. The advantage of RSSSM is especially obvious in arid or relatively cold areas, and during growing seasons. Moreover, the persistent high data quality as well as complete spatial coverage ensure the applicability of RSSSM to both the spatial and temporal pattern studies.*' We have also reported the nonlinearity of the relationship between our data and in-situ measurements, following: "*However, RSSSM overestimates soil moisture when it is low, which is a problem inherited from SMAP product (Figure 4), and is a bit nonlinearly correlated with the measured values (Figure 7a).*'

**Reviewer 3 Major comment 3.** The validation approach is based on site specific comparison. However, soil moisture being so spatially variable within a 0.1 degree pixel, validation based on single site observations within 0.1 degree pixels can be quite meaningless. This might be particularly true when one considers that site selection for in-situ measurement is rarely motivated by representativity of the surrounding landscape, but by specific ecological reasons.

**Response:** We agree that although only the 'good' quality data records were used, due to high spatial variability in surface soil moisture, it is not very reasonable to compare the 0.1° resolution soil moisture product against the ISMN site-scale measurements. However, the currently available global-scale soil moisture products are all in coarse resolution, usually about 0.25°. To evaluate these coarser resolution products, previous studies also have to rely on the site-measured soil moisture,

especially the ISMN dataset, while the validation process and the evaluation indicators are almost the same as this study (Al-Yaari et al., 2019; Albergel et al., 2012; Dorigo et al., 2015; Fernandez-Moran et al., 2017; Gao et al., 2020; Karthikeyan et al., 2017; Kerr et al., 2016; Kim et al., 2015b; Kolassa et al., 2018; Lievens et al., 2017; Zhang et al., 2019). For 29 ISMN networks used for validation in this study, 19 are dense networks (usually with multiple stations within one 0.1° pixel (Dorigo et al., 2015)), including AMMA-CATCH (Cappelaere et al., 2009; De Rosnay et al., 2009; Lebel et al., 2009; Mougin et al., 2009; Pellarin et al., 2009), BIEBRZA_S-1 (http://www.igik.edu.pl/en), BNZ-LTER (Van Cleve et al., 2015) (http://www.lter.uaf.edu/), CTP_SMTMN (Yang et al., 2013), FLUXNET-AMERIFLUX (http://ameriflux.lbl.gov/), FR_Aqui (Al-Yaari et al., 2018), HiWATER_EHWSN (Jin et al., 2014; Kang et al., 2014), HOBE (Bircher et al., 2012), HYDROL-NET_PERUGIA (Morbidelli et al., 2014), iRON (Osenga et al., 2019), MAQU (Su et al., 2011), OZNET (Smith et al., 2012; Young et al., 2008), REMEDHUS (http://campus.usal.es/~hidrus/), SASMAS (Rüdiger et al., 2007), SKKU (Hyunglok et al., 2016), SOILSCAPE (Moghaddam et al., 2010; Moghaddam et al., 2016), SWEX_POLAND (Marczewski et al., 2010), VAS (http://nimbus.uv.es/) and WSMN (http://www.aber.ac.uk/wsmn). This information has been added to Text S2. Therefore, the average of the data obtained from two or more stations within a 0.1° pixel, which was calculated in this study, can better represent the grid-scale soil moisture conditions (Gruber et al., 2020).

In addition, to avoid the errors induced by the high spatial variability of soil moisture as much as possible, we excluded the pixels with nonnegligible open water, wetland or inundated fields. In Supplementary Text- Text S2, the related details now read: '*It has been acknowledged that the scale difference between the records at ISMN sites and the 0.1° pixel-scale soil moisture data may lead to incomparability, especially for pixels with open water and inundated land (Loew, 2008). If the measurement site is located on land, away from water, yet the corresponding pixel contains much water, the pixel-scale soil moisture can be significantly higher than the site-measured values. Conversely, if the site is in or close to the open water or inundated areas, but land also exists in the pixel, the soil moisture measured at the station will be much higher than the pixel average value. Not only the absolute values are unmatchable, the temporal variations cannot be directly compared as well, because the moisture conditions of riverside (or wetland) soil and the land soil may change with precipitation differently. Therefore, the sites located in the pixels with average annual maximal water area fraction greater than 5% according to SWAMPS data are excluded (for example, some sites in wetlands in Canada)*' . We also added more explanations in the manuscript, following: 1) '*After data screening and processing (for example, in case of high spatial variability of soil moisture, we excluded the pixels with average annual maximal water area fraction greater than 5%, see Text S2), …*' ; 2) *More than 90% of the stations are located in relatively flat areas with topographic complexity lower than 10%;* and 3) '*Hence, to make full*

*use of all the good quality records, and to reduce the problem caused by the scale difference between simulation and*

1110 *measurement, the site-scale 10-day averaged soil moisture data are further aggregated to 0.1° pixel-scale by averaging all*

*the data (different stations or different sensors) within the pixel (Gruber et al., 2020).'*

**Reviewer 3 Specific comment 1:** "This new dataset, once validated against the International Soil Moisture Network (ISMN)

records, is supposed to be superior to the existing products" .

1115 Do you mean this validation hasn't been done yet? Superior in what way?

**Response:** Sorry for the confusion. Following this comment, we show the situation that our product is preferred, according to

the validation against site measurements, instead of the general description that our product is superior to other products. We

have made the clarification as: '*This new dataset, named RSSSM, is proved comparable to the in-situ surface soil moisture*

*measurements at sites of the International Soil Moisture Network (overall $R^2$ and RMSE values of 0.42 and 0.087 $m^3/m^3$),*

1120 *while the overall $R^2$ and RMSE values for the existing products (ASCAT-SWI, GLDAS Noah, ERA5-Land, CCI/ECV and*

*GLEAM) are within the range of 0.31~0.41 and 0.095~0.142 $m^3/m^3$, respectively. The advantage of RSSSM is especially*

*obvious in arid or relatively cold areas, and during growing seasons. Moreover, the persistent high data quality as well as*

*complete spatial coverage ensure the applicability of RSSSM to both the spatial and temporal pattern studies.'*

1125 **Reviewer 3 Specific comment 2:** "reveals that the surface moisture decline on rainless days is highest in summers over the

low-latitudes but highest in winters over most mid-latitude areas." Soil moisture being so spatially variable, I find the impact

of this statement quite limited – e.g. low latitude regions range from tropical/equatorial rain forests to deserts and one would

expect as much differences in the sensitivity of soil moisture to precipitation between a desert and a tropical forest than

between a tropical forest and a temperate prairie.

1130 **Response:** Sorry for the unclear sentences. In the calculation of soil moisture decline on consecutive rainless days, because

'*In desert areas, the random noise of the surface soil moisture product can hide the signal of moisture changes, while in wet*

*areas (e.g. rainforests), 20 days without effective precipitation seldom occurs, leading to no results over most areas.*', the

acquired results can only represent the condition in regions excluding deserts and rainforests. Following this comment, we

have revised the sentence as: '*It also reveals that without considering the deserts and rainforests, the surface moisture decline on consecutive rainless days is highest in summers over the low-latitudes (30°S~30°N) but highest in winters over most mid-latitude areas (30°N~60°N; 30°S~60°S)*'.

**Reviewer 3 Specific comment 3:** "L47: "due to various disturbances": what type of disturbances?

**Response:** Following your comment, we have revised it as '*due to various disturbances from for example, high vegetation cover, open water fraction and complex topography (Draper et al., 2012; Fan et al., 2020; Ye et al., 2015).*' for better clarity.

**Reviewer 3 Specific comment 4:** "L49: " Although new sensors, SMOS : : :." -> "Although new sensors such as SMOS: : :"

**Response:** We have revised it accordingly.

**Reviewer 3 Specific comment 5:** L 50: "better penetrability" -> please be more specific: what depth?

**Response:** Sorry for the unclear expression. It does not indicate the nominal soil depth of the microwave soil moisture. We have revised it to '*... because L-band microwaves (1~2 GHz) can better penetrate the vegetation canopy*', by referring to (Piles et al., 2018).

**Reviewer 3 Specific comment 6:** L66: "Because the temporal variation in soil moisture is often better captured by model simulations than remote sensing inversions": please include a reference that support this statement. L67: "CCI may undesirably combine the disadvantages of both." Be more specific here (low accuracy of temporal variations from remote sensing products and low spatial accuracy from model simulations – am I right?). And please include another reference here for this second statement.

**Response:** Sorry for the arbitrary statements. We agree that there is no strong evidence supporting the claim that the temporal variation in soil moisture is better captured by model simulations than remote sensing inversions. Following this comment, we have deleted these two sentences in the Introduction part of the manuscript. Thank you for reminding!

**Reviewer 3 Specific comment 7:** L70: "are assimilated instead": instead of what? this sentence is

not clear.

**Response:** Following this comment, we have revised the sentence as: '*Currently, anomalies of CCI soil moisture (the deviations to the seasonal climatology that indicate whether the soil moisture at a time point is more humid or drier than the multi-year average) are assimilated instead of the original CCI time series (Martens et al., 2017).*'

**Reviewer 3 Specific comment 8:** L85: "Among these three approaches, machine learning proves to be probably the best choice" based on what criteria – again, please be more specific

**Response:** Following this comment, we revised the sentence as: '*Among these three approaches, machine learning proves to be probably the best choice according to the connection between precipitation and the changes in soil moisture, evaluated through a data assimilation technique, and triple collocation analysis result (Van der Schalie et al., 2018).*'

**Reviewer 3 Specific comment 9:** "L102: "substantial success has not been achieved yet." This is a rather strong and yet vague statement that denies the merits of a large body of research. Please remove this statement.

**Response:** Sorry for the confusion. We have revised the sentence as: '*To be concluded, while previous studies have focused on developing long-term satellite-based surface moisture using machine learning, there remain some major concerns that need to be solved. 1)…*'.

**Reviewer 3 Specific comment 10:** "L102/103:" the high-quality microwave observations are not fully utilized": this is not clear from the literature review – please develop this point in earlier sections of the introduction (i.e. in what way high-quality microwave observations haven't been fully utilized, and how the authors are proposing to utilize them more efficiently).

**Response:** To avoid potential misunderstanding, we have revised the sentence as: '*1) the microwave observations from only at most three sensors are utilized, leading to large temporal and spatial gaps, and limited training efficiency*'. This point has

been illustrated in the earlier sections of the introduction: '*Global long-term observational-based soil moisture has been developed recently by building a neural network between the SMOS product and the Tb of AMSRE (2003~September 2011) and AMSR2 (July 2012~2015) (Yao et al., 2017)…. The gap between the temporal spans of AMSRE and AMSR2, and the lack*

1185  *of SMOS data in Asia resulted in large quantities of missing data.*'

**Reviewer 3 Specific comment 11:** "L106-107: This statement should be removed from the introduction section. - this is rather a concluding statement. Instead please describe your approach in a couple sentences and how this approach addresses the three major concerns identified.

1190  **Response:** Thank you for your advice. We have re-written the paragraph as: '*… there remain some major concerns that need to be solved. 1) the microwave observations from only at most three sensors are utilized, leading to large temporal and spatial gaps, and limited training efficiency; 2) it remains unclear which environmental factors should be incorporated as ancillary inputs, and why; and 3) the training designed for soil moisture estimation at global scale ought to be more complex than that for only a specific region to ensure a satisfying training efficiency. In this study, 11 high-quality microwave soil moisture*

1195  *products since 2003 are incorporated into 5 rounds of neural networks to achieve a spatially and temporally continuous simulation for 2003~2018, with as many sources of microwave observational data as possible used as predictors of each neural network. The quality impact factors of microwave soil moisture retrievals are also determined and then utilized as ancillary inputs to improve the training efficiency. Moreover, we designed localized subnetworks instead of only one global-scale neural network to account for the regional differences in training rules.*'

1200

**Reviewer 3 Specific comment 12:** "Section 2.1 L110: please add citation to literature supporting
this statement.

**Response:** In Introduction part, the best overall quality of SMAP soil moisture has been stated as follows: '*Although new sensors such as SMOS (Stillman and Zeng, 2018) and SMAP (Entekhabi et al., 2010), can produce significantly improved*

1205  *estimates because L-band microwaves (1~2 GHz) can better penetrate the vegetation canopy (Burgin et al., 2017; Chen et al.,*

*2018; Karthikeyan et al., 2017; Kerr et al., 2016; Kim et al., 2018; Leroux et al., 2014; Stillman and Zeng, 2018), the applicability of both products is still limited. SMOS data have too much noise and too many missing values in Eurasia due to high radio frequency interference (RFI) (Oliva et al., 2012). While SMAP has the highest quality (the unbiased RMSE of the passive product can be close to its target of 0.04 $m^3/m^3$) and has filtered RFI (Chen et al., 2018; Colliander et al., 2017), ...'*

Following your advice, we have revised this sentence and added two new references supporting the best performance of SMAP product. It reads: '*SMAP has currently the best quality of all remote sensing-based soil moisture products (Al-Yaari et al., 2019; Liu et al., 2019)...*'.

**Reviewer 3 Specific comment 13:** L115-118: This sentence is too long and too complex. Please split into shorter and clearer sentences.

**Response:** Sorry for the too complex sentence. We have revised it to: '*However, in this study, the well-acknowledged surface soil moisture products retrieved through mature algorithms (see Figure 1) are directly applied instead of Tb. This is because: 1) the primary goal of this study is to calibrate and then fuse the existing popular microwave soil moisture products; and 2) the Tb signals at multiple bands contain too much information that is not related to soil moisture, which may weaken the training efficiency and lead to over-fitting.*'

**Reviewer 3 Specific comment 14:** Section 2.2 L147/160: the purpose of this argumentation is quite unclear as the environmental predictors that are selected are also important drivers of soil moisture dynamic.

**Response:** Sorry for the arbitrary sentence. Here, we would like to express the idea that without the incorporation of any microwave remote sensing product, these factors (LAI, topographic complexity, LST, etc.) alone may not predict surface soil moisture very well. This is because although we agree that they are somewhat related to soil moisture (e.g. usually soil moisture is limited in areas with low vegetation cover), they can hardly be considered as direct indexes on surface soil moisture content (since the relationships are rather uncertain; for example, as found in this study, in Europe, the soil moisture is low in summers when vegetation peaks). On the other hand, however, we admit that water area fraction is a direct indicator of surface

1230  soil moisture, and have corrected by adding a sentence '*In addition, water fraction is a direct indicator of surface soil moisture.*'.

We removed the unclear argument following your advice. It now reads: '*Environmental factors, including DEM, LST and vegetation cover (indicated by NDVI, MVI, etc.), were used as ancillary neural network inputs for improved soil moisture simulation (Lu et al., 2015; Qu et al., 2019; Yao et al., 2017). According to these studies, these factors alone may not predict*

1235  *surface soil moisture well without the incorporation of any microwave remote sensing data, which can also be justified by the contribution analysis results (Figure S1a). This is because although they are somewhat related to soil moisture (e.g. soil moisture is limited in areas with low vegetation cover in general but high in forests (McColl et al., 2017)), the relationships are rather uncertain (e.g. at smaller scales, LAI may however have a negative influence on soil moisture due to the variation in evapotranspiration (Naithani et al., 2013), or without clear impacts (Zhao et al., 2010); also, soil moisture can be either*

1240  *high or low in summers when vegetation peaks (Baldocchi et al., 2006; Méndez-Barroso et al., 2009)). However, these factors are quite essential due to their direct impacts on soil moisture retrieval through radiative transfer model using microwave remote sensing data (Fan et al., 2020), and are retrieval quality impact factors. The detailed explanations are: 1) …*'.

**Reviewer 3 Specific comment 15:** L201: since precipitation is such an important driver of soil moisture, the reasons why this

1245  variable hasn't been included as a quality impact factor should be included in the main document.

**Response:** We have revised the sentence as: '*The contribution analysis results (Figure S1) show that because various microwave soil moisture data have already been included, precipitation data is not an essential indicator of soil moisture, and is not utilized as a physically-based 'quality impact factor' either (see Text S1 for detailed explanations).*'

The Text S1 is a bit long, so we did not move all the information to the manuscript, but we summarized the key reasons in the

1250  revised manuscript, following your advice. The Text S1 has been revised as follows:

'*Because it takes at least 3 days for a microwave sensor to cover the whole globe, for 11% of global land, there will be only 5 or less observations for random days within a 10-day period. By taking the average of these available data, this study only focuses on the mean soil moisture condition during that 10-day period. Then, to see how much can the incorporation of*

*precipitation data improve the neural network training efficiency, we calculated 10-day averaged GPM Final-Run*

1255 *precipitation, which can well indicate the overall precipitation water availability (the antecedent precipitation index is not used, because it must be calculated on daily scale, and the attenuation coefficient is hard to determine at global scale (Kohler and Linsley, 1951)). Taking the first primary independent neural network, NN1-1-1, as an example, we performed contribution tests on all the input features at the global scale (not for each separate zone), including 9 'quality impact factors', 4 predictor soil moisture products and precipitation - a potential ancillary soil moisture indicator. For each predictor, we added a random*

1260 *error that is controlled within the standard deviation of the predictor, and then the increased MSE in neural network training can indicate the relative contribution of that variable. The results (Figure S1a) show that precipitation will only contribute to 1.7% of the training efficiency, which is much lower than the contribution of any soil moisture product (the total contribution fraction of the four soil moisture products is 61.2%), and is also lower than that of most 'quality impact factor'. This suggests that various microwave soil moisture datasets together with several 'quality impact factors' of microwave soil moisture*

1265 *retrieval are enough to predict the training target- SMAP soil moisture, and there is no need to add precipitation as another ancillary indicator of soil moisture.*

*'Quality impact factors' are defined in this study as the variables that will have a significant impact on the retrieval errors of soil moisture by microwave remote sensing (section 2.1.2). Although the relative performances of different soil moisture products is related to surface moisture condition (Kim et al., 2015a), it is found mainly due to the less vegetation in arid areas.*

1270 *After all, no explicit mechanism can support the idea that the retrieval errors of soil moisture are significantly influenced by water availability. Even if this is true, the soil water availability can be already indicated by the microwave soil moisture products. So, it is unreasonable to incorporate the precipitation variable as a 'quality impact factor'. On the other hand, LAI, water area fraction, LST, land use cover, tree cover fraction, non-tree vegetation area fraction, topographic complexity, and soil sand/clay fractions all have direct impacts on the microwave soil moisture retrieval errors, with solid physical mechanism*

1275 *(see section 2.1.2). Therefore, theoretically they should be added to the neural network, even though the land use cover type and soil sand fraction data prove to have limited contributions to NN training efficiency.*

*One may argue that if NARX (nonlinear autoregressive with external input) is applied instead, in which the soil moisture in the previous 10-day period is also incorporated as a predictor, precipitation data can be very beneficial to the neural network training. This is true, because precipitation directly contributes to soil moisture increases. However, NARX is not suitable for*

1280 *global-scale long-term continuous soil moisture mapping, because the base map (i.e., soil moisture in the beginning of the*

*simulation period) is hard to determine. Moreover, in mid to high latitudes, the lack of soil moisture retrievals over frozen*

*ground in winters will lead to missing data there in summers when soil moisture data is otherwise available. So, if NARX is*

*adopted, we can only estimate long-term surface soil moisture in tropics and sub-tropics with air temperature consistently*

*higher than 0 ℃. Last but not least, if the soil moisture in the previous phase and the current precipitation amount are both*

1285 *incorporated, they will largely conceal the role of satellite-observed signals. As shown in Figure S1b, the total contribution*

*fraction of all four microwave soil moisture products is reduced to only 10.6%, while the roles of ASCAT, AMSR2-JAXA and*

*AMSR2-LPRM are all negligible. Without taking full advantage of remote sensing, the simulations based on previous soil*

*moisture and current precipitation will lead to errors over places where soil moisture gains are mostly driven by glacier*

*melting, or in places with high levels of radiation-driven soil evaporation. The reliability of the derived soil moisture will be*

1290 *lower in irrigated croplands and afforestation/deforestation areas as well.*

*On account of all above, precipitation data is neither included as an ancillary soil moisture indicator, nor added as a 'quality*

*impact factor' in this study.'*

[Figure]

*Figure R1 (Figure S1 in the revised Supplement): The roles of different input features in the soil moisture simulations based*

1295 *on BP neural networks and nonlinear autoregressive with external input (NARX) with microwave soil moisture products*

*incorporated: (a) the contributions of different input features of a primary neural network: NN1-1-1, including 4 predictor*

*soil moisture products, 9 quality impact factors of microwave soil moisture retrieval, plus 1 probable ancillary soil water*

*indicator: 10-day averaged precipitation, to the neural network training efficiency indicated by the increased MSE; (b) the*

*contributions of all the input features to the training efficiency, if NN1-1-1 is changed into a NARX, in which the SMAP soil*

1300 *moisture for the previous period is also applied as a predictor.*

**Reviewer 3 Specific comment 16:** L202/203: This sentence should start this section, not end it. A table similar to table 1 but

for the quality impact factors would be useful. The table would indicate the source of the data, the resolution and the temporal

span for the dynamic factors.

**Response:** Following your suggestion, we have revised this section, putting this concluding sentence to the start. It now reads

1305 as:

'*In this study, 9 quality impact factors: LAI, water fraction, LST, land use cover, tree cover fraction, non-tree vegetation*

*fraction, topographic complexity, and sand and clay fractions are selected and incorporated (see Figure 1). The reasons are*

*as follows.*

*Based on the two criteria above, the first environmental factor to be included is the 'vegetation factor' (i.e., vegetation water*

1310 *content, VWC). … Because the leaf area index (LAI) stands for the total leaf area per unit land, which is closely related to*

*VWC assuming a relatively stable leaf equivalent water thickness (Yilmaz et al., 2008), LAI is a suitable surrogate….*'

We also added a timeline figure (Figure 1 in the revised manuscript) to show the temporal spans, sources and spatial resolution

of all microwave soil moisture products, and the data for 9 quality impact factors. This could be clearer than a table.

[Figure]

Figure R2 (Figure 1 in the revised manuscript): The timeline figure showing the time periods of the soil moisture datasets and the 'quality impact factor' products (e.g. LAI dataset) used in this study (listed above the timeline), as well as the periods of data applied for the trainings of 67 independent neural networks and the neural network simulation outputs (i.e. simulated

*soil moisture) of eight substeps (listed below the timeline).*

1320    **Reviewer 3 Specific comment 17:** Section 2.3 Sections 2.4, 2.5 and 2.6 should be included in section 2.3 as it details the different steps of the calculation flow. A clear justification on why a neural network approach was adopted should be included in this section.

**Response:** We have made the revision accordingly. The structure for Data and Method Section is now as follows:

*'2 Data and Methods*

1325    *2.1 Data for the production of global long-term surface soil moisture data*

*2.1.1 The satellite-based surface soil moisture data products*

*2.1.2 The quality impact factors of soil moisture retrievals*

*2.2 Methods for the production of global long-term surface soil moisture data*

*The global long-term surface soil moisture data production includes three basic parts, which are as follows. 1) Preprocessing:*
1330    *the production of high-quality neural network inputs; 2) neural network operation: the network training and soil moisture simulation and 3) postprocessing: the correction of potential errors or deficiencies in the soil moisture simulation outputs. ...*

*2.2.1 Neural network design (1): localized neural networks*

*2.2.2 Preprocessing and postprocessing steps*

*2.2.3 Neural network design (2)- five rounds of simulations*

1335    *2.3 Methods for the validation of surface soil moisture products*

*2.4 Methods for the intra-annual variation analysis of surface soil moisture*'.

The justification of the neural network approach is now in the Introduction, '*Among these three approaches, machine learning proves to be probably the best choice according to the connection between precipitation and the changes in soil moisture, evaluated through a data assimilation technique, and triple collocation analysis result (Van der Schalie et al., 2018).*' In
1340    section 2.1.2, we have already mentioned the use of neural network approach in this study, '*... were used as ancillary neural network inputs for improved soil moisture simulation (Lu et al., 2015; Qu et al., 2019; Yao et al., 2017)*'.

**Reviewer 3 Specific comment 18:** L223: what is a hidden layer?

**Response:** In neural networks, hidden layer is located between the input and output layers. It is the result of nonlinear
transformations of the input data through activation function, and can also be transformed into the output data. We have revised
the sentence as: '*..., and the number of nodes in the hidden layer (between the input and output layers (Stinchcombe and
White, 1989)) of each subnetwork is set to 7.*'

**Reviewer 3 Specific comment 19:** L242: reference required for the "suspicious value removal".

**Response:** Sorry for the confusing phrase. Actually, it is a detailed method invented by this study, to make sure only the most
reliable estimates are applied as the training target of the next round of neural networks, in order to avoid significant error
propagation along with the neural network round. Because multiple rounds of neural network is a characteristic of this study,
this processing step was not found in previous researches (i.e., no reference). We have revised the sentence, removing the
phrase '*suspicious value removal*'. It now reads: '*On the other hand, to avoid error propagation with training times by
ensuring a high-quality training target for the next round's simulation, for every simulated result, we removed all the
suspicious values. This preprocessing step is performed by first obtaining the maximal and minimum values of SMAP_E soil
moisture in each pixel. If the simulated value is out of the range of the SMAP data during 2015~2018, the value is suspicious
and is not used as the training target.*'.

**Reviewer 3 Specific comment 20:** L259: Here you are referring to a 1 degree pixel a presume? Please specify.

**Response:** We regret for the confusing sentences. We have revised the sentences to: '*For data simulation in a 0.1° pixel, the
most preferable independent neural network is expected to be trained using all the available soil moisture data sources in that
pixel. However, in the 1° zone where it is located, the subnetwork belonging to that preferable independent neural network
may not exist due to limited valid data points (see section 2.2.1). Then, an alternative subnetwork driven by the combination
of fewer soil moisture data inputs should be applied instead.*' In the revised manuscript, the word 'pixel' only stands for 0.1-

degree resolution, while the word 'zone' indicates 1° scale.

**Reviewer 3 Specific comment 21:** Please specify. L268: why 74 networks? Please explain.

**Response:** We have corrected it to '67 independent neural networks' because the rest 7 networks are optional and not
independent. There are multiple independent networks included in each round. That is because '... *While increasing the sources of soil moisture data inputs can be beneficial to the training efficiency, the spatial coverage of the simulation output is sacrificed because the overlapping area of more soil moisture products is smaller. After all, most products have missing data in specific regions ... To solve that dilemma, we classified all 0.1° pixels according to the available predictor soil moisture products in it over a 10-day period (for example, if there are at most four predictor soil moisture data inputs in one round, there should be 4+3+2+1=10 combinations). However, to avoid soil moisture simulation under snow or ice cover (Section 2.2.2), not all combinations are considered. Then, corresponding to each selected combination, an independent neural network is trained*.' (Lines 269~277 in the revised manuscript). The revised explanations are clearer than the original version.*

In addition, the training periods of 67 independent neural network are now shown in Figure R2 (that is Figure 1 in the revised manuscript). So, this sentence is now revised to: '*Based on these principles, five rounds of neural networks are designed as follows, with 8 substeps containing a total of 67 independent neural networks. The training period of each neural network and the simulation period of each substep are shown in Figure 1 (below the timeline), and the details are as follows: ...*'.

**Reviewer 3 Specific comment 22:** Section 2.7 Figure S2 should be included in the main manuscript as the spatial distribution of validation data is critically important to evaluate the overall strength of this validation. It is surprising that none of the Canadian sites of the Russian sites made it to the final set of sites for validation.

**Response:** Thank you for this advice. We have moved the figure showing the spatial distribution of validation sites to the main manuscript (shown in Figure 3).

The soil moisture measurement depth for the Russian sites are 0~10 cm (RUSWET-GRASS) or 0~20 cm (RUSWET-AGRO and RUSWET-VALDAI) (Robock et al., 2000), which do not match the nominal simulation depth (0~5 cm) of our soil

1390    moisture product (constrained by SMAP).

On the other hand, the Canadian sites are actually very limited (Dorigo et al., 2011; Dorigo et al., 2013) (https://www.geo.tuwien.ac.at/insitu/data_viewer/), and are often wetland sites with nonnegligible water fraction, which should be dropped. Previous studies also utilized very few sites in Canada and Russia (Kolassa et al., 2018; Lievens et al., 2017; Zhang et al., 2019).

1395    In the revised manuscript, we added the information '*Accordingly, the measurements used for validation are limited to ≤ 5 cm in depth (e.g., the Russian networks were not applicable for this reason)*'.

**Reviewer 3 Specific comment 23:** L304/306: This is an important point. Soil moisture being so spatially variable, validation based on single site observations within 0.1 degree pixels can be quite meaningless. Especially when one considers that site

1400    selection for in situ measurement is rarely motivated by representativity of the surrounding landscape, but by specific ecological reasons.

**Response:** Thanks for careful thinking. Because grid-scale soil moisture measurements are unavailable, almost all the recent studies rely on the in-situ soil moisture records for the validation of remote sensing soil moisture products. Moreover, we have taken the average of the data at multiple stations in a 0.1° grid, and also excluded the grids with potentially high spatial

1405    heterogeneity of surface soil moisture. Please see the response to Major comment 3 for details.

**Reviewer 3 Specific comment 24:** Section 2.8 L335: "probably the best choice for periodic function fitting" : please support this statement by adequate reference to literature.

**Response:** We have revised it to '*Fourier functions can characterize the intra-annual variation well (Brooks et al., 2012;*

1410    *Hermance et al., 2007). Therefore, for the remaining areas ..., we fitted the intra-annual cycle of soil moisture using the Fourier function*'.

**Reviewer 3 Specific comment 25:** L344/345: I don't understand this argument. Why restricting this analysis to 10 consecutive

rainless days and not the whole range of 10-days sum of precipitation?

1415    **Response:** This study would like to explore the impact of dry periods (20 consecutive days without effective rainfall) on the surface soil moisture in different areas. Moreover, without effective precipitation, the surface soil moisture changes are mainly driven by evaporation and deep percolation, and thus should be negative. This is simpler and can help us exclude the unreliable soil moisture values from the analysis. However, if we consider the sum of precipitation over 10 days, the surface soil moisture changes will be rather complex, and there would be much more erroneous data included, leading to unreliable analysis results.

1420    We have revised the sentence to: '*Because RSSSM indicates the average soil moisture condition during every 10-day period, we studied the surface soil moisture decline after 20 consecutive days (i.e., two adjacent 10-day periods) without effective precipitation to explore the impact of dry periods on surface soil moisture.*'.

**Reviewer 3 Specific comment 26:** Tables 3 to 8 could be synthesized into only two tables: one for temporal accuracy

1425    assessment and one for spatial accuracy assessment for the three products comparisons with SIM. Similarly, it would be nice to have figures 3, 4, 6, 7, 9 and 10 summarized in 2 figures where all four products appear (SIM, SMAP, GLDAS and ASCAT). This would facilitate comparison between products.

**Response:** Thank you for this advice. It should be noted that the periods corresponding to these comparisons differ from each other. SMAP has data since March of 2015, ASCAT is available from 2007, while SIM (named 'RSSSM' in the revised

1430    manuscript), GLDAS, ERA5 Land and GLEAM v3.3a all cover the whole study period (2003~2018), but with missing values in different areas. Although CCI (ECV) also covers the entire period, it lacks data in many places, especially before 2007 (Figure 13d). Therefore, the in-situ surface soil moisture measurements entering the comparison between RSSSM and SMAP are limited to March 2015~2018, whereas only the ISMN sites' data during 2007~2018 were applied for the accuracy comparison between RSSSM and ASCAT. Also, when comparing RSSSM against CCI (ECV), we only included the soil

1435    moisture records in the grids with both RSSSM and CCI data during the specific period. As we can see, the overall accuracy of RSSSM in Figure 4c (for the comparison with SMAP, during April 2015~2018) is $R^2$=0.46, RMSE=0.083, but in Figure 10 (for comparison against GLDAS, during 2003~2018), the overall $R^2$ of RSSSM is 0.42, and the RMSE is 0.087. In addition,

the temporal accuracy of RSSSM in all climatic regions (Figure 5, Figure 8, Figure 11 in the revised paper), and its spatial accuracy during all seasons (Figure 6, Figure 9, Figure 12) are different among the comparisons against different soil moisture products. For that reason, if the 6 figures are combined into 2 figures, they will be too crowded, and the comparison will be not clear enough.

Following your comment, we have combined the tables as follows:

[revised manuscript text omitted]

**Reviewer 3 Specific comment 27:** L380-383: However, it looks like the relationship between SIM estimates and in situ observations is nonlinear (Figure 5a). Furthermore, SIM seems to overestimate soil moisture in the lowest range (winter?) when a density of pixels is quite high. Please include these remarks in the results.

**Response:** Thank you for reminding. We have added a sentence accordingly. It reads: '*However, RSSSM overestimates soil moisture when it is low, which is a problem inherited from SMAP product (Figure 4), and is a bit nonlinearly correlated with the measured values (Figure 7a).*'

1465 **Reviewer 3 Specific comment 28:** L393: the relationship between SIM and in situ measurements is also obviously nonlinear. Please include this remark in the text for fairness.

**Response:** We have revised the description accordingly. It now reads: '*While RSSSM is nonlinearly correlated with measured soil moisture, the relationship between GLDAS soil moisture and the measurements appears a bit more nonlinear, resulting in a smaller $R^2$ of 0.39 and higher RMSE of 0.097 for GLDAS product compared to RSSSM ($R^2$: 0.42; RMSE: 0.087, see Figure*

1470 *10).*'

**Reviewer 3 Specific comment 29:** Why table S19 and Figure S7 do not appear in the main document like the other product comparison? Please move them to the main manuscript

**Response:** We have combined Table S17, Table S19, Table S21, Table S22 in the original Supplement to Table 2 in the revised

1475 manuscript (see Table R1). Table S18, Table S20, Table S23 and Table S24 are integrated into Table 3 as well (see Table R2). For the reasons described in the response to Reviewer 3 Specific comment 26 (the time periods and spatial extents for the comparisons of RSSSM in this study against different existing surface soil moisture products are different), the figures were not combined together to avoid over-crowdedness and unnecessary confusion. Thus, we retained Figure S7, Figure S8, etc. in the Supplement to prevent the main manuscript from too long and too complex to read.

1480

**Reviewer 3 Specific comment 30:** L422-434: This belongs to the discussion section.

**Response:** We have moved the sentences belonging to the discussion section to the Discussion part. The revised sentences in the Result part now read: '*Next, we focus on the interannual change in data quality. According to Figure 13a~c, while the correlation coefficient for RSSSM does not vary significantly among different years, the RMSE and ubRMSE values in earlier*

[revised manuscript text omitted]

Kim, S., Liu, Y. Y., Johnson, F. M., Parinussa, R. M., and Sharma, A.: A global comparison of alternate AMSR2 soil
1600 moisture products: Why do they differ?, Remote Sens. Environ., 161, 43-62, https://doi.org/10.1016/j.rse.2015.02.002, 2015a

Kim, S., Parinussa, R. M., Liu, Y. Y., Johnson, F. M., and Sharma, A.: A framework for combining multiple soil moisture retrievals based on maximizing temporal correlation, Geophys. Res. Lett., 42, 6662-6670, http://doi.org/10.1002/2015GL064981, 2015b

1605 Kohler, M. A. and Linsley, R. K.: Predicting the runoff from storm rainfall, US Department of Commerce, Weather Bureau, 1951.

Kolassa, J., Reichle, R. H., Liu, Q., Alemohammad, S. H., Gentine, P., Aida, K., Asanuma, J., Bircher, S., Caldwell, T., Colliander, A., Cosh, M., Holifield Collins, C., Jackson, T. J., Martínez-Fernández, J., McNairn, H., Pacheco, A., Thibeault, M., and Walker, J. P.: Estimating surface soil moisture from SMAP observations using a Neural Network technique,
1610 Remote Sens. Environ., 204, 43-59, https://doi.org/10.1016/j.rse.2017.10.045, 2018

Lebel, T., Cappelaere, B., Galle, S., Hanan, N., Kergoat, L., Levis, S., Vieux, B., Descroix, L., Gosset, M., Mougin, E.,

Peugeot, C., and Seguis, L.: AMMA-CATCH studies in the Sahelian region of West-Africa: An overview, J. Hydrol., 375, 3-13, https://doi.org/10.1016/j.jhydrol.2009.03.020, 2009

Leroux, D. J., Kerr, Y. H., Bitar, A. A., Bindlish, R., Jackson, T. J., Berthelot, B., and Portet, G.: Comparison Between SMOS, VUA, ASCAT, and ECMWF Soil Moisture Products Over Four Watersheds in U.S, IEEE Trans. Geosci. Remote Sensing, 52, 1562-1571, https://doi.org/10.1109/TGRS.2013.2252468, 2014

Lievens, H., Martens, B., Verhoest, N. E. C., Hahn, S., Reichle, R. H., and Miralles, D. G.: Assimilation of global radar backscatter and radiometer brightness temperature observations to improve soil moisture and land evaporation estimates, Remote Sens. Environ., 189, 194-210, https://doi.org/10.1016/j.rse.2016.11.022, 2017

Liu, J., Chai, L., Lu, Z., Liu, S., Qu, Y., Geng, D., Song, Y., Guan, Y., Guo, Z., Wang, J., and Zhu, Z.: Evaluation of SMAP, SMOS-IC, FY3B, JAXA, and LPRM Soil Moisture Products over the Qinghai-Tibet Plateau and Its Surrounding Areas, Remote Sens., 11, http://doi.org/10.3390/rs11070792, 2019

Loew, A.: Impact of surface heterogeneity on surface soil moisture retrievals from passive microwave data at the regional scale: The Upper Danube case, Remote Sens. Environ., 112, 231-248, https://doi.org/10.1016/j.rse.2007.04.009, 2008

Lu, Z., Chai, L., Ye, Q., and Zhang, T.: Reconstruction of time-series soil moisture from AMSR2 and SMOS data by using recurrent nonlinear autoregressive neural networks, 26-31 July 2015 2015, 980-983, https://doi.org/10.1109/IGARSS.2015.7325932.

Marczewski, W., Slominski, J., Slominska, E., Usowicz, B., Usowicz, J., Romanov, S., Maryskevych, O., Nastula, J., and Zawadzki, J.: Strategies for validating and directions for employing SMOS data, in the Cal-Val project SWEX (3275) for wetlands, Hydrol. Earth Syst. Sci., 2010, 7007-7057, http://doi.org/10.5194/hessd-7-7007-2010, 2010

Martens, B., Miralles, D. G., Lievens, H., van der Schalie, R., de Jeu, R. A. M., Fernández-Prieto, D., Beck, H. E., Dorigo, W. A., and Verhoest, N. E. C.: GLEAM v3: satellite-based land evaporation and root-zone soil moisture, Geosci. Model Dev., 10, 1903-1925, https://doi.org/10.5194/gmd-10-1903-2017, 2017

McColl, K. A., Alemohammad, S. H., Akbar, R., Konings, A. G., Yueh, S., and Entekhabi, D.: The global distribution and dynamics of surface soil moisture, Nat. Geosci., 10, 100, https://doi.org/10.1038/ngeo2868, 2017

Méndez-Barroso, L. A., Vivoni, E. R., Watts, C. J., and Rodríguez, J. C.: Seasonal and interannual relations between precipitation, surface soil moisture and vegetation dynamics in the North American monsoon region, J. Hydrol., 377, 59-70, https://doi.org/10.1016/j.jhydrol.2009.08.009, 2009

Moghaddam, M., Entekhabi, D., Goykhman, Y., Li, K., Liu, M., Mahajan, A., Nayyar, A., Shuman, D., and Teneketzis, D.: A Wireless Soil Moisture Smart Sensor Web Using Physics-Based Optimal Control: Concept and Initial Demonstrations, IEEE J. Sel. Top. Appl. Earth Observ. Remote Sens., 3, 522-535, https://doi.org/10.1109/JSTARS.2010.2052918, 2010

Moghaddam, M., Silva, A. R., Clewley, D., Akbar, R., Hussaini, S. A., Whitcomb, J., Devarakonda, R., Shrestha, R., Cook, R. B., Prakash, G., Santhana Vannan, S. K., and Boyer, A. G.: Soil Moisture Profiles and Temperature Data from SoilSCAPE Sites, USA., ORNL DAAC, Oak Ridge, Tennessee, USA., https://doi.org/10.3334/ORNLDAAC/1339, 2016

Morbidelli, R., Saltalippi, C., Flammini, A., Rossi, E., and Corradini, C.: Soil water content vertical profiles under

natural conditions: matching of experiments and simulations by a conceptual model, Hydrol. Process., 28, 4732‑4742, https://doi.org/10.1002/hyp.9973, 2014

1650    Mougin, E., Hiernaux, P., Kergoat, L., Grippa, M., de Rosnay, P., Timouk, F., Le Dantec, V., Demarez, V., Lavenu, F., Arjounin, M., Lebel, T., Soumaguel, N., Ceschia, E., Mougenot, B., Baup, F., Frappart, F., Frison, P. L., Gardelle, J., Gruhier, C., Jarlan, L., Mangiarotti, S., Sanou, B., Tracol, Y., Guichard, F., Trichon, V., Diarra, L., Soumaré, A., Koité, M., Dembélé, F., Lloyd, C., Hanan, N. P., Damesin, C., Delon, C., Serça, D., Galy‑Lacaux, C., Seghieri, J., Becerra, S., Dia, H., Gangneron, F., and Mazzega, P.: The AMMA‑CATCH Gourma observatory site in Mali: Relating climatic variations to changes in
1655    vegetation, surface hydrology, fluxes and natural resources, J. Hydrol., 375, 14‑33, https://doi.org/10.1016/j.jhydrol.2009.06.045, 2009

        Naithani, K. J., Baldwin, D. C., Gaines, K. P., Lin, H., and Eissenstat, D. M.: Spatial Distribution of Tree Species Governs the Spatio‑Temporal Interaction of Leaf Area Index and Soil Moisture across a Forested Landscape, PLOS ONE, 8, e58704, 10.1371/journal.pone.0058704, 2013

1660    Oliva, R., Daganzo, E., Kerr, Y. H., Mecklenburg, S., Nieto, S., Richaume, P., and Gruhier, C.: SMOS Radio Frequency Interference Scenario: Status and Actions Taken to Improve the RFI Environment in the 1400–1427‑MHz Passive Band, IEEE Trans. Geosci. Remote Sensing, 50, 1427‑1439, https://doi.org/10.1109/TGRS.2012.2182775, 2012

        Osenga, E. C., Arnott, J. C., Endsley, K. A., and Katzenberger, J. W.: Bioclimatic and Soil Moisture Monitoring Across Elevation in a Mountain Watershed: Opportunities for Research and Resource Management, Water Resour. Res., 55,
1665    2493‑2503, https://doi.org/10.1029/2018WR023653, 2019

        Pellarin, T., Laurent, J. P., Cappelaere, B., Decharme, B., Descroix, L., and Ramier, D.: Hydrological modelling and associated microwave emission of a semi‑arid region in South‑western Niger, J. Hydrol., 375, 262‑272, https://doi.org/10.1016/j.jhydrol.2008.12.003, 2009

        Piles, M., Schalie, R. v. d., Gruber, A., Muñoz‑Marí, J., Camps‑Valls, G., Mateo‑Sanchis, A., Dorigo, W., and Jeu, R.
1670    d.: Global Estimation of Soil Moisture Persistence with L and C‑Band Microwave Sensors, 22‑27 July 2018 2018, 8259‑8262, 10.1109/IGARSS.2018.8518161.

        Qu, Y., Zhu, Z., Chai, L., Liu, S., Montzka, C., Liu, J., Yang, X., Lu, Z., Jin, R., Li, X., Guo, Z., and Zheng, J.: Rebuilding a Microwave Soil Moisture Product Using Random Forest Adopting AMSR‑E/AMSR2 Brightness Temperature and SMAP over the Qinghai–Tibet Plateau, China, Remote Sens., 11, https://doi.org/10.3390/rs11060683, 2019

1675    Robock, A., Vinnikov, K. Y., Srinivasan, G., Entin, J. K., Hollinger, S. E., Speranskaya, N. A., Liu, S., and Namkhai, A.: The Global Soil Moisture Data Bank, Bull. Amer. Meteorol. Soc., 81, 1281‑1300, http://doi.org/10.1175/1520‑0477(2000)081<1281:TGSMDB>2.3.CO;2, 2000

        Rüdiger, C., Hancock, G., Hemakumara, H. M., Jacobs, B., Kalma, J. D., Martinez, C., Thyer, M., Walker, J. P., Wells, T., and Willgoose, G. R.: Goulburn River experimental catchment data set, Water Resour. Res., 43,
1680    https://doi.org/10.1029/2006WR005837, 2007

        Smith, A. B., Walker, J. P., Western, A. W., Young, R. I., Ellett, K. M., Pipunic, R. C., Grayson, R. B., Siriwardena, L., Chiew, F. H. S., and Richter, H.: The Murrumbidgee soil moisture monitoring network data set, Water Resour. Res., 48, https://doi.org/10.1029/2012WR011976, 2012

Stillman, S. and Zeng, X.: Evaluation of SMAP Soil Moisture Relative to Five Other Satellite Products Using the Climate Reference Network Measurements Over USA, IEEE Trans. Geosci. Remote Sensing, 56, 6296-6305, https://doi.org/10.1109/TGRS.2018.2835316, 2018

Stinchcombe and White: Universal approximation using feedforward networks with non-sigmoid hidden layer activation functions, 1989 1989, 613-617 vol.611, http://doi.org/10.1109/IJCNN.1989.118640.

Su, Z., Wen, J., Dente, L., van der Velde, R., Wang, L., Ma, Y., Yang, K., and Hu, Z.: The Tibetan Plateau observatory of plateau scale soil moisture and soil temperature (Tibet-Obs) for quantifying uncertainties in coarse resolution satellite and model products, Hydrol. Earth Syst. Sci., 15, 2303-2316, https://doi.org/10.5194/hess-15-2303-2011, 2011

Van Cleve, K., Chapin, F. S. S., and Ruess, R. W.: Bonanza Creek Long Term Ecological Research Project Climate Database - University of Alaska Fairbanks., 2015

Van der Schalie, R., De Jeu, R., Parinussa, R., Rodríguez-Fernández, N., Kerr, Y., Al-Yaari, A., Wigneron, J.-P., and Drusch, M.: The Effect of Three Different Data Fusion Approaches on the Quality of Soil Moisture Retrievals from Multiple Passive Microwave Sensors, Remote Sens., 10, https://doi.org/10.3390/rs10010107, 2018

Yang, K., Qin, J., Zhao, L., Chen, Y., Tang, W., Han, M., Lazhu, Chen, Z., Lv, N., Ding, B., Wu, H., and Lin, C.: A Multiscale Soil Moisture and Freeze–Thaw Monitoring Network on the Third Pole, Bull. Amer. Meteorol. Soc., 94, 1907-1916, https://doi.org/10.1175/BAMS-D-12-00203.1, 2013

[revised manuscript text omitted]

删除了: indicates

删除了: it can provide …ne valid data point is provided. Se

删除了: For each…e subnetworks with, if the number of valid pixels (for a 0.1° pixel in a given 10-day period, if all the subnetwork inputs have valid values, it is seen as a valid pixel; the maximal number of valid pixels=100×the number of 10-day periods)

删除了: valid data points

删除了: exceeds

删除了: , the subnetwork is…ere considered valid…ropped. There are… leaving usually more than …15,000 zonal valid …ubnetworks within…included in an individual

删除了: i… performed in the

删除了: software

删除了: i

删除了: each

删除了: e

删除了: which …hat is a step in

[revised manuscript text omitted]

---

## Author Response (AR2)

Dear editor,

Thank you for editing our manuscript. The first part of this document includes the point-by-point responses to the reviews (Reviewer 1, Reviewer 2). Comments of the referees are marked as e.g. << Reviewer 1 Major comment 1>> followed by the answer from the authors, which includes the changes made in the manuscript to fulfill the referees' suggestions. The section of responses to the referees is followed by a marked-up version of the manuscript.

Best regards
Yongzhe Chen, Xiaoming Feng and Bojie Fu

To Reviewer 1:

**General comment:** This is a very valuable contribution to the research on long-term soil moisture data. The approach is technically sound and state-of-the-art.

**Response:** Thank you for the positive comments on this work. We have carefully revised the manuscript following your comments. Details of the changes are provided in the responses below.

**Major comment 1:** In the introduction the authors too strongly blame existing soil moisture products without discussing their advantages and drawbacks in detail. Also, other methods than NN to generate long-term time series such as Copulas are not mentioned. Introduction needs a much clearer structure.

**Response:** Following this comment, we have removed the incorrect phrasing throughout the Introduction and added detailed descriptions of the advantages and drawbacks of the existing soil moisture products. In addition, we have revised the paragraph illustrating the methods for generating long-term microwave soil moisture time series, and the Copulas method is included. Please review the Introduction in the revised manuscript.

The structure of the Introduction is as follows: the 1st paragraph introduces the significance of soil moisture; the 2nd and 3rd paragraphs illustrate the soil moisture products derived from land surface models and microwave remote sensing, respectively, with the advantages and drawbacks discussed; the 4th paragraph introduces the surface soil moisture datasets produced by combining land surface modeling and remote sensing, and the need for high-quality long-term surface soil moisture datasets derived from microwave remote sensing is highlighted; the 5th paragraph discusses the popular methods targeting the use of the information acquired by one sensor to produce soil moisture data compatible with those retrieved from another to generate long-term microwave soil moisture time series; the 6th paragraph introduces the existing long-term microwave soil moisture data developed by using the machine learning method and points out the major aspects that need to be improved; the 7th paragraph concludes the previous work and then proposes three major concerns in producing long-term microwave surface soil moisture, which we tried to solve in this study.

**Major comment 2:** Also the description of the neural net approach is not very clear. Figure 2 helps, but the full iterative and localized approach is still not clear. Also the separation into monthly ~10-day bins is not justified. Why not using a strict 10 day temporal resolution?

**Response:** We apologize for the confusion. Following this comment, we have revised the overall brief description of the iterative and localized neural network approach, and more details have been added. It now reads as follows: '*Global long-term surface soil moisture data production includes three basic parts: 1) preprocessing: the production of high-quality neural network inputs, including the training target soil moisture, predictor soil moisture products and the quality impact factors (i.e., 9 environmental factors);*

*2) neural network operation: the training of localized neural networks (i.e., the rules for soil moisture prediction are separately trained in different 1°×1° zones) followed by surface soil moisture simulation based on the localized neural networks; and 3) postprocessing: the correction of potential errors or deficiencies in the soil moisture simulation outputs.*

*The temporal span of the primary training target SMAP does not overlap with that of TMI, FY-3B, WindSat or AMSR-E (see Figure 1), while most microwave soil moisture products are not available from the beginning year 2003 (e.g., AMSR2 data are only available since July 2012). Therefore, to fully utilize the 10 predictor surface soil moisture products retrieved from 7 different microwave sensors and form a temporally continuous soil moisture dataset covering 2003~2018, several iterative rounds of simulations are performed. Here, 'iterative' means that the simulated soil moisture data in a round were also converted to part of the training targets of the next round's neural network (hereinafter the 'secondary training targets'), thus extending the potential temporal span of the target soil moisture data. Accordingly, the postprocessing steps which are intended to transform the simulation outputs to reliable secondary training targets can be seen as preprocessing steps as well. The basic flow of this process is shown in Figure 2.*' (Lines 246~260 in the revised manuscript)

In addition, we have revised the detailed explanation of the neural network design to help readers understand the specific operation processes. Please refer to Lines 265~269 and Lines 304~318 for these revisions.

Our dataset is separated into monthly ~10-day bins rather than a strict 10-day resolution because the key inputs, e.g., SPOT-VGT and PROBA-V LAI data, are available in monthly ~10-day bins. We added the explanation at the end of section 2.1.1, following: '… *Therefore, the temporal resolution of the dataset developed in this study is approximately 10 days, meaning that 3 data records are obtained within a month for days 1~10, 11~20 and from 21 to the last day of that month. This format is exactly the same as that of the ASCAT-SWI and many other products developed by the Copernicus Land Monitoring Service (https://land.copernicus.eu).*'.

**Major comment 3:** During validation the scale difference between coarse resolution of most satellite SSM products as compared to point-scale in situ measurements is not discussed.

**Response:** We thank the reviewer for this comment. We have added section 4.2 to the Discussion as follows: '… *However, we can neither conclude that our product is superior to the existing products, nor determine the performance of our product at the global scale. This is mainly because the ISMN measurements are unevenly distributed globally (Figure 3) and incompatible at a spatial scale with the scales of passive microwave observations and land surface modeling (0.1°~0.25°). We validated the soil moisture products against the ISMN's point-scale data just because only such in situ measurements are currently available, and the ISMN dataset (Dorigo et al., 2011; Dorigo et al., 2013) is the most frequently used in the assessments of large-scale soil moisture data (Al-Yaari et al., 2019; Albergel et al., 2012;*

*Dorigo et al., 2015; Fernandez-Moran et al., 2017; Gao et al., 2020; Karthikeyan et al., 2017b; Kerr et*
*al., 2016; Kim et al., 2015b; Kolassa et al., 2018; Lievens et al., 2017; Zhang et al., 2019). In this study,*
85 *to alleviate the impact of spatial scale differences on the evaluation, dense networks are more utilized*
*(19 out of 29 networks, see Text S2 for details) that contain multiple stations within the same 0.1° pixel.*
*The pixels with nonnegligible water area are also excluded in case of high spatial variability in surface*
*soil moisture. In addition, more than 90% of the selected stations are located in relatively flat areas with*
*a topographic complexity less than 10%. ...'*
90

**Major comment 4:** Also the discussion is very descriptive focusing on the statistics, but I would like to
see deeper interpretations by linking environmental characteristics, microwave observation methods and
NN predictions.

**Response:** We thank the reviewer for this comment. Following this advice, we have added section 4.1 to
95 the Discussion to discuss the roles of microwave-observed soil moisture data and environmental
characteristics on NN predictions as follows: '*The key algorithm calibrates and fuses various sources of*
*microwave surface soil moisture products through multiple neural networks. Several environmental*
*factors are also chosen as ancillary neural network inputs because they are quality impact factors of*
*microwave soil moisture retrievals, or also director indicators of surface soil moisture. To explore the*
100 *relative roles of soil moisture data retrieved from microwave observations and the environmental*
*characteristics, we performed contribution tests on all the input features at the global scale (for each*
*predictor, we added a random error that is controlled within the standard deviation of the predictor. Then*
*the increased mean squared error (MSE) in neural network training can be used to determine the relative*
*contribution of that variable). Taking the first independent neural network (NN1-1-1, a primary NN) as*
105 *an example, the results (Figure 16) indicate that SMOS soil moisture plays the dominant role in the neural*
*network training (55.5%), while the four predictor soil moisture products explained 62.7% in total. The*
*remaining 37.3% of the training efficiency could be attributed to the environmental characteristics,*
*among which the water fraction accounts for the most (13.4%) since it is both a quality impact factor*
*and a direct indicator of soil moisture. The tree cover fraction is an important neural network input as*
110 *well and reduces the MSE by 7.8%, which is probably due to the strong impact of forest cover on*
*microwave soil moisture retrievals.'*

[Figure]

*Figure R1 (Figure 16 in the revised manuscript): Relative contributions of the 13 input features (i.e., four predictor soil moisture products retrieved from microwave remote sensing and 9 environmental factors that are quality impact factors of microwave soil moisture retrieval or also indicators of soil moisture) to the training efficiency of the first round's primary neural network (NN1-1-1).*

**Major comment 5:** The language needs to be improved. All abbreviations need to be introduced first. The authors should speak about surface soil moisture, not surface moisture or similar.

**Response:** We have checked and corrected the language errors. The abbreviations are now introduced when they are first mentioned. For the remaining abbreviations of the names of satellites, remote sensors and missions, we added a table (Table 1 in the revised manuscript). Thank you for this reminder. We have corrected 'surface moisture' to 'surface soil moisture' in the manuscript accordingly.

**Major comment 6:** I am missing also a description of the data set itself, i.e. which format, auxiliary data etc.

**Response:** Thank you for this comment. We have added that information in the Data Availability section as follows: '*In the ZIP file, data maps are all provided in Geotiff format, and we also attached a csv table relating the filename and the nominal time period of the file.*'

**Specific comment 1:** L. 8: be more specific, what is lacking?

**Response:** We have changed the sentence to: '*However, long-term satellite monitoring of surface soil moisture at the global scale needs improvement.*' instead of saying that the long-term satellite monitoring of surface soil moisture at the global scale is lacking.

**Specific comment 2:** L. 12: elaborate?

**Response:** We have changed the phrase to 'complicated': '*The training efficiency was high ($R^2$ =0.95) due to ... and the complicated organizational structure of multiple neural networks.*'

**Specific comment 3:** L. 14: strange formulation. Iterative and localized?
**Response:** As the reviewer noted, the neural networks in this study are both iterative and localized. To clarify our meaning, we have changed the phrasing to '*5 rounds of iterative simulations; 8 substeps; 67 independent neural networks; and more than one million localized subnetworks*' accordingly.

**Specific comment 4:** L. 16: introduce RSSSM
**Response:** We have revised the text by adding an introduction as follows: '*Then, we developed the global Remote Sensing-based Surface Soil Moisture dataset (RSSSM) covering 2003~2018 at 0.1° resolution. The temporal resolution is approximately 10 days, meaning that 3 data records are obtained within a month, for days 1~10, 11~20 and from 21 to the last day of that month ...*'.

**Specific comment 5:** L. 19: why in cold and arid regions?
**Response:** We have revised the sentence as follows: '*RSSSM generally presents advantages over other products in arid and relatively cold areas, which is probably because of the difficulty in simulating the impacts of thawing and transient precipitation on soil moisture, and during the growing seasons…*'.

**Specific comment 6:** L. 21: which period? Is it valid to use the data set for trend analysis?
**Response:** We have revised this sentence as follows: '*Moreover, the persistent high data quality during 2003~2018 as well as the complete spatial coverage ensure the applicability of RSSSM to studies on both the spatial and temporal patterns (e.g., long-term trend).*' accordingly.

**Specific comment 7:** L. 33: where do the uncertainties come from?
**Response:** The uncertainties of microwave soil moisture products and land surface model products are described in detail in the following paragraphs. We agree that putting this sentence here will probably lead to confusion. Therefore, following this comment, we have deleted this sentence.
The uncertainties in land surface model products are described as follows: '*The uncertainties arise from meteorological forcing data, model parameters, as well as inadequacies in model physics (Cheng et al., 2017). Moreover, the anthropogenic impacts from irrigation and land cover changes are rarely considered (Kumar et al., 2015; Qiu et al., 2016).*'. On the other hand, for the microwave products, the description on the sources of uncertainties are as follows: '*Satellite-based soil moisture retrievals may also suffer from various disturbances, such as lower quality over dense vegetation cover, high open water fractions and complex topography (Draper et al., 2012; Fan et al., 2020; Ye et al., 2015). Difference in*

*the algorithms dealing with the disturbances make different microwave soil moisture products hardly comparable with each other (Kim et al., 2015a; Mladenova et al., 2014).'*

**Specific comment 8:** L. 47: explain abbreviations
**Response:** We have added a table for abbreviations for these sensors or satellites, see Table 1 in the revised manuscript. The sentence has been revised as follows: '… *SMOS, SMAP, see Table 1 for the full names*'.

**Specific comment 9:** L. 47: change to AMSR-E, also in the following
**Response:** We have changed it to AMSR-E accordingly.

**Specific comment 10:** L. 51: usually is wrong, reformulate. Reduced performance due to complex topography or dense vegetation
**Response:** This sentence may not have been clearly written. To avoid misunderstanding, we have changed it as follows: '*satellite-based soil moisture products usually have lower accuracies than modeled products … due to various disturbances, such as lower quality over high vegetation cover, high open water fractions and complex topography*'.

**Specific comment 11:** L. 56: reference for SMOS is Kerr et al. (2001)
**Response:** We have corrected the reference accordingly.

**Specific comment 12:** L. 57: better than shorter wavelengths
**Response:** We have revised it accordingly.

**Specific comment 13:** L. 61: reformulate: …and incorporated hardware RFI mitigation
**Response:** We have changed it accordingly.

**Specific comment 14:** L. 64: published a long-term surface soil moisture dataset under the Climate Change Initiative (CCI). Delete: or Essential Climate Variable (ECV)
**Response:** We have deleted it accordingly and deleted it in the Abstract.

**Specific comment 15:** L. 71: justify low quality, why?
**Response:** We apologize for the arbitrary phrasing. We have deleted this claim on low quality. The microwave sensors before 2003 may have had limited spatial coverage and coarser resolution (Karthikeyan et al., 2017b). Therefore, we have revised the sentence as follows: '*CCI utilized almost all the available microwave soil moisture datasets to form a long time series, and generally agrees well with*

*measured values at some sites, e.g.,, the Irish grassland sites and the grassland and agricultural fields in the United States, France, Spain, China and Australia (Albergel et al., 2013; An et al., 2016; Dorigo et al., 2017; Pratola et al., 2015). Valid microwave observations were quite limited before June 2002 due to satellite sensor constraints (Dorigo et al., 2017).'.*

**Specific comment 16:** L. 72: why is the merging algorithm probably too simple? Justify!
**Response:** We apologize for the incorrect phrasing. We have changed the sentence as follows: We have changed the sentence to: '*The temporal variation in each satellite product is retained, although the data averaging (Liu et al., 2012) cannot efficiently distinguish between the divergent interannual variations in various products (Feng et al., 2017).*'

**Specific comment 17:** L. 77: it is not true that temporal changes are mainly driven by model simulations, reformulate
**Response:** Since the anomalies (the deviations to the seasonal climatology, which indicate whether the soil moisture at a time point is more humid or drier than the multiyear average (Martens et al., 2017)) rather than the original CCI time series are assimilated into the GLEAM model now, the temporal changes (e.g., intra-annual variation) in the GLEAM v3 products will not learn much from satellite observations. Following this comment, to avoid suspicious negative claim and provide more positive assessments, we have corrected the sentences to: '*The general performance of the GLEAM soil moisture product is satisfactory (Beck et al., 2020). In the current version, the CCI soil moisture anomalies (the deviations to the seasonal climatology, which indicate whether the soil moisture at a time point is more humid or drier than the multiyear average) are assimilated instead of the original CCI time series (Martens et al., 2017). Therefore, satellite observations play a much smaller role than modelling in forming the GLEAM product.*'.

**Specific comment 18:** L. 85-90: it is not clear what the authors want to say here
**Response:** Here, we introduced the previous methods on calibrating the soil moisture (or Tb) data retrieved by one microwave sensor to make it matchable (compatible) with the soil moisture data retrieved from another (i.e., the approach for harmonizing two different microwave soil moisture datasets, except for CDF matching). We have revised these sentences to make it easier to understand, and added the Copulas function method as follows: '*In addition to the CDF matching algorithm, at least four methods have been proposed that target the use of the information acquired by one sensor to produce soil moisture data that are compatible with the data retrieved from another. Based on physical-based equations (Wigneron et al., 2004), the regression between SMOS soil moisture and dual-polarized brightness temperature (Tb) data from AMSR-E is applied to match the AMSR-E soil moisture time series to SMOS (R-square =0.36) (Al-Yaari et al., 2016). An example of the second method uses the Land*

*Parameter Retrieval Model (LPRM) (Owe et al., 2008) to retrieve soil moisture from SMOS and then*
*match the 'SMOS-LPRM' data with the AMSR-E-LPRM product by calibrating the LPRM parameters*
*and then applying a linear regression (Van der Schalie et al., 2017). Thirdly, Copulas functions allow to*
*model the structure of the dependence between two different Tb or soil moisture datasets and thus could*
*perform better for the extreme values, thereby reducing the RMSE (Gao et al., 2007; Leroux et al., 2014;*
*Lorenz et al., 2018; Verhoest et al., 2015).'*

**Specific comment 19:** L. 92: polarized reflectivity? Isn't it emission?
**Response:** We apologize for including the wrong information. There is no 'polarized reflectivity'. We
have corrected the sentence as follows: '*researchers built a neural network that links SMOS soil moisture*
*to the Tb at different polarizations and frequencies of AMSR-E to produce a calibrated soil moisture data*
*product that covers 9 years (2003~2011) (Rodríguez-Fernández et al., 2016).'* Thank you for the reminder.

**Specific comment 20:** L. 103: this data has been
**Response:** We have corrected the text as follows: '*SMAP soil moisture data have been chosen as …*'.

**Specific comment 21:** L. 115: iterative 5-step neural network
**Response:** We have revised it to '*iterative 5-round neural networks*'. because we would like to
distinguish between 'round' and 'substep', which is included in a 'round'.

**Specific comment 22:** L. 123: It has to be noted that the SMAP_E grid is 9km only, but that the spatial
resolution of that product is around ~20km. The 9km is misleading and should be clarified also to
correctly interpret the final product of this study.
**Response:** SMAP_E is the enhanced SMAP product (spatial resolution: 36 km). It is in the EASE-Grid
(equal-area scalable Earth) 2.0 projection, and the spatial resolution is 9.024 km (1623 rows×3855
columns at the global scale). Therefore, for most places, the spatial resolution of SPAM_E is
approximately at a 0.1° resolution in the WGS1984 coordinate system. To clarify our meaning, we added
the following sentence as follows: '*SMAP_E was reprojected from the EASE-Grid 2.0 projection with 9*
*km resolution to the WGS1984 geographic coordinate system with 0.1° resolution.*'

**Specific comment 23:** L. 126: The nominal penetration depth
**Response:** We have changed the text accordingly.

**Specific comment 24:** L. 134: ASCAT soil water index
**Response:** We have changed the text accordingly.

**Specific comment 25:** L. 135: not developed by Copernicus, but by EUMETSAT. It is provided by the ESA Copernicus Land Monitoring Service

**Response:** Thank you for reminding us of this mistake. We have corrected the information accordingly.

**Specific comment 26:** L. 135: why was the SMAP porosity used, and not the porosity provided with the ASCAT product?

**Response:** We used the SMAP porosity because porosity data is not included in the static layers of ASCAT-SWI product (https://land.copernicus.eu/global/products/swi). We added information to the revision.

**Specific comment 27:** L. 140: Highly questionable to use X-band for soil moisture retrieval, maybe that is the reason why the results of this study show improvements for cold and arid regions with low vegetation. C-band RFI is known to be high over the US, but also over the rest of the globe? L. 150: again, why not C-band also for Windsat?

**Response:** We agree that C-band retrievals will probably perform better than X-band soil moisture data in regions with high vegetation cover. To reduce the effect of this problem, we have incorporated LAI and vegetation continuous field (tree cover fraction) data as ancillary neural network inputs to consider the impact of vegetation on the retrievals from different frequency bands. According to Njoku et al., the C-band RFI is much higher than the X-band RFI. The C-band RFI is most densely concentrated in the United States, Japan, and the Middle East and sparsely distributed in Europe and other areas worldwide (Njoku et al., 2005).

[Figure]

*Figure R2: Classification maps of the global RFI. (a) 6.9-GHz RFI; and (b) 10.7-GHz RFI. This figure is directly obtained from (Njoku et al., 2005).*

305 In addition, the footprint size of C-band brightness temperature is much coarser than that of the X-band for AMSR-E, AMSR2 and WindSAT (~0.5°).

Following this comment, we revised the related sentence as follows: '*X-band retrievals may not perform well in high-vegetated areas, but C-band data such as AMSR2-LPRM-C or AMSR-E-LPRM-C were not applied due to the high RFI, especially in the United States, Japan, and the Middle East (Njoku et al.,*
310 *2005)*'.

For WindSAT, the RFI and coarser resolution may explain why no C-band product for WindSAT is available now (De Jeu and Owe, 2014a, b; Karthikeyan et al., 2017b).

However, we really appreciate your idea of incorporating both the X-band and C-band retrievals as predictors. This attempt may further improve the final data quality when the algorithm remains
315 unchanged.

**Specific comment 28:** L. 169: not all products are retrieved by a RTM. Additionally, typical all retrieval methods use a vegetation and/or a LST information. Don´t you add here those characteristics twice? Please discuss.
320 **Response:** We agree that not all soil moisture data are retrieved by physical models (e.g., RTM). Semi-empirical models, empirical models, vegetation contribution models and change detection models are used as well; however, vegetation cover or LST information is usually needed (Karthikeyan et al., 2017a). This sentence was not clearly written, and we have revised it as follows: '*However, these factors are quite essential due to their direct impacts on microwave-based soil moisture retrieval through the radiative*
325 *transfer model and other models (Fan et al., 2020; Karthikeyan et al., 2017a); thus, they are retrieval quality impact factors.*'

**Specific comment 29:** L. 182: sand and clay fractions can be named soil texture
**Response:** Following this comment, we have revised it as follows: '*... the 'soil texture factors' (two*
330 *factors, sand fraction and clay fraction) ...*' .

**Specific comment 30:** L. 189: the WARP change detection algorithm is applied to ASCAT after using the different angles to remove the vegetation contribution. Please check the ASCAT SM retrieval and modify accordingly.
335 **Response:** Thank you for this reminder. After checking the change detection algorithm, we have corrected the sentence as follows: '*..., whereas the TU-Wien change detection algorithm applied to ASCAT utilizes the quadratic polynomial dependence of backscatter on the incidence angle to better*

*characterize the vegetation effect on backscatter and then remove it by identifying the reference angles (Hahn et al., 2017; Vreugdenhil et al., 2016).*' (TU-Wien algorithm is used in WARP).

**Specific comment 31:** L. 211: that
**Response:** We have removed the duplicate 'that' accordingly.

**Specific comment 32:** L. 252: please justify this filtering.
**Response:** We have added an explanation for this filtering method to justify it as follows: '*the principle is that 99.87% of the data appear within this range for a normal distribution (Howell et al., 1998). Also note that the filter applied spatially rather than temporally to detect and delete the extreme values, which are usually noise in mountain areas. Therefore, the extreme climatic events will not be mistakenly removed*'.

**Specific comment 33:** L. 323: it is not clear why a NN sorting is necessary
**Response:** The reasons for sorting the independent NNs in each round are actually described in the previous paragraph (Lines 303~328). To make this part easier to understand, we have added the following details. It now reads: '*Although increasing the sources of soil moisture data inputs can improve the training efficiency, the spatial coverage of the simulation output is sacrificed because the overlapping area decreases as the number of soil moisture products increases. After all, most products have missing data in specific regions (e.g., mountains, wetlands and urban settlements), and some sensors are even unable to produce data at the global scale (e.g., TMI is limited to [N40°, S40°]; SMOS have many missing values in Eurasia). To resolve this dilemma, we classified all 0.1° pixels according to the predictor soil moisture products that have a valid value over a 10-day period (for example, if there are four predictor soil moisture datasets in one round, there should be 4+6+4+1=15 combinations. Here, '1' indicates the condition that all four products have a valid value in the 0.1° pixel, and there are '6' conditions when only two of the four predictors have valid value in the pixel). However, to avoid soil moisture simulation under snow or ice cover (see section 2.2.2), not all combinations are considered. Then, an independent neural network corresponding to each selected combination is trained. For data simulations in a 0.1° pixel, the most preferable independent neural network is expected to be trained using all the available soil moisture data sources in that pixel (i.e., if valid values are provided by three soil moisture products, then the preferable neural network is the one trained using those three predictors). However, in the 1° zone in which the 0.1° pixel is located, the subnetwork belonging to that preferable independent neural network may not exist due to limited valid data points (see section 2.2.1). Then, an alternative subnetwork driven by the combination of fewer soil moisture data inputs should be applied instead. Hence, we should determine the neural network collocation that is the best choice for every pixel. Apart from applicability,*

*the relative priority order of different neural networks was obtained by comprehensively considering the number and quality of input soil moisture products, ....'.*

375

**Specific comment 34:** L. 335: the Russian networks do not have any data in this period
**Response:** Thank you for this reminder. We have revised it as follows: '*Records outside of the RSSSM data period (2003~2018), such as those from Russian networks, are ignored as well.*'

380 **Specific comment 35:** L. 406: is that an independent comparison? If not, please indicate.
**Response:** We agree with you and have added the following note: '*these two datasets are not completely independent because SMAP data are used as the training target while RSSSM data are the simulation results*'.

385 **Specific comment 36:** L. 434: The ASCAT problem in arid areas is known and might be related to changes in soil scattering where the change detection method assumptions are not longer valid.
**Response:** Thank you for this suggestion. We have added the following sentence: '*This problem is known and might be related to the different scattering mechanisms in dry soils invalidating the assumptions of change detection method (Al-Yaari et al., 2014)*'.

[revised manuscript text omitted]

To Reviewer 2:

**General comment:** I've got the impression that the authors have sufficiently addressed my previous comments and questions. I support the publication of this data description paper after the following minor points had been considered.

**Response:** We thank the reviewer for the positive comments on our work as well as the last round of revision. Here, we have further addressed the minor points. Details of the changes are provided in the responses below.

**Specific comment 1:** L134: "The first is ASCAT" change to something like: "The first satellite soil moisture product", otherwise it reads as if it was a quality impact factor product.

**Response:** We have made the revision accordingly.

**Specific comment 2:** L180: You speak of 9 quality impact factors but you only list 8 and in the overview Fig. 1 there are only 7 different types distinguished.

**Response:** We have revised the sentence as follows: '*In this study, 9 quality impact factors are incorporated: LAI, water fraction, LST, land use cover, tree cover fraction, non-tree vegetation fraction, topographic complexity, soil sand fraction and clay fraction.*'. In addition, in Figure 1, '*sand and clay fractions*' are two quality impact factors, while '*vegetation continuous fields- tree cover and non-tree vegetation cover fraction*' indicate two quality impact factors as well.

**Specific comment 3:** L193: GEOV2-LAI is missing in the overview figure (Fig. 1)

**Response:** GEOV2-LAI indicates SPOT-VGT plus PROBA-V LAI. We added this information in the revised manuscript: '*The Copernicus global 1 km resolution LAI (called GEOV2-LAI, which consists of SPOT-VGT and PROBA-V LAI) data are adopted here ...*'

**Specific comment 4:** L211: that that

**Response:** We have removed the duplicate 'that'.

**Specific comment 5:** L223, L224, continue the enumeration of quality impact factors as you started with "first, second, third, ..." until you reach 9th as defined in the introduction

**Response:** We have made the revision accordingly.

**Specific comment 6:** L249: Still required to state the Toolbox used in Matlab 2016 and the name of the neural network training function

**Response:** We have revised the sentence as follows: '*The training was performed in MATLAB 2016a-*

*using the Neural network fitting toolbox, and the number of nodes in the hidden layer (between the input and output layers (Stinchcombe and White, 1989)) of each subnetwork was 7. We chose the gradient descent backpropagation algorithm as the training function.*'

**Specific comment 7:** L339: which as a -> which has a
**Response:** We apologize for the misspelling. We have revised the sentence as follows: '*... ~10-day-averaged soil moisture records obtained from 728 stations of 29 networks are applied for validation of the soil moisture products.*' for simplicity.

**Specific comment 8:** L395: "grid" please change all occurrences to "grid cell". A grid refers to an aggregate of cells, so a 0.5° x 0.5° unit corresponds to a grid cell.
**Response:** We have changed 'grid' to 'grid cell' accordingly.

**Specific comment 9:** L546: if you write "most" you should state which ones are better than yours.
**Response:** We have deleted the word 'most', since in this study we did not find datasets that are more comparable with the ISMN measurements than ours. However, we cannot conclude that our product is better than others due to the limitations in this validation method. Following this comment, we have revised the sentences as follows: '*Our product is generally more comparable to the in-situ measurements at ISMN stations than the existing global long-term surface soil moisture datasets in general, when all indicators on both spatial and temporal accuracy are considered. However, we can neither conclude that our product is superior to the existing products, nor determine the performance of our product at the global scale. This is mainly because the ISMN measurements are unevenly distributed globally (Figure 3) and incompatible at a spatial scale with the scales of passive microwave observations and land surface modeling (0.1°~0.25°). We validated the soil moisture products against the ISMN's point-scale data just because only such in situ measurements are currently available, and the ISMN dataset (Dorigo et al., 2011; Dorigo et al., 2013) is the most frequently used in the assessments of large-scale soil moisture data (Al-Yaari et al., 2019; Albergel et al., 2012; Dorigo et al., 2015; Fernandez-Moran et al., 2017; Gao et al., 2020; Karthikeyan et al., 2017; Kerr et al., 2016; Kim et al., 2015; Kolassa et al., 2018; Lievens et al., 2017; Zhang et al., 2019)...*'

**Specific comment 10:** L573: You could name the Cosmic-Ray Neutron Sensing method (CRNS) and the COSMSOS (http://cosmos.hwr.arizona.edu/) network here as a potential provider of root zone integrated large scale soil moisture with global coverage (Andreasen, M., Jensen, K.H., Desilets, D., Franz, T.E., Zreda, M., Bogena, H.R. and Looms, M.C. (2017), Status and Perspectives on the Cosmic‐Ray Neutron Method for Soil Moisture Estimation and Other Environmental Science Applications. Vadose Zone Journal, 16: 1-11 vzj2017.04.0086. doi:10.2136/vzj2017.04.0086)

**Response:** Following this comment, we added information to point out the significance of CRNS and COSMOS. In the Discussion in section 4.2 entitled 'Requirement of further validation', we wrote the following: '*The Cosmic-Ray Neutron Sensing method (CRNS) can provide soil moisture estimates at a scale of hundreds of meters in diameter (Andreasen et al., 2017). Hence, the in situ networks generated using this method, e.g., COSMOS, are more suitable for the validation of satellite-based or modeled coarse resolution soil moisture products. We hope that additional records obtained from cosmic-ray neutron stations become available in the future so that our product may be better evaluated.*'

**Specific comment 11:** L990: Figure caption: "Overview of the periods of the different soil moisture datasets..."
**Response:** We have made the revision accordingly.

**Specific comment 12:** L995: Figure caption: "Flow chart for the ..."
**Response:** We have changed it accordingly.

**Specific comment 13:** L1002: provide time range for (c) RSSSM vs site measured soil Moisture
**Response:** We have revised this phrasing as follows: '*RSSSM and the site-measured soil moisture from April 2015 to 2018*'.

**Specific comment 14:** Table 2 & 3: you could highlight the best performing dataset (per column) by setting their values bold
**Response:** We have revised the table and highlighted the better performing dataset for each comparison between RSSSM and the other datasets. Please note that the comparison period for different pairs of comparisons is not the same. Therefore, we could not identify the best performing dataset per column.

**Specific comment 15:** Language: the writing has improved since the initial version of the manuscript but some parts are still difficult to grasp at first reading.
**Response:** We have checked and corrected the errors in language again. In addition, many complicated parts, especially the descriptions of the methods, have been revised, and details have been added. Moreover, the abbreviations are now introduced when they are mentioned first. For the remaining abbreviations of the names of satellites, remote sensors and missions, we added a table (Table 1 in the revised manuscript). Thank you for your careful reading!

[revised manuscript text omitted]

删除了: dataset of satellite observation…emote Sensing-based Surface Soil Moisture (RSSSM) dataset -based global surface soil moisture …overing 2003~2018

删除了: is still lacking

删除了: SMAP

删除了: elaborate

删除了: zonal

删除了: W

删除了: achieved global satellite monitoring of surface soil moisture…during …

删除了: or to be specific, there are 3 data records

删除了: This new dataset, named RSSSM,

删除了: in-situ…n situ surface soil moisture measurements ofat…the International Soil Moisture Network sites (overall $R^2$ and RMSE values of 0.42 and 0.087 m$^3$/m$^3$), while the overall $R^2$ and RMSE values for the existing popular similar products (ASCAT-SWI, GLDAS Noah, ERA5-Land, CCI/ECV and GLEAM)…

删除了: The advantage of RSSSM is especially obvious …n arid and or

删除了: data …uality during 2003~2018 as well as the complete spatial coverage ensure the applicability of RSSSM to studies on both the spatial and temporal patterns (e.g., long-term trend). Our new data…SSSM data suggests an increase in the global mean surface soil moisture. These data…

删除了: also reveal that without …onsidering the deserts and rainforests, the surface soil moisture decline …oss on consecutive rainless days is highest in summers…over the low latitudes (30°S~30°N) but mostly highest …n winters…over most …he mid-latitude areas… (30°N~60°N; 30°S~60°S). Notably, the error propagation is well controlled with the extension of the simulation period to the past,

**1 Introduction**

Soil moisture plays an important role in modulating the exchange of water, carbon and energy between the land surface and atmosphere, and it also links the global water, carbon and energy cycles (Dorigo et al., 2012; Karthikeyan et al., 2017a). Soil moisture has been endorsed by the Global Climate Observing System (GCOS) as an essential climate variable (Bojinski et al., 2014), because it can indicate the climatic impact on the ecosystems, such as during ecological droughts (Martínez-Fernández et al., 2016; Samaniego et al., 2018). Current research requires high-quality soil moisture information in terms of data accuracy and spatial-temporal coverage (Hashimoto et al., 2015; Stocker et al., 2019).

Reanalysis-based land surface model products are frequently used, including the Global Land Data Assimilation System (GLDAS, with 0.25° resolution) (Rodell et al., 2004), European Reanalysis (ERA)-interim (0.75°) (Balsamo et al., 2015) and its successors, ERA5 (0.25°) and ERA5-Land (0.1°) (Hoffmann et al., 2019)). These products can often predict temporal variations well due to the incorporation of the time variance of environmental factors, e.g., precipitation. In addition, the modeling approach can also provide information on the soil moisture in soil layers deeper than the surface layer (< 5 cm). The uncertainties arise from meteorological forcing data, model parameters, as well as inadequacies in model physics (Cheng et al., 2017). Moreover, the anthropogenic impacts from irrigation and land cover changes are rarely considered (Kumar et al., 2015; Qiu et al., 2016).

With advances of remote sensing technology, microwave remote sensing became an alternative to soil moisture monitoring. Currently, global-scale soil moisture can be acquired from either passive sensors (e.g., SMMR, SSM/I, TMI, WindSAT, AMSR-E, AMSR2, SMOS, SMAP, see Table 1 for the full names) or active sensors (e.g., ERS and ASCAT), with that within the top 5 cm of soil being detectable (Feng et al., 2017; Jiao et al., 2016; Piles et al., 2018). The data quality and spatial coverage are improved step by step (Karthikeyan et al., 2017b). However, valid temporal spans of all these sensors are limited, and the data quality and spatial coverage were considered to be unsatisfactory until the launch of AMSR-E in June 2002 (Karthikeyan et al., 2017b; Kawanishi et al., 2003). Currently, ASCAT sensors have produced the longest continuous record of global surface soil moisture of microwave remote sensing (Bartalis et al., 2007), with the temporal span from 2007 until present. Satellite-based soil moisture retrievals may also suffer from various disturbances, such as lower quality over dense

删除了: as well as linking

删除了: as …ecause it is probably the best indicator of ecological droughts…

删除了: However, due to the large uncertainty in global-scale soil moisture data,(Sadeghi et al., 2020)(Cheng et al., 2017) the applicability of these data in global ecosystem models is currently limited

删除了: the most …requently used, mainly

删除了: -

删除了: Although t…hese products can often predict temporal variations well due to the incorporation of the time variance of environmental factors, e.g., high-quality …recipitation data,…

移动了(插入) [1]

删除了: Apart from surface soil moisture that can be observed by satellites, …he modeling method …pproach also

删除了: the bias and root mean square error (RMSE) may be large (Bi et al., 2016; Gu et al., 2019)

删除了: significant impacts of human activities…nthropogenic impacts such as…rom irrigation and land cover changes on soil moisture

上移了 [1]: Apart from surface soil moisture that can be

删除了: provides information on the moisture in deeper soil

删除了: the …dvances of remote sensing technology, soil

删除了: surface s…oil moisture monitoring (current satellite

删除了: but …owever, the …alid temporal spans of all these

删除了: the …SCAT sensors have product is the…roduced

删除了: and …ith the temporal span of this product is …rom

删除了: products…etrievals may also suffer from various

删除了: high

[revised manuscript text omitted]

删除了：H

删除了：most of the

删除了：s

删除了：Therefore, c

删除了：precipitation is highest in summer

删除了：  where plants often grow in all seasons

删除了：temporally

删除了：even

删除了：perhaps

删除了：with much higher evapotranspiration

删除了：proves

删除了：and

删除了：red

**4 Discussion and conclusions**

**4.1 Contributions of microwave observations and environmental characteristics to the neural network prediction**

In this study, we developed an improved global long-term remote sensing-based surface soil moisture dataset, named RSSSM. The key algorithm calibrates and fuses various sources of microwave surface soil moisture products through multiple neural networks. Several environmental factors are also chosen as ancillary neural network inputs because they are quality impact factors of microwave soil moisture retrievals, or also director indicators of surface soil moisture. To explore the relative roles of soil moisture data retrieved from microwave observations and the environmental characteristics, we performed contribution tests on all the input features at the global scale (for each predictor, we added a random error that is controlled within the standard deviation of the predictor. Then the increased mean squared error (MSE) in neural network training can be used to determine the relative contribution of that variable). Taking the first independent neural network (NN1-1-1, a primary NN) as an example, the results (Figure 16) indicate that SMOS soil moisture plays the dominant role in the neural network training (55.5%), while the four predictor soil moisture products explained 62.7% in total. The remaining 37.3% of the training efficiency could be attributed to the environmental characteristics, among which the water fraction accounts for the most (13.4%) since it is both a quality impact factor and a direct indicator of soil moisture. The tree cover fraction is an important neural network input as well and reduces the MSE by 7.8%, which is probably due to the strong impact of forest cover on microwave soil moisture retrievals.

**4.2 Requirement of further validations**

[revised manuscript text omitted]